# Sequential changes in ocean circulation and biological export productivity during the last glacial-interglacial cycle: a model-data study

Cameron M. O'Neill[1], Andrew McC. Hogg[1,2], Michael J. Ellwood[1], Bradley N. Opdyke[1], and Stephen M. Eggins[1]

[1]Research School of Earth Sciences, Australian National University, Canberra, Australia
[2]ARC Centre of Excellence for Climate Extremes, Australian National University, Canberra, Australia

*Correspondence to:* Cameron O'Neill (cameron.oneill@anu.edu.au)

**Abstract.**

We conduct a model-data analysis of the marine carbon cycle to understand and quantify the drivers of atmospheric $CO_2$ concentration during the last glacial-interglacial cycle. We use a carbon cycle box model "SCP-M", combined with multiple proxy data for the atmosphere and ocean, to test for variations in ocean circulation and Southern Ocean biological export productivity across marine isotope stages spanning 130 thousand years ago to the present. The model is constrained by proxy data associated with a range of environmental conditions including sea surface temperature, salinity, ocean volume, sea-ice cover and shallow water carbonate production. Model parameters for global ocean circulation, Atlantic meridional overturning circulation and Southern Ocean biological export productivity are optimised in each marine isotope stage against proxy data for atmospheric $CO_2$, $\delta^{13}C$ and $\Delta^{14}C$ and deep ocean $\delta^{13}C$, $\Delta^{14}C$ and $CO_3^{2-}$. Our model-data results suggest that global overturning circulation weakened during marine isotope stage 5d, coincident with a $\sim$25 ppm fall in atmospheric $CO_2$ from the last interglacial period. There was a transient slowdown in Atlantic meridional overturning circulation during marine isotope stage 5b, followed by a more pronounced slowdown and enhanced Southern Ocean biological export productivity during marine isotope stage 4 ($\sim$-30 ppm). In this model, the Last Glacial Maximum was characterised by relatively weak global ocean and Atlantic meridional overturning circulation, and increased Southern Ocean biological export productivity ($\sim$-20 ppm during MIS 3 and MIS 2). Ocean circulation and Southern Ocean biological export productivity returned to modern values by the Holocene period. The terrestrial biosphere decreased by 385 Pg C in the lead up to the Last Glacial Maximum, followed by a period of intense regrowth during the last glacial termination and Holocene ($\sim$600 Pg C). Slowing ocean circulation, a colder ocean and to a lesser extent shallow carbonate dissolution, contributed $\sim$-70 ppm to atmospheric $CO_2$ in the $\sim$100 thousand-year lead-up to the Last Glacial Maximum, with a further $\sim$-15 ppm contributed during the glacial maximum. Our model results also suggest that an increase in Southern Ocean biological export productivity was one of the ingredients required to achieve the Last Glacial Maximum atmospheric $CO_2$ level. We find the incorporation of glacial-interglacial proxy data into a simple quantitative ocean transport model, provides useful insights into the timing of past changes in ocean processes, enhancing our understanding of the carbon cycle during the last glacial-interglacial period.

# 1 Introduction

Large and regular fluctuations in the concentration of atmospheric $CO_2$ and ocean proxy signals for carbon isotopes and carbonate ion concentration during the last 800 kyr, are preserved in ice and marine core records. The most obvious of these fluctuations is the repeated oscillation of atmospheric $CO_2$ concentration over the range of $\sim$180-280 ppm every $\sim$100 kyr. The magnitude and regularity of these oscillations in atmospheric $CO_2$, combined with proxy observations for carbon isotopes, point to the quasi-regular transfer of carbon between the main earth reservoirs: the ocean, atmosphere, terrestrial biosphere and marine sediments (Broecker, 1982; Sigman and Boyle, 2000; Toggweiler, 2008; Hogg, 2008; Kohfeld and Ridgwell, 2009; Menviel et al., 2012; Kohfeld and Chase, 2017; Ganopolski and Brovkin, 2017). The ocean, given its large size as a carbon store and ongoing exchange of $CO_2$ with the atmosphere, likely plays the key role in changing atmospheric $CO_2$ (Broecker, 1982; Knox and McElroy, 1984; Siegenthaler and Wenk, 1984; Sarmiento and Toggweiler, 1984; Sigman and Boyle, 2000; Kohfeld and Ridgwell, 2009). Ocean-centric hypotheses for variation in atmospheric $CO_2$ concentration have been examined in great detail for the Last Glacial Maximum (LGM) and Holocene periods, supported by the abundance of paleo data from marine sediment coring and sampling activity (e.g. Sikes et al., 2000; Curry and Oppo, 2005; Kohfeld and Ridgwell, 2009; Oliver et al., 2010; Menviel et al., 2012; Peterson et al., 2014; Yu et al., 2014b; Menviel et al., 2016; Skinner et al., 2017; Muglia et al., 2018; Yu et al., 2019). However, the hypotheses for variation in atmospheric $CO_2$ across the LGM-Holocene remain debated (e.g. Kohfeld et al., 2005; Martinez-Garcia et al., 2014; Menviel et al., 2016; Skinner et al., 2017; Muglia et al., 2018; Khatiwala et al., 2019). Established hypotheses include those emphasising ocean biology (e.g. Martin, 1990; Martinez-Garcia et al., 2014), ocean circulation (e.g. Burke and Robinson, 2012; Menviel et al., 2016; Skinner et al., 2017), sea surface temperature (SST) (Khatiwala et al., 2019), or the aggregate effect of several mechanisms (e.g. Kohfeld and Ridgwell, 2009; Hain et al., 2010; Köhler et al., 2010; Menviel et al., 2012; Ferrari et al., 2014; Ganopolski and Brovkin, 2017; Muglia et al., 2018) to explain the LGM-Holocene carbon cycle transition. Hypotheses for an ocean biological role include the effects of iron fertilisation on biological export productivity (e.g. Martin, 1990; Watson et al., 2000; Martinez-Garcia et al., 2014), the depth of remineralisation of particulate organic carbon (POC) (e.g. Matsumoto, 2007; Kwon et al., 2009; Menviel et al., 2012), changes in the organic carbon:carbonate ("the rain ratio") or carbon:silicate constitution of marine organisms (e.g. Archer and Maier-Reimer, 1994; Harrison, 2000), and increased biological utilisation of exposed shelf-derived nutrients such as phosphorus (e.g. Menviel et al., 2012).

Several studies have attempted to solve the problem of glacial-interglacial $CO_2$ by modelling either the last glacial-interglacial cycle in its entirety, or multiple glacial-interglacial cycles (e.g. Ganopolski et al., 2010; Menviel et al., 2012; Brovkin et al., 2012; Ganopolski and Brovkin, 2017). These studies highlight the roles of orbitally-forced Northern Hemisphere ice sheets in the onset of the glacial periods, and important feedbacks from ocean circulation, carbonate chemistry and marine biological productivity throughout the glacial cycle (Ganopolski et al., 2010; Brovkin et al., 2012; Ganopolski and Brovkin, 2017). Menviel et al. (2012) modelled a range of physical, biological and biogeochemical mechanisms to deliver the full amplitude of atmospheric $CO_2$ variation in the last glacial-interglacial cycle, using transient simulations with the Bern3D model. According to Brovkin et al. (2012), a $\sim$50 ppm drop in atmospheric $CO_2$ concentration early in the last glacial-interglacial cycle

was caused by lower SST, increased Northern hemisphere ice sheet cover, and expansion of southern-sourced abyssal waters in place of North Atlantic Deep Water (NADW) formation. Ganopolski and Brovkin (2017) modelled the last four glacial-interglacial cycles with orbital forcing as the singular driver of carbon cycle feedbacks. They described the "carbon stew", a feedback of combined physical and biogeochemical changes in the carbon cycle driving the last four glacial-interglacial cycles

of atmospheric $CO_2$.

Kohfeld and Chase (2017) also extended the LGM-Holocene $CO_2$ debate further into the past by evaluating proxy data over the period 115-18 thousand years before present (ka), a time that encompasses the gradual fall in atmospheric $CO_2$ of ∼85-90 ppm from the last interglacial period until the last glacial termination. Kohfeld and Chase (2017) identified time periods during which $CO_2$ decreased and aligned these with concomitant changes in proxies for SST, sea-ice extent, deep Atlantic Ocean

circulation and mixing and ocean biological productivity. Kohfeld and Chase (2017) observed that the ∼100kyr transition to the LGM involved three discrete $CO_2$ reduction events. Firstly, a drop in atmospheric $CO_2$ of ∼35 ppm at ∼115-100 ka (marine isotope stage, or MIS, 5d) was accompanied by lower SST and the expansion of Antarctic sea-ice cover. A second phase of $CO_2$ drawdown between 72 and 65 ka (MIS 4), of ∼40ppm, likely resulted from a slowdown in deep ocean circulation (Kohfeld and Chase, 2017). Finally, during the period 40-18 ka (MIS 3-2) atmospheric $CO_2$ dropped a further 5-10 ppm, which according

to Kohfeld and Chase (2017) was the result of enhanced Southern Ocean biological productivity, continually intensifying deep ocean stratification, shoaling of NADW and northward extension of Antarctic Bottom Water (AABW).

In this paper we quantitatively test the Kohfeld and Chase (2017) hypothesis by undertaking model-data experiments in each MIS across the last glacial-interglacial cycle. We extend their analysis to include Pacific and Indian Ocean modelling and proxy data. We use the SST reconstructions compiled by Kohfeld and Chase (2017) and other proxy records presented in Kohfeld and

Chase (2017), covering the last glacial-interglacial cycle. We apply a carbon cycle box model (O'Neill et al., 2019) constrained by available atmospheric and oceanic proxy data, to solve for optimal model-data parameter solutions for ocean circulation and biological export productivity. We also present a qualitative analysis of the compiled proxy data to place the model-data experiment results in context. We thereby further constrain the timing and magnitude of posited $CO_2$ mechanisms operating during each MIS in the last glacial-interglacial cycle (e.g. Kohfeld and Ridgwell, 2009; Oliver et al., 2010; Menviel et al.,

2012; Brovkin et al., 2012; Yu et al., 2013; Eggleston et al., 2016; Yu et al., 2016; Kohfeld and Chase, 2017). This longer-dated analysis complements recent multi-proxy model-data studies of the LGM and Holocene (e.g. Menviel et al., 2016; Kurahashi-Nakamura et al., 2017; Muglia et al., 2018; O'Neill et al., 2019) by testing for changes in the ocean carbon cycle in the lead-up to the LGM, in addition to the LGM-to-Holocene. Our modelling approach differs from other model studies of the last glacial-interglacial cycle (e.g. Ganopolski et al., 2010; Menviel et al., 2012; Brovkin et al., 2012; Ganopolski and Brovkin, 2017)

because we constrain several physical processes from observations (SST, sea level, sea-ice cover, salinity, coral reef fluxes of carbon), then solve for the values of model parameters for ocean circulation and biology based on an optimisation against atmospheric and ocean proxy data.

## 2 Materials and methods

### 2.1 Model description

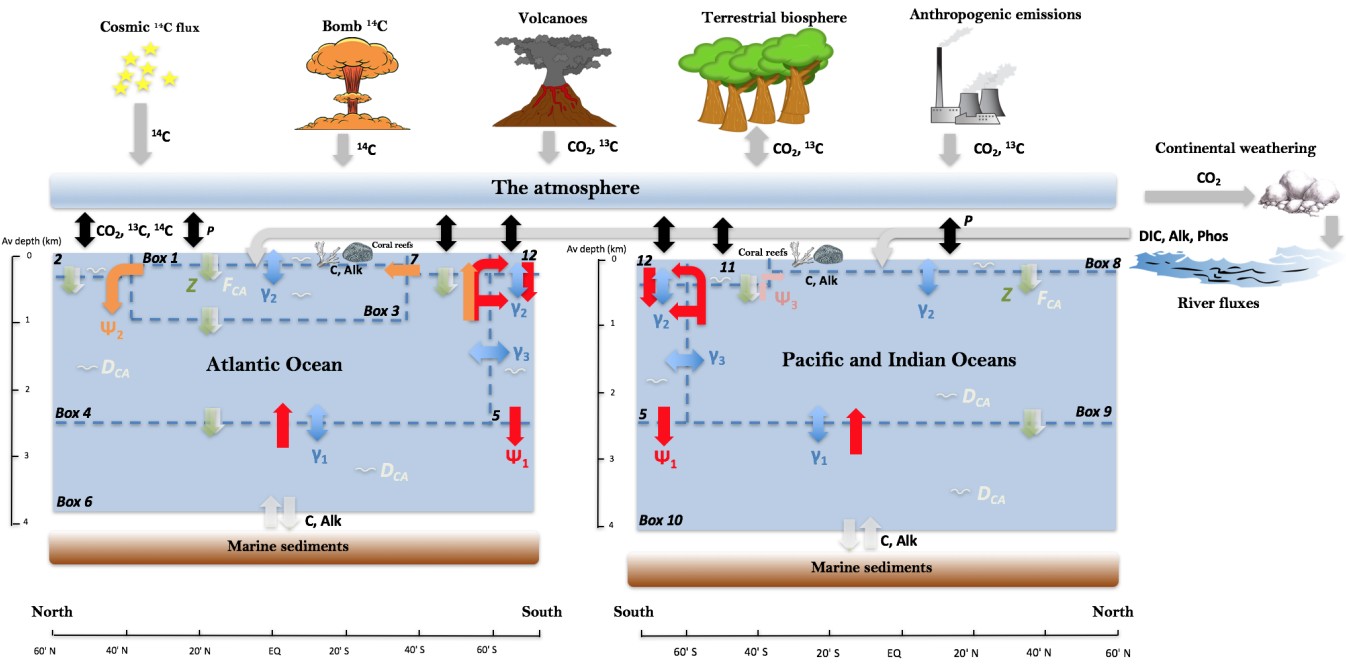

**Figure 1.** SCP-M configured as a twelve box ocean model-plus atmosphere with marine sediments, continents and the terrestrial biosphere. Exchange of elemental concentrations occur due to fluxes between boxes. $\Psi_1$ (red arrows) is global overturning circulation (GOC), $\Psi_2$ (orange arrows) is Atlantic meridional overturning circulation (AMOC). GOC upwelling in both basins is set by default to 50% split between upwelling into the subpolar and polar Southern Ocean. $\Psi_3$ (pink arrows) is Antarctic intermediate water (AAIW) and Subantarctic mode water (SAMW) formation in the Indian and Pacific Oceans (e.g. Talley, 2013). Blue arrows represent mixing fluxes between boxes. $\gamma_1$ and $\gamma_3$ parameterise deep-abyssal and Southern Ocean-deep topographically-induced mixing (e.g. De Boer and Hogg, 2014), while $\gamma_2$ is low-latitude thermohaline mixing (e.g. Liu et al., 2016). $Z$ (green downward arrows) is the biological pump, $F_{CA}$ (white downward arrows) is the carbonate pump, $D_{CA}$ (white squiggles) is carbonate dissolution and $P$ (black, bidirectional arrows) is the air-sea gas exchange. Key to boxes: Atlantic (box 1: low latitude/tropical surface ocean, 0-100m; box 2: northern surface ocean, 0-250m; box 3: intermediate ocean, 100-1,000m; box 4: deep ocean, 1,000-2,500m; box 6: abyssal ocean, 2,500-3,700m; box 7: subpolar southern surface ocean, 0-250m). Pacific-Indian (box 8: low latitude/tropical surface ocean, 0-100m; box 9: deep ocean, 100-2,500m; box 10: abyssal ocean, 2,500-4,000m; box 11: subpolar southern surface ocean, 0-250m). Southern Ocean (box 5: intermediate-deep; box 12: surface ocean). For a more detailed model description see O'Neill et al. (2019) and updated model code and data at https://doi.org/10.5281/zenodo.4084586.

We use the SCP-M carbon cycle box model in our model-data experiment (O'Neill et al., 2019). In summary, SCP-M contains simple parameterisations of the major fluxes in the Earth's surface carbon cycle (Fig. 1). SCP-M incorporates the

ocean, atmosphere, terrestrial biosphere and marine/continental sediment carbon reservoirs, weathering and river fluxes, and a number of variables including atmospheric $CO_2$, DIC, phosphorus, alkalinity, carbon isotopes ($^{13}C$ and $^{14}C$) and $CO_3^{2-}$. SCP-M calculates ocean $pCO_2$ using the equations of Follows et al. (2006), and applies the first and second "dissociation constants" of carbonic acid estimated by Lueker et al. (2000), to calculate $HCO_3^-$ and $CO_3^{2-}$ concentrations, respectively, in units of $\mu$mol kg$^{-1}$, in each ocean box. The model employs partial differential equations for determining the concentration of elements, with each box represented as a row and column in a matrix. In this paper, we extend SCP-M by incorporating a separate basin for the combined Pacific and Indian Oceans (Fig. 1) following the conceptual model of Talley (2013), to incorporate modelling and proxy data for those regions of the ocean. This version of SCP-M consists of 12 ocean boxes plus the atmosphere and terrestrial biosphere. SCP-M splits out depth regions of the ocean between surface boxes (100-250m average depth), intermediate (1,000m average depth), deep (2,500m average depth) and abyssal depth boxes (3,700 (Atlantic) - 4,000m (Pacific-Indian) average depth). The Southern Ocean is split into two boxes, including a polar box which covers latitude range 60-80 degrees South (box 12 in Fig. 1) and subpolar Southern Ocean boxes in the Atlantic (box 7) and Pacific-Indian (box 11) basins, which cover latitude range 40-60 degrees South. See O'Neill et al. (2019) for a discussion of the choice of box depth and latitude dimensions.

The major ocean carbon flux parameters of interest in this model-data study are global ocean circulation (GOC), $\Psi_1$, Atlantic meridional overturning circulation (AMOC), $\Psi_2$, and ocean biological export productivity, $Z$. The ocean circulation parameters $\Psi_1$ and $\Psi_2$ are simply prescribed in units of Sverdrups (Sv, $10^6$ m$^3$ s$^{-1}$). Ocean biological export productivity $Z$ is calculated using the method of Martin et al. (1987). The biological productivity flux at 100m depth is attenuated with depth for each box according to the decay rule of Martin et al. (1987). Each sub surface box receives a biological flux of an element at its ceiling depth, and loses a flux at its floor depth (lost to the boxes below it). The difference between influx and out-flux is the amount of element that is remineralised into each box. The input parameter is the value of export production at 100m depth, in units of mol C m$^{-2}$ yr$^{-1}$ as per Martin et al. (1987). Equation (1) shows the general form of the Martin et al. (1987) equation:

$$F = F_{100}(\frac{d}{100})^b \tag{1}$$

Where $F$ is a flux of carbon in mol C m$^{-2}$ yr$^{-1}$, $F_{100}$ is an estimate of carbon flux at 100m depth, $d$ is depth in metres and $b$ is a depth scalar. In SCP-M, the $Z$ parameter implements the Martin et al. (1987) equation. $Z$ is an estimate of biological productivity at 100m depth (in mol C m$^{-2}$ yr$^{-1}$), and coupled with the Martin et al. (1987) depth scalar, controls the amount of organic carbon that sinks from each model surface box to the boxes below.

Air-sea gas exchange is based on the relative $pCO_2$ between the surface ocean boxes and the atmosphere and is implemented in SCP-M by a parameter that sets its rate in m day$^{-1}$, $P$ (Fig. 1). SCP-M parameterises shallow water carbonate production, which is linked to the $Z$ parameter by an assumption for the relative proportion of carbonate vs organic matter in the biological export flux, known as "the rain ratio" (e.g. Archer and Maier-Reimer, 1994; Ridgwell, 2003). Carbonate dissolution is calculated based on the ocean box or marine surface sediment calcium carbonate concentration relative to a depth-dependant saturation concentration (Morse and Berner, 1972; Millero, 1983). The isotopes of carbon are calculated applying various

fractionation factors associated with the biological, physical and chemical fluxes of carbon (see Table S1 and O'Neill et al. (2019)).

We have added a simple representation of shallow water carbonate fluxes of carbon and alkalinity in SCP-M's low latitude surface boxes, to cater for this feature in theories for glacial-interglacial cycle $CO_2$ (e.g. Berger, 1982; Opdyke and Walker, 1992; Ridgwell et al., 2003; Vecsei and Berger, 2004; Menviel and Joos, 2012), using:

$$\left[\frac{dC_i}{dt}\right]_{reef} = C_{reef}/V_i \tag{2}$$

Where $C_{reef}$ is the prescribed flux of carbon out of/into the low latitude surface ocean boxes during net reef accumulation/dissolution, in mol C yr$^{-1}$, and $V_i$ is the volume of the low latitude surface box $i$. The alkalinity flux associated with reef production/dissolution is simply Eq. 2 multiplied by two (e.g. Sarmiento and Gruber, 2006).

SCP-M contains a simple parameterisation of the terrestrial carbon cycle. For continental rock weathering, we apply the simple scheme of Walker and Kasting (1992) as implemented in Toggweiler (2008), Hogg (2008) and Zeebe (2012). Weathering of silicate and carbonate rocks supplies DIC and alkalinity to the low latitude surface ocean boxes in each basin (boxes 1 and 8 in Fig. 1) as a function of a weathering constant and atmospheric $CO_2$, in units of mol m$^{-3}$ yr$^{-1}$. The parameter values used are shown in Table S1. For the SCP-M weathering equations please see O'Neill et al. (2019). $\delta^{13}$C fluxes for carbonate and silicate weathering are shown in Table S1. A volcanic flux of carbon (and $\delta^{13}$C) is also assumed which sets the rate of volcanic $CO_2$ outgassing roughly to the rate of silicate rock weathering (Walker and Kasting, 1992; Toggweiler, 2008; Hogg, 2008; Zeebe, 2012). Parameters for volcanic $CO_2$ and $\delta^{13}$C fluxes are shown in Table S1.

The terrestrial biosphere is represented in SCP-M as a stock of carbon (a box) that fluxes with the atmosphere, governed by parameters for net primary productivity (NPP) and respiration. In SCP-M, NPP is calculated as a function of carbon fertilisation, which increases NPP as atmospheric $CO_2$ rises via a simple logarithmic relationship, using the model of Harman et al. (2011). This is a simplified approach, which omits the effects of temperature and precipitation on NPP (François et al., 1999; van der Sleen et al., 2015). The terrestrial biosphere module in SCP-M assumes a fixed $\delta^{13}$C fractionation factor of -23‰ (Table S1).

The major fluxes of carbon are parameterised simply in SCP-M to allow them to be solved by model-data optimisation with respect to atmospheric and ocean proxy data. In this study the values for GOC, AMOC and biological export productivity at 100m depth are outputs of the model-data experiments, as they are deduced from a data optimisation routine. Their input values for the experiments are ranges, as described in 2.2.1. SCP-M's fast run time and flexibility renders it useful for long term paleo-reconstructions involving large numbers of quantitative experiments and data integration (O'Neill et al., 2019). SCP-M is a simple box model, which incorporates large regions of the ocean as averaged boxes and parameterised fluxes. It is an appropriate tool for this study, in which we evaluate many tens of thousands of simulations to explore possible parameter combinations, in conjunction with proxy data.

## 2.2 Model-data experiment design

We undertake series of model-data experiments to solve for the values of ocean circulation and biological parameters for each MIS during the last glacial-interglacial cycle (130-0 ka). We target these parameters due to their central role in many

LGM-Holocene $CO_2$ hypotheses (e.g. Knox and McElroy, 1984; Siegenthaler and Wenk, 1984; Toggweiler and Sarmiento, 1985; Martin, 1990; Kohfeld and Ridgwell, 2009; Hain et al., 2010; Sigman et al., 2010; Yu et al., 2014a; Menviel et al., 2016; Kohfeld and Chase, 2017; Muglia et al., 2018; Menviel et al., 2020). We force SST, salinity, sea volume and ice cover, and reef carbonate production, in each MIS (Section 2.2.1, Fig. 2), using values sourced from the literature (e.g. Opdyke

and Walker, 1992; Key, 2001; Adkins et al., 2002; Ridgwell et al., 2003; Kohfeld and Ridgwell, 2009; Rohling et al., 2009; Wolff et al., 2010; Muscheler et al., 2014; Kohfeld and Chase, 2017). Then, we optimise the model parameters for GOC, AMOC and Southern Ocean biological export productivity in each MIS time slice. We choose GOC and AMOC due to the prevalence of varying ocean circulation in many theories for glacial-interglacial cycles of $CO_2$ (e.g. Sarmiento and Toggweiler, 1984; Siegenthaler and Wenk, 1984; Toggweiler, 1999; Kohfeld and Ridgwell, 2009; Burke and Robinson, 2012; Freeman

et al., 2016; Menviel et al., 2016; Kohfeld and Chase, 2017; Skinner et al., 2017; Muglia et al., 2018; Menviel et al., 2020), and its key role in distribution of carbon and other elements in the ocean (Talley, 2013). We choose to vary Southern Ocean biological export productivity due to its long-standing place and debate among theories of atmospheric $CO_2$ during the LGM and Holocene (e.g. Martin, 1990; Knox and McElroy, 1984; Sarmiento and Toggweiler, 1984; Sigman and Boyle, 2000; Anderson et al., 2002; Kohfeld and Ridgwell, 2009; Martinez-Garcia et al., 2014; Menviel et al., 2016; Kohfeld and Chase,

2017; Muglia et al., 2018).

The GOC ($\Psi_1$), AMOC ($\Psi_2$) and Southern Ocean biology ($Z$) parameters are varied over ~9,000 possible combinations for each MIS, a total of ~80,000 simulations across MIS 5e-1. At the end of each experiment batch, the model results are solved for the best fit to the ocean and atmosphere proxy data using a least-squares optimisation and the parameter values for $\Psi_1$, $\Psi_2$ and $Z$ are returned. Our experiment time slices are the MIS of Lisiecki and Raymo (2005), with two minor modifications (see

Fig. 2). MIS 2 (14-29 ka) as per Lisiecki and Raymo (2005) straddles the LGM (18-24 ka) and the last glacial termination (15-18 ka), while MIS 1 (0-14 ka) incorporates the Holocene period (0-11.7 ka) and the end of the termination. We are interested in the LGM and Holocene as discrete periods, so our experiment time slice for MIS 2 is truncated at 18 ka and our MIS 1 simply covers the Holocene, removing overlaps with the glacial termination. Therefore, our modelling excludes the last glacial termination (~11-18 ka). The glacial termination period was highly transient with atmospheric $CO_2$ varying by ~85 ppm in

<10 kyr and large changes in carbon isotopes. Thus it is anticipated that in a model-data reconstruction model parameters would vary substantially for this period. Joos et al. (2004), Ganopolski et al. (2010), Menviel et al. (2012), Menviel and Joos (2012), Brovkin et al. (2012) and Ganopolski and Brovkin (2017) provide coverage of the termination period with transient simulations, using intermediate complexity models (more complex than our model). For MIS 5, we take the timing for peak glacial and interglacial substages of Lisiecki and Raymo (2005), ±5kyr for MIS 5c-5e, and ±2.5 kyr for MIS 5a-5b.

**2.2.1   Model forcings and parameter variations**

We take a reconstructed SST time series for the last 130 kyr (Kohfeld and Chase, 2017), map these to SCP-M's surface boxes and average the time series across each MIS (Fig. 2A). We extrapolate an Antarctic sea ice cover proxy as shown in Fig. 2B (Wolff et al., 2010) to the profiles for sea surface salinity (Fig. 2C) and the polar Southern Ocean box air-sea gas exchange parameter (Fig. 2D). For example, our notional reduction in the strength of the polar Southern Ocean box air-sea gas

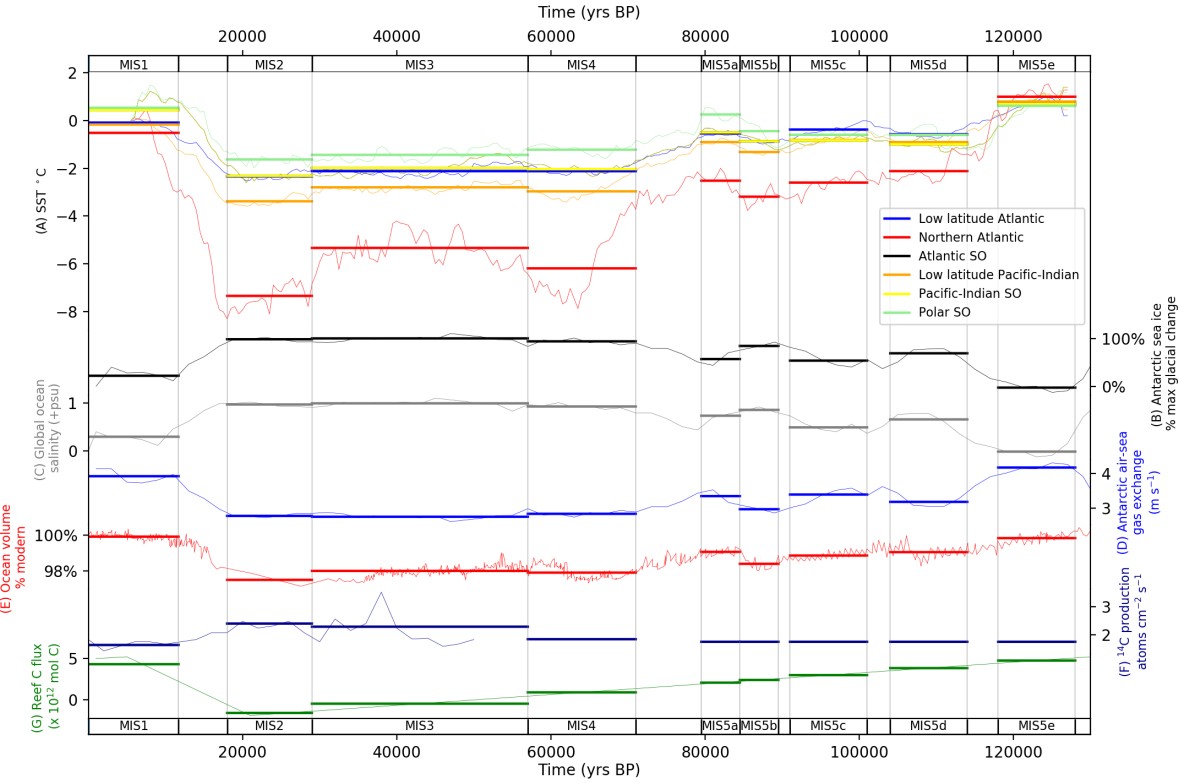

**Figure 2.** Model forcings for MIS across the last glacial-interglacial cycle. (A) sea surface temperature reconstruction of Kohfeld and Chase (2017), mean values mapped into SCP-M surface boxes (fine lines) and averaged across MIS (bold horizontal lines). (B) Proxy for Antarctic sea-ice extent using ssNa fluxes from the EPICA Dome C ice core (Wolff et al., 2010), used to temporally contour MIS model forcings for (C) salinity (Adkins et al., 2002) and (D) polar Southern Ocean air-sea gas exchange. Global ocean salinity is forced to a glacial maximum of +1 psu (shown in (C)) and the polar Southern Ocean is forced to +2 psu (not shown), as modified from Adkins et al. (2002). Ocean volume (E) forced using global relative sea level reconstruction of Rohling et al. (2009). (F) Atmospheric $^{14}$C production rate time series for 0-50 ka of Muscheler et al. (2014) . Long-term values assumed for >50 ka (Key, 2001). (G) Shallow water carbonate flux of carbon from Ridgwell et al. (2003) profiled across the glacial-interglacial cycle using a curve from Opdyke and Walker (1992). Fine lines are the time series data and bold lines are the model forcings in each MIS. Data behind the figure are shown in Tables S2 and S3.

exchange due to Antarctic sea ice cover (-30%) is linearly (negatively) profiled with the Antarctic sea ice proxy time series of Wolff et al. (2010). Note the polar Southern Ocean box, which is forced with reduced air-sea exchange, is separate from the subpolar Southern Ocean Box in which the biological export productivity parameter is varied in the model-data experiment. Our treatment of sea-ice cover is simply as a regulator of air-sea gas exchange in the polar Southern Ocean surface boxes in
5 each basin, not as a driver of other physical processes or biogeochemical feedbacks (e.g. Morrison and Hogg, 2013; Ferrari et al., 2014; Jansen, 2017; Kohfeld and Chase, 2017; Marzocchi and Jansen, 2017). Furthermore, our linear application of the

sea-ice proxy data of Wolff et al. (2010) to our air-sea gas exchange parameter (Fig. 2D) may overestimate its effect on the model results early in the glacial period (MIS 5d) and underestimate its effects during MIS 4-2 (Wolff et al., 2010).

Adkins et al. (2002) reconstructed LGM deep-sea salinity for the Southern, Atlantic and Pacific Oceans. They found increased salinity for the LGM at all locations across a range of +0.95-2.4 practical salinity units (psu) above modern values,
with an average value of +1.5 psu. The most saline LGM waters were in the Southern Ocean (+ 2.4 psu), with Atlantic and Pacific waters ranging +0.95-1.46 psu and a global ocean average of +1.2 psu. Adkins et al. (2002) also observed that within a (globally) more saline ocean, lower glacial temperatures would have caused less evaporation during the LGM, a negative feedback on salinity. We choose a forcing for LGM sea surface salinity of +1 psu for the global ocean and +2 psu for the polar Southern Ocean, relative to the interglacial period. These values conservatively reflect the hypothesis that surface evaporation
may have been less in the LGM, hence a lesser magnitude of change in salinity in the surface ocean relative to the deep ocean values estimated by Adkins et al. (2002), and also that the most voluminous parts of the ocean were less saline than the Southern Ocean (Adkins et al., 2002). In our model-data experiments, the estimated glacial change in sea surface salinity (Fig. 2C) is also contoured through time with the variation in Antarctic sea-ice cover of Wolff et al. (2010). Adkins et al. (2002) observed that glacial salinity is a poor predictor of global mean sea level, due to storage of saline waters in ice shelves and groundwater
reserves. Therefore, the proxy for Antarctic sea-ice cover may have a more direct linkage to sea surface salinity than using global sea level, for our purposes of estimating glacial-interglacial evolution in salinity.

Rohling et al. (2009) reconstructed global relative sea level (RSL) over the past five glacial-interglacial cycles. According to Rohling et al. (2009), the glacial RSL minimum was ∼-115m at ∼27 ka, immediately prior to the LGM. We perform a simple calculation to reduce ocean depth and volume in SCP-M, in line with the Rohling et al. (2009) time series. In a box model
this is only an approximation, given the lack of topographical detail. Varying ocean box volume and surface area affects the ocean surface area available for in-gassing and de-gassing, and overall ocean capacity to store $CO_2$, which impacts atmospheric $CO_2$, $\delta^{13}C$ and $\Delta^{14}C$ (Köhler et al., 2010; O'Neill et al., 2019). Opdyke and Walker (1992) reconstructed coral reef carbonate fluxes of $CaCO_3$ for the last glacial-interglacial cycle for the purposes of modelling the "coral reef hypothesis". According to Opdyke and Walker (1992), reef carbon fluxes (out of the ocean) declined through the glacial cycle, with net dissolution
in MIS 3 and MIS 2 leading to positive fluxes of carbon and alkalinity into the ocean in those periods. Fluxes of carbon and alkalinity out of the ocean into coral reefs, rebounded from the LGM (MIS 2) into the Holocene (MIS 1), driven by increased sea level and temperature (Kleypas, 1997). Given that Opdyke and Walker (1992) evaluated the possibility for coral reefs to drive the entire glacial-interglacial $CO_2$ variation, we take the more conservative modelling assumption of Ridgwell et al. (2003) of 0.5 x $10^{17}$ mol C for the postglacial accumulation of coral reefs. We profile this value across the glacial-interglacial
cycle accumulation/dissolution curve of Opdyke and Walker (1992) as shown in Fig. 2. We apply the estimated atmospheric production rate for $^{14}C$ for the last 50 kyr of Muscheler et al. (2014), with a long term average production rate of ∼1.7 atoms $cm^{-2}$ $s^{-1}$ assumed for 130-50 ka (Key, 2001). Model forcing values are shown in Tables S2 and S3.

The terrestrial biosphere module in SCP-M does not explicitly represent the carbon stored in buried peat, permafrost and also cold-climate vegetation that may have expanded its footprint in the glaciation, such as tundra biomes (e.g. Tarnocai et al., 2009;
Ciais et al., 2012; Schneider et al., 2013; Eggleston et al., 2016; Ganopolski and Brovkin, 2017; Treat et al., 2019). The freezing

and burial of organic matter across the glacial period sequesters carbon on land and may modify atmospheric $CO_2$ and $\delta^{13}C$ (Tarnocai et al., 2009; Ciais et al., 2012; Schneider et al., 2013; Eggleston et al., 2016; Ganopolski and Brovkin, 2017; Mauritz et al., 2018; Treat et al., 2019). Ganopolski and Brovkin (2017) incorporated permafrost, peat, and buried land carbon into their transient simulations of the last four glacial-interglacial cycles with the CLIMBER-2 model. Ganopolski and Brovkin (2017)

observed that these features dampened the amplitude of glacial-interglacial variations in terrestrial biosphere carbon stock and its effects on atmospheric $CO_2$. As a crude measure to account for this counter-$CO_2$ cycle storage of carbon in the terrestrial biosphere and frozen soils/buried carbon, we force the terrestrial biosphere productivity parameter in SCP-M in the range ∼+5-10 Pg C yr$^{-1}$ thoughout the last glacial-interglacial cycle, increasing into the LGM (MIS 2), and maintained in the Holocene (MIS 1). We maintain this forcing in the Holocene, as the posited effects of buried peat and permafrost storage of carbon

on atmospheric $CO_2$ and $\delta^{13}C$ during the lead-up to the LGM were likely not reversed after the glacial termination (Tarnocai et al., 2009; Eggleston et al., 2016; Mauritz et al., 2018; Lindgren et al., 2018; Treat et al., 2019). SCP-M calculates net primary productivity (NPP) using this productivity input parameter and a logarithmic function of carbon fertilisation (Harman et al., 2011).

More than 9,000 model simulations are undertaken across the parameter ranges in Table 1 for each MIS. Parameters are

varied simultaneously to allow coverage of all possible combinations of the parameter values within their respective experiment ranges. Within these ranges, values are incremented by 1 Sv for GOC ($\Psi_1$) and AMOC ($\Psi_2$), and ∼0.5 mol C m$^{-2}$ yr$^{-1}$ for Atlantic Southern Ocean biological export productivity ($Z$). Each simulation is run for 10 kyr to enable the model to achieve steady state. We show the experiment ranges for the biological export productivity parameter $Z$ for the Atlantic and Pacific-Indian sectors of the Southern Ocean (Table 1). In SCP-M, the Pacific-Indian Southern Ocean biological export productivity

parameter (in mol C m$^{-2}$ yr$^{-1}$) is set by default at a value of ∼70% of the corresponding Atlantic sector Southern Ocean box, to align with natural observations of variations in the Southern Ocean biological export productivity (e.g. Dunne et al., 2005; Sarmiento and Gruber, 2006; Henson et al., 2011; Siegel et al., 2014; DeVries and Weber, 2017). This variation is reflected in the values in Table 1. In the experiments, the values for $Z$ in the Pacific-Indian Southern Ocean surface box scale linearly with the values for the Atlantic Southern Ocean surface box (Table 1). Herein we focus our presentation and discussion of

the experiment results for the $Z$ parameter on the Atlantic Southern Ocean due to its prominence in glacial-interglacial cycle hypotheses for increased biological productivity (e.g. Martinez-Garcia et al., 2014; Lambert et al., 2015; Shaffer and Lambert, 2018; Muglia et al., 2018).

### 2.2.2 Optimisation procedure

We perform a least squares optimisation of the model experiment output against MIS data for atmospheric $CO_2$, atmospheric and deep and abyssal ocean $\Delta^{14}C$ and $\delta^{13}C$, and deep and abyssal ocean carbonate ion proxy, to source the best-fit parameter

**Table 1.** Free-floating parameter ranges in the model-data experiments for global overturning circulation (GOC, $\Psi_1$), Atlantic meridional overturning circulation (AMOC, $\Psi_2$) and Southern Ocean biological export productivity ($Z$). Parameters are varied simultaneously across these ranges and then optimised against proxy data in each MIS. Also shown are pre-industrial control values for GOC (Talley, 2013), AMOC (Talley, 2013) and Southern Ocean biological export productivity (Dunne et al., 2005; Sarmiento and Gruber, 2006; Henson et al., 2011; Siegel et al., 2014; DeVries and Weber, 2017). The Pacific-Indian Southern Ocean biology parameter is set at a base value of ~70% Atlantic Southern Ocean box, but scales linearly with the Atlantic Ocean parameter in the experiments. The smaller values for Pacific-Indian Southern Ocean takes account of natural observations of a relatively stronger biological export productivity in the Atlantic sector of the subpolar Southern Ocean (e.g. Dunne et al., 2005; Sarmiento and Gruber, 2006; Henson et al., 2011; Siegel et al., 2014; DeVries and Weber, 2017).

| Time period | GOC ($\Psi_1$) Sv | AMOC ($\Psi_2$) Sv | Southern Atlantic (Pacific-Indian) Ocean biology ($Z$) mol C m$^{-2}$ yr$^{-1}$ |
|---|---|---|---|
| PI control values | 29 | 19 | 3.2 (2.2) |
| MIS experiment ranges | 10-35 | 10-25 | 0.5-6.5 (0.3-4.5) |

values for GOC, AMOC and Southern Ocean biological productivity in each time slice - a brute force form of the *gradient descent* method for optimisation (e.g. Strutz, 2016). The equation for least fit applied is:

$$Opt_n = Min \sum_{i,k=1}^{N} (\frac{R_{i,k} - D_{i,k}}{\sigma_{i,k}})^2 \tag{3}$$

where: $Opt_n$ = optimal value of parameters $n$ (e.g. GOC, AMOC and Southern Ocean biological productivity), $R_{i,k}$ = model output for concentration of each element $i$ in box $k$, $D_{i,k}$ = average data concentration each element $i$ in box $k$ and $\sigma_{i,k}$ = standard deviation of the data for each element $i$ in box $k$. The standard deviation performs two roles. It normalises for different unit scales (e.g. ppm, ‰ and $\mu$mol kg$^{-1}$), which allows multiple proxies to be incorporated in the optimisation, and reduces the weighting of a proxy data point with a high standard deviation and therefore an uncertain value. The weighting by proxy data standard deviation also fulfils the important role of accounting for data variance in the optimised parameter results, such that the effects of data variance are embedded in the optimised parameter values. Where proxy data is unavailable for a box, that data and box combination is automatically omitted from the optimisation routine. The experiment routine returns the model run with the best fit to the data, and the model's parameters and results.

**Table 2.** Ocean and atmosphere proxy data sources for the last glacial-interglacial cycle

| Indicator | Time period coverage | Reference |
|---|---|---|
| Atmosphere $CO_2$ | 0-155 ka | Monnin et al. (2004), MacFarling Meure et al. (2006), Bereiter et al. (2012), Rubino et al. (2013), Schneider et al. (2013), Ahn and Brook (2014), Marcott et al. (2014), Bereiter et al. (2015), (all data found at https://www.ncdc.noaa.gov/paleo-search/study/17975) |
| Atmosphere $\delta^{13}C$ | 0-155 ka | Elsig et al. (2009), Schmitt et al. (2012), Schneider et al. (2013), Eggleston et al. (2016) |
| Atmosphere $\Delta^{14}C$ | 0-50 ka | Reimer et al. (2009) |
| Ocean $\delta^{13}C$ | 0-150 ka | Oliver et al. (2010), Govin et al. (2009), Piotrowski et al. (2009) |
| Ocean $\Delta^{14}C$ | 0-40 ka | Skinner and Shackleton (2004), Marchitto et al. (2007), Barker et al. (2010), Bryan et al. (2010), Skinner et al. (2010), Burke and Robinson (2012), Siani et al. (2013), Davies-Walczak et al. (2014), Skinner et al. (2015), Chen et al. (2015), Hines et al. (2015), Sikes et al. (2016), Ronge et al. (2016), Skinner et al. (2017), Zhao et al. (2017) |
| $CO_3^{2-}$ as deduced from B/Ca | 0-705 ka | Yu et al. (2010), Yu et al. (2013), Yu et al. (2014b), Yu et al. (2014a), Broecker et al. (2015), Yu et al. (2016), Qin et al. (2017), Qin et al. (2018), Chalk et al. (2019) |

## 2.3 Data

Our model-data optimisation rests on compilations of atmospheric and ocean paleo proxy data. We compile and apply published proxy data for atmospheric $CO_2$, $\delta^{13}C$ and $\Delta^{14}C$ and ocean $\delta^{13}C$, $\Delta^{14}C$ and $CO_3^{2-}$ concentration. We calculate the simple mean and standard deviation of data points for each model box and MIS. The proxy data for each ocean box is binned into model box based on depth, latitude and longitude which assigns the data to either the Atlantic or Pacific-Indian basin. The box-mapped data are binned into MIS age groups and the sample population is then averaged and the standard deviation is calculated. The standard deviation is then used as a weighting in the model-data optimisation procedure. Sources of proxy data are shown in Table 2 and data locations in Fig. 3. MIS and model box-averaged atmospheric and ocean proxy data and their respective standard deviations are shown in Tables S4-S7.

### 2.3.1 Ocean carbon isotopes

We gather published marine $\Delta^{14}C$ data extending back to ~40 ka (Table 2). Our dataset incorporates individual records contributed over the last ~thirty years and supplemented by the recent compilations of Skinner et al. (2017) and Zhao et al.

(2017). The data total ~75 individual location estimates for benthic and planktonic foraminifera, and deep sea corals. We have restricted our efforts to time series which contain independent calendar ages, and therefore corrections for radioactive decay in the time since the sample was deposited (yielding $\Delta^{14}C$). Figure 3 shows the geographic distribution of the $\Delta^{14}C$ data, which is generally concentrated on ocean basin margins. Some regions, such as the central Pacific, southern Indian and polar Southern Ocean, are devoid of data.

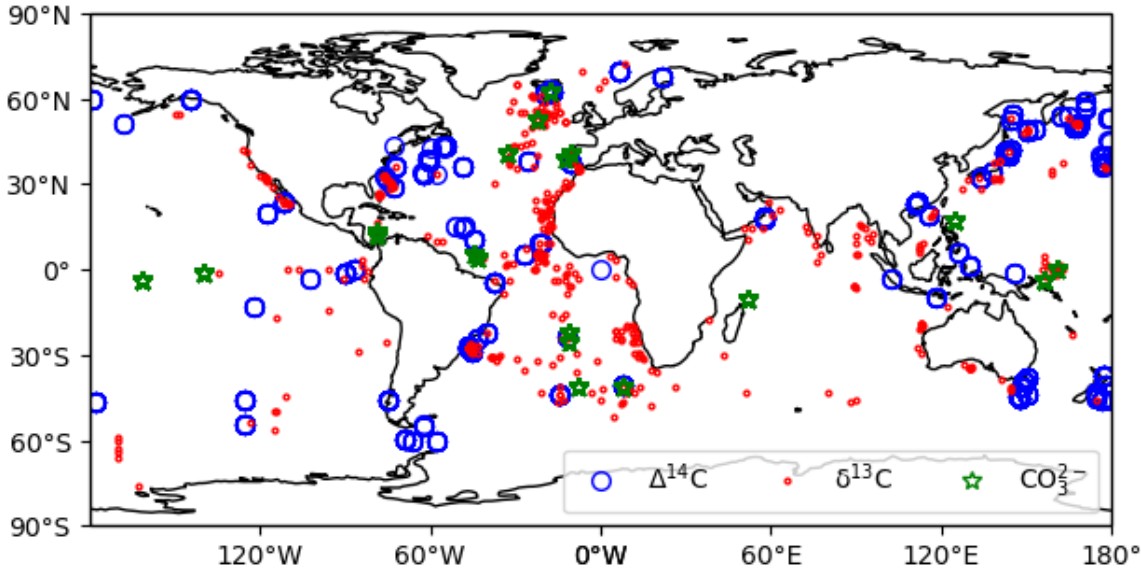

**Figure 3.** $\Delta^{14}C$, $\delta^{13}C$ and $CO_3^{2-}$ data locations. $\Delta^{14}C$ and $CO_3^{2-}$ data is compiled from published estimates. For $\delta^{13}C$ we take the compilation of Oliver et al. (2010). MIS and model box-averaged data and their respective standard deviations are shown in Tables S4-S7.

Oliver et al. (2010) compiled a global dataset of 240 cores of marine $\delta^{13}C$ data encompassing benthic and planktonic species for the last ~150 kyrs. Oliver et al. (2010) observed considerable uncertainties associated with the broad range of species included, particularly for the planktonic foraminifera. By comparison, Peterson et al. (2014) aggregated marine $\delta^{13}C$ for the LGM and late Holocene periods, as time period averages, exclusively sampling benthic *C. wuellerstorfi* data which is a more
10   reliable indicator of marine $\delta^{13}C$ (Oliver et al., 2010; Peterson et al., 2014). To narrow the range of uncertainty, we constrain our use of marine $\delta^{13}C$ data to the deep and abyssal (>2,500m) benthic *Cibicides* species foraminifera samples in the Oliver et al. (2010) dataset, supplemented with *Cibicides* species $\delta^{13}C$ proxy data from Govin et al. (2009) and Piotrowski et al. (2009) (Table 2). Figure 3 shows the $\delta^{13}C$ data locations from Oliver et al. (2010), which are concentrated in the Atlantic Ocean. We map and average the carbon isotope data into SCP-M's boxes on depth and latitude coordinates (Fig. 1), and averaged for each
15   MIS time slice.

### 2.3.2 Carbonate ion proxy

We aggregate ocean carbonate ion proxy data (as deduced from B/Ca) from the sources shown in Table 2 and locations in Fig. 3, map into SCP-M box coordinates and average the data across MIS. The data coverage for $CO_3^{2-}$ is relatively sparse, with $<20$ individual site locations across the global ocean. However, the depth and lateral coverage of SCP-M's boxes is large, particularly in the case of the deep ocean boxes, which cover the full lateral extent of the Pacific-Indian and Atlantic oceans, and depth ranges of 100-2,500m (Pacific-Indian) and 250-2,500m (Atlantic). $CO_3^{2-}$ can vary by more than 100 $\mu$mol kg$^{-1}$ across the depth range 100-2,500m, and can vary by up to $\sim$200 $\mu$mol kg$^{-1}$ in the shallow ocean (e.g. Sarmiento and Gruber, 2006; Yu et al., 2014b, a). Some boxes contain only one core, creating an exceptionally low standard deviation range relative to the other ocean proxies. In other cases, such as the deep Atlantic ocean, the data points are clustered within the 2,000-2,500m depth range, the bottom third of the corresponding SCP-M box. This clustering becomes a problem for the SCP-M box model, which outputs average concentrations over the complete depth range of each box - a drawback of using a large resolution box model to analyse proxy data at a global ocean level. Furthermore, the very low standard deviations associated with the $CO_3^{2-}$ data (shown in Table S6) cause it to assume a disproportionate weighting in the model-data optimisation, which uses standard deviation for weighting of proxies, relative to ocean $\delta^{13}C$ and $\Delta^{14}C$. The latter proxies often have box standard deviations up to 100% of their mean value, when averaged across a box. This issue is also an artefact of our procedure necessary to normalise the different proxies (each in unique units) in a multi-proxy model-data optimisation, by using the standard deviation as a weighting. To deal with this, we assign an arbitrary standard deviation (weighting) of 20 $\mu$mol kg$^{-1}$ to $CO_3^{2-}$ data observations in our model-data optimisations, which acts as a feasible weighting for the processing of $CO_3^{2-}$ relative to the other ocean proxy data. This value is a small fraction of the variation in $CO_3^{2-}$ concentrations observed over the depth range 100-2,500m in the modern ocean (e.g. Key et al., 2004; Yu et al., 2014b).

## 3 Data analysis

In this section we describe the proxy data used to constrain the glacial-interglacial model-data experiments. We depict the major changes in atmospheric $CO_2$, $\delta^{13}C$ and $\Delta^{14}C$, and ocean $\delta^{13}C$, $\Delta^{14}C$ and $CO_3^{2-}$ proxy data across the model box locations and MIS in the last glacial-interglacial cycle. We mainly refer to changes in the MIS-averaged proxy data.

Figure 4 shows the atmospheric data used to constrain the model-data experiments, averaged into MIS time slices. There are many fluctuations and transient changes throughout the last glacial-interglacial cycle, but there are three major sustained reductions in atmospheric $CO_2$ concentration in the lead-up to the LGM (Fig. 4A). An average drop of $\sim$25 ppm during MIS 5d (115-100 ka), a further average drop of $\sim$30 ppm during MIS 4 (72-65 ka) and finally a fall of $\sim$20 ppm in the period leading up to the LGM (during MIS 3 and 2, 40-18 ka). These are the three major $CO_2$ events described in Kohfeld and Chase (2017) (although MIS-averaged in our analysis), and, combined with additional reductions of $\sim$-10 ppm throughout the period, yield a total drop of $\sim$-85 ppm from the last interglacial to the LGM. Transient changes in atmospheric $CO_2$ concentration occur throughout the glacial cycle, including during MIS 5c-5a, MIS 4 and throughout MIS 3. As discussed in the Introduction, this sequence of $CO_2$ reductions is likely the result of oceanic drivers with biogeochemical and terrestrial feedbacks (e.g.

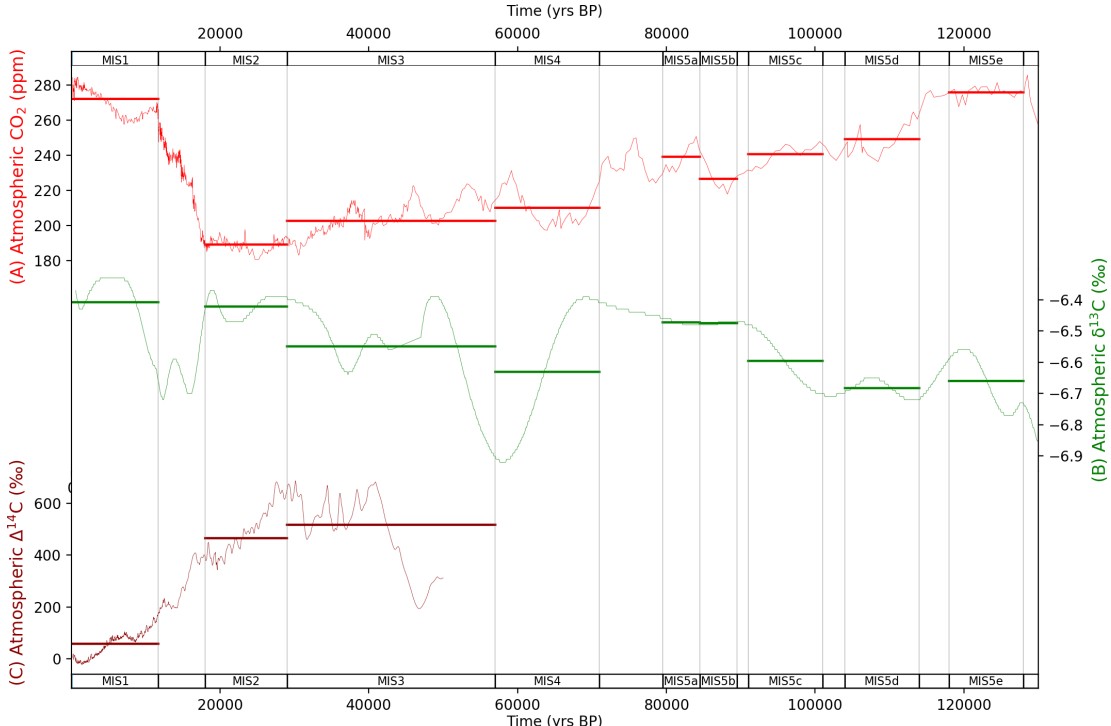

**Figure 4.** MIS atmosphere data for (A) atmospheric $CO_2$ (Monnin et al., 2004; MacFarling Meure et al., 2006; Bereiter et al., 2012; Rubino et al., 2013; Schneider et al., 2013; Ahn and Brook, 2014; Marcott et al., 2014), (B) $\delta^{13}C$ (Elsig et al., 2009; Schmitt et al., 2012; Schneider et al., 2013; Eggleston et al., 2016) and (C) $\Delta^{14}C$ (Reimer et al., 2009). Data are shown in fine lines, with bold horizontal lines for MIS-sliced data. Natural observations for $\Delta^{14}C$ do not exist beyond ∼50 ka due to the radioactive decay of $^{14}C$. Data behind the figure are shown in Table S4.

Ganopolski et al., 2010; Menviel et al., 2012; Brovkin et al., 2012; Ganopolski and Brovkin, 2017; Kohfeld and Chase, 2017). Atmospheric $CO_2$ concentration increases by ∼85 ppm in the glacial termination and Holocene periods, a transition in the carbon cycle which has occupied substantial research effort in the last four decades, but with a growing consensus of multiple physical and biogeochemical drivers and feedbacks. Kohfeld and Ridgwell (2009) and Köhler et al. (2010) provide summaries

5 of the potential candidate mechanisms to explain the glacial-interglacial changes in atmospheric $CO_2$, while recent model-data studies have attempted to explain the specific physical and biogeochemical drivers of the LGM-Holocene change in atmospheric $CO_2$ (Tagliabue et al., 2009; Menviel et al., 2016; Muglia et al., 2018; O'Neill et al., 2019).

Figure 4B shows atmospheric $\delta^{13}C$ over the last glacial-interglacial cycle. Eggleston et al. (2016) explained the glacial-interglacial atmospheric $\delta^{13}C$ pattern in terms of ongoing changes in SST, AMOC, Southern Ocean upwelling, dust-driven

10 Southern Ocean biological export productivity and the terrestrial biosphere. Atmospheric $\delta^{13}C$ (Fig. 4B) was ∼0.4‰ higher in the Holocene (MIS 1) and LGM (MIS 2) periods than in the last interglacial (MIS 5e) and penultimate glacial periods (MIS

6, not shown in Fig. 4B), as described in Schneider et al. (2013) and Eggleston et al. (2016). There were temporary falls in atmospheric $\delta^{13}$C between MIS 5e and 5d (between 120 and 110 ka), during MIS 4 (between 69 and 58 ka), during MIS 3 (between 50 and 35 ka) and in the last glacial termination between MIS 2 and 1 (between 19 and 16 ka). The cause of the observed increase in atmospheric $\delta^{13}$C across the last glacial-interglacial cycle may be the effect of accumulation and freezing or burial in glacial sediments, of peat and other soil organic matter at the high latitudes (e.g. Tarnocai et al., 2009; Ciais et al., 2012; Schneider et al., 2013; Eggleston et al., 2016; Ganopolski and Brovkin, 2017; Treat et al., 2019). According to Treat et al. (2019), peatlands and other vegetation accumulated carbon in the relatively warm periods, and these carbon stocks were then frozen and/or buried in glacial and other sediments during the cooler periods, throughout the last glacial-interglacial cycle. This buried or frozen stock of carbon mostly persists to the present day (Tarnocai et al., 2009; Ciais et al., 2012). Schneider et al. (2013) evaluated several possible candidates for the rising atmospheric $\delta^{13}$C pattern across the last glacial-interglacial cycle and could not discount any of (1) changes in the carbon isotope fluxes of carbonate weathering and sedimentation on the seafloor, (2) variations in volcanic outgassing or (3) peat and permafrost build-up throughout the last glacial-interglacial cycle.

The large drop in atmospheric $\delta^{13}$C observed during MIS 4 reverses in MIS 3 (Fig. 4B). This excursion in the $\delta^{13}$C pattern likely resulted from sequential changes in SST (cooling), AMOC, Southern Ocean upwelling and marine biological productivity (Eggleston et al., 2016). Eggleston et al. (2016) parsed the atmospheric $\delta^{13}$C signal into its component drivers across MIS 5a-3 using a stack of proxy indicators. Eggleston et al. (2016) highlighted the sequence of events between the end of MIS 5a and beginning of MIS 3 and their cumulative effects to deliver the full change in atmospheric $\delta^{13}$C. Our MIS-averaging approach as shown in Fig. 4B fails to capture the full amplitude of the changes in atmospheric $\delta^{13}$C during MIS 4 and MIS 3, and only captures the changes in the mean-MIS value, serving to understate the full extent of transient changes in responsible processes. In addition, the MIS-averaging approach misses the sequential timing of changes in processes within each MIS. These are limitations of our steady-state, MIS-averaging approach. The reduction in atmospheric $\delta^{13}$C at the last glacial termination, between the LGM and Holocene (Fig. 4B), coincident with a large atmospheric $CO_2$ increase, is attributed to the release of deep-ocean carbon to the atmosphere resulting from increased ocean circulation and Southern Ocean upwelling (Schmitt et al., 2012). The subsequent rebound of $\delta^{13}$C in the termination period and the Holocene is believed to result from terrestrial biosphere regrowth, in response to increased $CO_2$ and carbon fertilisation (Schmitt et al., 2012; Hoogakker et al., 2016).

Figure 4C shows atmospheric $\Delta^{14}$C over the last 50 kyr (Reimer et al., 2009). During this period $\Delta^{14}$C is heavily influenced by declining atmospheric $^{14}$C production (Broecker and Barker, 2007; Muscheler et al., 2014). In addition, an acceleration in atmospheric $\Delta^{14}$C decline at the last glacial termination is attributed to the release of old, $^{14}$C-depleted waters from the deep ocean, due mainly to increased Southern Ocean upwelling of $\Delta^{14}$C-depleted deep source waters (e.g. Marchitto et al., 2007; Skinner et al., 2010; Burke and Robinson, 2012; Siani et al., 2013). Broecker and Barker (2007) characterised the drop in atmospheric $\Delta^{14}$C at the last glacial termination as "the mystery interval" and questioned whether there existed a $\Delta^{14}$C-depleted ocean reservoir source of sufficient size to contribute to the drop.

Figure 5 shows deep and abyssal ocean $\delta^{13}$C data mapped into SCP-M box model space and averaged across MIS. The visual offset between deep and abyssal proxy data values is regularly interpreted as an indicator of the strength of deep ocean circulation and/or mixing, or biological productivity, during the LGM and the Holocene (e.g. Sikes et al., 2000; Curry and

Oppo, 2005; Marchitto et al., 2007; Oliver et al., 2010; Skinner et al., 2010; Burke and Robinson, 2012; Siani et al., 2013; Yu et al., 2013, 2014a; Skinner et al., 2015, 2017). The deep-abyssal Atlantic $\delta^{13}$C time series (Fig. 5A) exhibits modest widening in the MIS-average deep and abyssal offset between MIS 5e and 5d, again during MIS 5b, and then a more substantial widening during MIS 4 and during MIS 2 (the LGM). The widening of the offset during MIS 4 and MIS 2 is caused primarily by more

negative abyssal $\delta^{13}$C values. The offset is almost closed in MIS 1 (the Holocene). The deep Atlantic $\delta^{13}$C range itself also widens considerably from MIS 4, and narrows after the LGM. Oliver et al. (2010) and Kohfeld and Chase (2017) interpreted these patterns as the result of weakened deep Atlantic ocean circulation during MIS 4 and during the LGM, strengthening in the post glacial period.

The Pacific-Indian $\delta^{13}$C data (Fig. 5B) shows a drop in abyssal $\delta^{13}$C and widening in the MIS-average deep-abyssal off-

set between MIS 5e and 5d (Govin et al., 2009) which continued throughout the last glacial buildup. Importantly, the more negative abyssal $\delta^{13}$C values during MIS 5d-5a seen in Fig. 5B occur at the same time that deep ocean and atmospheric $\delta^{13}$C becomes more positive (Fig. 4B), suggesting that the abyssal Pacific-Indian ocean became more isolated from the deep ocean and atmosphere during this period. This is qualitative evidence for slowing ocean circulation or increased biological export productivity in the Pacific-Indian ocean, at that time (Govin et al., 2009). This also corresponds with a ~50 ppm fall in $CO_2$

across the period spanning MIS 5e to 5b (Fig. 4A). Abyssal Pacific-Indian $\delta^{13}$C drops further and most noticeably during MIS 4, again during the LGM, and then rebounds from the LGM into the Holocene period, as also observed in the Atlantic Ocean $\delta^{13}$C data. Statistical analysis of the $\delta^{13}$C data provided in Fig. S1 and Table S8, supports our qualitative interpretation of the Atlantic and Pacific-Indian $\delta^{13}$C proxy data.

Ocean $\Delta^{14}$C data covers the period MIS 1-3 and the LGM and Holocene in most detail (Fig. 6). We show ocean $\Delta\Delta^{14}$C,

which is ocean less atmospheric $\Delta^{14}$C. This calculation is made in attempt to normalise the effects of varying atmospheric $^{14}$C production through the glacial-interglacial cycle (Broecker and Barker, 2007; Muscheler et al., 2014), which imparts a dominant influence on the ocean $\Delta^{14}$C trajectory. Given the sparse data coverage for MIS 3 we focus our analysis on MIS 1 and 2. The $\Delta\Delta^{14}$C time series exhibits two key features across the MIS 2 (LGM) and MIS 1 (Holocene) periods. First, there is a narrowing in the spread of values between the shallow and abyssal ocean from the LGM to the Holocene, in both

the Atlantic (Fig. 6A) and Pacific-Indian (Fig. 6B) basins. Second, all ocean boxes display an increase in $\Delta\Delta^{14}$C from the LGM to the Holocene, towards equilibrium with the atmosphere. These patterns are believed to represent increased overturning circulation and Southern Ocean upwelling in the Atlantic and Pacific-Indian basins across the LGM-Holocene. Increased ocean overturning brought old, $\Delta^{14}$C-negative water up from the deep and abyssal oceans, mixing with shallow and intermediate waters, and eventually into the surface Southern Ocean and contact with the atmosphere (where $^{14}$C is produced) - known

as "increased ventilation" (e.g. Sikes et al., 2000; Marchitto et al., 2007; Bryan et al., 2010; Skinner et al., 2010; Burke and Robinson, 2012; Siani et al., 2013; Davies-Walczak et al., 2014; Skinner et al., 2014; Hines et al., 2015; Freeman et al., 2016; Sikes et al., 2016; Skinner et al., 2017).

The Atlantic Ocean $CO_3^{2-}$ time series shows a similar pattern to $\Delta\Delta^{14}$C and $\delta^{13}$C, with a wide dispersion of shallow-abyssal and deep-abyssal concentrations at the LGM that narrows by the Holocene (Fig. 7). This pattern has been interpreted as varying

strength and/or depth of AMOC and biological productivity in the Atlantic Ocean (e.g. Yu et al., 2013, 2014b, a, 2016). The

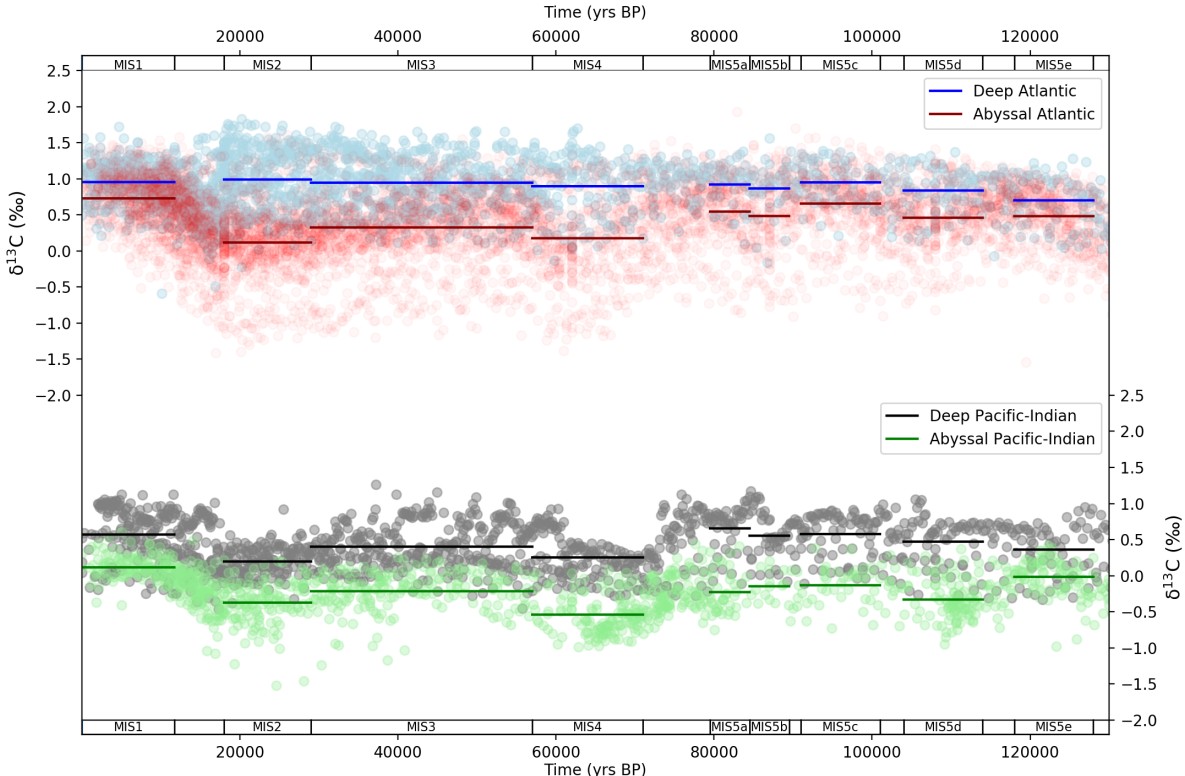

**Figure 5.** MIS ocean data mapped into SCP-M box model dimensions for $\delta^{13}$C (Govin et al., 2009; Piotrowski et al., 2009; Oliver et al., 2010). Data (round circles) are mapped into deep (2,500m average depth) and abyssal (3,700 (Atlantic) - 4,000m (Pacific-Indian) average depth) model boxes and averaged across MIS slices (bold lines). Data behind the figure are shown in Table S5.

abyssal Atlantic $CO_3^{2-}$ pattern, which spans the last glacial-interglacial cycle, is punctuated by two downward excursions (Fig. 7). These occur during MIS 4 and MIS 2, corresponding to the second and third major atmospheric $CO_2$ drops in the last glacial-interglacial cycle (Kohfeld and Chase, 2017), respectively (Fig. 4A). The lower deep Atlantic Ocean $CO_3^{2-}$ values during MIS 4 were interpreted by Yu et al. (2016) as shoaling of AMOC and increased carbon storage in the deep-abyssal

5    Atlantic Ocean. This signal is repeated at the LGM, where further shoaling and slowing AMOC contributed to deep oceanic drawdown of $CO_2$ from the atmosphere (Yu et al., 2013, 2014b, a). There is also a modest drop in abyssal Atlantic Ocean $CO_3^{2-}$ during MIS 5b (-13 $\mu$mol kg$^{-1}$ relative to MIS 5c), which coincides with a minor drop in abyssal Atlantic Ocean $\delta^{13}$C (-0.19‰) and atmospheric $CO_2$ (-14 ppm), indicating a common link. Menviel et al. (2012) modelled a transient slowdown in AMOC for this period, which could explain these features.

10    The Pacific Ocean is thought to partially buffer the effects of ocean circulation on $CO_3^{2-}$ concentrations (Fig. 7) via changes in shallow (reef) and deep carbonate production and dissolution, and therefore displays less variation across the MIS (Yu et al.,

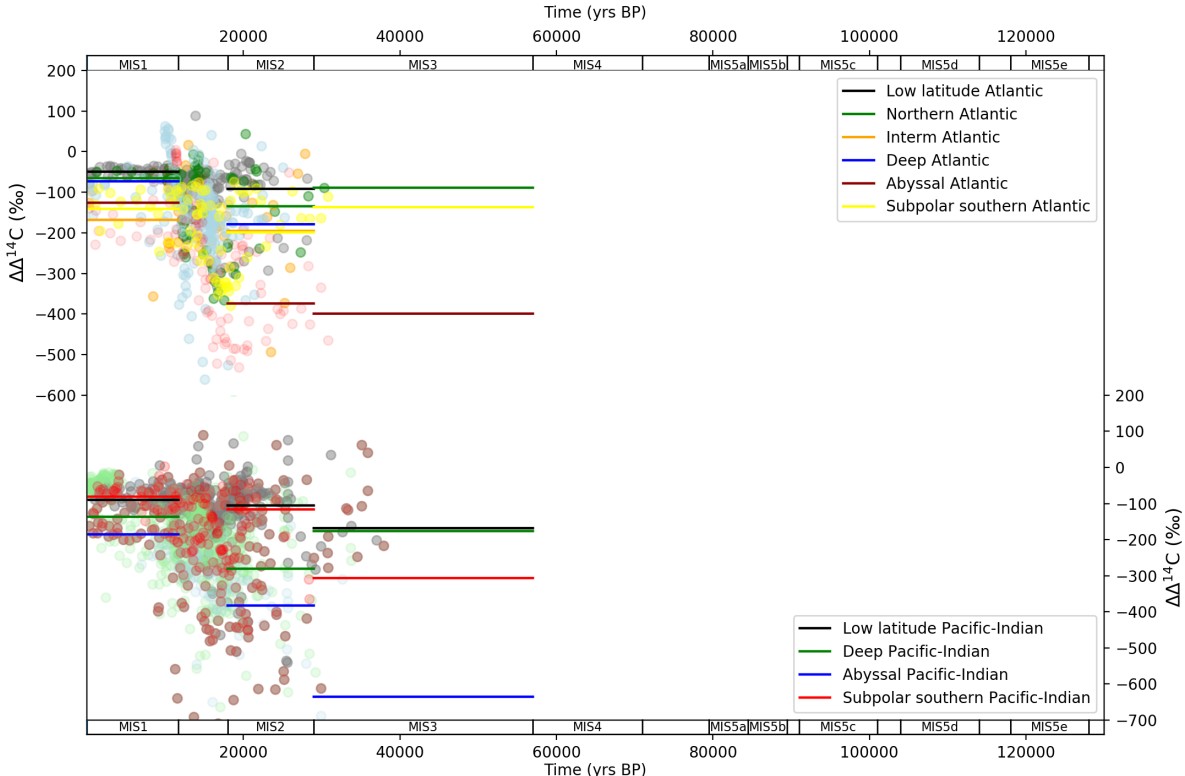

**Figure 6.** MIS stage ocean data mapped into box model dimensions for $\Delta\Delta^{14}$C. Data (round circles) are mapped into deep (2,500m average depth) and abyssal (3,700 (Atlantic) - 4,000m (Pacific-Indian) average depth) model boxes and averaged across MIS slices (bold lines). Natural observations do not exist beyond ~50 ka due to the radioactive decay of $^{14}$C. $\Delta\Delta^{14}$C is ocean minus atmosphere $\Delta^{14}$C. Note that this calculation is not done with the average ocean box and atmosphere values for each MIS, rather $\Delta\Delta^{14}$C represents the difference between each ocean data point and the contemporary atmospheric $\Delta^{14}$C value. Data behind the figure are shown in Table S7.

2014b; Qin et al., 2017, 2018). The deep and abyssal Pacific-Indian ocean data shows a gradual trend of increasing $CO_3^{2-}$ through the glacial-interglacial cycle (Fig. 7), suggesting that it is influenced more by variations in shallow/deep sea carbonate production/dissolution and less by deep ocean circulation (Yu et al., 2014b; Qin et al., 2017, 2018). Notable exceptions are during MIS 5d and MIS 4. Between MIS 5e and 5d, both deep and abyssal Pacific-Indian ocean $CO_3^{2-}$ drop (Fig. 7), aligning

5   with the contemporary drop in abyssal ocean $\delta^{13}$C and atmospheric $CO_2$ (Fig. 5 and Fig. 5B), suggesting a possible common driver, and providing additional qualitative evidence for changes in either Pacific-Indian ocean circulation or biology, at this time. During MIS 4, there is a drop in deep and abyssal Pacific-Indian $CO_3^{2-}$ and a modest widening in the average deep-abyssal offset from MIS 5b and 5a, also suggestive of the influence of deep ocean circulation and/or biological export productivity (Fig. 7). The widest Pacific-Indian deep-abyssal offset $CO_3^{2-}$ is observed during MIS 3, also seen in the $\Delta\Delta^{14}$C data (Figs 5-7),

10   indicating it is a persistent feature of the proxy records. This suggests MIS 3 may be the nadir of Pacific-Indian ocean circulation

and/or the peak in biological activity in the last glacial-interglacial cycle, or at least that important changes in this part of the ocean took place in MIS 3, prior to the LGM.

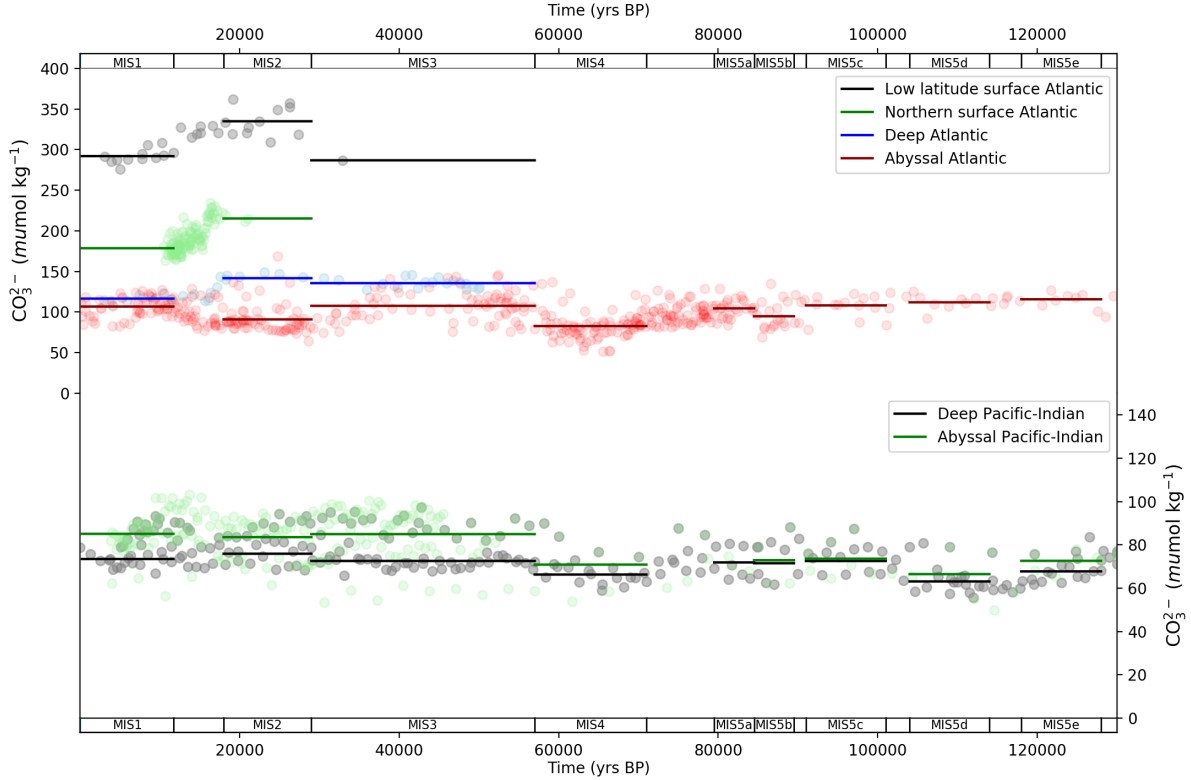

**Figure 7.** MIS stage ocean data mapped into box model dimensions for carbonate ion proxy. Data (round circles) are mapped into deep Data (round circles) are mapped into deep (2,500m average depth) and abyssal (3,700 (Atlantic) - 4,000m (Pacific-Indian) average depth) model boxes and averaged across MIS slices (bold lines). and abyssal (3,700 (Atlantic) - 4,000m (Pacific-Indian) average depth) model boxes and averaged across MIS slices (bold lines). Data behind the figure are shown in Table S6.

## 4 Results

Figure 8 shows the data-optimised MIS-average values returned from the model-data experiments for GOC, AMOC and At-
lantic Southern Ocean biological productivity parameters, in each MIS ("X" symbols). The optimised values take account of data variance, due to the weighting of proxy data points by their standard deviation in the model-data optimisation equation (Eq. 3). The full range of model-data experiment results are shown in Table S9. The GOC parameter ($\Psi_1$) value falls from 29 Sv to 22 Sv between MIS 5e and 5d, with gradual declines during MIS 5c-5a and a slight acceleration in the rate of decline during MIS 5a-3. GOC reaches its minimum glacial value (16 Sv) in MIS 3, then increases from 16 Sv to 29 Sv between MIS

2 (the LGM) and the Holocene. AMOC ($\Psi_2$) weakens modestly in MIS 5d (-2 Sv), with a further drop during MIS 5b (-2 Sv) that is partially reversed in MIS 5a. AMOC weakens further in MIS 4, achieving its glacial nadir (13 Sv), which is maintained until the LGM before increasing to 18 Sv in MIS 1. Importantly, $\Psi_2$ closely follows the abyssal Atlantic (>2,500 m, single box covering North and South Atlantic) $\delta^{13}$C and $CO_3^{2-}$ data patterns across the glacial-interglacial cycle, and $\Delta\Delta^{14}$C from the LGM to the Holocene (Figs 5-7). $\Psi_2$ remains near its modelled last interglacial value (MIS 5e, 18 Sv), during MIS 5d and 5c, before dropping in MIS 5b (abyssal Atlantic $\delta^{13}$C and $CO_3^{2-}$, and atmospheric $CO_2$, also drop at this point), before partly rebounding during MIS 5a and then falling synchronously with abyssal Atlantic $\delta^{13}$C and $CO_3^{2-}$ concentrations during MIS 4. Southern Ocean biological export productivity ($Z$) fluctuates around its last interglacial (MIS 5e) value during the time period spanning MIS 5d-5b, then increases during MIS 4. Atlantic (Pacific-Indian) Southern Ocean $Z$ spikes to 4.7 (3.3) mol C m$^{-2}$ yr$^{-1}$ in the LGM, then falls to 3.4 (2.4) mol C m$^{-2}$ yr$^{-1}$ in MIS 1.

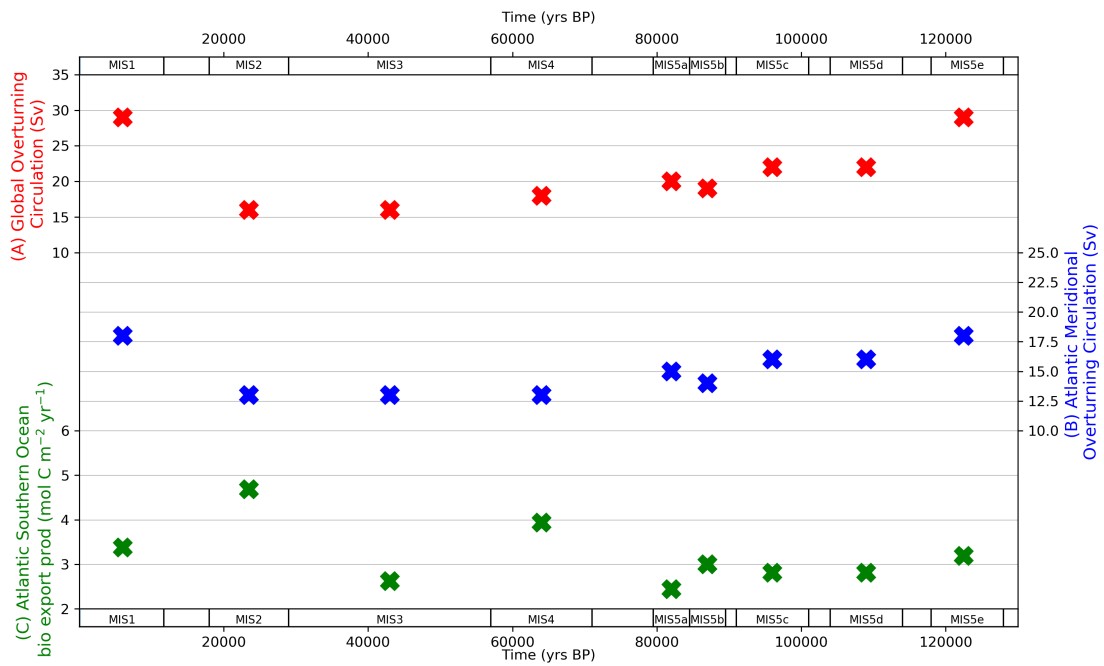

**Figure 8.** Model-data experiment results for global overturning circulation (A), Atlantic meridional overturning circulation (B) and Atlantic Southern Ocean biological export productivity (C). "X" symbols mark the optimal parameter values returned from the model-data experiments. The optimised values take account of data variance, due to the weighting of proxy data points by their standard deviation in the model-data optimisation equation (Eq. 3). Data for optimised parameter values shown in the figure are contained in Table S9.

Figure 9 shows the optimised model-data output for atmospheric $CO_2$ and ocean $CO_3^{2-}$ concentrations compared with the proxy data observations, in each MIS. This shows how well the model is constrained by the proxy data, and also how well the

model-data output of parameter values can explain the proxy data patterns as described in the data analysis section (Section 3). The model-data results fall within one standard deviation of atmospheric $CO_2$ and deep and abyssal $CO_3^{2-}$ data, and mostly on the MIS means, across the MIS periods (Fig. 9). The modelled abyssal Pacific-Indian $CO_3^{2-}$ falls close to the MIS proxy data means across the glacial-interglacial cycle, but misses some of the variations in the data - particularly between MIS 4 and MIS

5    3 (Fig. 9). This is a result of the abyssal ocean box carbonate dissolution equations in SCP-M, which effectively buffer changes in the relative balance of DIC and alkalinity from ocean physical and biological changes, and possibly the large box sizes in SCP-M which miss some detail for sparse $CO_3^{2-}$ data.

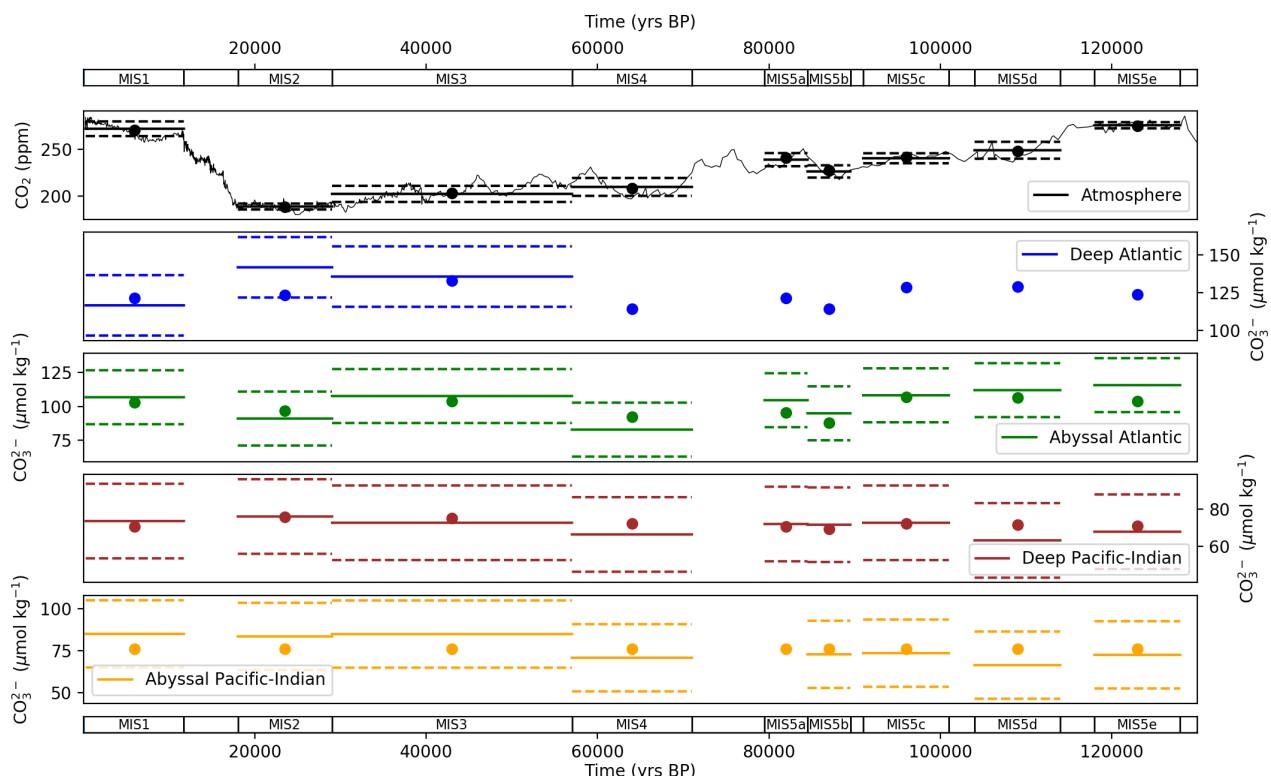

**Figure 9.** Values returned from the model-data experiment for (A) atmospheric $CO_2$ and carbonate ion proxy for (B) deep Atlantic (2,500m average depth), (C) abyssal Atlantic (3,700m average depth), (D) deep Pacific-Indian (2,500m average depth) and (E) abyssal Pacific-Indian (4,000m average depth). Model-data experiment results are shown as dots, with mean proxy data shown as solid lines, and one standard deviation range by dashed lines, in each MIS. A default standard deviation of 20 $\mu$mol kg$^{-1}$ is used as discussed in the text. $CO_3^{2-}$ data for the SCP-M deep Atlantic box in (B) does not extend beyond 50 ka. Model results for each box in each MIS are shown in Table S10 and S12.

The model-data results show good agreement with atmospheric, deep and abyssal $\delta^{13}$C data throughout the MIS (Fig. 10). The results mostly fall on the mean and all are within the standard deviation for atmospheric $\delta^{13}$C data in the MIS. Nearly all results fall within standard deviation for the deep and abyssal Atlantic and Pacific-Indian oceans. The modelled abyssal Pacific-Indian box $\delta^{13}$C underestimates mean MIS $\delta^{13}$C in most MIS time slices, which may reflect a discrepancy between the

5    average depth of the $\delta^{13}$C proxy data and SCP-M abyssal ocean box, or a bias in the model's equations.

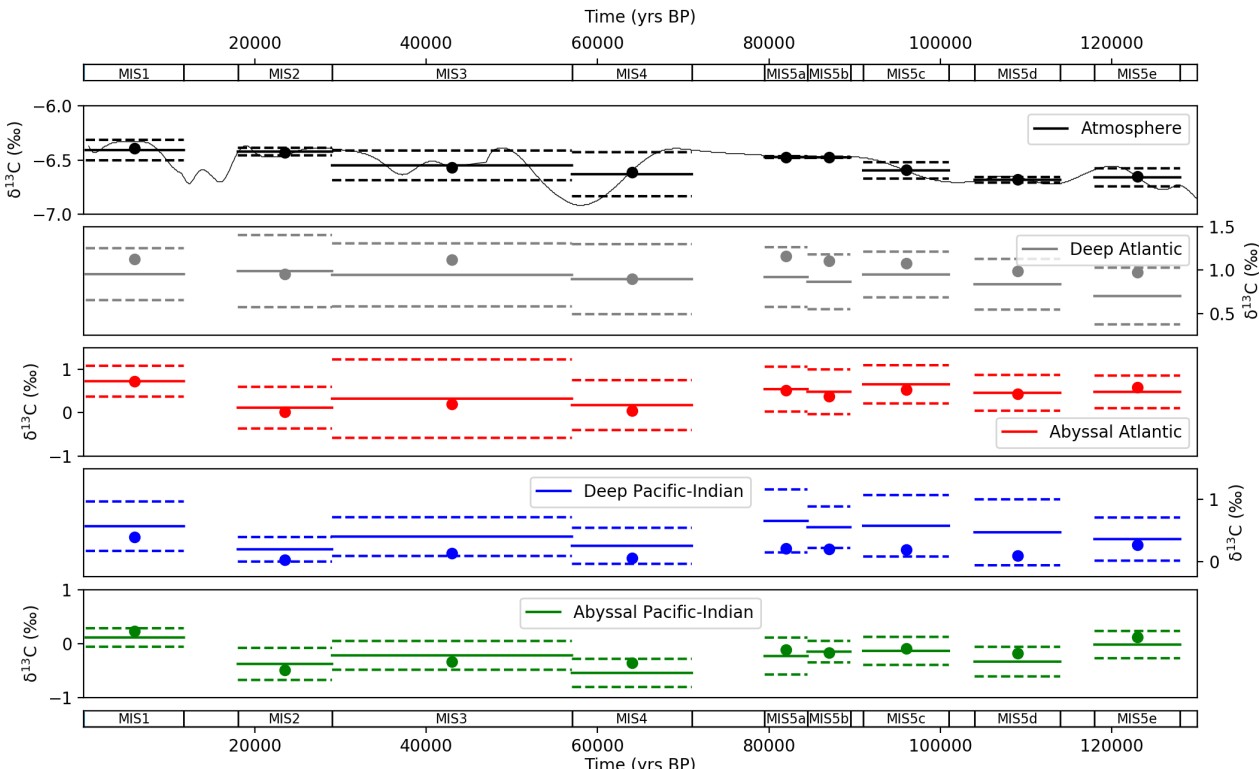

**Figure 10.** Values returned from the model-data experiment for $\delta^{13}$C for (A) atmosphere, (B) deep Atlantic (2,500m average depth), (C) abyssal Atlantic (3,700m average depth), (D) deep Pacific-Indian (2,500m average depth) and (E) abyssal Pacific-Indian (4,000m average depth). Model-data experiment results are shown as dots, with proxy data mean (solid lines) and one standard deviation (dashed lines) in each MIS. Model results for each box in each MIS are shown in Table S10 and S11.

Figure 11 shows model-data results for atmospheric $\Delta^{14}$C and ocean $\Delta\Delta^{14}$C compared with data, for MIS 1-3. Model-data results fall within one standard deviation of the data for all observations that were modelled and replicate the dramatic compression in deep-abyssal $\Delta\Delta^{14}$C and ocean-atmosphere offsets, between MIS 2 (LGM) and MIS 1 (the Holocene) as shown in the data (Fig. 11).

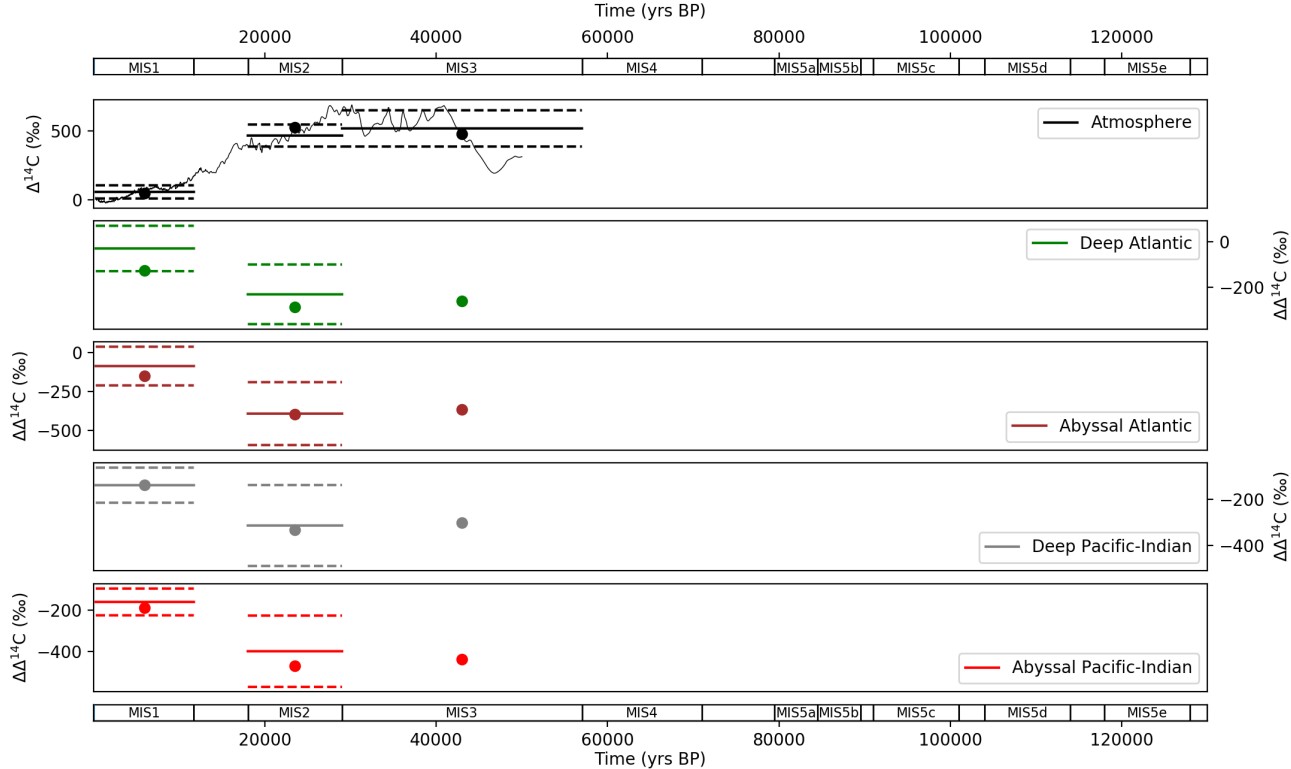

**Figure 11.** Values returned from the model-data experiment for (A) atmospheric $\Delta^{14}C$ and $\Delta\Delta^{14}C$ for (B) deep Atlantic (2,500m average depth), (C) abyssal Atlantic (3,700m average depth), (D) deep Pacific-Indian (2,500m average depth) and (E) abyssal Pacific-Indian (4,000m average depth). $\Delta\Delta^{14}C$ is ocean minus atmospheric $\Delta^{14}C$, calculated to correct for the varying atmospheric $\Delta^{14}C$ signal. Model-data experiment results are shown as dots, with proxy data mean (solid lines) and one standard deviation (dashed lines) in each MIS. Model-data experiment results prior to MIS 4 are omitted, due to the radioactive decay of $^{14}C$ which precludes natural observations prior to ~50 ka. Model results for each box in each MIS are shown in Table S10 and S13.

Figure 12 shows model-data output for the terrestrial biosphere net primary productivity (NPP) and carbon stock during the last glacial-interglacial cycle. The NPP and carbon stock follow atmospheric $CO_2$ downwards in the lead-up to the LGM and rebound from the LGM to the Holocene. In our model this is driven by carbon fertilisation from atmospheric $CO_2$ (Kaplan et al., 2002; Otto et al., 2002; Harman et al., 2011; Hoogakker et al., 2016). However, other studies emphasise the important role of temperature and precipitation in influencing NPP (François et al., 1999; van der Sleen et al., 2015). Notably, there is a distinct drop in NPP during MIS 4, a period where atmospheric $CO_2$ falls by ~30 ppm (Fig. 4A). Hoogakker et al. (2016) provided a reconstruction of NPP through the last glacial-interglacial cycle using pollen data and climate models, shown for

comparison in Fig. 12A. Our model-data results for NPP typically fall in the upper and lower end of the range of NPP values from Hoogakker et al. (2016). However, our model-data estimates of NPP for MIS 5d and 5e underestimate the NPP calculated by Hoogakker et al. (2016) (which extend only to 120 ka). We model the terrestrial biosphere carbon stock to fall by 385 Pg C from the last interglacial to the LGM, and increase by ∼600 Pg C from the LGM to the Holocene (Fig. 12B).

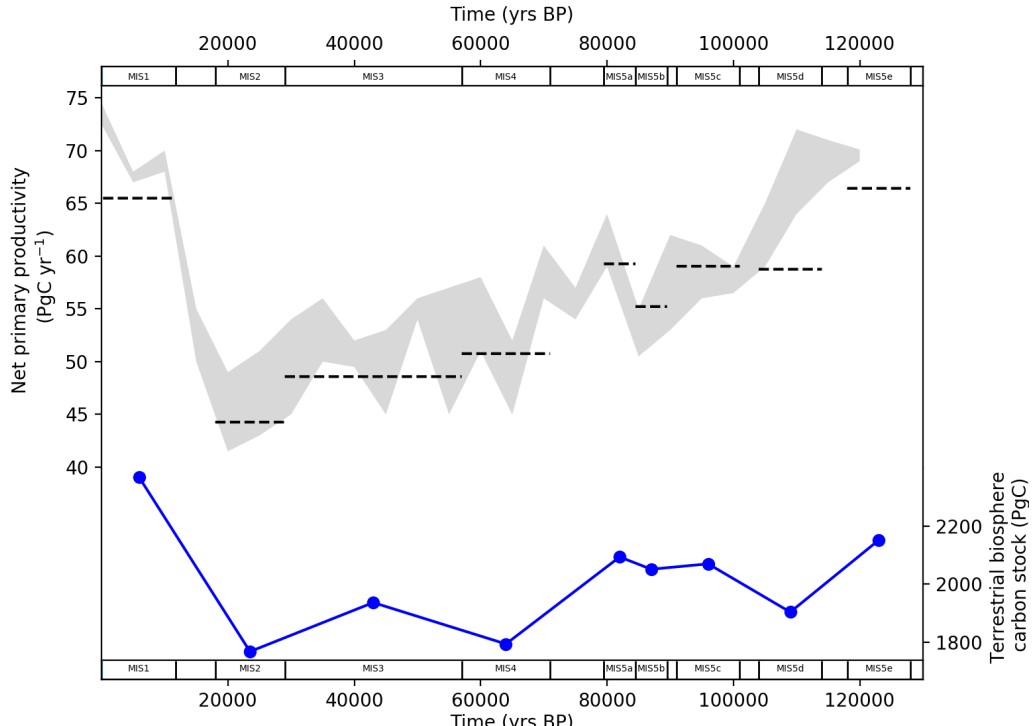

**Figure 12.** (A) Model-data output for the terrestrial biosphere net primary productivity (NPP) in each MIS time slice (black dashed lines) compared with the range of estimates provided by Hoogakker et al. (2016) (grey area). (B) model-data output for the terrestrial biosphere carbon stock for each MIS time slice.

## 5  Discussion

### 5.1  Last glacial-interglacial cycle

This study applies a carbon cycle box model to diagnose the values for ocean circulation and Southern Ocean biological export productivity during the last glacial-interglacial cycle, optimised for ocean and atmospheric proxy data. This study continues efforts to simulate the last glacial-interglacial cycle of atmospheric $CO_2$ (e.g. Ganopolski et al., 2010; Brovkin et al., 2012;

Menviel et al., 2012; Ganopolski and Brovkin, 2017), but with a simpler box model and using a non-transient model-data optimisation to estimate parameter values.

There were three major episodes in which atmospheric $CO_2$ concentration fell during the last glacial-interglacial cycle (Fig. 4A), accompanied by changes in atmospheric $\delta^{13}C$ (Fig. 4B), $\Delta^{14}C$ (Fig. 4C) and ocean $\delta^{13}C$, $\Delta^{14}C$ and $CO_3^{2-}$ (Figs. 5-7). Our model-data results show that glacial-interglacial atmospheric $CO_2$ and the other proxy patterns are delivered by a host of physical and biogeochemical changes. These changes include weakened GOC, AMOC and strengthened Southern Ocean biological export productivity (Figs. 8, 9, 10, 11), and changes in SST, salinity, ocean volume, the terrestrial biosphere, reef carbonates and atmospheric $^{14}C$ production (Figs. 2 and 12).

Our model-data results show that an initial fall in GOC took place during MIS 5d (Fig. 8), as MIS-average atmospheric $CO_2$ concentration fell by $\sim$25 ppm. This was also a time of substantial cooling in SST (Fig. 2A). GOC drifted lower until achieving its glacial minimum level in MIS 3 and MIS 2. AMOC weakened in MIS 4, at the same time that North Atlantic SST cooled dramatically (Fig. 2A) and MIS-average atmospheric $CO_2$ fell $\sim$30 ppm. GOC and AMOC were both equal to their glacial lows at the LGM, and accompanied by increased Southern Ocean biological export productivity, yielding the LGM minima in atmospheric $CO_2$ and the final fall major fall in $CO_2$ during the glacial cycle. We model elevated Southern Ocean biological productivity during MIS 4 and MIS 2, relative to interglacial values (MIS 5e and 1). Importantly, the transition from MIS 3 to MIS 2, which incorporates the LGM and increased Southern Ocean biological productivity, only accounted for an average 15 ppm reduction in $CO_2$ (Figs. 4, 9). Therefore, our results suggest an increase in Southern Ocean biological productivity during this period was an additional 'kicker' to achieve the LGM atmospheric $CO_2$ minima, following prior reductions of $\sim$70 ppm in the lead-up which were delivered mainly by ocean physical processes and SST. The finding of increased biological productivity, while mostly constrained in our model to MIS 4 and 2, and a modest yet essential contributor to the overall glacial $CO_2$ drawdown, corroborates proxy data (e.g. Martinez-Garcia et al., 2014; Lambert et al., 2015; Kohfeld and Chase, 2017; Shaffer and Lambert, 2018) and recent model-data exercises (e.g. Menviel et al., 2016; Muglia et al., 2018).

For the Holocene, we model GOC and AMOC returning to values similar to the modern ocean estimates of Talley (2013). Our Holocene result for Atlantic (Pacific-Indian) Southern Ocean biological export productivity, of 3.4 (2.4) mol C m$^{-2}$ yr$^{-1}$ (Fig. 8), falls within modern observations for the Southern Ocean of 0.5-6 mol C m$^{-2}$ yr$^{-1}$ (e.g. Lourey and Trull, 2001; Weeding and Trull, 2004; Ebersbach et al., 2011; Jacquet et al., 2011; Cassar et al., 2015; Arteaga et al., 2019). Our model-data experiment results also reproduce values that fall within one standard deviation of the mean value in nearly all model boxes, for all of the atmosphere and ocean proxies in each MIS (Figs. 9-11).

Kohfeld and Chase (2017) suggested that sequential falls in atmospheric $CO_2$ concentration were first the result of temperature, sea-ice cover and potentially sea-ice cover induced Atlantic Southern Ocean "barrier mechanisms" or shallow stratification during MIS 5d, and second, followed by falls in deep Atlantic ocean circulation and potentially dust-driven Southern Ocean biological productivity during MIS 4. Finally, a synthesis of those factors including enhanced Southern Ocean biology, delivered the LGM atmospheric $CO_2$ minimum. Our model-data results mostly agree with the Kohfeld and Chase (2017) hypothesis for glacial-interglacial $CO_2$, particularly with regard to lower SST early in the glacial inception, followed by weaker deep Atlantic ocean circulation and stronger Southern Ocean biological export productivity later in the glacial cycle. However, we also posit

a role for slowing GOC and no direct role for increased sea-ice cover, in delivering lower atmospheric $CO_2$ at the last glacial inception. Stephens and Keeling (2000) proposed that expanded glacial sea-ice cover around Antarctica could deliver LGM $CO_2$ changes on its own, as a result of reduced air-sea gas exchange or in combination with ice-driven ocean stratification. However, Köhler et al. (2010) demonstrated with a carbon cycle box model that increased sea-ice cover leads to increased

atmospheric $CO_2$, due to less in-gassing of $CO_2$ into the cold waters surrounding Antarctica. Kohfeld and Ridgwell (2009) reviewed estimates of the effects of *decreased* sea-ice cover at the last glacial termination and found a best estimate of -5 ppm within a range of -14-0 ppm, which is in the opposite direction to that envisaged by Stephens and Keeling (2000) and Kohfeld and Chase (2017). The modelling work by Stephens and Keeling (2000) was discounted by Kohfeld and Ridgwell (2009) because it assumed nearly all ocean-degassing of $CO_2$ was confined to the polar Antarctic region, when modern observations

suggest the locus of outgassing is in the equatorial ocean (Takahashi et al., 2003). In SCP-M, the effects of polar Southern Ocean sea-ice cover, modelled as a slowing down in air-sea gas exchange in the polar Southern Ocean surface box, are modest. This modelling result reflects the offsetting effects of upwelled nutrient- (and carbon) rich waters (degassing and higher $CO_2$), against the effects of lower temperatures and enhanced biological export productivity (in-gassing and lower $CO_2$). This finding may reflect our approach to treat Southern Ocean sea-ice cover simply as a regulator of the rate of air-sea gas exchange. Our

approach may neglect other effects of sea-ice cover including as a contributor to changes in Southern Ocean brine formation, buoyancy forcing, upwelling, mixing, deep ocean stratification and NADW formation rates (Morrison et al., 2011; Brovkin et al., 2012; Ferrari et al., 2014; Kohfeld and Chase, 2017; Jansen, 2017; Marzocchi and Jansen, 2017). For example, Brovkin et al. (2012) found that in the CLIMBER-2 model, atmospheric $CO_2$ was more sensitive to sea-ice cover when it was linked to weakened vertical diffusivity in the Southern Ocean of tracers such as DIC, thereby reducing outgassing of $CO_2$. The syner-

gistic effects of increased Antarctic Southern Ocean sea-ice cover discussed by Kohfeld and Chase (2017), in terms of reduced ocean vertical mixing rates to deliver reductions in atmospheric $CO_2$, could be tested with a more complex model than SCP-M.

In addition to lower SST, increased-sea ice cover and the other model forcings (Fig. 2), SCP-M requires additional changes in the ocean to deliver the ∼25 ppm fall in average $CO_2$ concentration during MIS 5d and satisfy the other atmospheric and ocean proxy data. We model a weakening in GOC of ∼7 Sv during MIS 5d and further weakening until the LGM, a substantial change

in the global ocean and not just the Atlantic Basin. This underscores the importance of the global ocean in any hypothesis for the last glacial-interglacial cycle or LGM-Holocene (Fig. 8). We note that our simplified representation of GOC, as per Talley (2013), includes features that may be separated out or characterised differently in other models or hypotheses, such as AABW formation rate, Southern Ocean upwelling or shallow mixing/stratification, Pacific and Indian deepwater formation (PDW/IDW), or northward extension of AABW versus NADW formation of abyssal waters in the Atlantic Ocean (e.g. Menviel

et al., 2016; Kohfeld and Chase, 2017).

The period MIS 5e-5d does not feature in some oceanographic theories of glacial inception atmospheric $CO_2$ decline, largely due to a focus on Atlantic ocean data and a lack of any obvious changes in the Atlantic shallow-deep-abyssal proxy offsets at that period, as observed clearly during MIS 4 and the LGM (e.g. Oliver et al., 2010; Yu et al., 2016; Kohfeld and Chase, 2017). However, Govin et al. (2009) proposed an expansion of AABW across the Southern Ocean and weakening of circumpolar deep

water upwelling during MIS 5d, based on deep ocean $\delta^{13}C$ from the Atlantic and Indian basins. The proxy evidence of Govin

et al. (2009) supports the concept of De Boer and Hogg (2014), that the glacial ocean could have exhibited slower and at the same time more expansive formation of AABW. Ganopolski et al. (2010) and Brovkin et al. (2012) modelled cooling SST and substitution of NADW by denser waters of Antarctic origin in the abyssal ocean, as the main drivers of falling atmospheric $CO_2$ at the last glacial inception. Menviel et al. (2012) modelled a transient slowdown in the rate of overturning circulation in the North Atlantic across MIS 5e-5d. Despite these findings, changes in ocean circulation at the last glacial inception are not obvious in Atlantic Ocean $\delta^{13}C$ proxy data (Oliver et al., 2010; Kohfeld and Chase, 2017).

To illustrate the plausibility of a slowdown in GOC during the last glacial inception in the context of deep ocean $\delta^{13}C$ proxy data, we show a model experiment testing the sensitivity of atmospheric $CO_2$ and abyssal ocean $\delta^{13}C$ to slowed GOC under MIS 5e and MIS 5d conditions (Fig. 13). Shown for comparison are the standard deviation of data values for abyssal ocean $\delta^{13}C$ for MIS 5e (Fig. 13B). The experiment shows that slowing GOC from the MIS 5e model-data optimised value of 29 Sv (e.g. Fig. 8), delivers lower values for atmospheric $CO_2$ (Fig. 13A) and more negative abyssal Pacific-Indian $\delta^{13}C$ (Fig. 13B). However, in the experiment of decreasing GOC, modelled atmospheric $CO_2$ crosses the $\sim$25 ppm change of the MIS 5e-5d transition, well before the model's abyssal Pacific-Indian box $\delta^{13}C$ breaches one standard deviation of the abyssal Pacific-Indian $\delta^{13}C$ data for MIS 5e (Fig. 13B). Changes in the deep-abyssal $\delta^{13}C$ offsets are also muted (Fig. 13C) relative to atmospheric $CO_2$, and particularly for the Atlantic Ocean. The observation is even more obvious when including other ocean changes for the MIS 5e-5d transition, such as SST. When these changes are incorporated (shown as the "x" symbols in Fig. 13A and B), the atmospheric $CO_2$ change across MIS 5e-5d is even more quickly satisfied by the modelled reduction in GOC, while abyssal ocean $\delta^{13}C$ remains near its MIS 5e box average and well within one standard deviation. Despite a range of GOC variation that surpasses the MIS 5e-5d atmospheric $CO_2$ reduction, the abyssal Atlantic $\delta^{13}C$ result hardly varies, a particularly interesting finding. In SCP-M this can be explained by a reduced rate of AABW formation as a part of slowing GOC, leading to relatively greater influence of other Atlantic Ocean processes such as the deep-abyssal mixing and AMOC, which mixes deep water with a more positive $\delta^{13}C$ into the abyssal Atlantic and offsets the effects of slowing GOC. Slowing GOC by itself leads to a more negative abyssal $\delta^{13}C$, as per the Pacific-Indian Basin results. This type of dynamic could help explain why hypothesised or modelled changes in the ocean during the last glacial inception (e.g. Govin et al., 2009; Menviel et al., 2012; Brovkin et al., 2012) don't show up more obviously in the deep and abyssal Atlantic Ocean $\delta^{13}C$ proxy data (Oliver et al., 2010; Kohfeld and Chase, 2017).

These observations from Fig. 13 could be exaggerated in SCP-M due to the large size of its ocean boxes and therefore relatively large spread of $\delta^{13}C$ values and standard deviations for each box. In addition, this experiment may reflect idiosyncrasies in the SCP-M model design and its simple parameterisation of ocean circulation and mixing. A finer resolution model may show a greater sensitivity of the ocean box $\delta^{13}C$ to variations in ocean circulation. Menviel et al. (2015) analysed the sensitivity of ocean and atmospheric $\delta^{13}C$ to variations in NADW, AABW and North Pacific Deep Water (NPDW) formation rates, in the context of past changes in atmospheric $\delta^{13}C$ and $CO_2$. Their modelling, using the more spatially-detailed LOVECLIM and Bern3D models, showed modest but location-dependent sensitivities of ocean $\delta^{13}C$ to slowing ocean circulation, and particular sensitivity to AABW. These models are higher resolution and show greater sensitivity of $\delta^{13}C$ to ocean circulation over depth intervals not differentiated in the SCP-M boxes. However, our simple experiment illustrated in Fig. 13 does highlight the

potential for important changes in the ocean during glacial-interglacial periods to go unnoticed, when focussed on one set of ocean proxy data and without validation by modelling.

As shown in Fig. 13, analysing Atlantic Ocean data in isolation, and only qualitatively assessing ocean proxy data offsets (e.g. solely relying on standard deviations), may obscure features that could have contributed meaningfully to glacial falls in

atmospheric $CO_2$ (e.g. GOC). According to (Talley, 2013), GOC is a key part of the global ocean carbon cycle, operating in the Atlantic, Pacific and Indian ocean basins. Given it is a global feature, spread across all basins, its global changes may not show up as dramatic changes in proxy data offsets in any particular basin, despite it exerting a strong influence on atmospheric $CO_2$. A number of authors highlight changes in $\Delta^{14}C$ distributions in the Pacific Ocean during the LGM and Holocene, providing qualitative evidence of changes in ocean circulation in this basin and of it being a potential driver for post-glacial increase in

atmospheric $CO_2$ concentration (e.g. Sikes et al., 2000; Marchitto et al., 2007; Stott et al., 2009; Cook and Keigwin, 2015; Skinner et al., 2015; Ronge et al., 2016; Skinner et al., 2017). Ocean $\Delta^{14}C$ values are particularly sensitive to ocean circulation rates (Broecker et al., 1980). However, $\Delta^{14}C$ proxy records in periods prior to the LGM and Holocene are sparse, because they can only extend to ∼50 ka due to their radioactive decay in nature, therefore cannot be applied to the glacial inception period.

There is qualitative multi-proxy evidence for a slowdown or shoaling of AMOC during MIS 4. Kohfeld and Chase (2017)

evaluated Atlantic basin $\delta^{13}C$ data and surmised that Atlantic deep ocean circulation slowed or shoaled during MIS 4. Yu et al. (2016) and Chalk et al. (2019) came to similar conclusions from analysis of carbonate proxy records. Piotrowski et al. (2009) further suggested a reduced proportion of AMOC-sourced waters in the deep Indian Ocean during MIS 4, as deduced from Indian Ocean $\delta^{13}C$ data. Our model-data results corroborate these findings, with a pronounced weakening in AMOC during MIS 4. SCP-M does not take explicit account of AMOC shoaling due to its rigid box boundaries, and therefore the change in

proxy data between MIS 5a and 4 is resolved as weakening AMOC, which could understate the importance of this event. We also model a drop in AMOC during MIS 5b which replicates abyssal Atlantic $\delta^{13}C$ and $CO_3^{2-}$ observations (Fig. 5 and Fig. 7), and also accompanies a transient fall in atmospheric $CO_2$ of 14 ppm during that period (Fig. 4). Menviel et al. (2012) modelled a transient, but more dramatic decline in AMOC rate during MIS 5b, and a more protracted but similarly large decline during MIS 4 (also modelled by Ganopolski et al. (2010)), in addition to a deepening in the remineralisation depth of organic carbon.

Our model-data results indicate a role for increased Southern Ocean biological export productivity in achieving glacial troughs in atmospheric $CO_2$ concentration during MIS 4 and MIS 2. Our finding of increased biological productivity, while constrained to MIS 4 and MIS 2, and a modest contributor to the overall glacial atmospheric $CO_2$ drawdown, aligns with proxy data for increased iron-rich continental dust supply to the Southern Ocean in these periods (e.g. Martinez-Garcia et al., 2014; Lambert et al., 2015; Kohfeld and Chase, 2017) and recent model-data exercises (e.g. Menviel et al., 2016; Muglia et al., 2018;

Khatiwala et al., 2019). Martin (1990) pioneered the "iron hypothesis", which invoked the increased supply of continent-borne dusts to the Southern Ocean in glacial periods. Increased dust supply stimulated more plankton productivity where plankton were bio-limited in nutrients supplied in the dust, such as iron (Martin, 1990). Since then, the iron hypothesis has retained an important place in the debate over glacial-interglacial cycles of $CO_2$. Watson et al. (2000) took experimental data on the effects of iron supply on plankton productivity in the Southern Ocean (Boyd, 2000) and applied this to a carbon cycle model

across glacial-interglacial cycles. Their modelling, informed by the ocean experiment data, suggested that variations in the

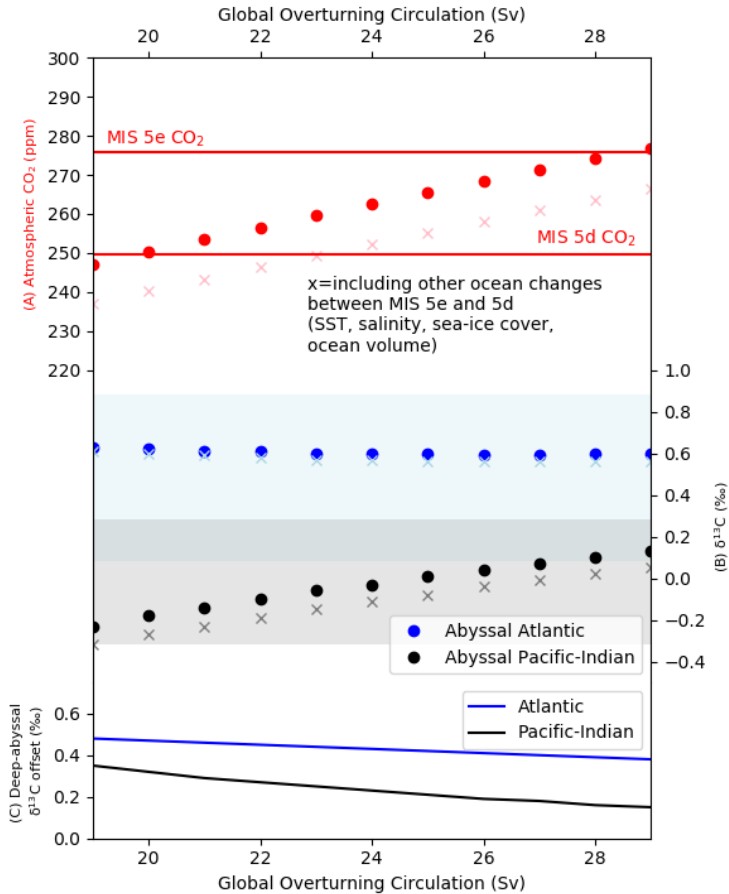

**Figure 13.** Sensitivity of atmospheric $CO_2$ concentration and ocean $\delta^{13}C$ to a downward variation in global ocean circulation parameter $\Psi_1$ in MIS 5e in SCP-M. x-axis shows the range of variation in $\Psi_1$ in Sv and the y-axes show the model results for (A) atmospheric $CO_2$ and (B) abyssal ocean $\delta^{13}C$ in each basin. Shaded areas are the $\pm$ standard deviations for abyssal $\delta^{13}C$ in MIS 5e. (C) shows the deep-abyssal $\delta^{13}C$ offset for each basin. Atmospheric $CO_2$ in MIS 5e and 5d is shown for reference. The "x" symbols in (A) and (B) show the same experiment including other changes in the ocean across MIS 5e-5d: SST, salinity, Antarctic sea-ice cover, ocean volume and coral reef carbonate production. Southern Ocean biological export productivity is not varied in this experiment.

Southern Ocean iron supply and plankton productivity could account for large (~40 ppm) swings in atmospheric $CO_2$, with peak activity in the last glacial cycle during MIS 4 and MIS 2. Debate has continued over the magnitude of the contribution of Southern Ocean biological productivity to the glacial $CO_2$ drawdown. According to Kohfeld et al. (2005), based on sediment data, enhanced Southern Ocean biological productivity could account for no more than half of the glacial $CO_2$ drawdown. Others emphasise that Southern Ocean biological export productivity fluxes may have been weaker in the LGM in absolute terms, but that with weaker Southern Ocean upwelling, the iron-enhanced productivity contributed to a stronger biological

pump of carbon and was a major contributor to the LGM $CO_2$ drawdown (Jaccard et al., 2013; Martinez-Garcia et al., 2014; Yamamoto et al., 2019). Importantly, our finding for increased biological export productivity during MIS 4 and 2 is delivered without any model-simulated iron dust fertilisation of the Southern Ocean and entirely on account of model results best-fit to the atmospheric and ocean proxy data used. Therefore the finding is a robust independently-derived support for increased

biological export productivity during MIS 4 and in particular MIS 2. It is important to note our model-data experiments assume unchanged biological export productivity in surface boxes outside of the Atlantic and Pacific-Indian subpolar Southern Ocean boxes across the last glacial-interglacial period. Some authors posit that low latitude biological export productivity may have been stronger at the LGM due to increased shelf-sourced phosporus (Broecker, 1981, 1982; Filippelli et al., 2007; Tamburini and Föllmi, 2009; Menviel et al., 2012) or increased biological matter remineralisation depth (Matsumoto, 2007; Menviel

et al., 2012). Others argue that low latitude biological export productivity was weaker at the LGM due to lesser upwelling of thermocline waters and lower shallow ocean nutrient levels (Calvo et al., 2011; Hayes et al., 2011; Winckler et al., 2016). Weaker (stronger) glacial biological export productivity in the low latitude surface boxes would reduce (increase) the sensitivity of atmospheric $CO_2$ to ocean circulation in our model-data experiments.

### 5.2 Contribution and attribution analysis

Table 3 shows a contribution analysis for the data observations in each MIS model-data optimisation of ocean parameter values. The ranking is based on the relative standard deviation (RSD) for each proxy data observation (or set of data observations) in each MIS, with the highest ranking (e.g. 1) given to the data observation with the lowest RSD in each model box/MIS. The contribution analysis shows that atmospheric $\delta^{13}C$ and $CO_2$ exert the greatest influence on the optimisation results throughout the MIS experiments. This reflects that each of these atmospheric data time series is derived from a single source and does not

require locational averaging as in the ocean boxes. For the atmosphere data, only MIS-averaging (not model box dimension) takes place and therefore there is a lower standard deviation of the data in most MIS time slices. For the ocean boxes, averaging on depth and latitude takes place as well as MIS-averaging to derive a box/MIS mean data value. Using a box model with large boxes such as SCP-M means that large parts of the ocean are averaged into the ocean box mean value and therefore there is an increased spread of data values around the mean for those boxes. Therefore, the model-data results show a precise fit to

the atmospheric $\delta^{13}C$ and $CO_2$ data as shown in Figs 9-11. The results for oceanic variables are typically less precise but also fall within the standard deviations of the data observations for each box and MIS (Figs 9-11). Others have attempted glacial-interglacial model-data studies focusing only on the ocean data without matching atmospheric data (e.g. LeGrand and Wunsch, 1995; Gebbie and Huybers, 2006; Hesse et al., 2011; Zhao et al., 2017; Kurahashi-Nakamura et al., 2017). While these studies could potentially elicit more detail on oceanic processes, they are also potentially fraught due to the high spread of data values

for the oceanic data and could return results that are not consistent with the relatively well constrained glacial-interglacial atmosphere data. For our study, the express purpose is to identify causes of changes in atmospheric $CO_2$ concentration, so it is appropriate that atmospheric data observations make an important contribution to the model results. However, as shown in Figs 9-11 this is not at the expense of providing plausible results for the ocean variables. Additional parameter sensitivity analysis of the model-data experiments is shown in Fig. S2.

**Table 3.** Contribution of proxy data observations to the model-data experiment results for ocean parameter values in each MIS. Each proxy data observation from each model box is ranked from 1 to 6 in each MIS based on the relative standard deviation (RSD) of its data points. A ranking of 1 is given to the data observation with the smallest RSD in each MIS. A smaller RSD gives the data observation a higher weighting in the model-data optimisation and therefore a greater contribution to the model results. $\Delta^{14}C$ proxy data does not exist for periods before MIS 3.

| MIS | Atmospheric $CO_2$ | Atmospheric $\delta^{13}C$ | Atmospheric $\Delta^{14}C$ | Ocean $\delta^{13}C$ | Ocean $CO_3^{2-}$ | Ocean $\Delta^{14}C$ |
|---|---|---|---|---|---|---|
| ~1 | 2 | 1 | 4 | 5 | 3 | 6 |
| ~2 | 2 | 1 | 3 | 6 | 4 | 5 |
| 3 | 2 | 1 | 3 | 5 | 4 | 6 |
| 4 | 2 | 1 | nan | 4 | 3 | nan |
| 5a | 2 | 1 | nan | 4 | 3 | nan |
| 5b | 2 | 1 | nan | 4 | 3 | nan |
| 5c | 2 | 1 | nan | 4 | 3 | nan |
| 5d | 2 | 1 | nan | 4 | 3 | nan |
| 5e | 1 | 2 | nan | 4 | 3 | nan |

Figure 14 shows the contribution to the glacial drawdown in atmospheric $CO_2$ by each mechanism we modelled, relative to the last interglacial period (MIS 5e), in SCP-M. Our model-data study finds that approximately half of the glacial atmospheric $CO_2$ drawdown is contributed by weakened ocean circulation (GOC and AMOC), with the other half contributed by a combination

5  of lower SST, increased Southern Ocean biological export productivity, varying coral reef carbonate production and dissolution, and increased polar Southern Ocean sea-ice cover. Weakened GOC delivers the highest contribution to falling $CO_2$, followed by lower SST, weakened AMOC and stronger Southern Ocean biological export productivity. Lower SST leads to modest reductions in atmospheric $CO_2$ concentration early in the glacial cycle, increasing as the ocean cools further during MIS 4, and is an important contributor to decreased $CO_2$ in the LGM (Kohfeld and Chase, 2017). Some studies observed that early

10  versions of box models tended to overstate the effects of SST and other processes at high latitudes on atmospheric $CO_2$, relative to general circulation models (GCMs) (Broecker et al., 1999; Archer et al., 2000; Ridgwell, 2001; Kohfeld and Ridgwell, 2009). However, our modelled estimate of 28 ppm for the contribution of SST to the glacial-interglacial atmospheric $CO_2$ change (Fig. 14) falls within the range of GCM-derived estimates of 21-30 ppm (mean value 26 ppm) compiled by Kohfeld and Ridgwell (2009), is similar to that of Menviel et al. (2012) (27.5 ppm) and substantially less than another recent GCM-derived estimate

15  of 44 ppm (Khatiwala et al., 2019). Southern Ocean biological export productivity strengthens during MIS 4, and contributes a peak of -13 ppm by MIS 2 (LGM).

The smaller glacial terrestrial biosphere contributes 13 ppm $CO_2$ to the atmosphere during the LGM (MIS 2), consistent with other modelled estimates (Köhler et al., 2010; Menviel et al., 2012; Ganopolski and Brovkin, 2017). Other parameters

contribute lesser increases in $CO_2$ (salinity, ocean volume) and decreases (Antarctic sea-ice, coral reefs) during the glacial cycle. Our estimate for coral reefs of -9 ppm $CO_2$ is at the lower end of the range of 6-20 ppm summarised in Kohfeld and Ridgwell (2009), suggesting that our simple parameterisation of the coral reef carbon and alkalinity fluxes could underestimate its effect, likely due to the assumed fast mixing rates of reef carbon and alkalinity into the surface boxes in SCP-M. Ridgwell

5  et al. (2003) modelled +20 ppm atmospheric $CO_2$ from coral reef carbonate accumulation in the Holocene period, noting a high sensitivity of their model to coral reef accumulation rates. It is likely that our model-data results underestimate the contribution of AMOC, because our model does not explicitly resolve AMOC shoaling (e.g. Menviel et al., 2012; Brovkin et al., 2012; Yu et al., 2016; Eggleston et al., 2016; Kohfeld and Chase, 2017; Menviel et al., 2020), other than a linear-positive linkage between the AMOC parameter and a deep-abyssal Atlantic box mixing term (less mixing between the deep and abyssal Atlantic boxes

10  as AMOC slows). Therefore, the analysis could miss additional features of the AMOC mechanism which could contribute to greater atmospheric $CO_2$ drawdown in Fig. 14 . The contribution of the model parameters to the glacial atmospheric $CO_2$ drawdown shown in Fig. 14, incorporate the effects of various feedbacks in the model such as continental weathering and calcium carbonate compensation.

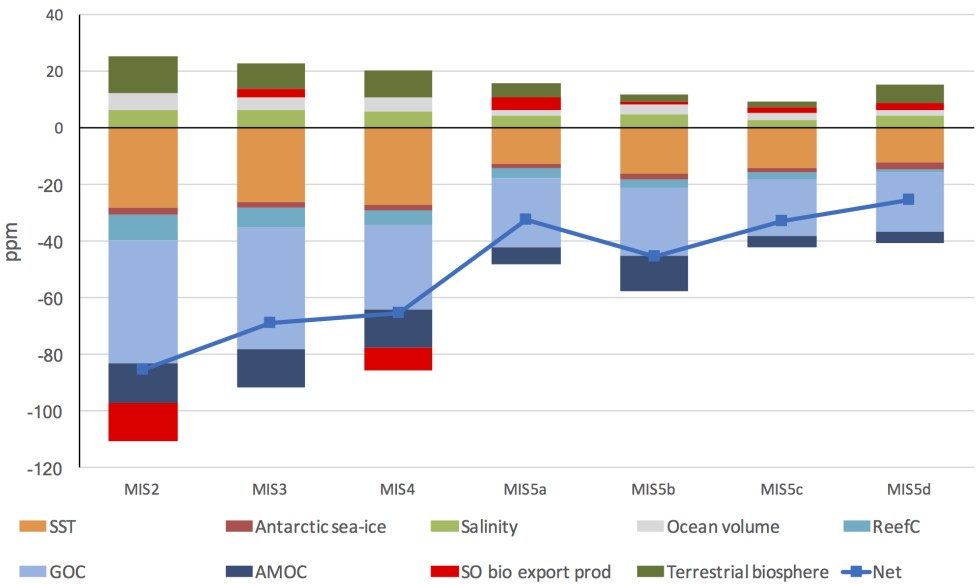

**Figure 14.** Impacts on atmospheric $CO_2$ concentration of model parameters in the model-data experiment results, from the last interglacial period (MIS 5e) to the Last Glacial Maximum (MIS 2). SST = sea surface temperature, ReefC = shallow carbonate production/dissolution, GOC = global ocean circulation, AMOC = Atlantic Meridional Overturning Circulation, SO Bio Export = Southern Ocean Biological export productivity.

## 5.3 The LGM and Holocene

Within the context of LGM-Holocene studies, our findings corroborate the hypothesis that a number of mechanisms, not one singular factor, delivered the ~85 ppm increase in atmospheric $CO_2$ concentration from the LGM to the Holocene (e.g. Kohfeld and Ridgwell, 2009; Köhler et al., 2010; Sigman et al., 2010; Hain et al., 2010; Menviel et al., 2012; Brovkin et al., 2012; Ferrari et al., 2014; Menviel et al., 2016; Ganopolski and Brovkin, 2017; Kohfeld and Chase, 2017; Muglia et al., 2018). This finding is more obvious when the sequential nature of changes is observed over the full glacial-interglacial cycle, as distinct from analysing the LGM and Holocene in isolation. Our model-data results agree with those of Menviel et al. (2016), who showed that the LGM oceanic $\delta^{13}C$ and $\Delta^{14}C$ records were most consistent with a weak GOC and AMOC. Menviel et al. (2016) further showed that this weak oceanic circulation would significantly increase the deep ocean carbon content (and thus significantly contribute to the $pCO_2$ decrease). The longer timescale of our analysis highlights that changes in GOC and AMOC took place earlier in the glacial cycle than the LGM, and were at or near their glacial minima prior to the LGM. However, some caution is required as our model-data results reflect the mean MIS state. For example, Menviel et al. (2014) modelled substantial variability in AMOC during Dansgaard-Oeschger events during MIS 3. Such variability is averaged out in our MIS state experiments. Our model-data results also support increased Southern Ocean biological export productivity in the LGM as a contributor to the lower LGM atmospheric $CO_2$ concentration (and in MIS 4), as well as lower SST and to a lesser extent coral reef carbonate dissolution.

## 5.4 The terrestrial biosphere

Our modelled increase in the terrestrial biosphere carbon stock from the LGM to Holocene, of ~600 Pg C (Fig. 12), falls within but towards the upper end of recent estimates of this change of 300-850 Pg C (e.g. Joos et al., 2004; Brovkin et al., 2007; Köhler et al., 2010; Prentice et al., 2011; Brovkin et al., 2012; Ciais et al., 2012; Peterson et al., 2014; Menviel et al., 2016; Jeltsch-Thommes et al., 2019). Brovkin et al. (2007), Brovkin et al. (2012) and Köhler et al. (2010) all modelled ~500-550 Pg C increase in the terrestrial biosphere between the LGM and Holocene (Prentice et al. (2011) estimated (550-694 Pg C)). According to François et al. (1999), palynological and sediment data infer that the terrestrial biosphere carbon stock was 700-1350 Pg C smaller in the LGM than the present. Ciais et al. (2012) pointed to a growth of a large carbon pool in steppes and tundra during the LGM as an offsetting feature to the declining tropical biosphere, leading to a smaller estimate of ~330 Pg C (Ganopolski and Brovkin (2017) modelled a similar estimate of 350 Pg C). Jeltsch-Thommes et al. (2019) estimated a glacial-interglacial change in terrestrial biosphere of 850 Pg C (median estimate; range 450 to 1250 Pg C), a similar estimate to that of Joos et al. (2004) of 820-850 Pg C. Jeltsch-Thommes et al. (2019) demonstrated the importance of including ocean-sediment and weathering fluxes in their modelling estimates, and suggested other studies may underestimate the full deglacial change in the terrestrial biosphere carbon stock. While our model results (~600 Pg C) are higher than some estimates of the LGM-Holocene change in the terrestrial biosphere (e.g. Ciais et al., 2012; Menviel et al., 2016; Ganopolski and Brovkin, 2017), they are mostly in good agreement (e.g. Joos et al., 2004; Brovkin et al., 2007; Köhler et al., 2010; Prentice et al., 2011; Brovkin et al., 2012; Peterson et al., 2014; Jeltsch-Thommes et al., 2019), and our NPP estimates mostly align with the

glacial-interglacial cycle NPP reconstruction of Hoogakker et al. (2016) as shown in Fig. 12. The driver for NPP in the simple terrestrial biosphere module in SCP-M is atmospheric $CO_2$ concentration via carbon fertilisation (e.g. Otto et al., 2002; Kaplan et al., 2002; Joos et al., 2004; Hoogakker et al., 2016). Temperature and precipitation also exert important controls on NPP (e.g. François et al., 1999; van der Sleen et al., 2015), which are not accounted for in our model.

The isotopic fractionation behaviour of the terrestrial biosphere may also vary on glacial-interglacial timeframes. This has been studied for the LGM, Holocene and the present day (e.g. Collatz et al., 1998; François et al., 1999; Kaplan et al., 2002; Köhler and Fischer, 2004; Joos et al., 2004; Kohn, 2016). The variation in isotopic fractionation within the terrestrial biosphere reflects changes in the relative proportions of plants with the $C_3$ and $C_4$ photosynthetic pathways, but also strong variations within the same photosynthetic pathways themselves (François et al., 1999; Kohn, 2010; Schubert and Jahren, 2012; Kohn,

2016). The drivers for these changes include relative sea level and exposed land surface area (François et al., 1999), global tree-line extent (Köhler and Fischer, 2004), atmospheric temperature and $CO_2$ (Collatz et al., 1998; François et al., 1999; Köhler and Fischer, 2004; Kohn, 2010; Schubert and Jahren, 2012), global and localised precipitation and humidity (Huang et al., 2001; Kohn, 2010; Schubert and Jahren, 2012; Kohn, 2016), and also changes in the intercellular $CO_2$ pressure in the leaves of $C_3$ plants (François et al., 1999). Estimated changes in average terrestrial biosphere $\delta^{13}C$ signature between the

LGM and the Holocene fall in the range -0.3-1.8‰ (less negative $\delta^{13}C$ signature in the LGM), with further changes estimated from the onset of the Holocene to the pre-industrial, and even greater changes to the present day (due to rising atmospheric $CO_2$). This feature has been covered in detail in studies that focussed on the terrestrial biosphere between the LGM and Holocene, but less so in modelling and model-data studies of the last glacial-interglacial cycle. Menviel et al. (2016) provided a sensitivity of -0.7+0.5‰ around an average LGM terrestrial biosphere $\delta^{13}C$ value of -23.3‰, based on previous modelling of

the LGM-Holocene timeframe by Joos et al. (2004). Another modelling study (Menviel and Joos, 2012), assessed the variation in LGM-Holocene $\delta^{13}C$ of the terrestrial biosphere to be a minor factor and it was not considered. Köhler and Fischer (2004) assessed the changing $\delta^{13}C$ signature of plants between the LGM and Holocene to be a minor factor in setting $\delta^{13}C$ of marine DIC, compared to changes in the absolute size of the terrestrial biosphere across this period. Given the uncertainty and ranges of starting estimates of terrestrial biosphere $\delta^{13}C$ (for example, the very large range in present day estimates of $C_3$ plant $\delta^{13}C$

(Kohn, 2010, 2016)), the uncertain LGM-Holocene changes, the large number of potential drivers of relative $C_3$ and $C_4$, and the further uncertainty in extrapolating the posited LGM-Holocene changes back for the preceding 100 kyr, and finally the modest changes relative to the average $\delta^{13}C$ signature, we omit this feature with the caveat that there is added uncertainty in our terrestrial biosphere results with respect of the $\delta^{13}C$ signature applied. Our choice of a constant terrestrial biosphere $\delta^{13}C$ signature of -23‰ is similar to values assumed by Menviel et al. (2016) and Jeltsch-Thommes et al. (2019) (-23.3‰,

-24‰ respectively), but more negative than assumed in Brovkin et al. (2002), Köhler and Fischer (2004) and Joos et al. (2004) (-16‰, -17‰). In summary, our aim is not to contribute new findings of the terrestrial biosphere, but to ensure that the simple representation of the terrestrial biosphere in SCP-M provides the appropriate feedbacks to our (exhaustive) glacial-interglacial cycle model-data optimisation experiments, that are in line with the published estimates discussed above.

## 5.5 Advantages and limitations of this study

The use of a simple box model for this model-data study, SCP-M, enabled a range of proxies to be incorporated into MIS data reconstructions, and a large number of simulations ($\sim$9,000 in each MIS) to explore possible parameter combinations in each MIS. However, the use of a simple box model means that some details are lost in the analysis. Given the large spatial coverage of the SCP-M boxes, data for large areas of the ocean are averaged. In the case of carbonate ion proxy, we apply a default estimate of standard deviation to account for the large volume of ocean covered by SCP-M's boxes relative to the proxy data locations, and to enable the normalisation of the carbonate ion proxy data in a procedure that uses the data standard deviation as a weighting. Despite this caveat, we believe that the model-data experiment results provide a good match to the data across the various atmospheric and ocean proxies as shown in Figs 9-11.

Most major processes in the SCP-M model are simply parameterised, allowing them to be free-floated in model-data experiments. However, the driving factors behind parameter value changes can only be speculated. For example, slowdown in GOC may be the result of changing wind patterns or buoyancy fluxes around Antarctica (Morrison and Hogg, 2013), Antarctic sea-ice cover (Ferrari et al., 2014), or may be the result of shoaling AMOC leading to extensive filling of the abyssal ocean by waters sourced from GOC (Curry and Oppo, 2005; De Boer and Hogg, 2014; Jansen, 2017). Probing the root cause of our model-data findings would require a more detailed physical and/or biogeochemical model. Furthermore, we apply a simple representation of the terrestrial biosphere in our model-data experiments, relying primarily on atmospheric $CO_2$ as the driver for NPP. This approach provided reasonable results for the terrestrial biosphere carbon stock and NPP, on the whole, but may miss some detail in the terrestrial biosphere during the last glacial-interglacial cycle. Our MIS time-slicing obscures details in the proxy records within MIS. For example, Yu et al. (2013) observed a transient drop in carbonate ion concentrations in the deep Pacific Ocean during MIS 4, and there are large transient changes in atmospheric $\delta^{13}C$ during MIS 4 and MIS 3. Ganopolski et al. (2010) and Menviel et al. (2012) modelled transient collapses and rebounds in AMOC during MIS 4 (and other short-term changes in atmospheric dust supply and depth of biological nutrient remineralisation), which could have contributed to the full observed magnitude of changes in atmospheric $\delta^{13}C$ across this period (e.g. Eggleston et al., 2016) - not captured with our MIS-averaging approach. We omitted the last glacial termination from our analysis, a period in which atmospheric $CO_2$ concentration increased by $\sim$85 ppm in 8 kyr. Future model-data optimisation work could probe this period at 1 kyr intervals, or with transient, data-optimised simulations, to profile the unwinding of processes that led to the last glacial cycle $CO_2$ drawdown.

In summary, while the carbon cycle box model we applied is high level in nature and there are caveats, the modelling itself is heavily constrained by natural observations and proxy data from the carbon cycle. Therefore, this work presents a plausible set of modelled outcomes for the last glacial-interglacial cycle.

## 6 Conclusions

Multiple processes drove changes in atmospheric $CO_2$ concentration during the last glacial-interglacial cycle. Against a backdrop of varied SST, salinity, sea-ice cover, ocean volume and reef carbonates, we modelled sequentially weaker GOC (first) and

AMOC (second) to reduce atmospheric $CO_2$ concentration in the lead up to the LGM. During the LGM, increased Southern Ocean biological export productivity delivered an incremental fall in atmospheric $CO_2$ concentration, resulting in the glacial cycle $CO_2$ minimum. GOC, AMOC, Southern Ocean biology and SST rebounded to modern values between the LGM and Holocene, contributing to the sharp post-glacial increase in atmospheric $CO_2$ concentration. The terrestrial biosphere played an important negative feedback role during the glacial-interglacial cycle, releasing $\delta^{13}$C-depleted $CO_2$ to the atmosphere at times during the glaciation, and taking up $CO_2$ during the termination and Holocene. These model-data results were achieved with a simple carbon cycle box optimised for proxy data for atmospheric $CO_2$, atmospheric and ocean $\delta^{13}$C and $\Delta^{14}$C, and ocean $CO_3^{2-}$. Our results agree with hypotheses for glacial-interglacial cycle $CO_2$ that include varying ocean circulation, Southern Ocean biological export productivity and other physical and biogeochemical changes in the marine and terrestrial carbon cycle (e.g. Kohfeld and Ridgwell, 2009; Sigman et al., 2010; Ganopolski et al., 2010; Brovkin et al., 2012; Menviel et al., 2012; Ferrari et al., 2014; Menviel et al., 2016; Kohfeld and Chase, 2017; Ganopolski and Brovkin, 2017). We emphasise the need to include the Pacific and Indian oceans in evaluation of the oceanic carbon cycle, particularly in relation to the last glacial-interglacial cycle and the LGM-Holocene transition.

Many uncertainties exist in the data and the prescribed nature of carbon cycle processes in a box model. However, such uncertainty is largely inescapable when dealing with models and proxy data. We propose these model-data results as one set of plausible results for the last glacial carbon cycle, in agreement with available proxy data, and see them as encouraging for the use of models and data to help constrain hypotheses for past changes in the Earth's carbon cycle.

## 7 Code and data availability

The model code, processed data files, model-data experiment results, and any (published) raw proxy data gathered in the course of this work, are located at https://doi.org/10.5281/zenodo.4084586. No original data was created, or unpublished data used, in this work. Fig. S3 contains an overview of the files contained in the repository. For more detail on the SCP-M equations, see O'Neill et al. (2019).

*Author contributions.* CO undertook model development work, data-gathering, modelling and model-data experiments. AH provided the oceanographic interpretation and guided modelling and data analysis. ME designed model-data experiments, provided input into data analysis and guided the modelling of the marine biology and carbon isotopes. BO contributed glacial-interglacial cycle model forcings and input to modelling of the reef carbonates. SE contributed to the modelling of the marine biology and carbonate pump. All authors contributed to drafting and reviewing the document.

*Competing interests.* The authors declare that they have no conflict of interest.

*Acknowledgements.* Stewart Fallon provided input to the processing of radiocarbon data. Malcolm Sambridge provided input on model-data optimisation and inversions. Jimin Yu provided helpful discussions and published carbonate ion proxy data.

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
