# Peer review of "Sequential changes in ocean circulation and biological export productivity during the last glacial-interglacial cycle: a model-data study"

_Climate of the Past, 2019_

## Referee Comment (RC1) · Anonymous Referee #1 · 18 Feb 2020

This paper describes the application of a carbon cycle box model on the last glacial cycle (last 130 kyr). The model has been described before, but is modified here for the application to paleo-timescales. A great part of the effort is the compilation of available paleo-data to which simulations results are compared.

For this effort steady-state results for mean values for each of the Marine Isotope Stages (MIS) from 5e-1 are evaluated, while periods with rapid changes (eg glacial inception, Termination I) are not investigated. An optimisation approach is then used to derive the parameter values of a few important processes, namely Global Ocean Circulation, Atlantic Meridional Ocean Circulation, and Southern Ocean biological pro-

ductivity.

The approach is in itself an interesting piece of work, that combines data analysis with modelling, but I have two major concerns, that need the attention of the authors: (1) the shortcomings of the steady-state approach, and (2) the $\delta^{13}$C cycle.

**Shortcomings of the steady-state approach**

The chosen approach of steady-state analysis combined with optimization is a way, which certainly has benefits, but also shortcomings. I believe the benefits lie in the possibility to test a great number of parameter values, and this is certainly analysed with great effort and detail and worth publishing (but see my recommendation on shortenings of certain parts below). However, there is little learned on the potential shortcomings and pitfalls, which in my view need to be discussed more deeply. I believe where this approach is falling to short is the following: By analysing only steady-state the authors miss out the opportunities to judge the results based on the timing (when do processes change leading to what results).

I provide one example where the article nicely fails, producing a potenially right answer for very likely the wrong reason: One of the dominant features of atmospheric $\delta^{13}$C during the last glacial cycle is a drop by about 0.5‰ during MIS4. The steady-state approach now leads to the evalution of a mean value of atmospheric $\delta^{13}$C which does not really cover this decrease at all, it shows about a decline by about 0.2‰ from MIS5a to MIS4 (Fig 4). So, any explanation for this drop would be falling too short in the observed amplitude by 0.3‰. Note that this $\delta^{13}$C feature is not rapid, it is an anomaly that has been detected from raw data by spline smoothing and is alltogether nearly 20 kyr long, however the decreasing flank falls in MIS4, the increasing flank in MIS3, thus the signal is largely smoothed out in the chosen MIS-centric analysis. The analysis of the results now comes to the conclusion that very likely changes in terrestrial carbon storage was responsible for a change in atmospheric $\delta^{13}$Căof -0.2‰ (as said explaining

a too little amplitude), it is furthermore said that the drop is accompanied by a 30 ppm fall in $CO_2$ (page 12, lines 1-5), citing Hoogakker et al., 2016. I believe this is entirely wrong: The drop in $CO_2$ happens clearly a few kyr **before** the drop in atmospheric $\delta^{13}$C, as seen in Fig. 4. Furthermore, since both $CO_2$ and $\delta^{13}$C are meassured at the same samples and are both derived from gases in ice cores, this temporal offset between $CO_2$ and atmospheric $\delta^{13}$C can not be explained by chronological issues. The anomalies in biosphere as documented by Hoogakker et al., 2016 all fall in line with the $CO_2$ changes, but not with the $\delta^{13}$C changes, also note that Hoogakker et al., 2016 was published before the atmospheric $\delta^{13}$C data set of Eggleston et al (2016). In that respect citations from Hoogakker on page 19 are also missing the correct timing: In Hoogakker NPP drops between around 70 ka (parallel to the drop in $CO_2$), while the $\delta^{13}$C drop occurs 5 ka later. Also note, that in Eggleston et al. (2016) the authors of this atmospheric $\delta^{13}$C record tried to make sense of it by focusing on the part in which $\delta^{13}$C falls, but $CO_2$ rises again (Fig 2 in that paper) focusing on an opposite behaviour than described here.

The second most dramatic change in atmospheric $\delta^{13}$C is a sharp drop by 0.2‰ during Termination I, a time window which has been chosen to be not be included in this steady-state analysis, again missing the opportunity to use $^{13}$C to pin down responsible processes. Only the long-term trend in $\delta^{13}$C of +0.2‰ from the penultimate interglacial to the Holocene seemed to be meaningful covered by the approach.

**The $\delta^{13}$C cycle**

As already seen above the steady-state approach might not be the best way to tackle atmospheric $\delta^{13}$C. Furthermore, for an evaluation of $\delta^{13}$C in general in such steady-state experiments as performed here the fluxes (e.g. as mol C/yr) and $\delta^{13}$C-signatures in/out of the simulated atmosphere/ocean carbon cycle are essential: atmosphere-land carbon fluxes, volcanic $CO_2$ outgassing, weathering, and burial of organic and inorganic carbon in the sediments. Little to non of those fluxes (and $\delta^{13}$C-signatures)

are given in the text itself. If I dig into the python source code of the model (or the description of version 1 in O'Neill et al. (2019)) I find a few information, but the source code is difficult to interpret as a non-user and some information seemed to be either misleading or wrong. An examples:

Continental weathering consists of two different processes depending on the rock type that is weathered. In carbonate weathering 1 mol of $CaCO_3$ together with 1 mol of $CO_2$ from the atmosphere leads to the entry of 2 mol of $HCO_3^-$ into the surface ocean. In silicate weathering 2 mol of atmospheric $CO_2$ǎare necessary to weather 1 mol of $CaSiO_3$ leading again to the entry of 2 mol of $HCO_3^-$ into the surface ocean. For details see, for example Lord et al. (2016). From the description of weathering in O'Neill et al. (2019) I have the impression that the carbonate weathering is not depicted correctly (no consumption of atmospheric $CO_2$). Furthermore, from the python code I learned that weathering (probably meaning carbonate weathering, since in silicate weathering all $CO_2$ comes from the atmosphere with its $\delta^{13}$C-signature) has a $\delta^{13}$C-signature of $-6.9$‰, similarly as volcanic $CO_2$. While the volcanic $\delta^{13}$C seems to be in the expected range (although on the lower side) I believe the weathering $\delta^{13}$C-signature is wrong, since carbonate rocks have a typical $\delta^{13}$C-signature of about +1-2‰, see for example Sano and Williams (1996); Mook (1986).

I also do not understand how their approach with not explicitly considering terrestrial carbon change (terrestrial carbon to my understanding is covered as externally to the atmosphere/ocean system, fluxes in/out of it prescribed by optimization) covers changes in C3 vs C4 photosynthesis (which have a significantly different isotopic fractionation) on glacial/interglacial timescales (Collatz et al., 1998; Köhler and Fischer, 2004) which leads to differences in the mean terrestrial $\delta^{13}$C and therefore also the changes in the $\delta^{13}$C-cycle as a whole (Kaplan et al., 2002).

**Overall recommendation**

My recommendation therefore is, that the model in its present form might be a useful tool for evaluating marine processes, and might be well used together with the available marine data (apart from $\delta^{13}$C), but fails to give meaningfull results for the $\delta^{13}$C cycle. This includes atmospheric and marine $\delta^{13}$C. I urge the authors to get those parts out of the manuscript. If they wish to further analyse the $\delta^{13}$C-cycle I believe fundamental model improvements are necessary, that can not be obtained by a major revision, but by a revised model version. Besides this, shortcomings of the steady-state approach should be discussed in more detail and the unclear (wrong?) aspects of carbonate weathering and annual fluxes in/out of the simulated system (atmosphere/ocean) need to be clarified for each MIS, maybe in a table or a new figure.

**Minor issues in chronological order:**

1. Figure 1: It is not clear, how GOC (red arrows) is split up in the part upwelling in Atlantic and Indo-Pacific Ocean.

2. Figure 1: Does your approach focusing on changes in GOC, AMOC and export production inply, all other proceses (fluxes) stay constant in time?

3. Figure 12: x-axis is wrong, eg. MIS5e is between ∼114-122 ka, while it has been between ∼118-128 ka in other figures.

4. With respect to iron fertilisation you might check on Shaffer and Lambert (2018).

5. The fact that not one single process is needed to explain LGM-Holocene carbon cycle changes is long known, and has been called "the carbon stew" by some authors. You might want to check and discuss in more detail earlier modelling approaches on one glacial cycle (or longer), for example in Ganopolski and Brovkin (2017).

6. Figure 14: Changes in $CO_2$ caused by changes in terrestrial NPP and carbon stocks are missing in this figure. Please add.

7. Section 5.3. You might want to check on recent finding of terrestrial carbon storage from $\delta^{13}C$ in Jeltsch-Thömmes et al. (2019).

**References**

Collatz, G. J., Berry, J. A., and Clark, J. S.: Effects of climate and atmospheric $CO_2$ partial pressure on the global distribution of $C_4$ grasses: present, past and future, Oecologia, 114, 441–454, 1998.

Eggleston, S., Schmitt, J., Bereiter, B., Schneider, R., and Fischer, H.: Evolution of the stable carbon isotope composition of atmospheric $CO_2$ over the last glacial cycle, Paleoceanography, 31, 434–452, doi:10.1002/2015PA002874, 2016.

Ganopolski, A. and Brovkin, V.: Simulation of climate, ice sheets and $CO_2$ evolution during the last four glacial cycles with an Earth system model of intermediate complexity, Climate of the Past, 13, 1695–1716, doi:10.5194/cp-13-1695-2017, 2017.

Jeltsch-Thömmes, A., Battaglia, G., Cartapanis, O., Jaccard, S. L., and Joos, F.: Low terrestrial carbon storage at the Last Glacial Maximum: constraints from multi-proxy data, Climate of the Past, 15, 849–879, doi:10.5194/cp-15-849-2019, 2019.

Kaplan, J. O., Prentice, I. C., Knorr, W., and Valdes, P. J.: Modeling the dynamics of terrestrial carbon storage since the Last Glacial Maximum, Geophysical Research Letters, 29, 2074, doi: 10.1029/2002GL015 230, 2002.

Köhler, P. and Fischer, H.: Simulating changes in the terrestrial biosphere during the last glacial/interglacial transition, Global and Planetary Change, 43, 33–55, doi:10.1016/j.gloplacha.2004.02.005, 2004.

Lord, N. S., Ridgwell, A., Thorne, M. C., and Lunt, D. J.: An impulse response function for the long tail of excess atmospheric CO2 in an Earth system model, Global Biogeochemical Cycles, 30, 2–17, doi:10.1002/2014GB005074, 2016.

Mook, W. G.: $^{13}C$ in atmospheric $CO_2$, Netherlands Journal of Sea Research, 20, 211–223, 1986.

O'Neill, C. M., Hogg, A. M., Ellwood, M. J., Eggins, S. M., and Opdyke, B. N.: The [simple carbon project] model v1.0, Geoscientific Model Development, 12, 1541–1572, doi:10.5194/gmd-12-1541-2019, 2019.

Sano, Y. and Williams, S. N.: Fluxes of mantle and subducted carbon along convergent plate boundaries, Geophysical Research Letters, 23, 2749–2752, doi:10.1029/96GL02260, 1996.

Shaffer, G. and Lambert, F.: In and out of glacial extremes by way of dust-climate feedbacks, Proceedings of the National Academy of Sciences, 115, 2026–2031, doi:10.1073/pnas.1708174115, 2018.

---

## Referee Comment (RC2) · Anonymous Referee #2 · 21 Feb 2020

SUMMARY

This paper uses the SCP-M model - an 11-box model of the ocean carbon cycle - to simulated changes in ocean circulation, marine productivity, and resulting changes in atmospheric carbon dioxide during distinct Marine Isotope Stages (5a-e, 4, 3, 2, and 1) over the past 130,000 years. The model is forced using reconstructed quantities for SST, salinity, ocean volume, sea ice cover, and reef carbonate productivity. The model outputs for atmospheric CO2, atmospheric d13C, D14C, oceanic d13C, and deep/abyssal carbonate ion concentrations were optimized against estimates for these parameters that were obtained from global reconstructions. Following optimization,

optimized estimates of global and Atlantic meridional overturning rates and Southern Ocean biological productivity are presented. These results suggest: (a) global overturning rates responded early (MIS5d) in the glacial cycle; the largest response in the AMOC occurs between MIS5a and MIS4, with low rates also during MIS2; Southern Ocean biological productivity descrease early during glaciation (MIS5d) and recover gradually during each successive MIS, exceeding interglacial values during MIS3 and reaching peak values during MIS2. An interesting contribution of this paper is the use of previously compiled ocean tracers for the last glacial-interglacial cycle as a means of optimizing these box model results. In particular, the inclusion of sparse Indo-Pacific data in the optimization is an interesting and useful expansion from the predominantly Atlantic-centric perspective on glacial-interglacial global ocean circulation changes. This inclusion leads to the interesting result that early changes in the global overturning circulation is a possible strong contributor to early changes in carbon dioxide. The authors support their simulations by suggesting that previously hypothesized ideas such as early AABW expansion and weakened circumpolar deep water upwelling could result in reduced global overturning, which influenced the abyssal deep Pacific carbonate ion concentrations and d13C values. They suggest with their box model simulations that reduced GOC can influence d13C in the abyssal Indian and Pacific Ocean sectors without necessarily resulting in a strong Atlantic abyssal d13C response. Obviously, the use of box-model simulations has its limitations (i.e. the generalization of the whole ocean in a box model context, which necessarily brushes over many processes and regional variability). Furthermore, the authors base these conclusions of Indo-Pacific change in carbonate ion concentrations on a single core. That said, their proposal that an early glacial response in GOC could be responsible for early CO2 drawdown is an interesting new contribution to the field.

I provide detailed comments below. Of substantial concern is the qualitative treatment of data in the Methods section, which I think could be more rigourous given that the authors base some of their most important conclusions on optimizing their results to these data. First, the authors should filter their d13C results to include only species known

to represent deep ocean d13C changes and classify what they mean by "abyssal" and "deep" in their data (comments 7-8); second a better quantification of their observed changes for the MIS periods is warranted (I suggest examination of probability density functions and whether differences are statistically significant) (see comments 9-10, 13). Second, I think a more thorough description of their model would be useful to readers who wish to understand, e.g., what drives biological productivity export in their model, and what their southern ocean box actually represents in terms of the real ocean (comment 6).

COMMENTS

1. The authors base their paper on a recently published carbon cycle box model (O'Neill et al. 2019). They provide a brief description of the model but I found that this manuscript would benefit better description of some of the key parameters that are quite important to this paper, such as the controls on Z (biological productivity). It was very unclear to me on first reading how values of Z were ascertained.

2. Figure 1 – this graphic, while nice and colourful, is challenging for reading the actual numbers and symbols (especially the white ones which do not show up at all on my colour print). Readability is more important than colour! I suggest making box numbers, symbols all BLACK using larger fonts so that they are readable.

3. Pg 5 lines 15-17. This sentence seems out of place: "Therefore, our modelling excludes the last glacial termination (∼11-18 ka)." Should it occur before the previous sentence?

4. Section 2.2.1 Model forcings: Although the authors ultimately conclude that sea ice cover – as a barrier mechanism constraining air-sea $CO_2$ exchange – is not that important, they authors should emphasize limitations of their use of the ice core sea ice proxy. First, this proxy is non-linear, so their simulations probably over estimate early (MIS5d) sea ice cover and underestimate later (MIS4-2) sea ice cover. This point is made very clearly by Wolff et al. 2010 (and supports) the authors' assertion that the

barrier effect of sea ice early in the glaciation is probably small.

5. Furthermore, it is worth pointing out somewhere in the discussion that this modeling exercise only examines the potential role of sea ice as a barrier to CO2 exchange, and not its synergistic (and likely more important) roles in influencing nutrient distributions, marine productivity, and a trigger for deep ocean circulation changes. The authors state this somewhat in their "Advantages and limitations" section, but I think that this point could be made more explicitly.

6. Another larger issue that the sea ice proxy highlights is the spatial heterogeneity of the Southern Ocean and how the model results are linked with reality: the sea ice proxy likely represents changes very close to the continent and early glacial changes in sea ice are not well reproduced in the few long sea ice records that are found near the APF. This not only suggests that a barrier effect of sea ice would be limited to only part of the Southern Ocean, it points to larger issues with treating the Southern Ocean as one box, with an unclear delineation of how much of the S. Oc. this box is presumed to cover. If the box is supposed to ONLY cover those areas close to the continent where AABW and Circumpolar Deepwater processes that influence GOC are most important, then the authors' main conclusion of increases in S. Oc. export production aren't well supported by paleoceanographic data which show reductions in export South of the APF for the majority of the glacial cycle between MIS5d and MIS2. Some discussion of what the Southern Ocean box actually represents - and this potential disconnect with paleoceanographic data - is warranted.

7. Throughout the paper the authors refer to "abyssal" and "deep" water masses for all basins, but I was never able to find the depth cut-offs that were used to distinguish these depths in the different basins. Please put them in the figure captions and text (not just supplemental information, if it is there.)

8. The authors discuss briefly that previous studies have only used the C. wueller-storfi data to reconstruct deep ocean d13C (Peterson et al. study; Kohfeld and Chase

study). Which data did these authors select from Oliver et al. (2010)? They mention only using "deep" and "abyssal" sites (again, depths undefined) on page 11, but they do not indicate whether they have filtered the data to only include C. wuellerstorfi (or even Cibicidoides spp), which they SHOULD be doing if they haven't. Otherwise, the changes in d13C described on page 12 are invalid as descriptions of deep ocean circulation changes in d13C.

9. On Page 12, the authors qualitatively describe the differences between "deep" and "abyssal" changes in d13C. Why leave this discussion qualitative, when the data are available and quantification would be hugely useful. These data in the Pacific that are described are the data that pin the authors' entire argument surrounding early changes in GOC. I think that this warrants a bit more quantification of these data (once species other than Cibicidoides are filtered out of the dataset). I would be interested to know if the differences between deep and abyssal d13C in the Indo-pacific are statistically significant, and I think plots of the probability distribution functions of these data would be very useful.

10. Some type of quantification would also be very useful for the authors' description of the "transient drop in abyssal Atlantic ocean CO3= at MIS5b" on page 14. I was not convinced that this transient drop exists from the figure presented.

11. Please note on the bottom of page 13 and top of page 14 that the authors mean to refer to Figure 7 (not 6) to describe carbonate ion concentration data.

12. Last sentence before Results section: Please cite the figures you are using to make these observations about changes in d13C and DD14C

13. Similar quantification would be useful in the comparison between the carbonate ion concentration model output and data in Figure 9 and in the discussion on page 16-17.

---

## Referee Comment (RC3) · Anonymous Referee #3 · 2 Mar 2020

O'Neil et al. try and attribute the different changes in pCO2 that occurred from the LIG to the Holocene to changes in oceanic circulation and Southern Ocean (SO) biology. For this, they perform a series of simulations of all the MIS 5 through 1 with a box model forced by changes in SST, salinity, and sea-ice as derived from proxy records. They further vary the values of the Atlantic and Global overturning circulation as well as SO biology over a wide range of possible values. They then chose the simulations,which best fit with paleo-records (D14C, d13C, CO3(2-)...). It is an interesting attempt, which can inform on carbon cycle changes from the LIG to the Holocene. Please find below some comments that should be addressed.

[Figure]

Major comments: 1) The "data analysis" section 3 presents the changes in atm. CO2, d13CO2, oceanic d13C, D14C and CO3(2-) as inferred from proxy records from the LIG to the LGM. This is obviously a huge task, but which I am afraid can give rise to approximations and simplifications. I would consider seriously amending this section. How can the "increase in d13C across the glacial cycle be attributed to the growth of tundra at high latitudes"? (p12, L. 2-3). p12, L. 11-14: How were the values for MIS3 DD14C in the Atlantic derived? From Fig. 6a, it looks like there is no data across MIS3. This is quite a shortcut to explain the deglacial D14C decrease, and maybe you want to check the references and include " increase in Southern Ocean ventilation" above anything else. P14, L. 5-6: This reads like speculation.

2) Fit with the data: 50 umol/L as an "arbitrary standard deviation' for [CO3] is huge and represents more than the [CO3] changes (0-30 umol/L) recorded across the G-IG cycles. How much was taken for the standard deviation for d13C adn D14C? It looks quite large. Figures 9-11 would gain in having a more appropriate range in the y axis. At the moment the ranges and std are large, so that it almost looks like there are no changes from MIS5 to MIS 2.

3) References: In general I find that only a few references are used over and over and sometimes not appropriately. A few additional references are included in this review. Please note the typo throughout the document in "Ridgwell".

Specific comments: 1) Abstract: The first line does not make sense. Please reformulate. L. 3 Please add "SO" in front of "biological productivity"

2) Introduction: - L.15-19: please be more specific. Instead of "Ocean biology" you might want to refer to "iron fertilisation and its impact on nutrient utilisation", or changes in remineralisation depth (e.g. Kwon et al. 2009). What do you mean by composite mechanisms? - It would be good to also introduce the numerous modelling studies that have been done on the topic of G-IG changes in pCO2, and notably transient simulations of the G-IG trying to understand the changes in pCO2 (e.g. Ganopolski &

Brovkin 2017, Menviel et al., 2012).

3) Methods: - Variables included in the model: surely the model includes Dissolved Inorganic Carbon. By "CO2", do you mean atmospheric CO2? Does the model really includes "carbonate ions" as a prognostic tracer? - p4, L. 2: please refer to section 2.2.1 and Figure 2. - p7: I am very confused by the treatment of the terrestrial biosphere in the model and the paragraph L. 19-27. It reads like there is an interactive terrestrial module. But how can NPP be calculated with significance if there is no atm. Temperature or precipitation in the model? Why is "tundra"discussed with such emphasis in this paragraph? Tundra is not an "inert" carbon pool, and I don't think "permafrost" is a vegetation type. What is "pre-carbon fertilisation" - p8: what is the point of Table 1 if all the values of GOC, AMOC, biology are the same? It would be interesting to mention the PI control values though. - p10-11: The 'depth issue" should also be discussed in 2.3.1 and 2.3.2.

4) Discussion: p20, L. 3-6: It is not what the simulations tell you, but the proxy data! p21, L. 1-2: This is wrong → you are forcing your model with SST, Sea-ice.... so all these factors contribute to the pCO2 decrease. The experiments show that changes in oceanic circulation and SO biological productivity also contribute to that pCO2 decrease. Please take into consideration that G-IG pCO2 changes have been previously successfully simulated with models of intermediate complexity (e.g. e.g. Ganopolski & Brovkin 2017, Menviel et al., 2012) and box models. p21, L. 3-4: I don't understand the meaning p21, L. 7: Might want to check Piotrowski et al., 2008, Yu et al., 2016. (Nat. Geo). p21, L. 10 –p22, L. 5: This section really has to be discussed in light of all the work that has been done on the impact of iron fertilisation in the Southern Ocean. Some work on the topic: Watson et al., 2000, Nature; Jaccard et al., 2013, Science; Yamamoto et al., 2019, Climate of the Past; p22, L. 18: "sea-ice cover" p23, L. 1-12: Figure 13 is interesting but care has to be taken here given the large size of the "boxes". This should at least be discussed in light of previous modelling studies on the subject (e.g. Menviel et al., 2015, GBC).

---

## Author Comment (AC1) · 11 Jun 2020

CP reviewer comments #3 and author responses

AC: We thank the reviewer for their comments, suggestions and input into this manuscript. These comments make a strong contribution to improving the quality of our work. Please see below our responses to the individual comments.

We have made reference to changes to the manuscript, which are included as a supplement to the author comments, in track changes. Page and line references below refer to locations in the revised document with track changes.

[Figure]

Please note that we have changed our treatment of ocean $\delta$13C proxy data, stemming from one of the other reviewer comments, to only include $\delta$13C from Cibicides species of benthic foraminifera. We have also made some small changes to the parameterisation of the volcanic and weathering isotopic signatures in the model, from reviewer comments. These changes required the re-calibration of our model and re-running of the model-data experiments. The model-data results changed modestly. We have updated the figures and text (tracked in the attachment) in the manuscript, accordingly.

Major comments:

RC 1) The "data analysis" section 3 presents the changes in atm. CO2, d13CO2, oceanic d13C, D14C and CO3(2-) as inferred from proxy records from the LIG to the LGM. This is obviously a huge task, but which I am afraid can give rise to approximations and simplifications. I would consider seriously amending this section. How can the "increase in d13C across the glacial cycle be attributed to the growth of tundra at high latitudes"? (p12, L. 2-3).

AC: Thanks for the comment. In this instance (P12, L2-3 in the original manuscript) and throughout our manuscript, we have been a bit loose with our references to tundra, permafrost and peat, as you point out in this comment and a few below.

What we mean to refer to here is the storage of carbon by the accumulation and freezing, or burial, of peat and other soil organic matter under soil overburden, and growth of cold-climate vegetation, throughout the glacial cycle (e.g. Tarnocai et al., 2009; Ciais et al., 2012; Schneider et al., 2013; Eggleston et al., 2016; Treat et al., 2019).

We have corrected the statement on P12 L2-3, and expanded a bit, including a few more references and other possible causes of the atmospheric $\delta$13C pattern, now at P15 L10:

"Atmospheric $\delta$13C (Fig. 4(B)) increased by âĹij0.4‰ between the penultimate inter-glacial (MIS 5e) and the Holocene (MIS 1), with temporary falls at MIS 5d, MIS 4 and in

the last glacial termination (between MIS 1 and 2). The cause of the observed increase in atmospheric $\delta$13C across the last glacial-interglacial cycle may be the effect of accumulation and freezing, or burial in glacial sediments, of peat and other soil organic matter at the high latitudes (e.g. Tarnocai et al., 2009; Ciais et al., 2012; Schneider et al., 2013; Eggleston et al., 2016; Ganopolski and Brovkin, 2017; Treat et al., 2019). According to Treat et al. (2019), peatlands and other vegetation accumulated carbon in the relatively warm periods, and these carbon stocks were then frozen and/or buried in glacial and other sediments during the cooler periods, throughout the last glacial cycle. This buried or frozen stock of carbon persists to the present day (Tarnocai et al., 2009), although according to Ciais et al. (2012) it may be smaller now than in the LGM. Schneider et al. (2013) evaluated several possible candidates for the rising atmospheric $\delta$13C pattern across the last glacial-interglacial cycle and could not discount any of (1) changes in the carbon isotope fluxes of carbonate weathering and sedimentation on the seafloor, (2) variations in volcanic outgassing or (3) peat and permafrost build-up throughout the last glacial-interglacial cycle.

The large drop in $\delta$13C in MIS4, reverses in MIS 3 (Fig. 4(B)). This excursion in the $\delta$13C pattern likely resulted from sequential changes in SST (cooling), AMOC, Southern Ocean upwelling and marine biological productivity (Eggleston et al., 2016). Eggleston et al. (2016) parsed the atmospheric $\delta$13C signal into its component drivers across MIS 3-5, using a stack of proxy indicators, and highlighted the sequence of events between the end of MIS 5 and beginning of MIS 3, and their cumulative effects to deliver the full change in atmospheric $\delta$13C. Our MIS-averaging approach fails to capture the full amplitude of the changes in atmospheric $\delta$13C during MIS 3-5, and only captures the changes in the mean-MIS value, serving to understate the full amount of transient changes in responsible processes. In addition, the MIS-averaging approach misses the sequential timing of changes in processes within each MIS. These are limitations of our steady-state, MIS-averaging approach. The reduction in atmospheric $\delta$13C at the last glacial termination, between MIS 1 and MIS 2, coincident with a large atmospheric $CO_2$ increase, is attributed to the release of deep-ocean carbon to the atmosphere resulting from increased ocean circulation and Southern Ocean upwelling (Schmitt et al., 2012). The subsequent rebound of $\delta$13C in the termination period and the Holocene is believed to result from terrestrial biosphere regrowth, in response to increased CO2 and carbon fertilisation (Schmitt et al., 2012; Hoogakker et al., 2016). " Other amendments to this section are shown in track changes. RC: p12, L. 11-14: How were the values for MIS3 DD14C in the Atlantic derived? From Fig. 6a, it looks like there is no data across MIS3.

AC: Thanks, this was a charting error and now the chart has been corrected to show the data for MIS 3.

RC: This is quite a shortcut to explain the deglacial D14C decrease, and maybe you want to check the references and include " increase in Southern Ocean ventilation" above anything else.

AC: P16, L4 modified to "….an acceleration in atmospheric $\Delta$14C decline at the last glacial termination is attributed to the release of old, 14C -depleted waters from the deep ocean, due mainly to increased Southern Ocean upwelling (e.g. Sikes et al., 2000; Marchitto et al., 2007; Skinner et al., 2010; Burke and Robinson, 2012; Siani et al., 2013; Skinner et al., 2017)." RC: P14, L. 5-6: This reads like speculation.

AC: This sentence re-worded as: (P17, L35) "There is a modest drop in abyssal Atlantic Ocean CO2-3 at MIS 5b (-13 $\mu$mol kg-1 relative to MIS 5c), which coincides with a minor drop in abyssal Atlantic Ocean $\delta$13C (-0.19‰ and atmospheric CO2 (-14 ppm), indicating a common link. Menviel et al. (2012) modelled a transient slowdown in North Atlantic overturning circulation for this period, which could explain these features. "

RC: 2) Fit with the data: 50 umol/L as an "arbitrary standard deviation' for [CO3] is huge and represents more than the [CO3] changes (0-30 umol/L) recorded across the G-IG cycles. How much was taken for the standard deviation for d13C and D14C? It looks quite large. Figures 9-11 would gain in having a more appropriate range in the y axis. At the moment the ranges and std are large, so that it almost looks like there are

no changes from MIS5 to MIS 2.

AC: Re CO2-3. In response to this reviewer's comments, and a change to our data approach from the other reviewer comments (using only Cibicides species for $\delta$13C), we have been able to reduce our default standard deviation for ocean CO2-3 from 50 umol kg-1 to 15 umol kg-1, a substantial improvement. The rationale for setting the CO2-3 SD at an artificial level for the weighting in our model-data optimisation is dealt with in Section 2.3.2. This is an unfortunate feature of using a box model with large boxes and applying sparse proxy data. The relatively small number of CO2-3 data points in clustered locations leaves relatively small standard deviations, giving CO2-3 a disproportionate weighting in the model-data optimisation versus the other proxies. Therefore, we overcome the issue by scaling up the CO2-3 standard deviations and applying as default across all boxes and MIS time slices.

Re $\delta$13C and $\Delta$14C. The standard deviations are calculated from box-averaged published proxy data and shown in the supporting information. The standard deviations look large for these box-averaged and MIS-averaged values, because the boxes in the box model are large. The ocean box $\delta$13C standard deviation is now lower in the revised manuscript due to filtering out only Cibicides species, from the other reviewer comments.

The issue of box size and standard deviation is addressed again in the discussion of limitations of the study (P34 L7):

"However, given the large spatial coverage of the SCP-M boxes, data for large areas of the ocean are averaged, and some detail is lost. For example, in the case of the carbonate ion proxy, we apply a default estimate of standard deviation to account for the large volume of ocean covered by SCP-M's boxes relative to the proxy data locations, and to enable the normalisation of the carbonate ion proxy data in a procedure that uses the data standard deviation as a weighting. Despite this caveat, we argue that the model-data experiment results provide a good match to the data across the various

atmospheric and ocean proxies as shown in Figs 9-11."

Re Figs 9-11. The standard deviation ranges for CO2-3 and $\delta$13C are now narrower following the improvements we have made, which improves the resolution of Figs 9-11. In addition, we have expanded y-axes where we can to help with reading the figures.

RC: 3) References: In general I find that only a few references are used over and over and sometimes not appropriately. A few additional references are included in this review. Please note the typo throughout the document in "Ridgwell".

AC: Thanks, we've now added the references suggested by the reviewer, throughout the manuscript, and we corrected the typo for Ridgwell throughout.

References added following this reviewers' comments:

Watson et al., 2000 Joos et al, 2004 Tarnocai et al, 2009 Ganopolski et al., 2010 Menviel et al., 2012 Menviel and Joos, 2012 Brovkin et al, 2012 Jaccard et al, 2013 Schneider et al., 2013 Menviel et al, 2015 Yu et al, 2016 Ganopolski and Brovkin, 2017 Lindgren et al, 2018 Mauritz et al, 2018 Yamamoto et al, 2019 Treat et al, 2019

Specific comments:

RC: 1) Abstract: The first line does not make sense. Please reformulate. L. 3 Please add "SO" in front of "biological productivity"

AC: Re-formulated as: "We conduct a model-data analysis of the marine carbon cycle to understand and quantify the drivers of atmospheric CO2 during the last glacial cycle".

Southern Ocean added to the sentence P1 L3.

RC: 2) Introduction: - L.15-19: please be more specific. Instead of "Ocean biology" you might want to refer to "iron fertilisation and its impact on nutrient utilisation", or changes in remineralisation depth (e.g. Kwon et al. 2009).

AC: text modified to (P2 L18):

"Hypotheses for an ocean biological role include the effects of iron fertilisation on biological export productivity (e.g. Martin, 1990; Watson et al., 2000; Martinez-Garcia et al., 2014), the depth of remineralisation of particulate organic carbon (POC) (e.g. Matsumoto, 2007; Kwon et al., 2009; Menviel et al., 2012), changes in the organic carbon:carbonate ("the rain ratio") or carbon:silicate constitution of marine organisms (e.g. Archer and Maier-Reimer, 1994; Harrison, 2000), and increased biological utilisation of exposed shelf-derived nutrients such as phosphorus (e.g. Menviel et al., 2012)."

RC: What do you mean by composite mechanisms?

AC: we have amended this to "the aggregate effects of several mechanisms" throughout the document

RC: It would be good to also introduce the numerous modelling studies that have been done on the topic of G-IG changes in pCO2, and notably transient simulations of the G-IG trying to understand the changes in pCO2 (e.g. Ganopolski & Brovkin 2017, Menviel et al., 2012).

AC: Thanks, we have added to our introduction (P2 L23):

"Several studies have attempted to solve the problem of glacial-interglacial CO2 by modelling either the last glacial-interglacial cycle in its entirety, or multiple glacial-interglacial cycles (e.g. Ganopolski et al., 2010; Menviel et al., 2012; Brovkin et al., 2012; Ganopolski and Brovkin, 2017). These studies highlight the roles of orbitally-forced Northern Hemisphere ice sheets in the onset of the glacial periods, and important feedbacks from ocean circulation, carbonate chemistry and marine biological productivity throughout the glacial cycle (Ganopolski et al., 2010; Brovkin et al., 2012; Ganopolski and Brovkin, 2017). Menviel et al. (2012) modelled a range of physical and biogechemical mechanisms to deliver the full amplitude of atmospheric CO2 variation in the last glacial-interglacial cycle, using transient simulations with the Bern3D

model. According to Brovkin et al. (2012), a ∼50 ppm drop in atmospheric CO2 early in the last glacial cycle was caused by cooling sea surface temperatures (SST), increased Northern hemisphere ice sheet cover, and expansion of southern-sourced abyssal waters in place of North Atlantic Deep Water (NADW) formation. Ganopolski and Brovkin (2017) modelled the last four glacial cycles with orbital forcing as the singular driver of carbon cycle feedbacks. They described the "carbon stew", a feedback of combined physical and biogeochemical changes in the carbon cycle, to drive the last four glacial-interglacial cycles of atmospheric CO2."

And also, a few lines down to explain how our approach differs (P3 L23):

"Our modelling approach differs from other model studies of the last glacial-interglacial cycle (e.g. Ganopolski et al., 2010; Menviel et al., 2012; Brovkin et al., 2012; Ganopolski and Brovkin, 2017), in that we constrain several physical processes from observations (SST, sea level, sea-ice cover, salinity, coral reef fluxes of carbon), then solve for the values of model parameters for ocean circulation and biology based on an optimisation against atmospheric and ocean proxy data. " And at P8 L14:

"Joos et al. (2004), Ganopolski et al. (2010), Menviel et al. (2012), Menviel and Joos (2012), Brovkin et al. (2012) and Ganopolski and Brovkin (2017) provide coverage of the termination period with transient simulations of the last glacial-interglacial cycle, using intermediate complexity models (more complex than our model). "

RC: 3) Methods: - Variables included in the model: surely the model includes Dissolved Inorganic Carbon.

AC: yes, the model includes DIC and we have added DIC to the sentence.

RC: By "CO2", do you mean atmospheric CO2?

AC: yes, we have added "atmospheric" to the sentence at P3 L33.

RC: Does the model really includes "carbonate ions" as a prognostic tracer?

AC: Yes. SCP-M calculates CO2-3 concentration in umol kg-1, by calculating the three species of DIC. First, pCO2 is calculated using the method of Follows et al (2006) which takes as inputs DIC, alkalinity, pH, SST, salinity and phosphorus in each box in the model. Then H2CO3, HCO3- and CO2-3 are calculated using coefficients for the solubility of CO2 (K0) and coefficients for carbonic acid of K1 and K2 using Lueker et al (2000). In the model documentation paper (O'Neill et al, 2019) the SCP-M model estimates for CO2-3 in a modern ocean setting are demonstrated to align with modern data from the ocean, using data from Key at al (2004).

We have added a summary sentence to describe this, in section 2.1 "Model description" on P4.

RC: p4, L. 2: please refer to section 2.2.1 and Figure 2.

AC: Added

RC: p7: I am very confused by the treatment of the terrestrial biosphere in the model and the paragraph L. 19-27. It reads like there is an interactive terrestrial module. But how can NPP be calculated with significance if there is no atm. Temperature or precipitation in the model?

Our box model applies a simple representation of the terrestrial biosphere, whereby biological productivity responds to carbon fertilisation. Therefore, CO2 is the driver of terrestrial biosphere productivity in this model. We use a two-box terrestrial box model scheme, presented in Harman et al (2011). The inputs are starting estimates of net primary productivity (NPP), the terrestrial biosphere carbon stock, plant respiration rate and atmospheric CO2. The approach of Harman et al (2011) is to split the terrestrial biosphere into two boxes, a fast-response (grasslands and grassy components of savannah systems) and a slow-response (woody trees) component. In this model, the productivity is mostly focussed on the plants/grasses component.

The formula is shown in the model documentation paper (O'Neill et al, 2019) and Harman et al (2011), and extract is reproduced here:

dAtCO2/dt = −NpreRP[1+$\beta$LN(AtCO2)] + Cstock/k + Dforest

Where Npre is NPP at a reference pre-industrial level of atmospheric CO2, RP is a parameter to split NPP between short-term terrestrial biosphere carbon stock and the longer term stock (Cstock1 and Cstock2). B is a parameter with a value typically in the range 0.4-0.8 (Harman et al, 2011). Cstock is the carbon stock in each terrestrial biosphere box, k is the respiration timeframe for each box. Dforest is the prescribed rate of deforestation emissions for present day simulations and projections. A terrestrial biosphere fractionation factor is applied for the carbon isotopes.

Harman et al (2011) model the terrestrial biosphere primarily as a function of atmospheric CO2. They also incorporate an optional temperature dependency. This is the same approach used in the simplest 4Box terrestrial biosphere module of the Bern Simple Carbon Model (Strassman and Joos, 2018; Seigenthaler and Joos, 1992; Kicklighter et al, 1999; Meyer et al, 1999), and described by Enting (1994) – although we understand that there are various terrestrial biosphere modules applied with the Bern models, and most are more complex. As far as we can discern, the simple carbon fertilisation approach is also used in Jelstch-Thommes et al (2019), which also applies the simplest 4Box terrestrial biosphere of the simple Bern model.

There are other possible drivers of the NPP – temperature, precipitation, soil nutrient levels. In the context of our simple carbon cycle model, we are mainly interested in CO2. We don't model atmospheric temperature, and if we were to try to incorporate atmospheric temperature as a driver of terrestrial biosphere, we would also need to incorporate it for terrestrial weathering. There is a limit to how much detail we want to include in the model given we are conducting many simulations (∼80,000) in our model-data optimisations across the MIS of the last glacial-interglacial cycle.

We do note that there are studies devoted to determining whether the CO2 fertilisation effect or climate is the dominant control on terrestrial biosphere NPP and the size of the

terrestrial biosphere carbon stock. According to Hoogakker et al (2016), CO2 fertilization, rather than climate, is the primary driver of lower glacial net primary productivity by the terrestrial biosphere, accounting for around 85% of the reduction in global NPP at the LGM. Kaplan et al (2002) also concluded that over glacial-interglacial timescales, global terrestrial carbon storage is controlled primarily by atmospheric CO2, while the climate has more influence on the isotopic composition. Otto et al (2002) also found that the CO2 fertilization effect is mostly responsible for the total increase in vegetation and soil carbon stocks since the last glacial maximum. Kohler et al (2010) prioritised CO2 fertilisation as the driver of terrestrial biosphere in their "control" main simulation scenario for glacial-interglacial cycles over the last 740 kyr, but also ran scenarios with a climatic driver for the terrestrial biosphere to estimate the effects of "fast" climate changes on atmospheric $\delta$13C. Other studies arguing that atmospheric CO2 is an important, or is the main driver of terrestrial biosphere productivity include Kicklighter et al (1999), Joos et al. (2001), Schimel et al. (2015), Sitch et al. (2008), Arneth et al (2017)). This view has been contested by Francois et al (1999) and van der Sleen et al. (2015).

Given we don't model the atmospheric temperature or precipitation, we saw limited additional benefit to introduce them into our model of the terrestrial biosphere, although it would not be difficult to do this. Finally, given that CO2 and atmospheric temperature co-vary closely, across glacial cycles, it seems of limited benefit to split these effects out in our simple carbon cycle modelling exercise. For example, Meyer et al (1999) found similar results for modelling carbon uptake in the terrestrial biosphere whether only CO2 fertilisation, or CO2 fertilisation + climate, were included as drivers of NPP – but noting this was not tested for the LGM.

In summary, our aim is not to contribute new findings on the terrestrial biosphere, but we present the behaviour of the terrestrial biosphere in our manuscript to confirm that our exhaustively multi-proxy constrained model-data output is consistent with the range of literature estimates of variations in the terrestrial biosphere in the last

glacial-interglacial cycle and LGM-Holocene period, and we show this. For example, our experiment shows a change in the terrestrial biosphere carbon stock of +630 PgC between the MIS 2 (LGM) and MIS 1 (Holocene) period. This compares with other estimates of +540 PgC (Brovkin et al, 2007), +~820-850 PgC (Joos et al, 2004) – with the majority by CO2 fertilisation, +~500 PgC (Kohler et al, 2010), +~500 PgC (Brovkin et al, 2012), +850 PgC (Jeltsch-Thommes et al, 2019), +511 +/- 289 PgC (Peterson et al, 2014), +378 +/- 88 PgC (Menviel et al, 2016). Another estimate of the LGM-Holocene terrestrial biosphere change is 550-694 Pg C, which our result of 630 Pg C sits comfortably within (Prentice et al, 2011) Our estimate is actually towards the upper end of the literature ranges, suggesting if anything we could exaggerate the effects of the terrestrial biosphere from the LGM to the Holocene period, with perhaps little to gain by splitting out temperature and precipitation effects. If did, we would probably also need to consider other important features such as soil nutrients and local humidity. While we have a simple, but explicit two-box representation of the terrestrial biosphere, we don't believe that this detracts from our model-data results, as shown in Figures 9-11 and Figure 12 specifically for the terrestrial biosphere.

If there is some reason to examine the terrestrial biosphere in more detail, we suggest for our study this would be done simply by a sensitivity, as applied in Menviel et al (2016) with regard to C3/C4 plants and the relative proportional influence of C3 and C4 plants on terrestrial biosphere $\delta$13C fractionation.

We have added some text to explain that we have a simplified representation of the terrestrial biosphere employing CO2 fertilisation, and that we don't take account of temperature and precipitation, in the methods section, P5 L24. This also includes discussion of the isotopic fractionation factor in response to one of the other reviewers:

[revised manuscript text omitted]

We have also updated the discussion of our model results for the terrestrial biosphere, to provide a bit more detail and some additional references (Section 5.3), plus an additional caveat in the "advantages and limitations section" (P34, L18). "Furthermore, we apply a simple representation of the terrestrial biosphere in our model-data experiments, relying primarily on atmospheric CO2 as the driver for NPP. This approach

provided reasonable results for the terrestrial biosphere carbon stock and NPP, on the whole, but may miss some detail in the terrestrial biosphere during the last glacial-interglacial cycle."

RC: Why is "tundra" discussed with such emphasis in this paragraph?.

AC: Thanks for picking up on this. We have substantially revised this paragraph as follows (P10 L25):

"The terrestrial biosphere module in SCP-M does not explicitly represent the carbon stored in buried peat, permafrost and also cold-climate vegetation that may have expanded its footprint in the glaciation, such as tundra biomes (e.g. Tarnocai et al., 2009; Ciais et al., 2012; Schneider et al., 2013; Eggleston et al., 2016; Ganopolski and Brovkin, 2017; Treat et al., 2019). The freezing and burial of organic matter across the glacial cycle may significantly imprint the terrestrial biosphere CO2 size and $\delta$13C signature (Tarnocai et al., 2009; Ciais et al., 2012; Schneider et al., 2013; Eggleston et al., 2016; Ganopolski and Brovkin, 2017; Mauritz et al., 2018; Treat et al., 2019). Schneider et al. (2013) and Eggleston et al. (2016) both observed a permanent increase in atmospheric $\delta$13C during the last glacial cycle, of âĹij0.4‰ and attributed its cause likely due to soil storage of carbon in peatlands which were buried or frozen as permafrost as the glacial cycle progressed. Ganopolski and Brovkin (2017) incorporated permafrost, peat, and buried carbon into their transient simulations of the last four glacial- interglacial cycles, observing that these features dampened the amplitude of glacial-interglacial variations in terrestrial biosphere carbon stock, in the CLIMBER-2 model. As a crude measure to account for this counter-CO2 cycle storage of carbon in the terrestrial biosphere and frozen soils, we force the terrestrial biosphere productivity parameter in SCP-M in the range âĹij+5-10 PgC yr−1, increasing into the LGM (MIS 2), and maintained in the Holocene (MIS 1). We maintain the forcing of the terrestrial biosphere in the Holocene, as the posited effects of buried peat and permafrost storage of carbon on atmospheric CO2 and $\delta$13C during the lead-up and into the LGM, were likely not fully reversed after the glacial termination (Tarnocai et al., 2009; Eggleston et al., 2016; Mauritz et al., 2018; Treat et al., 2019), and were partially or wholly replaced by other soil stocks of carbon (e.g. Lindgren et al., 2018). SCP-M calculates net primary productivity (NPP) using this productivity input parameter, as a function of carbon fertilisation (Harman et al., 2011)."

RC: Tundra is not an "inert" carbon pool

AC: we've modified the sentence as per above excerpt to refer to carbon stored in frozen peat, permafrost soils.

RC: and I don't think "permafrost" is a vegetation type

AC: We've modified this sentence as per above excerpt, to remove the reference to permafrost as a vegetation type.

RC: What is "pre-carbon fertilisation"?

AC: This is just the Npre in the equation for NPP from the model documentation, re-produced above. We can refer to this as "undisturbed" (by CO2) NPP. The equations for NPP takes an input value Npre, which is subsequently varied due to any change in atmospheric CO2. This is our model representation of CO2 fertilisation of the terrestrial biosphere.

RC: p8: what is the point of Table 1 if all the values of GOC, AMOC, biology are the same? It would be interesting to mention the PI control values though.

AC: Thanks, we've consolidated Table 1 to show the MIS model-data experiment ranges and the PI control values.

RC: - p10-11: The 'depth issue" should also be discussed in 2.3.1 and 2.3.2.

AC: Re 2.3.1 – there is a much greater coverage of $\delta$13C and $\Delta$14C data for the ocean boxes so we have not applied a default weighting for those data in our model-data optimisation. For CO2-3, a problem presents because there are only 1 or 2 data points in some boxes, and they are clustered near the box boundary, so we end up with

unrepresentative data for some boxes for CO2-3. So, we applied a larger weighting for CO2-3 data, as discussed in 2.3.2.

4) Discussion:

RC: p20, L. 3-6: It is not what the simulations tell you, but the proxy data!

AC: We've removed this reference to the modelling and replaced with reference to the proxy data shown in Figure 4 (P23, L7).

RC: p21, L. 1-2: This is wrong → you are forcing your model with SST, Sea-ice. . ... so all these factors contribute to the pCO2 decrease. The experiments show that changes in oceanic circulation and SO biological productivity also contribute to that pCO2 decrease.

AC: We have reworded this sentence to list the full set of changes modelled (P24 L7)

RC: Please take into consideration that G-IG pCO2 changes have been previously successfully simulated with models of intermediate complexity (e.g. e.g. Ganopolski & Brovkin 2017, Menviel et al., 2012) and box models.

AC: We have added a sentence at the start of the discussion to reference these studies (P23, L5) and they are referenced throughput the Discussion.

RC: p21, L. 3-4: I don't understand the meaning

AC: This sentence has been reworded (P24 L6).

RC: p21, L. 7: Might want to check Piotrowski et al., 2008, Yu et al., 2016. (Nat. Geo).

AC: We have picked up the citation of Yu et al (2016) in reference to AMOC in the MIS 4, a little further down in the manuscript (P29 L28). We have added a reference to Piotrowski et al (2009) in the same place (P29 L29).

We have also added the Piotrowski et al (2009) $\delta$13C data to our dataset and cited it in the manuscript (Table 2).

RC: p21, L. 10 –p22, L. 5: This section really has to be discussed in light of all the work that has been done on the impact of iron fertilisation in the Southern Ocean. Some work on the topic: Watson et al., 2000, Nature; Jaccard et al., 2013, Science; Yamamoto et al., 2019, Climate of the Past;

AC: Text added (P31 L2):

"Our finding of increased biological productivity, while mostly constrained to MIS 2 and MIS 4, and a modest contributor to the overall glacial $CO_2$ drawdown, corroborates proxy data (e.g. Martinez-Garcia et al., 2014; Lambert et al., 2015; Kohfeld and Chase, 2017) and recent model-data exercises (e.g. Menviel et al., 2016; Muglia et al., 2018; Khatiwala, 2019). Martin (1990) pioneered the "iron hypothesis", which invoked the increased supply of continent-borne dusts to the Southern Ocean in glacial periods. Increased dust supply stimulated more plankton productivity where plankton were bio-limited in nutrients supplied in the dust, such as iron (Martin, 1990). Since then, the iron hypothesis has retained an important place in the debate over glacial-interglacial cycles of $CO_2$. Watson et al. (2000) took experimental data on the effects of iron supply on plankton productivity in the Southern Ocean (Boyd, 2000) and applied this to a carbon cycle model across glacial- interglacial cycles. Their modelling, informed by the ocean experiment data, suggested that variations in the Southern Ocean iron supply and plankton productivity could account for large (âĹij40 ppm) swings in atmospheric $CO_2$, with peak activity in the last glacial cycle at MIS 2 and MIS 4. Debate has continued over the magnitude of the contribution of Southern Ocean biological productivity to the glacial $CO_2$ drawdown. According to Kohfeld et al. (2005), based on sediment data, the Southern Ocean biological productivity mechanism could account for no more than half of the glacial $CO_2$ drawdown. Others emphasise that Southern Ocean biological export productivity fluxes may have been weaker in the LGM, in absolute terms, but that with weaker Southern Ocean upwelling, the iron-enhanced productivity contributed to a stronger biological pump of carbon and was a major contributor to the LGM $CO_2$ drawdown (Jaccard et al., 2013; Martinez-Garcia et al., 2014; Yamamoto et al., 2019).

" RC: p22, L. 18: "sea-ice cover"

AC: Thanks, corrected

RC: p23, L. 1-12: Figure 13 is interesting but care has to be taken here given the large size of the "boxes". This should at least be discussed in light of previous modelling studies on the subject (e.g. Menviel et al., 2015, GBC).

AC: This figure has changed from the original manuscript due to a change in our data method for $\delta$13C, stemming from the other reviewer comments. We are now only using Cibicides species $\delta$13C data, and we re-ran our model-data experiments. There are only slight variations to our model-data results. However, a narrower spread of standard deviations of the $\delta$13C data necessitates us to change this Figure. We do think it's an important figure that provides some insights into our model, the results in this manuscript and how they might differ from other studies that simply rely on qualitative and simple statistical analysis of proxy data (without models).

Text added P29 L3:

"These observations from Fig. 13 could be exaggerated in SCP-M due to the large size of its ocean boxes and therefore relatively large spread of $\delta$13C values and standard deviations for each box. In addition, this experiment may reflect idiosyncrasies in the SCP-M model design and its simple parameterisation of ocean circulation and mixing. A finer resolution model may show a greater sensitivity of the ocean box $\delta$13C to variations in ocean circulation. Menviel et al. (2015) analysed the sensitivity of ocean and atmospheric $\delta$13C to variations in NADW, AABW and North Pacific Deep Water (NPDW) formation rates, in the context of rapid changes in atmospheric $\delta$13C and $CO_2$ observed during the last glacial termination. Their modelling, using the more spatially-detailed LOVECLIM and Bern3D models, showed modest but location-dependent sensitivities of ocean $\delta$13C to slowing ocean circulation, and particular sensitivity to AABW. These models are much higher resolution and show greater sensitivity of $\delta$13C to ocean circulation over depth intervals not differentiated in the SCP-M boxes,

but also quite a variation across the LOVECLIM and Bern3D models. However, our simple experiment illustrated in Fig. 13 does highlight the potential for important changes in the ocean during glacial-interglacial periods to go unnoticed, when focussed on one set of ocean proxy data and without validation by modelling."

[Figure]

**Supplement:**

[revised manuscript text omitted]

---

## Author Comment (AC2) · 11 Jun 2020

CP reviewer comments #2 and author responses

AC: We thank the reviewer for their comments, suggestions and input into this manuscript. These comments make a substantial contribution to improving the quality of our work, particularly with reference to our treatment of the oceanic $\delta$13C data. Please see below our responses to the individual comments.

We have made reference to changes to the manuscript, which are included as a supplement to the author comments, in track changes. Page and line references below

[Figure]

refer to locations in the revised document with track changes.

RC 1. The authors base their paper on a recently published carbon cycle box model (O'Neill et al. 2019). They provide a brief description of the model but I found that this manuscript would benefit better description of some of the key parameters that are quite important to this paper, such as the controls on Z (biological productivity). It was very unclear to me on first reading how values of Z were ascertained.

AC: To address this comment we have added the following text to (P3, L30). In addition to the biological productivity, it includes a bit more detail on some other processes, stemming from the other reviewer comments:

"We used the SCP-M carbon cycle box model in our model-data experiment (O'Neill et al., 2019). In summary, SCP-M contains simple parameterisations of the major fluxes in the Earth's surface carbon cycle (Fig. 1). SCP-M incorporates the ocean, atmosphere, terrestrial biosphere and marine/continental sediment carbon reservoirs, weathering and river fluxes, and a number of variables including atmospheric $CO_2$, DIC, phosphorus, alkalinity, carbon isotopes (13C and 14C) and the carbonate ion.

SCP-M calculates ocean $pCO_2$ using the equations of Follows et al. (2006), and applies the first and second "dissociation constants" of carbonic acid estimated by Lueker et al. (2000), to calculate $HCO-$ and $CO2-3$ concentrations, respectively, in units of $\mu$mol kg$-1$, in each ocean box. The model employs partial differential equations for determining the concentration of elements in each box, with each box represented as a row and column in a matrix. In this paper, we extend SCP-M by incorporating a separate basin for the combined Pacific and Indian Oceans (Fig. 1), following the conceptual model of Talley (2013), to incorporate modelling and proxy data for those regions of the ocean. This version of SCP-M consists of 12 ocean boxes plus the atmosphere and terrestrial biosphere. SCP-M splits out depth regions of the ocean between surface boxes (100-250m average depth), intermediate (1,000m average depth), deep (2,500m average depth) and abyssal depth boxes (3,700 (Atlantic) - 4,000m (Pacific-
Indian) average depth). The Southern Ocean is split into two boxes, including a polar box which covers latitude range 60-80 degrees South (box 12 in Fig. 1) and sub polar boxes in the Atlantic (box 7) and Pacific-Indian (box 12) basins, which cover latitude range 40-60 degrees South. See O'Neill et al. (2019) for a discussion of the choice of box depth and latitude dimensions.

The major ocean carbon flux parameters of interest in this model-data study, are global ocean circulation (GOC), $\Psi 1$, Atlantic meridional overturning circulation (AMOC), $\Psi 2$, and ocean biological export productivity, Z. The ocean circulation parameters $\Psi 1$ and $\Psi 2$ are simply prescribed in units of Sverdrups (Sv, 106 m3 s−1). Ocean biological export productivity Z is calculated using the method of Martin et al. (1987). The biological productivity flux, at 100m depth, is attenuated with depth for each box according to the decay rule of Martin et al. (1987). Each sub surface box receives a biological flux of an element at its ceiling depth, and loses a flux at its floor depth (lost to the boxes below it). The difference is the amount of element that is remineralised into each box. The input parameter is the value of export production at 100m depth, in units of mol C m−2 yr−1 as per Martin et al. (1987). Equation (1) shows the general form of the Martin et al. (1987) equation:

$F=F100(d/100)b$ (1)

Where F is a flux of carbon in mol C m−2 yr−1, F100 is an estimate of carbon flux at 100m depth, d is depth in metres and 20 b is a depth scalar. In SCP-M, the Z parameter implements the Martin et al. (1987) equation. Z is an estimate of biological productivity at 100m depth (in mol C m−2 yr−1), and coupled with the Martin et al. (1987) depth scalar, controls the amount of organic carbon that sinks from each model surface box to the boxes below. Each subsurface ocean box receives a flux of carbon from the box above it, at its ceiling depth (also the floor of the overlying box), and loses carbon as a function of the depth of the bottom of the box. Remineralisation in each box is accounted for as the difference between the influx and out-flux of organic carbon.  
[revised manuscript text omitted]

RC: Figure 1 – this graphic, while nice and colourful, is challenging for reading the actual numbers and symbols (especially the white ones which do not show up at all on my colour print). Readability is more important than colour! I suggest making box numbers, symbols all BLACK using larger fonts so that they are readable.

AC: Thanks. To address this comment, for Figure 1 we have upgraded the box number font size, in bold and black, and we increased font sizes for text elsewhere in the diagram. We would like to retain the colour coding of parameter symbols with their associated flux arrows. To address the RC, we have expanded the font size of these to help with readability (please see attached revised manuscript at Figure 1).

RC: Pg 5 lines 15-17. This sentence seems out of place: "Therefore, our modelling excludes the last glacial termination (âĹij11-18 ka)." Should it occur before the previous sentence?

AC: Thanks, we have relocated the misplaced sentence (P8, L14).

RC: Section 2.2.1 Model forcings: Although the authors ultimately conclude that sea ice cover – as a barrier mechanism constraining air-sea $CO_2$ exchange – is not that important, the authors should emphasize limitations of their use of the ice core sea ice proxy. First, this proxy is non-linear, so their simulations probably over estimate early (MIS5d) sea ice cover and underestimate later (MIS4-2) sea ice cover. This point is made very clearly by Wolff et al. 2010 (and supports) the authors' assertion that the barrier effect of sea ice early in the glaciation is probably small.

Text added in the methodology section (P8, L28).

"Our treatment of sea-ice cover is simply as a regulator of air-sea gas exchange in the polar ocean surface boxes. This treatment misses important linkages that likely exist between sea-ice cover and Southern Ocean upwelling, wind-sea surface interactions,

NADW formation, deep ocean stratification, nutrient distributions and biological productivity (Morrison and Hogg, 2013; Ferrari et al., 2014; Jansen, 2017; Kohfeld and Chase, 2017; Marzocchi and Jansen, 2017). Furthermore, our linear application of the sea-ice proxy data of Wolff et al. (2010) to our air-sea gas exchange parameter may serve to overestimate its effect on the model results early in the glacial period (MIS 5d), and underestimate it during MIS 2-4 (Wolff et al., 2010)."

RC: Furthermore, it is worth pointing out somewhere in the discussion that this modelling exercise only examines the potential role of sea ice as a barrier to CO2 exchange, and not its synergistic (and likely more important) roles in influencing nutrient distributions, marine productivity, and a trigger for deep ocean circulation changes. The authors state this somewhat in their "Advantages and limitations" section, but I think that this point could be made more explicitly.

AC: To address this comment, we have added the following (P27, L23):

"This finding may reflect our approach to treat polar sea-ice cover simply as a regulator of the rate of air-sea gas exchange in the polar oceans. This approach may neglect other effects of sea-ice cover including as a trigger for changes in Southern Ocean upwelling, NADW formation rates, deep ocean stratification, nutrient distributions and biological productivity (Morrison et al, 2011; Brovkin et al, 2012; Ferrari et al, 2014; Kohfeld and Chase, 2017; Jansen, 2017; Marzocchi and Jansen, 2017). For example, Brovkin et al (2012) found that in the CLIMBER-2 model, atmospheric CO2 was more sensitive to sea ice cover when it was linked to weakened vertical diffusivity in the Southern Ocean of tracers such as DIC, thereby reducing outgassing of CO2."

RC: Another larger issue that the sea ice proxy highlights is the spatial heterogeneity of the Southern Ocean and how the model results are linked with reality: the sea ice proxy likely represents changes very close to the continent and early glacial changes in sea ice are not well reproduced in the few long sea ice records that are found near the APF. This not only suggests that a barrier effect of sea ice would be limited to only part

of the Southern Ocean, it points to larger issues with treating the Southern Ocean as one box, with an unclear delineation of how much of the S. Oc. this box is presumed to cover. If the box is supposed to ONLY cover those areas close to the continent where AABW and Circumpolar Deepwater processes that influence GOC are most important, then the authors' main conclusion of increases in S. Oc. export production aren't well supported by paleoceanographic data which show reductions in export South of the APF for the majority of the glacial cycle between MIS5d and MIS2. Some discussion of what the Southern Ocean box actually represents - and this potential disconnect with paleoceanographic data - is warranted.

AC: SCP-M has two Southern Ocean boxes in each basin: a polar and sub polar Southern Ocean box. These are: polar Southern Ocean box for both basins (box 12 in Figure 1) which covers 60-80 deg S, sub polar Atlantic box (box 7 in Figure 1, 40-60 deg S) and sub polar Pacific-Indian box (box 11, 40-60 deg S). The sea ice forcing/air-sea gas exchange is undertaken for the polar Southern Ocean box. The biological export productivity experiment is undertaken for the sub polar Southern Ocean boxes in each basin, as per the regions highlighted for increased glacial period biological activity by Martinez-Garcia (2014) and Lambert et al (2015), Shoenfelt et al (2018). Put another way, our Southern Ocean biological flux experiments are not concerned with the APF, but with the open Southern Ocean box.

We have added the following text in the model description in Section 2.1 (P5, L6):

"The Southern Ocean is split into two boxes, including a polar box which covers latitude range 60-80 degrees South (box 12 in Fig. 1) and sub polar boxes in the Atlantic (box 7) and Pacific-Indian (box 12) basins, which cover latitude range 40-60 degrees South. See O'Neill et al (2019) for a discussion of the choice of box depth and latitude dimensions."

We have also added the following text in the first paragraph of Section 2.2.1 Model parameters and forcing (P8, L26):

"Note the polar Southern Ocean box which is forced with reduced air-sea exchange, is separate from the sub polar Southern Box in which the biological export productivity parameter is varied in the model-data experiment."

RC: Throughout the paper the authors refer to "abyssal" and "deep" water masses for all basins, but I was never able to find the depth cut-offs that were used to distinguish these depths in the different basins. Please put them in the figure captions and text (not just supplemental information, if it is there.)

AC: We have added the following text in the model description in Section 2.1 (P5, L4):

"SCP-M splits out depth regions of the ocean between surface boxes (100-250m average depth), intermediate (1,000m average depth), deep (2,500m average depth) and abyssal depth boxes (3,700-4,000m average depth). The Southern Ocean is split into two boxes, including a polar box which covers latitude range 60-80 degrees South (box 12 in Fig. 1) and sub polar boxes in the Atlantic (box 7) and Pacific-Indian (box 12) basins, which cover latitude range 40-60 degrees South. See O'Neill et al (2019) for a discussion of the choice of box depth and latitude dimensions."

We have also added depth references to the Caption on Figure 1, Figures 5-7, Figures 9-11

RC: The authors discuss briefly that previous studies have only used the C. wuellerstorfi data to reconstruct deep ocean $\delta$13C (Peterson et al. study; Kohfeld and Chase study). Which data did these authors select from Oliver et al. (2010)? They mention only using "deep" and "abyssal" sites (again, depths undefined) on page 11, but they do not indicate whether they have filtered the data to only include C. wuellerstorfi (or even Cibicidoides spp), which they SHOULD be doing if they haven't. Otherwise, the changes in $\delta$13C described on page 12 are invalid as descriptions of deep ocean circulation changes in $\delta$13C.

AC: The work of Oliver et al (2010) was to aggregate ocean $\delta$13C data, estimate and

correct for species-related problems or errors, and thereby provide a dataset to be used for assessing ocean circulation changes. The Oliver et al (2010) dataset is split into Planktonic and Benthic species data. We had used the benthic datasets. We had given Oliver et al (2010) the benefit of the doubt, in our first manuscript, as they had gone to substantial effort to produce a $\delta$13C dataset for paleooceanographic purposes.

However, on the suggestion of the reviewer, we have revisited the data and filtered the Cibicides species for the $\delta$13C dataset, which also includes Cibicides data contributed by Govin et al (2009) and Piotrowski et al (2009).

We have re-constructed our ocean $\delta$13C database using only the Cibicides species $\delta$13C data, re-calibrated the model for a new set of (penultimate) interglacial starting data, and re-run all of our model-data experiments. The revised manuscript (attached) incorporates these changes in the text, charts and tables.

The data section is updated as follows (P13, L4):

"Oliver et al. (2010) compiled a global dataset of 240 cores of marine $\delta$13C data encompassing benthic and planktonic species over the last âĹij150 kyrs. Oliver et al. (2010) observed considerable uncertainties associated with the broad range of species included, particularly for the planktonic foraminifera. By comparison, Peterson et al. (2014) aggregated marine $\delta$13C for the LGM and late Holocene periods, as time period averages, exclusively sampling benthic C. wuellerstorfi data, which is a more reliable indicator of marine $\delta$13C (Oliver et al., 2010; Peterson et al., 2014). To narrow the range of uncertainty, we constrain our use of marine $\delta$13C data to the deep and abyssal benthic Cibicides species foraminifera samples in the Oliver et al. (2010) dataset, supplemented with Cibicides species $\delta$13C proxy data from Govin et al. (2009) and Piotrowski et al. (2009) (Table 2). Figure 3 shows the $\delta$13C data locations from Oliver et al. (2010), which are concentrated in the Atlantic Ocean. We mapped and averaged the carbon isotope data into SCP-M's boxes on depth and latitude coordinates (Fig. 1), and averaged for each MIS time slice."

RC: On Page 12, the authors qualitatively describe the differences between "deep" and "abyssal" changes in $\delta$13C. Why leave this discussion qualitative, when the data are available and quantification would be hugely useful. These data in the Pacific that are described are the data that pin the authors' entire argument surrounding early changes in GOC. I think that this warrants a bit more quantification of these data (once species other than Cibicidoides are filtered out of the dataset). I would be interested to know if the differences between deep and abyssal $\delta$13C in the Indo-pacific are statistically significant, and I think plots of the probability distribution functions of these data would be very useful.

AC: Thanks. Following from this comment we've investigated a number of ways to analyse the data. We have focussed on the $\delta$13C data (Cibicides, as above) only, for this analysis, as there is continuous coverage for deep and abyssal boxes for the Atlantic and Pacific-Indian oceans across all of the MIS stages we are interested in.

We applied some tests for statistical significance of the various boxes throughout the MIS stages. We used a Welch's paired unequal variance t-test for statistically different mean $\delta$13C between deep and abyssal boxes, and also for differences in the offsets in mean $\delta$13C between deep and abyssal boxes, between MIS stages. We have added this to the supplementary information file and referenced its location from the main document (P17 L13).

As per the reviewer comment, we first plot the distribution of mean $\delta$13C values for each of the deep and abyssal boxes across the MIS stages.

Figure 1: Distribution histograms of $\delta$13C data for the Pacific-Indian (left column) and Atlantic Ocean (right column) deep (100/1,000-2,500m) and abyssal (>2,500m) boxes. Plots also show the mean $\delta$13C for each box (vertical dashed lines), and the calculated offset between the deep and abyssal mean $\delta$13C values (CP_RC2_Fig1.png). (see below for Figure 1)

We applied a Welch's paired t-test to test for statistical independence of the means of

$\delta$13C in the deep and abyssal ocean boxes for Atlantic and Pacific-Indian, within each MIS. This returns p-values very close to zero for every box pair and every MIS. A p-value <0.05 means that we reject the null hypothesis that the abyssal and deep ocean boxes are statistically the same. That is, our deep and abyssal boxes in the model are statistically independent of each other, in terms of mean $\delta$13C. This simply confirms that our abyssal and deep ocean boxes are not the same in terms of mean $\delta$13C in each MIS.

Table 1: Tests for statistical independence of the mean $\delta$13C between deep and abyssal boxes (CP_RC2_Tab1.png). (see below for Table 1)

Given we discuss in the manuscript (qualitatively) the changes in the offset between deep and abyssal ocean $\delta$13C through the MIS, we can test to see if the changes in deep-abyssal offset from the penultimate interglacial (MIS 5e) to the glacial periods are statistically significant. The chart below shows the deep-abyssal offsets in $\delta$13C for the Pacific-Indian and Atlantic Ocean boxes through each MIS of the last glacial-interglacial cycle. We show the absolute deep-abyssal $\delta$13C offsets for Pacific-Indian and Atlantic Ocean boxes, for each MIS (columns). We also show the deep-abyssal $\delta$13C offsets relative to the penultimate interglacial in MIS 5e (lines).

The Pacific-Indian $\delta$13C offset shows a widening in MIS 5d, relative to MIS 5e, which is maintained until MIS 5a, and then begins a slow decline. The offset declines to a similar value to MIS 5e, by MIS 1 (the Holocene). The Atlantic deep-abyssal $\delta$13C offset does not increase meaningfully until MIS 4, and then peaks at MIS 2 (the LGM), before contracting at MIS 1 to a value almost the same as MIS 5e.

Figure 2: Offsets between mean deep and abyssal box $\delta$13C for each MIS in the last glacial-interglacial cycle for the Pacific-Indian (blue columns) and Atlantic Ocean (grey columns). Changes in the offsets from the penultimate interglacial (MIS 5e) are shown by the blue (Pacific-Indian) and grey (Atlantic) lines (CP_RC2_Fig2.png).(see below for Figure 2)

We further undertook Welch's paired T-tests for the independence of deep-abyssal offsets in mean $\delta$13C with respect of the penultimate interglacial period (MIS 5e), for the periods MIS 1-5e. The null hypothesis is that the deep-abyssal offset in mean $\delta$13C in each MIS is not statistically independent of MIS 5e (i.e. statistically the same and not supportive of a change in deep-abyssal $\delta$13C distribution that may be delivered by a changed ocean process). p-values>0.05 lead to the null hypothesis being accepted, whereas p-values <0.05 lead to the null hypothesis being rejected and confirm statistical independence of the deep-abyssal offsets relative to MIS 5e (perhaps supportive of a changed ocean distributive process in the glacial period). Deep-abyssal offsets for the Pacific-Indian during MIS 2-MIS5d are statistically independent of MIS 5e, supportive of a changed oceanic distribution of $\delta$13C throughout the glacial period. The MIS 1 Pacific-Indian deep-abyssal $\delta$13C offset is not statistically independent of MIS 5e, indicating a similar deep-abyssal $\delta$13C distribution between the last and penultimate interglacial periods. For the Atlantic Ocean, deep-abyssal mean $\delta$13C offsets are not statistically independent with respect to MIS 5e (p-value >0.05, accept null hypothesis), until the period MIS 2-4. Atlantic deep-abyssal mean $\delta$13C offset in MIS 1 is not statistically different from MIS 5e.

Table 2: Statistical tests for significance of difference in deep-abyssal $\delta$13C offsets versus penultimate interglacial (MIS 5e). 'Accept'/red is to accept the null hypothesis - no statistically significant difference, 'Reject'/green is to reject the null hypothesis – statistically significant difference with respect of MIS 5e (CP_RC2_Tab2.png). (see below for Table 2)

The statistical analysis above is helpful and provides support for our model-data experiment results – that GOC slowed in MIS 5d and AMOC slowed in MIS 4. However, we do want to make the point that our model-data results don't hang on one particular data point to deliver these findings, in any MIS. They are constrained and optimised with many observations. The model-data results in the first instance are telling us that, the many observational forcings we have imposed in each MIS (SST, salinity, sea-ice

cover proxy, coral reef carbonates) are not enough to deliver the change in atmospheric CO2, atmospheric and ocean $\delta$13C, D14C and CO23 proxy data. Changes from within the set of ocean circulation, mixing and/or biology parameters are needed. Note the result for GOC that is hinted at by the $\delta$13C data, that we model in our experiments, is sustained throughout the last glacial cycle, not just at MIS 5d.

The main point of our work, and what has taken substantial effort, is to undertake an exhaustive model-data optimisation using a carbon cycle box model and multiple atmospheric and ocean proxy data. The model-data results don't just rely on one data point, the results need to be the best fit for all the data used, in each MIS. This is where this model-data experiment differentiates itself from many others.

We have included the distribution plot and T-test table above, in the manuscript's Supporting Information. We make reference to this material in the manuscript when discussing the data charts in the "Data Analysis" section. This chart/table provide supplemental support to the model-data analysis, the latter being the focus of our manuscript. We feel that the manuscript is becoming very voluminous and we also think that this analysis would require its own section in the manuscript (However, it is presented in this response to the discussion (preserved online) and in the SI).

RC: Some type of quantification would also be very useful for the authors' description of the "transient drop in abyssal Atlantic ocean CO3= at MIS5b" on page 14. I was not convinced that this transient drop exists from the figure presented.

AC: Yes, the axes on these charts are a little difficult to decipher small changes in the data. We wish to show the range of shallow-deep CO23 data (not just deep-abyssal), as the pattern is quite interesting at the LGM-Holocene. Our suggestion is to add the changes in units for the pattern that we wish to describe (P17, L35):

"There is a modest drop in abyssal Atlantic Ocean CO2-3 at MIS 5b (-13 $\mu$mol kg$-1$ relative to MIS 5c), which coincides with a minor drop in abyssal Atlantic Ocean $\delta$13C (-0.19‰ and atmospheric CO2 (-14 ppm), suggesting a possible common link." RC:

Please note on the bottom of page 13 and top of page 14 that the authors mean to refer to Figure 7 (not 6) to describe carbonate ion concentration data.

AC: Thank you, we have corrected these references in the manuscript.

RC: Last sentence before Results section: Please cite the figures you are using to make these observations about changes in $\delta$13C and DD14C

AC: We have added the figure references

RC: Similar quantification would be useful in the comparison between the carbonate ion concentration model output and data in Figure 9 and in the discussion on page 16-17.

AC: Figure references for Figure 9 added to the text here

  References

Govin, A., Michel, E., Labeyrie, L., Waelbroeck, C., Dewilde, F., and Jansen, E.: Evidence for northward expansion of Antarctic Bottom Water mass in the Southern Ocean during the last glacial inception, Paleoceanography, 24, doi:10.1029/2008PA001 603, 2009.

Lambert, F., Tagliabue, A., Shaffer, G., Lamy, F., Winckler, G., Farias, L., Gallardo, L., and Pol-Holz, D.: Dust fluxes and iron fertilization in Holocene and Last Glacial Maximum climates, Geophysical Research Letters, 42, 6014–6023, 2015.

Martinez-Garcia, A., Sigman, D., H.Ren, Anderson, R., Straub, M., Hodell, D., Jaccard, S., Eglinton, T., and Haug, G.: Iron Fertilization of the Subantarctic Ocean During the Last Ice Age, Science, 343, 1347–1350, 2014.

Oliver, K., Hoogakker, B., Crowhurst, S., Henderson, G., Rickaby, R., Edwards, N., and Elderfield, H.: A synthesis of marine sediment core d13C data over the last 150 000 years, Climate of the Past, 6, 645–673, 2010.

Piotrowski, A., Banakar, V., Scrivner, A., Elderfield, H., Galy, A., and Dennis, A.: Indian Ocean circulation and productivity during the last glacial cycle, Earth and Planetary Science Letters, 285, 179–189, 2009.

Shoenfelt, E.M., Winckler, G., Lamy, F., Anderson, R.F., and Bostick, B.C. Highly bioavailable dust-borne iron delivered to the Southern Ocean during glacial periods. PNAS 115 (44) 11180-11185, 2018. https://doi.org/10.1073/pnas.1809755115.

Please also note the supplement to this comment:
https://www.clim-past-discuss.net/cp-2019-146/cp-2019-146-AC2-supplement.pdf

[Figure]

**Fig. 1.** Figure 1: Distribution histograms of $\delta$13C data for the Pacific-Indian (left column) and Atlantic Ocean (right column) deep (100/1,000-2,500m) and abyssal (>2,500m) boxes. Plots also show the mean $\delta$13C

| MIS | Abyssal-deep Pacific-Indian | | Abyssal-deep Atlantic | |
|---|---|---|---|---|
| | t-statistic | p-value | t-statistic | p-value |
| MIS5e | -7.0 | 0 | 5.1 | 0 |
| MIS5d | 16.1 | 0 | 9.8 | 0 |
| MIS5c | 13.0 | 0 | 7.8 | 0 |
| MIS5b | 9.5 | 0 | 6.3 | 0 |
| MIS5a | 13.2 | 0 | 6.9 | 0 |
| MIS4 | 24.0 | 0 | 17.6 | 0 |
| MIS3 | 23.3 | 0 | 21.6 | 0 |
| MIS2 | 18.8 | 0 | 31.6 | 0 |
| MIS1 | 14.2 | 0 | 11.9 | 0 |

**Fig. 2.** Table 1: Tests for statistical independence of the mean $\delta$13C between deep and abyssal boxes (CP_RC2_Tab1.png)

[Figure]

[Figure]

**Fig. 3.** Figure 2: Offsets between mean deep and abyssal box $\delta$13C for each MIS in the last glacial-interglacial cycle for the Pacific-Indian (blue columns) and Atlantic Ocean (grey columns). Changes in the off

| MIS | Pacific-Indian (vs MIS 5e) | | | Atlantic (vs MIS 5e) | | |
|---|---|---|---|---|---|---|
| | t-statistic | p-value | Accept/ reject null | t-statistic | p-value | Accept/ reject null |
| MIS 5e | 0.0 | 0.500 | Accept | 0.0 | 0.500 | Accept |
| MIS5d | 3.8 | 0.000 | Reject | 1.4 | 0.079 | Accept |
| MIS5c | 3.0 | 0.002 | Reject | 0.7 | 0.253 | Accept |
| MIS5b | 2.9 | 0.002 | Reject | 1.5 | 0.074 | Accept |
| MIS5a | 4.5 | 0.000 | Reject | 1.4 | 0.082 | Accept |
| MIS4 | 3.7 | 0.000 | Reject | 4.5 | 0.000 | Reject |
| MIS3 | 2.1 | 0.017 | Reject | 3.6 | 0.000 | Reject |
| MIS2 | 1.7 | 0.044 | Reject | 5.9 | 0.000 | Reject |
| MIS1 | 0.7 | 0.246 | Accept | 0.1 | 0.478 | Accept |

**Fig. 4.** Table 2: Statistical tests for significance of difference in deep-abyssal $\delta13C$ offsets versus penultimate interglacial (MIS 5e). 'Accept'/red is to accept the null hypothesis - no statistically signif

**Supplement:**

[revised manuscript text omitted]

---

## Author Comment (AC3) · 11 Jun 2020

CP reviewer comments #1 and author responses

AC: We thank the reviewer for their comments and the opportunity to explore some important issues in greater detail. We feel the review comments make a strong contribution to improving the quality of our work. We have followed up on the reviewer comments in detail and have attempted to address them.

We have made reference to changes to the manuscript, which are included as a supplement to the author comments, in track changes. Page and line references below

refer to locations in the revised document with track changes.

We have addressed the minor comments first, and then provided a longer discussion on the major comments.

Please note that we have changed our treatment of ocean $\delta$13C proxy data, stemming from one of the other reviewer comments, to only include $\delta$13C from Cibicides species of benthic foraminifera. We have also made some small changes to the parameterisation of the volcanic and weathering isotopic signatures in the model, from reviewer comments. These changes required the re-calibration of our model and re-running of the model-data experiments. The model-data results changed modestly. We have updated the figures and text (tracked in the attachment) in the manuscript, accordingly.

Minor comments

RC: "Figure 1: It is not clear, how GOC (red arrows) is split up in the part upwelling in Atlantic and Indo-Pacific Ocean."

AC: In the SCP-M simple carbon cycle box model GOC is split between a part that upwells into the subpolar Southern Ocean, and a part which transports directly into the polar Southern Ocean. This is an attempt to represent the GOC model of Talley (2013). This split is arbitrarily set at 50%. We have added this information to the caption for Figure 1 "GOC upwelling in both basins is set by default to 50% split between upwelling into the subpolar and polar Southern Ocean."

RC: "Figure 1: Does your approach focusing on changes in GOC, AMOC and export production inply, all other proceses (fluxes) stay constant in time?"

AC: In the model-data experiments we allow GOC, AMOC and biological export production parameter values to vary, and we solve for them in the optimisation. The experiments include specified forcings of SST, salinity, ocean volume, polar Southern Ocean air-sea gas exchange, coral reef carbonate accumulation and cosmogenic 14C production, guided by proxy observations. Other input parameter values are held constant

in the experiments.

RC: Figure 12: x-axis is wrong, eg. MIS5e is between 114-122 ka, while it has been between 118-128 ka in other figures.

AC: Thank you. We have fixed the chart.

RC: With respect to iron fertilisation you might check on Shaffer and Lambert (2018)

AC: Thank you, we have added it to the references throughout the document (P11, L22; P26, L7) and an additional sentence at P26, L8 (note the LaTeXdiff/track changes program struggles to fit reference changes onto the page):

"According to Shaffer and Lambert (2018), fertilisation of the surface ocean, and dust scattering effects on solar radiation, helped to push atmospheric $CO_2$ into and out of their glacial minima, for example at the LGM and last glacial termination."

RC: The fact that not one single process is needed to explain LGM-Holocene carbon cycle changes is long known, and has been called "the carbon stew" by some authors. You might want to check and discuss in more detail earlier modelling approaches on one glacial cycle (or longer), for example in Ganopolski and Brovkin (2017).

AC: Thanks. We have added to the introduction (P2, L23):

"Several studies have attempted to solve the problem of glacial-interglacial $CO_2$ by modelling either the last glacial-interglacial cycle in its entirety, or multiple glacial-interglacial cycles (e.g. Ganopolski et al., 2010; Menviel et al., 2012; Brovkin et al., 2012; Ganopolski and Brovkin, 2017). These studies highlight the roles of orbitally-forced Northern Hemisphere ice sheets in the onset of the glacial periods, and important feedbacks from ocean circulation, carbonate chemistry and marine biological productivity throughout the glacial cycle (Ganopolski et al., 2010; Brovkin et al., 2012; Ganopolski and Brovkin, 2017). Menviel et al. (2012) modelled a range of physical and biogechemical mechanisms to deliver the full amplitude of atmospheric $CO_2$ variation in the last glacial-interglacial cycle, using transient simulations with the Bern3D model.

According to Brovkin et al. (2012), a ∼50 ppm drop in atmospheric CO2 early in the last glacial cycle was caused by cooling sea surface temperatures (SST), increased Northern hemisphere ice sheet cover, and expansion of southern-sourced abyssal waters in place of North Atlantic Deep Water (NADW) formation. Ganopolski and Brovkin (2017) modelled the last four glacial cycles with orbital forcing as the singular driver of carbon cycle feedbacks. They described the "carbon stew", a feedback of combined physical and biogeochemical changes in the carbon cycle, to drive the last four glacial-interglacial cycles of atmospheric CO2."

We have also added the following to our discussion (P28, L10):

"Ganopolski et al. (2010) and Brovkin et al. (2012) modelled cooling SST and substitution of North Atlantic Deep Water by denser waters of Antarctic origin, in the abyssal ocean, as the main drivers of falling atmospheric CO2 at the last glacial inception. Menviel et al. (2012) modelled a transient slowdown in the rate of overturning circulation in the North Atlantic across MIS 5d-5e."

RC: Section 5.3. You might want to check on recent finding of terrestrial carbon storage from $\delta$13C in Jeltsch-Thömmes et al. (2019).

AC: Thanks, we have added this to our discussion of the terrestrial biosphere (P33, L24):

Jeltsch-Thommes et al. (2019) estimated a glacial-interglacial change in terrestrial biosphere of 850 Pg C (median estimate; range 450 to 1250 Pg C). Jeltsch-Thommes et al. (2019) demonstrated the importance of including ocean-sediment and weathering fluxes in their modelling estimates, and suggested other studies may underestimate the full deglacial change in the terrestrial biosphere carbon stock.

RC: Figure 14: Changes in CO2 caused by changes in terrestrial NPP and carbon stocks are missing in this figure. Please add.

AC: We have incorporated the contribution of the terrestrial biosphere to the glacial

CO2 drawdown in Figure 14. We have shown the effect of the model run with- and without the terrestrial biosphere to estimate its effects, as per Ganopolski and Brovkin (2017) Figure 9b, and we have compared with their model output.

Major comments

RC: Overall recommendation

My recommendation therefore is, that the model in its present form might be a useful tool for evaluating marine processes, and might be well used together with the available marine data (apart from $\delta$13C), but fails to give meaningfull results for the $\delta$13C cycle. This includes atmospheric and marine $\delta$13C. I urge the authors to get those parts out of the manuscript. If they wish to further analyse the $\delta$13C-cycle I believe fundamental model improvements are necessary, that can not be obtained by a major revision, but by a revised model version. Besides this, shortcomings of the steady-state approach should be discussed in more detail and the unclear (wrong?) aspects of carbonate weathering and annual fluxes in/out of the simulated system (atmosphere/ocean) need to be clarified for each MIS, maybe in a table or a new figure.

AC: We've discussed the comments of the reviewer, and clarified various parts of the modelling referred to by the reviewer, in quite some detail below. We've also made some small adjustments to $\delta$13C parameters for volcanic source carbon and silicate weathering, in the model, and incorporated those in the revised modelling results in the updated manuscript. We've clarified what our model-data results are saying about $\delta$13C in MIS 3-5, and clarified how our model deals with carbonate weathering, with specific reference to the literature and the model code (annotations provided in the Attachments to our responses). We've also discussed features of the terrestrial biosphere in more detail. We've discussed the issues associated with MIS-averaging, revised our wording in the manuscript, and also provided a better description of the model-data results. There is an updated model code to upload with the finalised manuscript.

As a more general comment on $\delta$13C, we demonstrated in O'Neill et al. (2019) that the model we have used, replicates the modern atmosphere and ocean $\delta$13C data time series, and replicates the effects of anthropogenic emissions on ocean and atmospheric $\delta$13C, including matching atmospheric data time series for the last 250 years, and GLODAPv2 data for the present, and matches Holocene data, and successfully matched LGM proxy data.

We argue that the model-data results and manuscript are best left with the $\delta$13C material retained, however with appropriate caveats to describe the shortcomings, as laid out below.

RC: "The chosen approach of steady-state analysis combined with optimization is a way, which certainly has benefits, but also shortcomings. I believe the benefits lie in the possibility to test a great number of parameter values, and this is certainly analysed with great effort and detail and worth publishing (but see my recommendation on shortenings of certain parts below). However, there is little learned on the potential shortcomings and pitfalls, which in my view need to be discussed more deeply. I believe where this approach is falling to short is the following: By analysing only steady-state the authors miss out the opportunities to judge the results based on the timing (when do processes change leading to what results). I provide one example where the article nicely fails, producing a potenially right answer for very likely the wrong reason: One of the dominant features of atmospheric $\delta$13C during the last glacial cycle is a drop by about 0.5‰ during MIS4. The steady-state approach now leads to the evalution of a mean value of atmospheric $\delta$13C which does not really cover this decrease at all, it shows about a decline by about 0.2‰ from MIS5a to MIS4 (Fig 4). So, any explanation for this drop would be falling too short in the observed amplitude by 0.3‰."

AC: This is an interesting debate, and we thank the reviewer for the opportunity to explore this.

The aim of our study is to help diagnose the causes of the major changes in atmospheric CO2 during the last glacial cycle. As we identify in our manuscript, there are three particularly large, sustained falls in atmospheric CO2 between the penultimate interglacial (∼125 ka) and the LGM (18-24 ka). These three major changes in atmospheric CO2 are summarised well in the literature, for example, in Kohfeld and Chase (2017). Our aim, with our model-data study, is to understand if plausible changes in ocean circulation (GOC and AMOC) and marine biological productivity can explain the major falls in atmospheric CO2. Other proxy data (e.g. $\delta$13C) provide useful data constraints for a model-data study to help solve this problem. Our approach is to apply a model-data optimisation with a simple carbon cycle box model, to solve for major ocean carbon cycle parameter values during the last glacial-interglacial cycle, and to explain the major, non-transient, falls in atmospheric CO2. To our knowledge, this is the first time that someone has attempted a multiple proxy model-data optimisation, that is optimised against atmospheric CO2, $\delta$13C, $\Delta$14C, ocean $\delta$13C, ocean $\Delta$14C, and carbonate ion proxy data, and hard-constrained by many observational data (SST, salinity, ocean volume, sea-ice cover proxy, coral reef carbonates, atmospheric 14C production rate) for the last glacial-interglacial cycle of 130 kyr. This study is quite different and unique in this regard.

This is done in an average sense across each MIS (nine of them over the last 130 kyr), using average proxy data values for each MIS and solving for the average parameter values at each MIS over the last 130 kyr. The MIS timeframes were chosen as an accessible reference point to the scientific community and because they are also simple reference points for the major atmospheric CO2 declines in the last glacial cycle. In this way, we may not solve for maximum or minimum values in the parameters, "overshoots" and "undershoots", within each MIS, but the changes in the average values across the last glacial-interglacial cycle. We think the article has been successful in achieving what it set out to do.

The aim of our study is not to disentangle the transient or shorter-term changes in the carbon cycle within MIS stages. Other studies (e.g. Eggleston et al., 2016) have done

that excellently for their area of focus (e.g. MIS 3-4 atmospheric $\delta$13C), and other mod-elling studies have attacked this using transient simulations (e.g. Ganopolski, 2010; Menviel et al., 2012). Our study does successfully diagnose the timing of changes in major oceanic processes that drive major changes in atmospheric CO2 during the last glacial-interglacial cycle, that are hard-constrained/brutally optimised by a host of data and observations.

With regard to your comments about MIS 3-5 and our manuscript and modelling, below, we address that in more detail to clarify what it is (and isn't) that our model-data results are telling us, and what we should have said about transient changes in our original manuscript. We clarify things quite substantially, but we are still very happy to explore the shortcomings of the approach and we have amended the manuscript with additional caveats, as explained below.

RC: Note that this $\delta$13C feature is not rapid, it is an anomaly that has been detected from raw data by spline smoothing and is alltogether nearly 20 kyr long, however the decreasing flank falls in MIS4, the increasing flank in MIS3, thus the signal is largely smoothed out in the chosen MIS-centric analysis. The analysis of the results now comes to the conclusion that very likely changes in terrestrial carbon storage was responsible for a change in atmospheric d13Ca of $\sim$0.2‰ (as said explaining a too little amplitude), it is furthermore said that the drop is accompanied by a 30 ppm fall in CO2 (page 12, lines 1-5), citing Hoogakker et al., 2016. I believe this is entirely wrong: The drop in CO2 happens clearly a few kyr before the drop in atmospheric $\delta$13C, as seen in Fig. 4. Furthermore, since both CO2 and $\delta$13C are meassured at the same samples and are both derived from gases in ice cores, this temporal offset between CO2 and atmospheric $\delta$13C can not be explained by chronological issues. The anomalies in biosphere as documented by Hoogakker et al., 2016 all fall in line with the CO2 changes, but not with the $\delta$13C changes, also note that Hoogakker et al., 2016 was published before the atmospheric $\delta$13C data set of Eggleston et al. (2016). In that respect citations from Hoogakker on page 19 are also missing the correct timing: In

Hoogakker NPP drops between around 70 ka (parallel to the drop in CO2), while the $\delta$13C drop occurs 5 ka later. Also note, that in Eggleston et al. (2016) the authors of this atmospheric $\delta$13C record tried to make sense of it by focusing on the part in which $\delta$13C falls, but CO2 rises again (Fig 2 in that paper) focusing on an opposite behaviour than described here.""

AC: There is a major confusion here, that we will spend some time below to help with.

There are two minor comments in our original manuscript about the terrestrial biosphere and atmospheric $\delta$13C at MIS4. They are not conclusions of our work, nor are they a result of our model-data experiments, and thus don't reflect any obvious or glaring deficiency in our modelling or model-data analysis that we are aware of. The two comments are just peripheral statements we made about the terrestrial biosphere in MIS 4, with a very quick look at the atmospheric $\delta$13C data, without looking in any detail, as this excursion in the $\delta$13C pattern is not the focus of our work. In the two short sentences in our manuscript, we made casual reference of the transient, reversing change in atmospheric $\delta$13C across MIS 4 and MIS 3 (termed an "excursion" by Eggleston et al., 2016). We stated that the transient drop in $\delta$13C probably reflects a weaker terrestrial biosphere, based on reconstruction of the terrestrial biosphere for the same period, by Hoogakker et al. (2016). The first of these comment is in the "Data Analysis" section, and was just the result of a quick eyeballing of the atmospheric $\delta$13C data and another study on the terrestrial biosphere covering approximately the same time period (Hoogakker et al., 2016). You correctly point out our oversimplification of the true complexity of MIS 3-5, in our short statements.

This is simply an oversight on our part, in drafting the text and not joining the dots between the various data sources we have gathered, and our model-data results. If we look at this in a little more detail, with reference to the figures and tables in our manuscript (drawing the reviewer's attention to them here), we can provide the following (below).

If we look at Figure 2 in the manuscript, we can see that there are dramatic changes in SST (top panel), and less dramatic changes in salinity, sea-ice proxy, sea level/ocean volume, and reef C carbon between MIS 3-5. These data are well incorporated in our model-data experiments as forcings – or constraints, or, another way of saying it - values fed into the model - they hard hard-baked into our model-data results and are influencing the results. With those forcings included, our model-data experiments solve for changes in GOC, AMOC and SO Bio across MIS 3-5, and we find important changes in these parameters across MIS 3-5 (see Figure 8 where GOC, AMOC drop and Southern Ocean biological productivity increases). Therefore, our model-data experiments, and what we should say in our manuscript, is that there are large changes in SST and other observations in the ocean during MIS 4 and MIS 3 (Figure 2 top panel), as well as the changes we estimate for GOC, AMOC and Southern Ocean biological export productivity. It is likely that the combination of these features, led to the $\delta$13C pattern during MIS 3 and 4. We also note that Eggleston et al. (2016) posited changes in SST, ocean biological productivity, AMOC and Southern Ocean upwelling to explain the $\delta$13C "excursion" at MIS 3-4. There are also changes in the terrestrial biosphere, but as per the reviewer comments about timing with regards of changes in the terrestrial biosphere, atmospheric CO2 and $\delta$13C, and on closer inspection we can see that this is perhaps not a dominant driver but some background factor or simply a part of the MIS 3-4 $\delta$13C pattern.

However, it is clear, as you point out, that our MIS-averaging approach does not capture the full extent, the overshoots and undershoots, of the changes in atmospheric $\delta$13C across this period of MIS 3-5.

To address the reviewer comments, we simply reword the sentence in "Data analysis" you refer to (i.e. before the model results section), to better reflect the data we have used and how that data is described (e.g. Eggleston et al., 2016) and also the literature that has focussed in detail on atmospheric $\delta$13C at MIS 3-4.

Original text: "The large drop in $\delta$13C in MIS 4 accompanies a $\sim$30 ppm fall in CO2.

The drop in $\delta$13C is likely caused by a reduction in the terrestrial biosphere, itself driven by the fall in CO2 (Hoogakker et al., 2016)."

We reword this and include the caveat about how our MIS-averaging does not include the full amplitude of changes within MIS 4 and 3, at P15 L23:

"The large drop in $\delta$13C in MIS4, reverses in MIS 3 (Fig. 4(B)). This excursion in the $\delta$13C pattern likely resulted from sequential changes in SST (cooling), AMOC, Southern Ocean upwelling and marine biological productivity (Eggleston et al., 2016). Eggleston et al. (2016) parsed the atmospheric $\delta$13C signal into its component drivers across MIS 3-5, using a stack of proxy indicators, and highlighted the sequence of events between the end of MIS 5 and beginning of MIS 3, and their cumulative effects to deliver the full change in atmospheric $\delta$13C. Our MIS-averaging approach fails to capture the full amplitude of the changes in atmospheric $\delta$13C during MIS 3-5, and only captures the changes in the mean-MIS value, serving to understate the full extent of transient changes in responsible processes. In addition, the MIS-averaging approach misses the sequential timing of changes in processes within each MIS. These are limitations of our steady-state, MIS-averaging approach."

Then on P22 L24, in reference to the terrestrial biosphere:

Original text: "Notably, there is a distinct drop in NPP at MIS 4, a period where atmospheric CO2 falls by ∼30 ppm (Fig. 4(A)). Falling NPP and persistent respiration of the terrestrial biosphere carbon stock during MIS 4, which releases $\delta$13C-negative carbon to the atmo- sphere, can explain the steep drop in atmospheric $\delta$13C during the same period (Fig. 4(B))."

We simply delete the reference to the terrestrial biosphere and atmospheric $\delta$13C.

Plus, we have added a caveat to the discussion on limitations of the work that our MIS-averaging approach misses the full amplitude of transient changes (P34, L21):

"Our MIS time-slicing obscures details in the proxy records within MIS. For example, Yu

et al. (2013) observed a transient drop in carbonate ion concentrations in the deep Pacific Ocean during MIS 4, and there are large transient changes in atmospheric $\delta$13C during MIS 3-4. Ganopolski et al. (2010) and Menviel et al. (2012) modelled transient collapses and rebounds in AMOC during MIS 4 (and other short-term changes in atmospheric dust supply and depth of biological nutrient remineralisation), which could have contributed to the full observed magnitude of changes in atmospheric $\delta$13C across this period (e.g. Eggleston et al., 2016) - not captured with our MIS-averaging approach."

However, what we are getting at in our response here, is that although our MIS-averaging strategy misses the full amplitude of transient changes or "excursions" in the proxy record, this doesn't mean that we don't meaningfully capture the data signals across the glacial-interglacial cycle, in our model-data experiments, as data constraints on our model-data experiments. This is a more nuanced but very important point, that we explore in more detail in the following.

A closer look at MIS 3-5 atmospheric CO2 and $\delta$13C and our model-data results.

Our model-data experiments at MIS 3,4 and 5, contain forcings of the model with observationally-derived SST, salinity, sea-ice proxy, sea level/ocean volume, and reef carbonate carbon fluxes. In addition to the model forcings, our MIS-averaged model-data results show a fall in GOC and AMOC, and an increase in Southern Ocean biological export productivity from MIS 5 into MIS 4. This outcome is supported by many proxy observations from the ocean for this time period including ocean carbonate ion proxy, ocean $\delta$13C and dust records for the Southern Ocean and intense cooling in the North Atlantic Ocean (e.g. Oliver et al., 2010; Yu et al., 2016; Kohfeld and Chase, 2017). In addition, our results agree with transient modelling of the last glacial-interglacial cycle, across this period MIS 3-5. Ganopolski et al. (2010), Brovkin et al. (2012) and Menviel et al. (2012) all show a slowdown in AMOC at this time. Ganopolski and Brovkin (2017), in Figure 9(c) in their paper, model a contribution to atmospheric CO2 drawdown from dust iron-fertilisation of Southern Ocean marine biological productivity in MIS 4. Therefore, our model-data conclusions for MIS 4 are consistent with the proxy

data and also transient modelling exercises for this period.

Further to the review comments, we undertook a simple reconnaissance modelling experiment to test our MIS-average model-data results, at more detailed time intervals, against the non-MIS averaged data for atmospheric CO2 and $\delta$13C through the MIS 3-5 period, to see if they hold up.

Eggleston et al. (2016) attempted to disentangle transient changes in the atmospheric $\delta$13C pattern during MIS 4 and MIS 3 (Heading 4.2 in their paper "Transient Changes at the Onset and End of MIS 4"). The first process they identify is iron fertilisation from dust over the Southern Ocean and a possible increase in SO biological export productivity (as above, we modelled increased SO biological export in this period too) to lower atmospheric CO2 (but this would increase atmospheric $\delta$13C). Then, Eggleston et al. (2016) mention cooling SST (a key part of our model constraints), where they plot a global average (we model latitude bands), which would also lead to lower atmospheric CO2 as well as lower $\delta$13C (colder ocean fractionates more $\delta$13C). Then, they mention slowing AMOC as a minor cause of lower atmospheric CO2 and higher $\delta$13C. Then, Eggleston et al. (2016) mention the effects of carbonate compensation and ocean alkalinity in lowering atmospheric CO2, and with minor effects on $\delta$13C (captured in our model in MIS average).

Then, Eggleston et al. (2016) mention that Nd isotope and Pa/Th ratios in proxy data support a more pronounced slowdown in AMOC, which lasted until the end of MIS 4 (also in our model-data result for MIS 4, as discussed above and in the manuscript). Eggleston et al. (2016) discuss a weakening and shoaling of AMOC and expansion of AABW at this time, and quote the hypothesised changes to AABW and AMOC of Ferrari et al. (2014). This would have lowered atmospheric CO2 but increased atmospheric $\delta$13C.

Then, Eggleston et al. (2016) mention that iron dust fertilisation may have reduced at the end of MIS 4 (showing the dust proxy data as evidence), leading to a drop in

SO biological productivity, which would increase CO2 and lower atmospheric $\delta$13C, reversing the hypothesised changes in early part of MIS 4 (note our results show SO biological export productivity drops off from MIS 4 levels at MIS 3 – Figure 8 in the manuscript). At this time, SST warmed a small amount in the SO, and cooled in the North Atlantic, with presumably offsetting effects. According to Eggleston et al. (2016), quoting opal flux data, a short-term increase in SO upwelling likely led to the final drop in atmospheric $\delta$13C to reach the trough of the $\delta$13C pattern near the boundary if MIS 3-4 (not captured in our MIS-averaged modelling).

We forced our model with the data in Figure 2 in the manuscript, without averaging for the MIS stages, over 1kyr intervals for the period 47-75 ka. We then took our model-data results for the average parameter values across MIS 3-5 as shown in the manuscript, and profiled them to vary within each MIS according to the hypothesised changes from Eggleston et al. (2016), described above, also at 1 kyr intervals. In this way, we allow the parameters to vary within the MIS stages, but constrained to meet the MIS-average values, in their average, from our model-data experiments as shown in the manuscript.

Figure 1 below (top panel) shows model-data results compared with the proxy data for atmospheric CO2 and Figure 1 (bottom panel) shows the same for atmospheric $\delta$13C (see figures below, at the bottom of this response document). This shows that taking the forcings for SST, salinity, sea-ice cover proxy, sea level/volume and coral reef carbonates (as per Table 2 of the manuscript), and time-profiled average MIS values for MIS 3-4-5 from the model-data experiments (taking the averages from Figure 8 in the manuscript), accounts for the full amplitude of changes in atmospheric CO2 and atmospheric $\delta$13C across this period, the overshoots and undershoots. The time-profiled model-data results shown in Fig. 1 below also account for the MIS-averaged proxy data across MIS 3, MIS 4 and MIS 5. The model results oscillate relative to the $\delta$13C data (Figure 1 below), due to the 1 kyr data slice intervals we have applied (we understand the $\delta$13C data has been smoothed), but it is easy to make this a 1 year/1

second interval exercise and it will produce a smoother set of results, for future analysis (in another body of work).

Completing this analysis shown in Figure 1 below, to match the transient atmospheric $CO_2$ and $\delta 13C$ data across MIS 3-5, does not change our findings as presented in the manuscript, but actually reinforces them.

Therefore, while the model-data results we present in the manuscript do not describe fully the transient or short-term changes in the carbon cycle within each MIS, they are not inconsistent with the transient data observations – as evidenced by a 1 kyr-interval extension of our model-data results for MIS 3-5 (Figure 1), and comparison with proxy data and other modelling studies. Our model-data results show that, on average, GOC and AMOC weakened in MIS 4, and SO biology on average, was stronger, although these values fluctuated around their mean values within the MIS. We emphasise that these findings above are still peripheral to the main objective of our manuscript (major, sustained changes in atmospheric $CO_2$ through the last glacial-interglacial cycle), although this helps shed some more light on our model-data results in the context of the reviewer comments.

RC: "The second most dramatic change in atmospheric $\delta 13C$ is a sharp drop by 0.2‰ during Termination I, a time window which has been chosen to be not be included in this steady-state analysis, again missing the opportunity to use 13C to pin down responsible processes.."

AC: We disagree. We argue that the responsible processes for the major and sustained changes in atmospheric $CO_2$ over the last glacial-interglacial cycle (e.g. SST, ocean circulation, biological export productivity, sea level, coral reefs, salinity, terrestrial biosphere) actually show themselves much more clearly, in sequence over the last 130 kyr, than the very short last glacial termination – whereby many processes were interacting in a relatively, very short period of time, and not easily untangled.

Our approach to attempting to solve for large changes in atmospheric $CO_2$, is to study

the 100 kyr lead-up to the LGM, where the large changes separate out much more clearly into unique events over 100 kyr. Many studies have attempted to answer the problem of glacial-interglacial CO2 by focussing on the LGM and Holocene periods alone (e.g. Peterson et al., 2014; Menviel et al. 2016; Muglia et al., 2018). Others may try to get at this by looking at 10-18 ka period with transient modelling, where all the changes in the carbon cycle rapidly unwound (e.g. Menviel et al., 2012; Joos et al., 2004, Ganopolski and Brovkin (2017)), but that's not our paper. That's almost an entirely different approach to the explicit approach of our paper which was NOT to focus on the transient termination of the last glacial maximum, which has been studied in great detail elsewhere. Our paper is focussed on the major, non-transient, drops in atmospheric CO2 in the lead-up to the last glacial maximum over 100 kyr – a much longer period that nicely shows up the sequential changes in the carbon cycle.

We have added references to the studies mentioned, to point readers in that direction if that is their area of focus, at P8 L7.

"We are interested in the LGM and Holocene as discrete periods, so our experiment time slice for MIS 2 is truncated at 18 ka, and our MIS 1 simply covers the Holocene, removing overlaps with the glacial termination. Therefore, our modelling excludes the last glacial termination (âĹij11-18 ka). The glacial termination period was highly transient, with atmospheric CO2 varying by âĹij85 ppm in <10 kyr, and large changes in carbon isotopes. Thus, it is anticipated that in a model-data reconstruction, model parameters would vary substantially for this period. Our strategy of integrating the model forward to an equilibrium state for each MIS as intervals of discrete climate and CO2, would be unsuitable when applied to the last glacial termination. Joos et al. (2004), Ganopolski et al. (2010), Menviel et al. (2012), Menviel and Joos (2012), Brovkin et al. (2012) and Ganopolski and Brovkin (2017) provide coverage of the termination period with transient simulations of the last glacial-interglacial cycle, using intermediate complexity models (more complex than our model)."

Further, in our discussion of limitations of the study (P34 L27): "We omitted the transient last glacial termination from our analysis, a period in which atmospheric CO2 rose ∼85 ppm in 8 kyr. Future model-data optimisation work could probe this period at 1 kyr intervals, or with transient, data-optimised simulations, to profile the unwinding of processes that led to the last glacial cycle CO2 drawdown."

RC: Only the long-term trend in $\delta$13C of +0.2‰ from the penultimate interglacial to the Holocene seemed to be meaningful covered by the approach

AC: The change in $\delta$13C across the last glacial-interglacial cycle is a bit larger than +0.2 per mil, as stated by the reviewer comment. The change in atmospheric $\delta$13C is quoted as +0.4 per mil in the literature (e.g. Schneider et al., 2013; Eggleston et al., 2016) and is a very important feature of the last glacial-interglacial cycle of atmospheric $\delta$13C. Noted our MIS-averaging also understates this full variation, but it is a very important long-term and sustained feature of the last glacial-interglacial cycle, so it is important that our analysis meaningfully captures this feature. As per Eggleston et al. (2016):

"Due to the lack of a complete d13C(atm) record connecting the various data sets, unanswered questions remained. Most importantly, the penultimate glacial maximum (PGM) was found to be 0.4‰ isotopically lighter in d13C(atm) than the Last Glacial Maximum (LGM), and the penultimate warm period (marine isotope stage (MIS) 5e) was also more negative in d13C(atm) by a similar amount. This is a surprisingly large difference, on the order of the changes in d13C(atm) observed during glacial terminations."

While we don't focus on the MIS 3-5 transient $\delta$13C excursion, with better explanation provided above we can demonstrate our approach produces results that are consistent with more detailed interpretations of the transient proxy record, such as Eggleston et al. (2016).

Reviewer comments: the $\delta$13C cycle

AC: We address the reviewer comments individually, and then provide annotated snap-

shots of the model code as supporting evidence in the Attachment A to our author comments.

RC: "As already seen above the steady-state approach might not be the best way to tackle atmospheric $\delta$13C. Furthermore, for an evaluation of $\delta$13C in general in such steady- state experiments as performed here the fluxes (e.g. as mol C/yr) and $\delta$13C-signatures in/out of the simulated atmosphere/ocean carbon cycle are essential: atmosphere- land carbon fluxes, volcanic CO2 outgassing, weathering, and burial of organic and inorganic carbon in the sediments. Little to non of those fluxes (and $\delta$13C-signatures) are given in the text itself. If I dig into the python source code of the model (or the description of version 1 in O'Neill et al. (2019)) I find a few information, but the source code is difficult to interpret as a non-user and some information seemed to be either misleading or wrong. An examples: Continental weathering consists of two different processes depending on the rock type that is weathered. In carbonate weathering 1 mol of CaCO3 together with 1 mol of CO2 from the atmosphere leads to the entry of 2 mol of HCO−3 into the surface ocean. In silicate weathering 2 mol of atmospheric CO2 are necessary to weather 1 mol of CaSiO3 leading again to the entry of 2 mol of HCO−3 into the surface ocean. For details see, for example Lord et al. (2016). From the description of weathering in O'Neill et al. (2019) I have the impression that the carbonate weathering is not depicted correctly (no consumption of atmospheric CO2). "

AC: Re carbon fluxes/$\delta$13C. We have added the below table (see Figure 2/Table 1 below, at the bottom of our response) to the Supplementary Information to describe the various prescribed fluxes of C and $\delta$13C signatures in our (revised) modelling exercise. This includes some changes from the original model/and model-data runs, from this set of review comments, and also the other reviewer comments. These changes in the model from the revised model-data experiments, will be uploaded with the final manuscript to a new Zenodo link.

Further below, we have clarified our treatment of carbonate and silicate weathering,

carbonate weathering $\delta$13C signature, and we have modified our volcanic $\delta$13C and silicate weathering $\delta$13C signatures in the model (incorporated in the revised manuscript model-data results provided in the revised manuscript), in response to the reviewer comments.

Re carbonate and silicate weathering fluxes.

We consulted Lord et al. (2016), as recommended by the reviewer in the RC above, and we note that the approach to carbonate weathering of Lord et al. (2016) is identical to ours, in that the activity of carbonate rock weathering simply transfers fluxes of DIC and Alk (in ratio 1:2) to the ocean via rivers, which causes a sink of CO2 to the ocean, and their treatment of silicate weathering is very similar to ours (see below, where we looked into more detail in the rock weathering model of Lord et al. (2016), which is described in detail in Colbourn et al. (2013)).

For example, in Lord et al. (2016):

"In all schemes, the terrestrial rock-weathering module calculates global fluxes of ALK and DIC from carbonate and silicate rock weathering and routes them to the coastal ocean". And importantly, as described in in Colbourn et al. (2013), the carbonate weathering model used in Lord et al. (2016):

"Note that there is only one mole of DIC for each mole of Ca2+; this is a short-circuiting of the atmosphere based on the assumption that the atmosphere and surface ocean are well mixed on the timescales considered here. Instead of removing one mole of CO2 from the atmosphere – and by implication the ocean – and adding two moles of bicarbonate to the ocean nothing is taken from the atmosphere and one mole of bicarbonate is added to the ocean." In addition to Lord et al. (2016), we also found our approach for carbonate and silicate weathering to be identical to a range of other studies – they are shown and referenced below. We also found our approach to $\delta$13C in carbonate weathering, as shown by the model code as shown in the Attachment to these comments below (with line-by-line annotation) was identical to that used in Sano

and Williams (1996) and Mook (1986), the references suggested by the reviewer for us to consult (see RC below).

As pointed out by the reviewer, some confusion for the reader about carbonate and silicate weathering, is perhaps contributed from our simple, high level model description paper which glosses over some details (O'Neill et al., 2019) and perhaps non-user friendliness of the model code. We will add better descriptive text in our model code for the final model upload to this manuscript upon finalisation. We've provided line-by-line references to our model code in the Attachment to these responses, to help understanding.

Further on the treatment of carbonate and silicate weathering in SCP-M.

The treatment of carbonate and silicate weathering in SCP-M is described in O'Neill et al. (2019) and mainly takes into account Walker and Kasting (1992), Toggweiler (2008) and Zeebe (2012) for its basis. Walker and Kasting (1992) provides the theoretical basis for treatment of carbonate and silicate rock weathering/river fluxes in many carbon cycle models (e.g. Zeebe, 2012; Colbourn et al., 2013; Lord et al., 2016). For example, Zeebe (2012) applies to the LOSCAR carbon cycle model a simple, parameterised weathering scheme based on Walker and Kasting (1992) and the same scheme was applied in simple carbon cycle feedback modelling applied by Toggweiler (2008) and Hogg (2008). An almost identical approach, was also applied by Lenton and Britton (2006), and Colbourn et al. (2013) and Lord et al. (2016). The only difference with Lenton and Britton (2006) and Colbourn et al. (2013) from our simple model, is that they applied additional temperature and terrestrial biosphere dependencies for rock weathering.

In summary, continental silicate and carbonate rock weathering are both represented in the SCP-M model. Both supply alkalinity and carbon to the surface ocean in ratio 2:1 (e.g. more alkalinity than DIC).

The weathering equation used in the model, is as per the model documentation (O'Neill

et al., 2019), and the original model code at (https://doi.org/10.5281/zenodo.1310161), and is reproduced here:

dC/dt_weath = (WSC + (WSV + WCV )AtCO2)

where WSC is a constant silicate weathering term set at $0.75 \times 10^{-4}$ mol m$^{-3}$ year$^{-1}$, WSV is a variable rate of silicate weathering per unit of atmosphere CO2 (ppm), set to 0.5 mol m$^{-3}$ atm$^{-1}$ CO2 year$^{-1}$ and WCV is the variable rate of carbonate weathering with respect to atmosphere CO2, set at 1.5-2.0 mol m$^{-3}$ atm$^{-1}$ CO2 year$^{-1}$ (Toggweiler, 2008).

There is a slight difference between carbonate weathering versus silicate weathering, in our model, in terms of the direct consumption of CO2 from the atmosphere when weathering takes place. This direct consumption of CO2 is assumed to be fully reversed in the case of carbonate weathering, but is only partially reversed in the case of silicate weathering. The main CO2 sink activity of the carbonate weathering, is therefore is in the alkalinity fluxes to the ocean and its effects on relative pCO2 in the ocean versus the atmosphere (e.g. Colbourn et al., 2013).

Carbonate weathering

Weathering of carbonate rocks initially takes up CO2 from the atmosphere (one mol), and supplies calcium and bicarbonate ions to the ocean (an additional mol of carbon), as per the following equation:

$CaCO_3 + H_2O + CO_2 = Ca^{2+} + 2HCO^-_3$

Therefore, two moles of carbon and one mole of calcium enter the ocean for each mole of CaCO3 weathered. This raises ocean carbon and alkalinity by two units each. In steady state, subsequent precipitation of CaCO3 releases the same amount of CO2 back to the atmosphere that was consumed by weathering (Zeebe, 2012) – a short-term circular loop that leads to a net zero direct consumption of CO2 from the atmosphere from carbonate weathering (e.g. Colbourn et al., 2013; Lord et al., 2016). This

is described in detail in Zeebe (2012) - see Fig. 3 extract of the Zeebe (2012) schematic description of carbonate weathering.

This return of $CO_2$ to the atmosphere (one mol of carbon, as per Fig. 3 below from Zeebe (2012)) leaves a net addition to the ocean of carbon and alkalinity from carbonate weathering in 1:2 ratio (Zeebe, 2012). The ocean carbon and alkalinity balance is later restored due to subsequent burial and $CaCO_3$ and carbonate compensation (Zeebe, 2012).

According to Zeebe (2012):

"As a result, although the addition of $Ca^{2+}$ and 2 $HCO_3$ increases ocean $TCO_2$ : TA in a 2:2 ratio, on a net basis $CaCO_3$ weathering increases ocean $TCO_2$ : TA in a 1:2 ratio because one mole of $CO_2$ returns to the atmosphere. If influx equals burial, carbonate weathering thus represents a zero net balance for atmospheric $CO_2$."

For our steady state modelling, we assume the $CO_2$ consumed directly by the carbonate weathering process is returned to the atmosphere – a net zero of direct consumption of $CO_2$ from the atmosphere. This is a short-circuiting of the process, but not incorrect (refer Colbourn et al. (2013) quote reproduced above, about "short-circuiting" direct atmospheric $CO_2$ effect of carbonate weathering). Therefore, the fluxes associated with carbonate weathering are those of DIC and alkalinity into the surface ocean boxes of the model. This is the same approach applied by Toggweiler (2008), and Lenton and Britton (2006), and identical to the approach of Lord et al. (2016), and Colbourn et al. (2013). For these studies, the sink of atmospheric $CO_2$ from carbonate weathering comes indirectly through the effects of alkalinity supplied to the surface ocean which lowers $pCO_2$ and draw $CO_2$ into the ocean. Some interesting quotes from those references below, with bolded parts for emphasis.

The approach for carbonate weathering in Lord et al. (2016) (the reference suggested by the reviewer), is referenced in that study to Colbourn et al. (2013), and is described in Colbourn et al. (2013) as:

"Note that there is only one mole of DIC for each mole of Ca2+; this is a short-circuiting of the atmosphere based on the assumption that the atmosphere and surface ocean are well mixed on the timescales considered here. Instead of removing one mol of CO2 from the atmosphere – and by implication the ocean – and adding two moles of bicarbonate to the ocean (as in Eq. 1), nothing is taken from the atmosphere and one mole of bicarbonate is added to the ocean."

And "The fluxes are then used to calculate fluxes of DIC (FDIC) and Alkalinity (FAlk)".

We note further from Colbourn et al. (2013):

"In the case of carbonate weathering there is an overall null cycle for CO2, whereas silicate weathering transfers CO2 to the Earth's crust."

In summary, our simple box modelling representation of carbonate weathering is consistent with the theory of carbonate chemistry, and the literature on modelling of carbonate weathering. Our calculated estimate of 10 Tmol C yr-1 from carbonate weathering supplied to the ocean at 275 ppm atmospheric CO2, is comparable to that of 12 Tmol mol C yr-1 in Morse and Mackenzie (1990) and Zeebe (2012), Archer et al. (1998), but higher than that assumed by Colbourn et al. (2013) and Lord et al. (2016) of 5 Tmol C yr-1. In those latter two studies, they simply assume an even, equal split of the fluxes of silicate and carbonate weathering in their model spin-up Fsil=Fcarb=5 Tmol C yr-1. However, Colbourn et al. (2013) quote post-spin-up, pre-industrial total flux of weathering of 12-20 mol C C yr-1, split equally between carbonate and silicate weathering (6-10 mol C yr-1 each).

We have supplied an annotated snapshot of our model code, in the Attachment A to these comments (below).

Silicate weathering

Silicate rock weathering can be described by the following chemical equation:

CaSiO3 + H2O + 2 CO2 = Ca2+ + 2HCO−3 +SiO2

Silicate rock weathering removes 2 mols of CO2 from the atmosphere for each mole of CaSiO3 weathered. The subsequent precipitation of CaCO3 in the ocean releases one mole of CO2 back to the atmosphere, with the other mole of CO2 consumed by the atmosphere, taken up in CaCO3, which may end up buried in the marine sediments (Zeebe, 2012). In steady state, over long timeframes, the silicate weathering direct consumption of atmospheric CO2 balances out volcanic emissions of CO2 (Berner et al., 1983; Zeebe and Caldeira, 2008, Zeebe, 2012). Because the steady state in the silicate weathering is achieved over a much longer timeframe (1e5-1e6 years), it is appropriate to model a direct sink of CO2 from the atmosphere associated with silicate weathering. The steady state atmosphere-ocean response to carbonate weathering only requires a relatively short timeframe, hence we can model the steady state assumption of carbonate weathering returning its direct consumption of CO2 to the atmosphere (Walker and Kasting, 1992; Lenton and Britton, 2006; Toggweiler, 2008, Zeebe, 2012).

Therefore, relative to carbonate weathering, there is an additional step applied with silicate weathering. To account for the unit of CO2 consumed directly from the atmosphere in silicate weathering that is not returned (one more unit than carbonate weathering, as per Zeebe, 2012), and using the approach of Toggweiler (2008), we also subtract an amount equal to a unit of silicate weathering directly from the atmosphere. This is the same approach of Zeebe (2012) who applies a doubling of the molar flux of silicate weathering (to replicate two mols of CO2 initially drawn from the atmosphere), and that of Toggweiler who subtracts a flux of CO2 directly from the atmosphere (but no direct consumption of CO2 in the case of carbonate weathering) to account for the additional unit of CO2 consumed by silicate weathering (when compared with carbonate weathering). This flux is subtracted directly from Atmospheric CO2 in SCP-M as referenced in the model equation above (and described in the code in Attachment A to these comments). This flux, subtracted from the atmosphere, negates the effects on atmospheric CO2 of the units of C added to the ocean by the silicate weathering flux of C. Volcanic CO2 emissions are set equal to the amount of CO2 taken directly from
the atmosphere by silicate weathering, to reflect the long-term offset of volcanic emissions by silicate weathering (Walker and Kasting, 1992; Archer et al., 1998, Toggweiler, 2008; Zeebe, 2012, Colbourn et al., 2013; Brault et al., 2017).

As described in Walker and Kasting (1992), Toggweiler (2008), Zeebe (2012) Brault et al. (2017), Colbourn et al. (2013, 2015) and Lord et al. (2016), in steady state the silicate weathering flux feedback for $CO_2$ matches the volcanic $CO_2$ emissions, which we have set in SCP-M. Note, for anthropogenic scenarios we separate volcanic emissions from weathering flux, because the silicate weathering feedback under the forcing of atmospheric $CO_2$, is expected to increase at a greater rate than volcanic emissions (volcanic emissions do not respond to anthropogenic emissions of $CO_2$).

Our calculation for silicate weathering yields a flux of carbon to the oceans 6.3 T mol C yr-1 at 275 ppm atmospheric $CO_2$. Our volcanic emissions rate is set to this figure, which is in good agreement with Lord et al. (2016) who set their volcanic C flux at 5.6 Tmol yr-1 to balance the silicate weathering component.

RC: Furthermore, from the python code I learned that weathering (probably meaning carbonate weathering, since in silicate weathering all $CO_2$ comes from the atmosphere with its $\delta$13C-signature) has a $\delta$13C-signature of $-6.9$‰ similarly as volcanic $CO_2$. While the volcanic $\delta$13C seems to be in the expected range (although on the lower side) I believe the weathering $\delta$13C-signature is wrong, since carbonate rocks have a typical $\delta$13C-signature of about +1-2‰ see for example Sano and Williams (1996); Mook (1986).

AC: Re carbonate weathering. The $\delta$13C of carbonate weathering in our model is not -6.90 per mil, as stated in the reviewer comment above, but it is 0 per mil, via our application of the reference standard value for $\delta$13C (the Pee Dee Belemnite) = 0. This feature is shown clearly in the annotated excerpt of the model code in the Attachment A. 0 per mil is the identical value for carbonate weathering used in the first reference provided by the reviewer (Sano and Williams, 1986), and precisely in the middle of the

range (+/- 1 per mil) used in the second reference provided by reviewer (Mook, 1986). We have added text to our model code to make this more obvious (for final upload).

With regards to silicate weathering $\delta$13C. In the SCP-M model the $\delta$13C of silicate weathering CO2 drawdown was originally set at -6.90 per mil, which is the same as the volcanic $\delta$13C we had assumed. This approach was consistent with offsetting volcanic CO2 emissions with silicate weathering (Zeebe, 2012; Toggweiler, 2008, Lord et al., 2016; Colbourn et al., 2013, 2015; Walker and Kasting, 1991). This is a simplification with regards to the $\delta$13C, and therefore we have changed this, and now applied the atmospheric $\delta$13C signature output from the model to the silicate weathering flux (this is now updated in our model results/re-runs).

As per the reviewer comments we have now set the $\delta$13C of the direct consumption of CO2 by silicate weathering, to take atmospheric $\delta$13C value. This is a modest change, however, as atmospheric $\delta$13C is in the range -6.3-7 per mil in the last glacial-interglacial cycle, and we had initially assumed a fixed value of -6.90 per mil.

We also note the reviewer comment that our assumption of -6.90 per mil for volcanic CO2 emissions is at the low end of literature estimates. We have modified this to -4.5 (compared with -4.0 in Zeebe, 2012).

In summary, the changes we have incorporated in the final set of model runs for this manuscript, guided by the reviewer comments:

-We have changed the $\delta$13C of silicate weathering direct consumption of CO2 from the atmosphere, to the atmospheric $\delta$13C signature outputted from model at each time step, as suggested in the reviewer comments (previously it was set as -6.90 per mil).

-We have adjusted our $\delta$13C of volcanic emissions from -6.90 to -4.50 per mil, which is more of a "middle of the range" value.

-We have tidied up the model code description of carbonate weathering and its $\delta$13C (for upload to the Zenodo repository upon finalisation of the manuscript).

RC: I also do not understand how their approach with not explicitly considering terrestrial carbon change (terrestrial carbon to my understanding is covered as externally to the atmosphere/ocean system, fluxes in/out of it prescribed by optimization) covers changes in C3 vs C4 photosynthesis (which have a significantly different isotopic fractionation) on glacial/interglacial timescales (Collatz et al., 1998; Köhler and Fischer, 2004) which leads to differences in the mean terrestrial $\delta$13C and therefore also the changes in the $\delta$13C-cycle as a whole (Kaplan et al., 2002).

AC: Thanks for the comment. Our response is broken in two parts 1) the terrestrial biosphere and 2) C3 vs C4 photosynthesis.

In summary, the terrestrial biosphere is explicitly considered in our modelling. It is two boxes within the carbon cycle box model we have used. It is not prescribed by optimisation.

We have decided not to assess C3 versus C4 photosynthesis and its effects on $\delta$13C fractionation.

We discuss both of these points in more detail below.

Terrestrial biosphere in SCP-M

The terrestrial biosphere is treated in SCP-M as two boxes that exchange carbon with the atmosphere based on fluxes of net primary productivity (NPP) (carbon in) and respiration (carbon out). It is part of the carbon cycle that includes the terrestrial biosphere-atmosphere-ocean-sediments-volcanoes etc. Our box model applies a simple representation of the terrestrial biosphere, whereby biological productivity responds to carbon fertilisation. Therefore, CO2 is the driver of terrestrial biosphere productivity in this model. We apply the two-box terrestrial box model scheme of Harman et al. (2011). The inputs are starting estimates of net primary productivity (NPP), the terrestrial biosphere carbon stock, plant respiration rate and atmospheric CO2. The approach of Harman et al. (2011) is to split the terrestrial biosphere into a

fast-response (grasslands and grassy components of savannah systems) and a slow-response (woody trees) component. In this model, the productivity is mostly focussed on the plants/grasses component.

The formula is shown in the model documentation paper (O'Neill et al., 2019) and Harman et al. (2011), and is reproduced here:

dAtCO2/dt = $-$NpreRP[1+$\beta$LN(AtCO2)] + Cstock/k + Dforest

Where Npre is NPP at a reference pre-industrial level of atmospheric CO2, RP is a parameter to split NPP between short-term terrestrial biosphere carbon stock and the longer term stock (Cstock1 and Cstock2). B is a parameter with a value typically in the range 0.4-0.8 (Harman et al., 2011). Cstock is the carbon stock in each terrestrial biosphere box, k is the respiration timeframe for each box. Dforest is the prescribed rate of deforestation emissions for present day simulations and projections. A terrestrial biosphere fractionation factor is applied for the carbon isotopes.

This flux out of the atmosphere feeds into the two terrestrial biosphere stocks of carbon (Cstock1 and Cstock2), and the boxes lose carbon to the atmosphere by respiration, as per the equation above. This differential equation for NPP, respiration, and the net flux into and out of the terrestrial biosphere (increase or decrease in the terrestrial biosphere carbon stock), solves at each time step of the model, taking the model's output of CO2 and then calculating the NPP, respiration, Cstock1 and Cstock2 and calculating a new atmospheric CO2. The time step of the model is one year, with 10,000 years for each model-data simulation.

Harman et al. (2011) model the terrestrial biosphere primarily as a function of atmospheric CO2. They also incorporate an optional temperature dependency. This is the same approach used in the simplest 4Box terrestrial biosphere module of the Bern Simple Carbon Model (Strassman and Joos, 2018; Seigenthaler and Joos, 1992; Kicklighter et al., 1999; Meyer et al., 1999), and described by Enting (1994) – although we understand that there are various terrestrial biosphere modules applied with the Bern

models, and most are more complex. As far as we can discern, the simple carbon fertilisation approach is also used in Jelstch-Thommes et al. (2019), which also applies the simplest 4Box terrestrial biosphere of the simple Bern model.

There are other possible drivers of the NPP – temperature, precipitation, soil nutrient levels. In the context of our simple carbon cycle model, we are mainly interested in CO2. We don't model atmospheric temperature, and if we were to try to incorporate atmospheric temperature as a driver of terrestrial biosphere, we would also need to incorporate it for terrestrial weathering. There is a limit to how much detail we want to include in the model given we are conducting many simulations ($\sim$80,000) in our model-data optimisations across the MIS of the last glacial-interglacial cycle.

We do note that there are studies devoted to determining whether the CO2 fertilisation effect or climate is the dominant control on terrestrial biosphere NPP and the size of the terrestrial biosphere carbon stock. According to Hoogakker et al. (2016), CO2 fertilization, rather than climate, is the primary driver of lower glacial net primary productivity by the terrestrial biosphere, accounting for around 85% of the reduction in global NPP at the LGM. Kaplan et al. (2002) also concluded that over glacial-interglacial timescales, global terrestrial carbon storage is controlled primarily by atmospheric CO2, while the climate has more influence on the isotopic composition. Otto et al. (2002) also found that the CO2 fertilization effect is mostly responsible for the total increase in vegetation and soil carbon stocks since the last glacial maximum. Kohler et al. (2010) prioritised CO2 fertilisation as the driver of terrestrial biosphere in their "control" main simulation scenario for glacial-interglacial cycles over the last 740 kyr, but also ran scenarios with a climatic driver for the terrestrial biosphere to estimate the effects of "fast" climate changes on atmospheric $\delta$13C. Other studies arguing that atmospheric CO2 is an important, or is the main driver of terrestrial biosphere productivity include Kicklighter et al. (1999), Joos et al. (2004), Schimel et al. (2015), Sitch et al. (2008), Arneth et al. (2017). This view has been contested by van der Sleen et al. (2015).

Given we don't model the atmospheric temperature or precipitation, we saw limited

additional benefit to introduce them into our model of the terrestrial biosphere, although it would not be difficult to do this. Finally, given that CO2 and atmospheric temperature co-vary closely, across glacial cycles, it seems of limited benefit to split these effects out in our simple carbon cycle modelling exercise. For example, Meyer et al. (1999) found similar results for modelling carbon uptake in the terrestrial biosphere whether only CO2 fertilisation, or CO2 fertilisation + climate, were included as drivers of NPP – but noting this was not tested for the LGM.

Our aim is not to contribute new findings on the terrestrial biosphere, but we present the behaviour of the terrestrial biosphere in our manuscript to confirm that our exhaustively multi-proxy constrained model-data output is consistent with the range of literature estimates of variations in the terrestrial biosphere in the last glacial-interglacial cycle and LGM-Holocene period, and we show this. For example, our experiment shows a change in the terrestrial biosphere carbon stock of +630 PgC between the MIS 2 (LGM) and MIS 1 (Holocene) period. This compares with other estimates of +540 PgC (Brovkin et al., 2007), +∼820-850 PgC (Joos et al., 2004) – with the majority by CO2 fertilisation, ∼+500 PgC (Kohler et al., 2010), +∼500 PgC (Brovkin et al., 2012), +850 PgC (Jeltsch-Thommes et al., 2019), +511 +/- 289 PgC (Peterson et al., 2014), +378 +/- 88 PgC (Menviel et al., 2016). Another estimate of the LGM-Holocene terrestrial biosphere change is 550-694 Pg C (Prentice et al., 2011), which our result of 630 Pg C sits comfortably within. Our estimate is actually towards the upper end of the literature ranges, suggesting if anything we could exaggerate the effects of the terrestrial biosphere from the LGM to the Holocene period, with perhaps little to gain by splitting out temperature and precipitation effects. If did, we would probably also need to consider other important features such as soil nutrients and local humidity. While we have a simple, but explicit two-box representation of the terrestrial biosphere, we don't believe that this detracts from our model-data results, as shown in Figures 9-11 and Figure 12 specifically for the terrestrial biosphere.

C3 and C4 photosynthesis.

In summary, our model exercise doesn't take account of C3 versus C4 photosynthesis in the terrestrial biosphere, or consider its effects on the $\delta13C$ signature of the terrestrial biosphere. In response to the reviewer comments, we looked into this in more detail to see if we can improve our modelling – noting that it is very easy to update the model code for something like this. For example, we re-ran the model-data experiments as part of one of the other reviewer comments, so could easily incorporate more detail for the terrestrial biosphere, such as C3%/C4% variation in $\delta13C$.

Our approach was to understand and quantify the references provided by the reviewer, review approaches by other modelling exercises for the glacial-interglacial cycle of last 130 kyr, and decide whether we should re-run the modelling with an alternative treatment of the terrestrial biosphere to cater for C3%/C4% and $\delta13C$. As part of investigation, we also constructed the C3/C4 model of Collatz et al. (1998)/Kohler and Fischer (2004) in a python module that easily fits into the carbon cycle box model, to evaluate whether it would improve our modelling (described below and attached to these comments).

Kohler and Fischer (2004), suggested reading by the reviewer, in their excellent paper do make a very good point about C3/C4 photosynthesis in the context of glacial-interglacial $\delta13C$, that is worth reproducing here as a summary:

"Oceanic inorganic carbon is becoming 0.4 heavier during the G/IG transition, which is in good agreement with both modelling studies and data constraints (Curry et al., 1988; Duplessy et al., 1988; Michel et al., 1995). It should be noted that 85% of this calculated oceanic change in $\delta13C$ can be explained by the increase in the terrestrial carbon stock and only the missing fraction of 15% by changes in the abundance of the two photosynthetic pathways. Thus, uncertainties in the current knowledge on C3/C4 plant distribution during the LGM are of minor importance for the overall simulation results."

Collatz et al. (1998)/Kohler and Fischer (2004) modelling approach

The Collatz et al. (1998) approach to modelling C3 vs C4 %, is based on the estimation of a "cross-over temperature" for dominance of C4 or C3 plants. Above the cross-over temperature, C4 plants are favoured. Below the cross-over temperature, C3 plants are favoured. Collatz et al. (1998) derived a simple equation for the cross-over temperature of C3 vs C4. The cross-over temperature exhibits a positive relationship with atmospheric CO2. Therefore, as CO2 goes up, the cross-over "hurdle" temperature for C4 dominance also increases, so C4% has a negative relationship with CO2. While increasing temperatures may favour C4 plants, if CO2 was also increasing, this would tip the advantage back towards C3 plants. The cross-over temperature calculation of Collatz et al. (1998) is shown as:

$T50(degC) = (10/\ln Q10)\ln(pO2(1 + 0.5\alpha C3/\alpha C4)/(0.8 \times pCO2 \times s25(\alpha C3/\alpha C4 - 1))) + 25$

Where T50 is the crossover temperature for C4 and C3 dominance, where aC3 is the "intrinsic quantum yield for C3 photosynthesis" and pi is the leaf internal pCO2, assumed to be equal to 0.8 x atmospheric pCO2. s25 is the value of s at 25°C and Q10 is the relative change in s for a change in temperature.

s is defined as:

$s = 2600 Q10^{(Tx-25/10)}$

where Q10 is the relative change in s for a 10°C change in temperature.

To analyse C4%, Kohler and Fisher (2004) extended the Collatz et al. (1998) equation and provide a simple set of equations to estimate C4% and C3% between the glacial and interglacial periods, using the change in temperature relative to changes in the cross-over temperature between the two periods :

$C4\% = C4\%^{*} \times (1 - \alpha C3/C4 \times (\Delta T - \Delta T50))$

$C3\% = C3\%^{*} \times (1 + \alpha C3/C4 \times (\Delta T - \Delta T50))$

We reconstructed this C3%/C4% model of Collatz et al. (1998)/Kohler and Fischer (2004), to investigate the reviewer comments.

We have uploaded the Python script and data for last glacial-interglacial cycle atmospheric CO2 and temperature, at https://zenodo.org/record/3889704#.XuH3Ji1L0_U.

We use the cross-over temperature calculation of Collatz et al. (1998), the C4% model of Kohler and Fischer (2004), and estimate an average terrestrial biosphere $\delta$13C using the C4% and C3% output from this model and estimates of $\delta$13C for C4 and C3 plants.

To test our simple model works, we satisfy the estimate of T50 of 22 deg C at atmospheric pCO2 of 350 ppm from Collatz et al. (1998), and, as per Kohler and Fischer (2004) Figure 4, ∼18 deg C at atmospheric pCO2 of ∼280 ppm, and ∼11 deg C for atmospheric pCO2 of ∼190 ppm. We forced our version of the C4% model of Kohler and Fischer (2004)/Collatz et al. (1998) with atmospheric temperature and CO2 through the last glacial-interglacial cycle (Figure 3 below). The atmospheric temperature data of Jouzel et al. (2007) is derived from Antarctic ice cores, so it likely overstates the amplitude global average temperature cooling during the glacial period. Jouzel et al. (2007) show peak cooling of ∼11 degrees C, which is greater than global estimates in the range 3-6 deg C (Schneider von Deimling et al., 2006a; Holden et al., 2009; Schmittner et al., 2011; Annan and Hargreaves, 2013). We take an intermediate average global LGM cooling of 4.5 degrees, and scale the profile of Jouzel et al. (2007) to the average global amplitude of cooling of 4.5 deg C for the LGM, which is the middle of the range of global estimates. This is a simplification, but appropriate for our reconnaissance exercise. We also apply the last glacial-interglacial cycle atmospheric CO2 data of Bereiter et al. (2015) – Figure 3 below.

In terms of what starting values to use for $\delta$13C for the C3 and C4 plants, we note a huge variation in the possible values to use for $\delta$13C of C3 plants, and also note a large variation in the estimates for average $\delta$13C of the terrestrial biosphere applied in terrestrial biosphere and carbon cycle modelling exercises for the last glacial-interglacial

cycle. We discuss in more detail below, but flag that natural variation in the average values assumed for $\delta$13C fractionation of the terrestrial biosphere, and variation in $\delta$13C values assumed between modelling studies, greatly outweigh the posited variation in $\delta$13C fractionation from C4% vs C3%.

Carbon cycle modelling exercises show a large range (e.g. Brovkin et al., 2002 (-16 per mil), Menviel et al. (2016) (-23.3 per mil), Jelstch-Thommes et al. (-24 per mil), and the study of Kohler and Fisher applied an average of -16 per mil (C3 -19 per mil, C4 -5 per mil). For this simple exercise, we take the starting average $\delta$13C for terrestrial biosphere taken from Jeltsch-Thommes et al. (2019) (this text was a suggested reference by the reviewer) of -24 per mil, and back out the average starting C3 and C4 $\delta$13C assuming the PI value of C4% of 20% applied in Kohler and Fischer (2004) (the reference suggested by the reviewer). This yields a starting $\delta$13C for C3 plants of -27 per mil, and -14 per mil for C4 plants. For comparison, Kohn et al. (2010) provided a range of $\delta$13C estimates for C3 plants of -20 to -37 per mil, with a global average of -27 per mil. O'Leary et al. (1988) provided a synthesis of global data of -27.1 per mil for C3 plants and -13 per mil for C4 plants.

We model C4% to vary from the preindustrial starting estimate of 20% (Kohler and Fischer, 2004), up to an average of 25% during the LGM (Figure 4) (see figures below, at the bottom of this response document). We model average $\delta$13C for the terrestrial biosphere to vary between the range -24.2-23.6 per mil during the last glacial-interglacial cycle, a variation of 0.6 per mil (Figure 4 below).

Our estimated C4% from using the Collatz et al. (1998) equation (25% as per Fig. 4) is a little higher than Kohler and Fischer (2004) (24%) and this likely reflects differing atmospheric CO2 and temperature assumptions. For example, Kohler and Fischer (2004) take average northern hemisphere average temp change of -5 degrees, and southern of -8 degrees. We have inputted a global average change of -4.5 degrees C as per the literature range of 3-6 degrees C cooling (Fig. 4 below). However, there must be something else being applied by Kohler and Fischer (2004) to achieve their

LGM "target" C4% of 30-33%.

The approach of Kohler and Fischer (2004) was to establish a target variation of C4% between the LGM and the PI and then to see what parameterisations of their model runs could reach that target. Our estimate of LGM C4% is of 25% is far below the "targeted" C4% of 30-33% from Kohler and Fischer (2004). Their study found that varying the C4% amplitude in the Collatz et al. (1998) C3/C4% share model could increase C4% from 20% to 24%, but increasing the grassland succession amplitude increased the C4% up to 42%, a much bigger change than the C3/C4% share model alone. Furthermore, according to Huang et al. (2001), local moisture conditions might be even more important than any temperature or $CO_2$ effects on C3/C4%.

The grassland succession factor is an equation contributed by Kohler and Fischer (2004) to estimate the effects of changes in the tree-line (the divide between where trees and grasses grow) as a function of changes in temperature, between the LGM and PI. According to Kohler and Fisher (2004), this is the main driver for the C4% change and change in the terrestrial biosphere $\delta$13C fractionation, perhaps not the temperature and $CO_2$-dependant equation of Collatz et al. (1998). Kaplan et al. (2002) posit something different again, that the major driver of changed terrestrial biosphere $\delta$13C discrimination since the LGM is retreating ice sheets, with an additional or ancillary role for C3/C4 plant substitution.

Our estimated change in terrestrial biosphere $\delta$13C fractionation of ~+0.6 per mil, is below the estimate from Kohler and Fischer of 1.3 per mil, and that reflects that they include the grassland succession factors in their LGM-PI analysis. The offset in assumed $\delta$13C fractionation between C3 and C4 of -13 per mil (-27 per mil less -14 per mil) is very similar to their chosen -14 per mil (-19 per mil less -5 per mil), suggesting that the differences reflect the use of another factor outside C3/C4%, the grassland succession factors, to drive their results.

Beyond the simple exploratory attempt above, modelling highly uncertain grassland

succession factors, or ice sheet retreat/advance, or localised moisture and temperature changes, to try and explain uncertain changes in C3% vs C4%, for which the starting values themselves could fall within huge ranges of uncertainty, looks beyond the scope of our study.

We note that, with regard to the estimates of C4% used by Kohler and Fisher to create "targets" for pre-industrial and LGM periods, Kohler and Fisher (2004) say the following:

"NPP and fC4 for the LGM are based on modelling studies only and, thus, represent only weak indicators which were only used for uncertainty estimates." And furthermore, on P16:

"However, because the constraints on NPP and the fraction of C4 plants were based on only a few mostly modelling studies, we merely interpret those as a model evaluation." These findings underscore the uncertainty of estimates for quantifying C4/C3 and therefore $\delta$13C of the terrestrial biosphere. This uncertainty is amplified in the actual estimates of $\delta$13C for C3 and C4 plants, as we discuss below.

Fig. 5 below is reproduced from Kohn (2010), and shows the range in $\delta$13C fractionation for C3 plants alone, which spans -20 to -37 per mil, and is impacted by many factors including temperature, precipitation, and effects of canopies and new growth.

Furthermore, more recently, Kohn (2016) attempted to estimate the change in $\delta$13C for C3 plants from the LGM to modern day, based on atmospheric CO2, and also to quantify the effects of precipitation on C3 plant $\delta$13C. This shows the variation in C3 $\delta$13C discrimination itself, is even bigger than the posited effect of C3/C4% (see Fig. 6 below extract from Kohn (2016)

To model C3 and C4 $\delta$13C properly, there are other important effects in C3 plants (on their own), that would need to be taken into account. For example, Francois et al. (1999) point out that changes in the $\delta$13C fractionation from a changing C4% were partially offset by changes in the opposite sign in the fractionation of C3 plants due to

the modification of the intercellular CO2 pressure within their leaves.

Peer group/modelling approaches

In exploring this issue of C3 vs C4% and $\delta$13C further, and to benchmark our work against the peers who are modelling and analysing the last glacial-interglacial cycle (0-130 ka), we investigated the literature. C3 versus C4 fraction in photosynthesising plants is not discussed much in the literature of modelling of the last 130 kyr glacial-interglacial cycle of carbon. We couldn't find any reference to C3/C4 photosynthesis and $\delta$13C in Eggleston et al. (2016), who contributed the atmospheric $\delta$13C data we used in our model-data analysis. We don't find any mention of C3/C4 photosynthesis and its effects on $\delta$13C in any of Ganopolski et al. 2010, Brovkin et al. 2012, Ganopolski and Brovkin, 2017; Kohfeld and Chase, 2017.

Brovkin et al. (2002) simply state, with reference to their CLIMBER-2 model of the last glacial-interglacial cycle:

"Most of the carbon (ca. 85%) is allocated to the C3 photosynthesis pathway and the remaining carbon (15%) to the C4 pathway. The globally averaged $\delta$13C fractionation factor for terrestrial biosphere is 0.984." (-16 per mil).

We find no reference to any changes for glacial interglacial C3 and C4 and terrestrial biosphere $\delta$13C modelled in Brovkin et al. (2007, 2012), or Ganopolski (2010, 2017).

The transient modelling of the last glacial-interglacial cycle, undertaken by Menviel et al. (2012b), does not mention C3 and C4 photosynthesis, or its effects on $\delta$13C fractionation.

We note that Kohler et al. (2010), mention the parameterisation of C4% in the terrestrial biosphere in their 740 kyr transient simulations with the BICYCLE model. In their control simulation (CTRL) they had a representation of the terrestrial biosphere that emphasised CO2 fertilisation as the dominant control on terrestrial biosphere NPP, and limited or no change (hard to tell from reading) in C4% on the glacial-interglacial

$\delta$13C of the terrestrial biosphere. There is an extended scenario TB+ which emphasises climate as the driver of the terrestrial biosphere, faster response of NPP/terrestrial biosphere and parameterises higher C4% in the LGM (and associated change in the $\delta$13C of the terrestrial biosphere), leading to a combined small effect on deep Pacific $\delta$13C of 0.1 per mil.

However, in discussing the all-important drivers of the changes in atmospheric pCO2, $\delta$13C and deep Indo-Pacific $\delta$13C, and mean ocean $\delta$13C, for termination I, as listed in Kohler et al. (2010) Table 3, C4% and terrestrial $\delta$13C changes are not mentioned. The features listed by Kohler et al. (2010) as the drivers are: lower ocean temperatures, smaller terrestrial carbon storage, lower sea level, weaker NADW formation, enhanced marine export production, larger sea ice cover (gas exchange), higher Southern Ocean stratification.

There is a little more discussion of the C4% and terrestrial biosphere in terms of the LGM and Holocene, which unfortunately is only a small fraction of our 130 kyr period of interest.

For example, Joos et al. (2004) modelled a change in terrestrial biosphere $\delta$13C between the LGM and Holocene of 0.5 per mil. However, they observed the following:

"Changes in the mean terrestrial isotopic signature have a minor impact on the modeled changes in $\delta$13C of DIC. . . . . . . . . .The estimated oceanic $\delta$13C shift is 0.05% smaller than in the standard case, if the land biosphere-atmosphere $\delta$13C difference is kept at the Holocene value of -17 per mil." Menviel et al. (2012a) provided an interesting quote and the following caveat with their modelling of the last glacial termination and Holocene:

"A caveat is that a constant atmosphere-land isotopic fractionation factor is applied in the inverse approach by Elsig et al. [2009] and in this study, therefore not taking into account any relative changes in the occurrence of C3/C4 plants and other influences on fractionation. However, using the LPJ-DGVM vegetation model, Joos et al. [2004]

found that changes in fractionation and C3/C4 plant abundance due to climate and $CO_2$ changes lead to a decrease in $\delta$13C signature of the terrestrial biosphere of about 0.5 per mil from the early Holocene (10 ka B.P.) to pre-industrial times. A 0.5 permil decrease in biosphere $\delta$13C translates into an atmospheric $\delta$13C decrease of about 0.02 permil. This suggests that changes in the atmosphere-land isotopic fractionation have a small influence on the results presented above. "

We note another paper relevant to our manuscript, by Menviel et al. (2016) and focussed on the LGM (18-24 ka), made brief mention of C3/C4, and described that they undertook a sensitivity of -0.7 per mil and +0.5 mil around their average estimate of -23.3 per mil $\delta$13C for the terrestrial biosphere, but the modelling results of that sensitivity are not discussed further in the paper. That type of sensitivity is pretty easy to undertake for analysing only the LGM and the Holocene, as any studies on C3 vs C4 (Kohler and Fisher, 2004; Kaplan et al. 2002, Francois et al., 1999; Joos et al., 2004) have looked at this time period – even though they produce uncertain estimates for the % C3 vs C4 and therefore $\delta$13C fractionation factor. It is a much more difficult proposition to come up with values for a sensitivity for the last glacial-interglacial cycle in its entirety (130,000 years), but that may be an interesting piece of work on its own – future work.

Studies focussed on the terrestrial biosphere

We note the references that focussed specifically on the terrestrial biosphere in detail as the major focus of their work, in the early 2000's, or example those provided by the reviewer (e.g. Collatz et al., 1998; Kaplan et al., 2002, Kohler and Fisher, 2004), and another (e.g. Francois et al., 1999), focused only for the Last Glacial Maximum and PI/modern periods None of them examined the last glacial-interglacial cycle which was ∼130 kyrs in duration. All of these studies above, to our understanding, produced uncertain results.

A recent study devoted to analysing the terrestrial biosphere in detail/major focus

(Jeltsche-Thommes et al. (2019) - suggested by the reviewer), does not mention this feature C3 vs C4%. Jeltsche-Thommes (2019), in their study focussed on the terrestrial biosphere from the last glacial maximum to the Holocene, simply state:

"The $\delta$13C signature of terrestrial carbon is set to $-24$ ‰." (at the top of page 856).

We wondered whether we can contribute something important here with regard to C3%/C4% and the terrestrial biosphere $\delta$13C that has not been considered by any of our peer group of model-data analysis of the last glacial-interglacial cycle.

In summary, there are studies that focussed specifically on the terrestrial biosphere, using dedicated vegetation models. We see that these studies had great detail for the terrestrial biosphere, but were very light on detail for other features of the carbon cycle (ocean circulation and biology, volcanism, weathering, the effects of calcium carbonate compensation). In reviewing these papers, and consistent with our prior understanding, there is not great confidence on quantifying the change in C3 and C4 proportions during the LGM and Holocene, and this is particularly worse during the time period we have analysed up to 130 ka. The papers of Collatz et al. (1998), Kaplan et al. (2002), Kohler and Fisher (2002), all focus on the period LGM-present. There is no coverage of the last glacial cycle 130-20 ka, which is the focus of our study. Furthermore, studies that do focus on the last glacial-interglacial cycle of atmospheric CO2, eg Brovkin, Ganopoloski, do not mention C3 versus C4 fractionation in their papers – making difficult any comparison. We even note that a paper we have referenced in our manuscript, Hoogakker et al. (2016), a paper devoted entirely to the terrestrial biosphere in the last glacial-interglacial cycle, does not address C3 versus C4 plant composition.

As shown above, it is actually an easy process to add the C3 and C4 equations of Collatz et al. (1999) and Kohler and Fischer (2004), and also a temperature dependency for NPP, as we have shown above and with the attached code (Attachment B). We could do this and then re-do the simulations as an appendix or addendum (or a sensitivity).

We could even just apply a sensitivity on the $\delta$13C of terrestrial biosphere of +1/-1 per mil change between LGM and Holocene. However, that's a straightforward exercise for the LGM and Holocene comparison, but it would involve us trying to fit the uncertain LGM-Holocene changes back for the entire last glacial cycle, which is another highly uncertain exercise. We note all of the studies referenced in the reviewer comments and described here, considered C3 vs C4 only for the LGM to Holocene-modern period, but we've explicitly looked at the lead-up to the LGM over the period from 130 ka. We would not like to try to extrapolate changes in C3 v C4 for the LGM over the entire last glacial-interglacial cycle, and implementing the Collatz et al. (1998)/Kohler and Fischer (2004) module would not help us much in that regard as it only explains less than half of the change $\delta$13C of terrestrial biosphere from the LGM (the rest explained by changes in grassland vs forest succession).

There is huge uncertainty around average $\delta$13C factors for plants, and that extends even further to C3 and C4 $\delta$13C, and their possible respective shares and variations. The indicated changes of 0.3-1.8 per mil terrestrial biosphere $\delta$13C between the Holocene and LGM, from the literature described above, are very minor compared to the absolute uncertainties and range in $\delta$13C of the terrestrial biosphere itself.

Summary on terrestrial biosphere and C3 vs C4 photosynthesis

We investigated these topics enthusiastically, based on the reviewer's comments. We're very confident, based on our assessment of the papers above, that our model results will not change by much at all, and the paper conclusions by nothing at all, by varying our approach to the terrestrial biosphere (equally for rock weathering as discussed above). If the CP Journal Editors and the reviewer feel greatly compelled that we need to modify our modelling approach, we certainly can (these would not be major model revisions, only minor adjustments). Our preferred approach, is to simply add a caveat that our model-data experiments don't consider the effects of C3/C4 photosynthesis on $\delta$13C fractionation of the terrestrial biosphere.

We reconstructed this C3%/C4% model of Collatz et al. (1998)/Kohler and Fischer (2004), to investigate the reviewer comments. We have uploaded the Python script of the simple model and data for last glacial-interglacial cycle atmospheric CO2 and temperature, at https://zenodo.org/record/3889704#.XuH3Ji1L0_U.

Amendments to the manuscript

We have added the following text to the model description (P5 L24):

"The terrestrial biosphere is represented in SCP-M as a stock of carbon that fluxes with the atmosphere, governed by parameters for net primary productivity (NPP) and respiration. In SCP-M, NPP is calculated as a function of carbon fertilisation, which increases NPP as atmospheric CO2 rises via a simple logarithmic relationship, using the model of Harman et al. (2011). This is a simplified approach, which omits the contribution of temperature and precipitation on NPP. Other, more complex models of the carbon cycle applied to glacial-interglacial cycles have a more detailed treatment of the terrestrial biosphere, including climate dependencies (e.g. Brovkin et al., 2002; Menviel et al., 2012). A number of studies emphasise the role of atmospheric CO2 as the driver of terrestrial biosphere NPP on glacial-interglacial cycles (Kaplan et al., 2002; Otto et al., 2002; Joos et al., 2004; Hoogakker et al., 2016), although other studies cast doubt on the relative importance of atmospheric CO2 versus temperature and precipitation (Francois et al., 1999; van de Sleen et al., 2015).

The isotopic fractionation behaviour of the terrestrial biosphere may also vary on glacial-interglacial timeframes. This has been studied for the LGM, Holocene and the present day (e.g. Collatz et al., 1998; Francois et al., 1999; Kaplan et al., 2002; Kohler and Fischer, 2004; Joos et al., 2004; Kohn, 2016). The variation in isotopic fractionation within the terrestrial biosphere reflects changes in the relative proportions of plants with the C3 and C4 photosynthetic pathways, but also strong variations within the same photosynthetic pathways themselves (Francois et al., 1999; Kohn, 2010; Schubert and Jahren, 2012; Kohn, 2016). The drivers for these changes include relative sea level

and exposed land surface area (Francois et al., 1999), global tree-line extent (Kohler and Fischer, 2004), atmospheric temperature and CO2 (Collatz et al., 1998; Francois et al., 1999; Kohler and Fischer, 2004; Kohn, 2010; Schubert and Jahren, 2012), global and localised precipitation and humidity (Huang et al., 2001; Kohn, 2010; Schubert and Jahren, 2012; Kohn, 2016), and also changes in the intercellular CO2 pressure in the leaves of C3 plants (Francois et al., 1999).

Estimated changes in average terrestrial biosphere $\delta$13C signature between the LGM and the Holocene fall in the range -0.3-1.8‰ (less negative $\delta$13C signature in the LGM), with further changes estimated from the onset of the Holocene to the pre-industrial, and even greater changes to the present day (due to rising atmospheric CO2). This feature has been covered in detail within studies that focussed on the terrestrial biosphere between the LGM and Holocene, but less so in modelling and model-data studies of the last glacial-interglacial cycle. Menviel et al. (2016) provided a sensitivity of -0.7+0.5‰ around an average LGM value of -23.3‰ for the LGM, based on previous modelling of the LGM-Holocene timeframe by Joos et al. (2004). Another modelling study (Menviel and Joos, 2012), assessed the variation in LGM-Holocene $\delta$13C of the terrestrial biosphere to be a minor factor and it was omitted. Kohler and Fischer (2004) assessed the changing $\delta$13C signature of plants between the LGM and Holocene to be a minor factor in setting $\delta$13C of marine DIC, compared to the change in the absolute size of the terrestrial biosphere across this period.

Given the uncertainty around the starting estimates of $\delta$13C, the uncertain LGM-Holocene changes, the large number of potential drivers, and the further uncertainty in extrapolating the posited LGM-Holocene changes back for the preceding 100 kyr, and the modest changes relative to the average $\delta$13C signature (and the very large range in, for example, present day estimates of C3 plant $\delta$13C (Kohn, 2010, 2016), we omit this feature with the caveat that there is added uncertainty in our terrestrial biosphere results with respect of the $\delta$13C signature applied. We apply an average $\delta$13C signature of -23‰ similar to values assumed by Menviel et al. (2016) and Jeltsch-Thommes

et al. (2019) (23.3‰ -24‰ respectively), but more negative than assumed in Brovkin et al. (2002), Kohler and Fischer (2004) and Joos et al. (2004) (-16-(-17)‰. Our aim is not to contribute new findings of the terrestrial biosphere, but to ensure that the simple representation of the terrestrial biosphere in SCP-M provides the appropriate feedbacks to our (exhaustive) glacial-interglacial cycle model-data optimisation experiments, that are in line with published estimates."

We have also updated the discussion of our model results for the terrestrial biosphere, to provide a bit more detail and some additional references (Section 5.3), plus an additional caveat in the "advantages and limitations section" (P34, L18).

"Furthermore, we apply a simple representation of the terrestrial biosphere in our model-data experiments, relying primarily on atmospheric CO2 as the driver for NPP. This approach provided reasonable results for the terrestrial biosphere carbon stock and NPP, on the whole, but may miss some detail in the terrestrial biosphere during the last glacial-interglacial cycle."

Future work could enhance this set of modelling results with more detail in the terrestrial biosphere. For example, the modelling values for ocean circulation and biology derived here, could be used to solve for the optimal data-matching values for C3 and C4 plant productivity, with separate $\delta$13C-fractionation factors, to help inform that area of study.

Attachment A

Carbonate rock weathering in SCP-M

The reviewer mentioned the model code. In terms of the model equations and model code, the flux of carbon to the ocean from carbonate weathering is set in our model by the following equation (please see O'Neill et al. (2019) and the annotated model code snapshot below):

RVCARB=WCARB*AtCO2 (1)

Where WCARB is a weathering parameter with respect of atmospheric CO2 and is set at 1.5-2.0 mol C/m3/atmosphere. At 275 ppm atmospheric CO2, this is a flux of 10 x1012 mol C annum. (for comparison, this flux is 12 x 1012 mol C annum in Morse and Mackenzie (1990), Zeebe (2012) and Archer et al. (1998), and 14.9 Tmol C annum in Toggweiler (2008). This flux of carbon is added to the low latitude surface box of the model (as per Toggweiler (2008), Zeebe (2012), Hogg (2008)), and alkalinity is added in the ratio ALK:DIC 2:1 (as per Toggweiler (2008), Zeebe (2012), Colbourn et al.,2013) by multiplying RVCARB by 2.0 to create the river flux of alkalinity to the ocean surface boxes. This 2:1 flux of alkalinity:carbon reflects that the initial one mol of CO2 consumed by the carbonate weathering equation, has been returned to the atmosphere (the DIC proportion of 1 is 2 mols less one mol returned to the atmosphere) as per Zeebe (2012) and Lenton and Britton (2006).

The fluxes of DIC and Alk from carbonate weathering are added to the ocean via the river fluxes of C and Alk (see below). This lowers pCO2 in the ocean surface box and therefore draws CO2 from the atmosphere into the ocean, a net sink of CO2 from carbonate rock weathering. We do not subtract a mol of CO2 directly from the atmosphere in our equation for atmospheric CO2, as for the time scale modelled ∼10 kyr, we are taking the short-cut of assuming the CO2 taken up directly from the atmosphere from carbonate weathering, is released back to the atmosphere upon precipitation of CaCO3 into the ocean (Zeebe (2012), Toggweiler (2008)). Carbonate weathering is therefore a flux of carbon and alkalinity to the surface ocean via a river flux, leading to lowering of pCO2 in the surface ocean box and subsequent drawdown of CO2 from the atmosphere. An almost identical approach to ours, was applied by Lenton and Britton (2006), a paper devoted to the study of rock weathering as a sink of atmospheric CO2. The only difference is that Lenton and Britton (2006) applied an additional temperature and terrestrial biosphere dependency on weathering.

We consulted Lord et al. (2016) as suggested in the reviewer comments. Lord et al. (2016) use the cGenie model to estimate weathering feedbacks from atmospheric

CO2 emissions. Lord et al. (2016) is a paper that is devoted to the feedback of rock weathering on atmospheric CO2. The treatment of carbonate weathering, in terms of setting this simply as fluxes of DIC and Alk in the ratio of 1:2 to the surface ocean box, is identical to ours. Where they differ, is because they are looking in much more detail at the effects of terrestrial rock weathering, they also explore other dependencies for rock weathering, such as temperature, terrestrial biosphere productivity and run-off rates. Ours has an atmospheric CO2 dependency, as per Zeebe (2012), Toggweiler (2008), Walker and Kasting (1992).

Silicate rock weathering in SCP-M

The treatment of silicate weathering in the SCP-M model is:

RVSIL=(BSIL+WSIL*AtCO2) (2)

Where BSIL is a constant weathering rate of 0.75 e-4 mol/m3/yr (Toggweiler, 2008), and WSIL is a rate varying with atmospheric CO2, set at 0.5 mol/m3/atmosphere as per Toggweiler (2008). For atmospheric CO2 of 275 ppm, this is a weathering flux of 5.7 x 1012 mol C annum (5 x 1012 mol in Zeebe (2012) and 5.63 x 1012 mol annum in Toggweiler (2008)).

The silicate and carbonate weathering fluxes of carbon, are added to the surface ocean boxes of the box model. Alkalinity is also added, in a ratio of 2:1 to the carbon fluxes (Sarmiento and Gruber (2006), Toggweiler (2008), Zeebe (2012)).

However, there is an additional step applied with silicate weathering. To account for the unit of CO2 consumed directly from the atmosphere in silicate weathering (one more unit than carbonate weathering, as per Zeebe, 2012), and using the approach of Toggweiler (2008), we also subtract an amount equal to a unit of silicate weathering directly from the atmosphere. This is the same approach of Zeebe (2012) who applies a doubling of the flux of silicate weathering, and that of Toggweiler who subtracts a flux of CO2 directly from the atmosphere to account for the additional unit of CO2 con-

Interactive
comment

sumed by silicate weathering (when compared with carbonate weathering). This flux is subtracted directly from Atmospheric CO2 in SCP-M. This flux subtracted from the atmosphere negates the effects on atmospheric CO2 of the units of C added to the ocean by the silicate weathering flux of C. The effect of the more alkaline ocean (alk:C is 2:1 in the silicate weathering flux) is to draw down the volcanic emissions of CO2. Volcanic CO2 emissions are set equal to the amount of CO2 taken directly from the atmosphere by silicate weathering, to reflect the long-term offset of volcanic emissions by silicate weathering (Walker and Kasting, 1992; Archer et al., 1998, Toggweiler, 2008; Zeebe, 2012, Brault et al., 2017). In Walker and Kasting, 1992; Toggweiler, 2008; Zeebe, 2012; Brault et al., 2007, volcanic emissions are also set to the silicate weathering drawdown of CO2.

As described in Walker and Kasting, 1992; Toggweiler, 2008; Zeebe, 2012; Brault et al., 2007, Colbourn et al. (2013, 2015); Lord et al. (2016), in steady state the silicate weathering flux feedback for CO2 matches the volcanic CO2 emissions, which we have set in SCP-M. Note, for anthropogenic scenarios we separate weathering flux from volcanic emissions, as it is clearly a non-steady state simulation, and the silicate weathering feedback, under the forcing of atmospheric CO2, is expected to increase at a greater rate than volcanic emissions.

We note that Zeebe (2012) implements the scheme slightly differently to ours, by subtracting fluxes of carbonate and silicate weathering from the atmosphere, but by doubling the silicate flux to account for the net removal of CO2 from the atmosphere (balanced by volcanic emissions). In Zeebe (2012) when CO2 is returned to the atmosphere from precipitation of CaCO3 in the ocean surface boxes, there is a net zero direct flux of CO2 from the atmosphere from carbonate weathering and a direct flux of CO2 from the atmosphere of 1 mol from silicate weathering.

The SCP-M model code for carbonate and silicate weathering

Below is the description and extract of the original model code presented in O'Neill

et al. (2019) as referenced by the reviewer. A revised model code, incorporating the changes described in this response, will be uploaded with the final manuscript.

Fig. 7 (see figures below) first model code extract annotations:

Line 418 shows the equation (1) above, where the carbonate rock weathering (RV-CARB) is calculated from atmospheric CO2 with the WCARB parameter.

Line 419 shows the equation (X) above where the silicate rock weathering (RVSIL) is calculated from atmospheric CO2 and a constant.

Line 420 the silicate weathering amount to be directly subtracted from the atmosphere, as described above, "weaths", is identified.

Line 423 Volcanic emissions is set to equal "weaths", the direct (net) amount of CO2 taken from the atmosphere by silicate weathering, as described above.

In line 425-428 there is the option to apply an input value for volcanic emissions instead of setting it to equal silicate weathering. This is for the model runs with analysis of anthropogenic emissions/short time frames and is switched off for our experiments.

In Line 431 the net effect of volcanic emissions and silicate weathering on atmospheric CO2 is calculated

In Line 432 the above terrestrial fluxes of carbon can be disabled by a switch (for sensitivities and model testing) via "TerrestrialGeo" (1 is on, 0 = off).

In line 435 the $\delta$13C of silicate weathering drawdown of CO2 from the atmosphere is set to the hardwired value of -6.90. In the revised model code it is now set to atmospheric $\delta$13C within each model time step.

In line 436 the $\delta$13C for silicate weathering direct atmospheric CO2 flux and volcanic emissions of CO2 are applied to their fluxes of carbon and converted to molar concentrations in the atmosphere. (we have now amended the volcanic emissions $\delta$13C to a value of -4.5 per mil).

In line 437 the terrestrial $\delta$13C fluxes can be switched on or off (for model testing or sensitivity) via "TerrestrialGeo" (1 is on, 0 = off).

In line 438 the radiocarbon content (zero, dead) of volcanic emissions and weathering fluxes is applied.

In line 442 both RVCARB and RVSIL fluxes of carbon are added to the surface ocean box via river flux.

In line 443 alkalinity flux is added to the surface box in ration 2:1, leading to a lowering in pCO2 in the surface box and a drop in atmospheric CO2.

Fig. 8 below, second model code extract line by line decscriptions:

In line 475 ocean $\delta$13C is calculated. The river flux of C (derived from weathering) is introduced to the surface ocean box with a $\delta$13C of the standard value "Sstand" ($\delta$13C=0) as discussed above. The dissolution of marine carbonates also introduces carbon with the standard value for $\delta$13C ($\delta$13C = 0) to the ocean boxes.

In line 481 the net fluxes of volcanic emissions and silicate weathering drawdown of CO2 are added to the equation for atmospheric CO2.

In line 484 the $\delta$13C of net fluxes of volcanic emissions and silicate weathering drawdown of CO2 are added to the equation for atmospheric $\delta$13C.

The confusion with the reviewer likely comes from our comment in the model code in line 416 "# As per Toggweiler (2008) only silicate weathering is a sink of CO2 from the atmosphere". We will delete this statement as it is a poor descriptor.

In addition, we should modify the following comment "# Weathering of carbonate rocks is a source of carbon to the low latitude surface ocean via rivers" with "...source of carbon and alkalinity" Therefore, it is indeed the case in SCP-M that both carbonate and silicate weathering ultimately work as sinks of atmospheric CO2 by altering the surface boxes' alkalinity.

[Figure]

The third code extract below (Fig. 9) shows the values chosen for weathering input parameters, as described in the text above. At line 321, weath$\delta$13C is the value that was applied to silicate weathering, NOT carbonate weathering as assumed by the reviewer.

Attachment B

Collatz/Kohler and Fischer terrestrial biosphere C3%/C4% python script and last glacial-interglacial cycles of atmospheric CO2 and temperature data, constructed for this author response:

https://zenodo.org/record/3889704#.XuIDri1L0_V

––––––––––––––––––––––

**Fig. 1.** 1kyr-interval model results for MIS 3-5 compared to proxy data for atmospheric CO2 (top panel) and atmospheric $\delta13C$ (bottom panel), using non MIS-averaged model inputs from the manuscript

| Parameter (units) | Value |
|---|---|
| Terrestrial biosphere $\delta^{13}$C (‰) | -23 |
| Marine biological productivity $\delta^{13}$C (‰) | -19 |
| Carbonate weathering DIC flux $\delta^{13}$C (‰) | 0 |
| Silicate weathering $CO_2$ flux $\delta^{13}$C (‰) | Atmosphere $\delta^{13}$C |
| Volcanic $CO_2$ $\delta^{13}$C (‰) | -4.5 |
| Marine carbonate $\delta^{13}$C (‰) | 0 |
| Air-sea $\delta^{13}$C fractionation factors | 0.9989-0.999 |
| Air-sea D14C fractionation factors | 0.98-0.998 |
| Volcanic $CO_2$ emissions (mol (GtC) yr$^{-1}$) | 6 x $10^{12}$ (0.1) |
| Carbonate weathering flux of C (mol m$^{-3}$ yr$^{-1}$) | 1.5 |
| Silicate base weathering flux of C (mol m$^{-3}$ yr-1) | 7.5 x $10^{-3}$ |
| Silicate weathering slope with respect of atmospheric $CO_2$ (m$^{-3}$ yr$^{-1}$) | 0.7 |
| Calculated carbonate weathering flux of C at 275 ppm atmospheric $CO_2$ (Tmol yr$^{-1}$) | 10 |
| Calculated carbonate weathering flux of C at 190 ppm atmospheric $CO_2$ (Tmol yr$^{-1}$) | 7 |
| Calculated silicate weathering flux of C at 275 ppm atmospheric $CO_2$ (Tmol yr$^{-1}$) | 6 |
| Calculated silicate weathering flux of C at 190 ppm atmospheric $CO_2$ (Tmol yr$^{-1}$) | 5 |
| Terrestrial NPP interglacial base rate (PgC yr$^{-1}$) | 66 |

**Fig. 2.** Table 1: parameterisation of various fluxes of C and $\delta$13C in the modelling experiments

**Carbonate Weathering**

Atmosphere

$(X,Y) = (TCO_2, TA)$

$CO_2$  (1,0)

$CO_2$  (1,0)

$CaCO_3 + CO_2 + H_2O$

$\rightleftharpoons$

$Ca^{2+} + 2\,HCO_3^-$

(2,2)

$Ca^{2+} + 2\,HCO_3^- \rightleftharpoons CaCO_3 + CO_2 + H_2O$

Ocean

$CaCO_3$  (1,2)

Sediment

**Fig. 3.** Extract of the Zeebe (2012) schematic description of carbonate weathering

**Fig. 4.** Modelling of the share of C4 photosynthetic plants (C4%) (bottom panel) and average terrestrial biosphere $\delta$13C fractionation factor (middle panel) as a function of atmospheric CO2 and temperature

[Figure]

**Fig. 1.** *(A)* Histogram of MAP values for isotopically charact.erized C3 plants, showing emphasis on relatively arid ecosystems (MAP ≤ 500 mm/yr) and tropical rainforests (spikes at MAP ~ 2,000, 3,000 mm/yr). *(B)* Histogram of $\delta^{13}$C values of modern C3 plants. Data compiled in this study average −27.0‰, excluding analyses from the understory of closed-canopy forests. Estimated global average composition, based on global trends in precipitation and vegetation, is approximately −28.5‰, significantly lower than typically assumed. An accurate average $\delta^{13}$C value for C3 plants is needed for accurate models of carbon fluxes, atmospheric $CO_2$ compositions, and soil organic matter. *(C)* $\delta^{13}$C values vs. MAP showing increasing $\delta^{13}$C with aridity. Data sources are listed in *SI Text*. White dots are average compositions of data from a large collection made in a single month during a wet year (35).

**Fig. 5.** The Figure below is reproduced from Kohn (2010), and shows the range in $\delta13$C fractionation for C3 plants alone, which spans -20 to -37 per mil, and is impacted by many factors including temp, precip

[Figure]

**Figure 1** Proposed models for factors that influence δ¹³C of C3 plants. **(a)** $p_{CO2}$. Differences are illustrated between geological conditions vs. AD 2000 ($p_{CO2}$ = 370 ppmv, average δ¹³C = -28.5 for C3 biomass). LGM = Last Glacial Maximum. Note inverse relationship between δ¹³C and Δ¹³C. Experiments are for above-ground biomass (Schubert and Jahren, 2012), shifted to fit preferred curve. **(b)** Mean annual precipitation (data and data averages from Kohn, 2010).

**Fig. 6.** Extract from Kohn (2016) showing the range d13C of C3 plants and effects of pCO2 and precipitation

[Figure]

```
414  ## Weathering, river fluxes and volcanic emissions------------------------
415
416      # As per Toggweiler (2008) only silicate weathering is a sink of CO2 from the atmosphere
417      # Weathering of carbonate rocks is a source of carbon to the low latitude surface ocean via riv
418      RVCARB=WCARBs*AtCO2
419      RVSIL=(BSILs+WSILs*AtCO2)
420      weaths=RVSIL*Varr[0,0] #silicate rock weathering sink of CO2, carbonate weathering is a source
421
422      # Volcanic carbon emissions
423      volcs=weaths # volcanic emissions in step with silicate weathering as per Toggweiler (2008)
424      # unless anthropocene scenario, hardwired estimate
425      if AnthEmits==1:
426          volcs=volcs1
427      else:
428          volcs=weaths
429
430      # net source/sink of terrestrial carbon
431      TerrC=(volcs-weaths)/Varrat
432      TCflux=TerrC*TerrestrialGeo
433
434      # weathering and volcanism 13C and 14C fluxes
435      weathd13C=weathd13C
436      TerrSC=(-weaths*weathd13C+volcs*volcd13C)/Varrat
437      TSCflux=TerrSC*TerrestrialGeo
438      TRCflux=TerrC*TerrRC*TerrestrialGeo
439
440      # River fluxes
441      RiverCflux=np.zeros([7,1])
442      RiverCflux[0,0]=(RVCARB+RVSIL) # mol/m3 Incorporates source of carbonate weathering
443      RiverAlkflux=RiverCflux*2.0 #Alk:C ratio 2:1 as per Toggweiler (2008)
444      RiverPflux=np.zeros([7,1])
445      RiverPflux[0,0]=RiverP_mols/Varr[0,0]
446      PSedflux=np.zeros([7,1])
447      PSedflux[5,0]=RiverP_mols/Varr[5,0]
448
```

**Fig. 7.** Original model code extract showing silicate and carbonate weathering equations (see line by line descriptions in Attachment A)

```
464    ## Step forward model calculations--------------------------------------
465
466        # Model equations
467
468        # Ocean boxes
469        Parr = Parr +dt*secsyr*(np.dot(PhysMat, Parr)+BioP+RiverPflux*Rivers-PSedflux*Rivers)
470        Carr = Carr + dt*secsyr*(np.dot(PhysMat, Carr)+BioC+cflux+NetCflux+RiverCflux*Rivers)
471        Alkarr = Alkarr + dt*secsyr*(np.dot(PhysMat, Alkarr)+NetAlkflux+RiverAlkflux*Rivers)
472        Fearr = Fearr + dt*secsyr*(np.dot(PhysMat, Fearr)+BioFe)
473        Siarr = Siarr + dt*secsyr*(np.dot(PhysMat, Siarr)+BioSi)
474        Oarr = Oarr + dt*secsyr*(np.dot(PhysMat, Oarr)+BioO)
475        SCarr = SCarr+ dt*secsyr*(np.dot(PhysMat, SCarr)+BioSC+Scflux+NetCflux*Sstand+
476                           RiverCflux*Sstand*Rivers)
477        SCratio=SCarr/Carr
478        RCarr = RCarr + dt*secsyr*(np.dot(PhysMat, RCarr)+BioRC+Rcflux+NetCflux*(RCarr/Carr)-(RCD1*R
479
480        # Atmosphere
481        AtCO2 = AtCO2 + dt*secsyr*(Atcflux+TCflux-((CFert-Respire)/Varrat)+
482                           ((AnthEmit+DeforestC)/Varrat)*AnthEmits)
483        pCO2a=np.append(pCO2,AtCO2) # create an array of all pCO2
484        SCAt = SCAt+dt*secsyr*(AtSCflux+TSCflux-((CFert-Respire)/Varrat)*TerrBioSC*
485                           TerrestrialBios+(AnthSC1+DeforestSC)/Varrat
486                           *AnthEmits)
487        RCAt = RCAt+dt*secsyr*(AtRCflux+RCS1At/Varrat-RCD1At*RCAt*Varrat/Varrat-TRCflux-((CFert-Resp
488                           *TerrestrialBios+((AnthRC1+DeforestRC)/Varrat)*AnthEmits+(Bomb14C/Var
```

**Reference standard
for d13C=0per mil**

**Fig. 8.** Original model code extract showing application of reference standard d13C (0 per mil) to weathering fluxes to the ocean (see line by line descriptions in Attachment A)

```
310 # Continental weathering atmospheric CO2 sink and flux into oceans
311 # Generally following Toggweiler (2008)
312 WCARB=2.0 # mol/m3/atm/yr as per Toggweiler (2008)
313 WCARBs=WCARB/secsyr
314 BSIL=0.75e-4 # mol/m3/yr as per Toggweiler (2008)
315 BSILs=BSIL/secsyr
316 WSIL=0.5 #mol/m3/atm/yr as per Toggweiler (2008)
317 WSILs=WSIL/secsyr
318
319 # Carbon isotops for volanic emissions and weathering
320 volcd13C=-6.90 #per mil
321 weathd13C=-6.90 #per mil
322 volcd13C=(volcd13C/1000+1)*Sstand
323 weathd13C=(weathd13C/1000+1)*Sstand
324 TerrRC=0.0 # 14C dead
325
```

**If you look above you will see this was applied to silicate weathering direct consumption of atmospheric CO2, not carbonate weathering. We have amended silicate weathering direct CO2 consumption to take atmospheric d13C**

**Fig. 9.** Original model code showing the d13C signature (-6.90 per mil) originally applied to silicate weathering

**Supplement:**

[revised manuscript text omitted]

---

## Author Comment (AC4) · 18 Jun 2020

See below corrected Figure 12 from the manuscript, as per Reviewer #1 comments.

[Figure]

**Fig. 1.** Corrected Figure 12 from the manuscript, as per Reviewer #1 comments. Note the data reconstruction from Hoogakker et al (2016) (grey-shaded area) only partially covers MIS 5e

---

## Author Comment (AC5) · 18 Jun 2020

Please see below corrected Figure 12 from the manuscript, as per Reviewer #1 comments.

[Figure]

[Figure]

**Fig. 1.** Corrected Figure 12 from the manuscript, as per Reviewer #1 comments. Note the data reconstruction from Hoogakker et al. (2016) (grey-shaded area) only partially covers MIS 5e

---

## Author Response (AR1)

Key to Author Responses to Reviews of "Sequential changes in ocean circulation and biological export productivity during the last glacial cycle: a model-data study"

**CP reviewer comments #1 and author responses**

**AC:** We thank the reviewer for their comments and the opportunity to explore some important issues in greater detail. We feel the review comments make a strong contribution to improving the quality of our work. We have followed up on the reviewer comments in detail and have attempted to address them.

We have made reference to changes to the manuscript, which are included as a supplement to the author comments, in track changes. Page and line references below refer to locations in the revised document with track changes.

We have addressed the minor comments first, and then provided a longer discussion on the major comments.

Please note that we have changed our treatment of ocean $\delta^{13}C$ proxy data, stemming from one of the other reviewer comments, to only include $\delta^{13}C$ from *Cibicides* species of benthic foraminifera. We have also made some small changes to the parameterisation of the volcanic and weathering isotopic signatures in the model, from reviewer comments. These changes required the re-calibration of our model and re-running of the model-data experiments. The model-data results changed modestly. We have updated the figures and text (tracked in the attachment) in the manuscript, accordingly.

**Minor comments**

**RC: "Figure 1: It is not clear, how GOC (red arrows) is split up in the part upwelling in Atlantic and Indo-Pacific Ocean."**

AC: In the SCP-M simple carbon cycle box model GOC is split between a part that upwells into the subpolar Southern Ocean, and a part which transports directly into the polar Southern Ocean. This is an attempt to represent the GOC model of Talley (2013). This split is arbitrarily set at 50%. We have added this information to the caption for Figure 1 "GOC upwelling in both basins is set by default to 50% split between upwelling into the subpolar and polar Southern Ocean."

**RC: "Figure 1: Does your approach focusing on changes in GOC, AMOC and export production inply, all other proceses (fluxes) stay constant in time?"**

**AC**: In the model-data experiments we allow GOC, AMOC and biological export production parameter values to vary, and we solve for them in the optimisation. The experiments include specified forcings of SST, salinity, ocean volume, polar Southern Ocean air-sea gas exchange, coral reef carbonate accumulation and cosmogenic 14C production, guided by proxy observations. Other input parameter values are held constant in the experiments.

**RC: Figure 12: x-axis is wrong, eg. MIS5e is between ~114-122 ka, while it has been between ~118-128 ka in other figures.**

**AC**: Thank you. We have fixed the chart.

**RC: With respect to iron fertilisation you might check on Shaffer and Lambert (2018)**

**AC**: Thank you, we have added it to the references throughout the document (P11, L22; P26, L7) and an additional sentence at P26, L8 (note the LaTeXdiff/track changes program struggles to fit reference changes onto the page):

"According to Shaffer and Lambert (2018), fertilisation of the surface ocean, and dust scattering effects on solar radiation, helped to push atmospheric $CO_2$ into and out of their glacial minima, for example at the LGM and last glacial termination."

**RC: The fact that not one single process is needed to explain LGM-Holocene carbon cycle changes is long known, and has been called "the carbon stew" by some authors. You might want to check and discuss in more detail earlier modelling approaches on one glacial cycle (or longer), for example in Ganopolski and Brovkin (2017).**

**AC**: Thanks. We have added to the introduction (P2, L23):

"Several studies have attempted to solve the problem of glacial-interglacial $CO_2$ by modelling either the last glacial-interglacial cycle in its entirety, or multiple glacial-interglacial cycles (e.g. Ganopolski et al., 2010; Menviel et al., 2012; Brovkin et al., 2012; Ganopolski and Brovkin, 2017). These studies highlight the roles of orbitally-forced Northern Hemisphere ice sheets in the onset of the glacial periods, and important feedbacks from ocean circulation, carbonate chemistry and marine biological productivity throughout the glacial cycle (Ganopolski et al., 2010; Brovkin et al., 2012; Ganopolski and Brovkin, 2017). Menviel et al. (2012) modelled a range of physical and biogechemical mechanisms to deliver the full amplitude of atmospheric $CO_2$ variation in the last glacial-interglacial cycle, using transient simulations with the Bern3D model. According to Brovkin et al. (2012), a ~50 ppm drop in atmospheric $CO_2$ early in the last glacial cycle was caused by cooling sea surface temperatures (SST), increased Northern hemisphere ice sheet cover, and expansion of southern-sourced abyssal waters in place of North Atlantic Deep Water (NADW) formation. Ganopolski and Brovkin (2017) modelled the last four glacial cycles with orbital forcing as the singular driver of carbon cycle feedbacks. They described the "carbon stew", a feedback of combined physical and biogeochemical changes in the carbon cycle, to drive the last four glacial-interglacial cycles of atmospheric $CO_2$."

We have also added the following to our discussion (P28, L10):

"Ganopolski et al. (2010) and Brovkin et al. (2012) modelled cooling SST and substitution of North Atlantic Deep Water by denser waters of Antarctic origin, in the abyssal ocean, as the main drivers of falling atmospheric $CO_2$ at the last glacial inception. Menviel et al. (2012) modelled a transient slowdown in the rate of overturning circulation in the North Atlantic across MIS 5d-5e."

**RC: Section 5.3. You might want to check on recent finding of terrestrial carbon storage from δ13C in Jeltsch-Thömmes et al. (2019).**

**AC**: Thanks, we have added this to our discussion of the terrestrial biosphere (P33, L24):

Jeltsch-Thommes et al. (2019) estimated a glacial-interglacial change in terrestrial biosphere of 850 Pg C (median estimate; range 450 to 1250 Pg C). Jeltsch-Thommes et al. (2019) demonstrated the importance of including ocean-sediment and weathering fluxes in their modelling estimates, and suggested other studies may underestimate the full deglacial change in the terrestrial biosphere carbon stock.

**RC: Figure 14: Changes in CO2 caused by changes in terrestrial NPP and carbon stocks are missing in this figure. Please add.**

**AC**: We have incorporated the contribution of the terrestrial biosphere to the glacial $CO_2$ drawdown in Figure 14. We have shown the effect of the model run with- and without the terrestrial biosphere to estimate its effects, as per Ganopolski and Brovkin (2017) Figure 9b, and we have compared with their model output.

**Major comments**

**RC: Overall recommendation**

**My recommendation therefore is, that the model in its present form might be a useful tool for evaluating marine processes, and might be well used together with the available marine data (apart from δ13C), but fails to give meaningfull results for the δ13C cycle. This includes atmospheric and marine δ13C. I urge the authors to get those parts out of the manuscript. If they wish to further analyse the δ13C-cycle I believe fundamental model improvements are necessary, that can not be obtained by a major revision, but by a revised model version. Besides this, shortcomings of the steady-state approach should be discussed in more detail and the unclear (wrong?) aspects of carbonate weathering and annual fluxes in/out of the simulated system (atmosphere/ocean) need to be clarified for each MIS, maybe in a table or a new figure.**

**AC**: We've discussed the comments of the reviewer, and clarified various parts of the modelling referred to by the reviewer, in quite some detail below. We've also made some small adjustments to $\delta^{13}C$ parameters for volcanic source carbon and silicate weathering, in the model, and incorporated those in the revised modelling results in the updated manuscript. We've clarified what our model-data results are saying about $\delta^{13}C$ in MIS 3-5, and clarified how our model deals with carbonate weathering, with specific reference to the literature and the model code (annotations provided in the Attachments to our responses). We've also discussed features of the terrestrial biosphere in more detail. We've discussed the issues associated with MIS-averaging, revised our wording in the manuscript, and also provided a better description of the model-data results. There is an updated model code to upload with the finalised manuscript.

As a more general comment on $\delta^{13}C$, we demonstrated in O'Neill et al. (2019) that the model we have used, replicates the modern atmosphere and ocean $\delta^{13}C$ data time series, and replicates the effects of anthropogenic emissions on ocean and atmospheric δ13C, including matching atmospheric data time series for the last 250 years, and GLODAPv2 data for the present, and matches Holocene data, and successfully matched LGM proxy data.

We argue that the model-data results and manuscript are best left with the $\delta^{13}C$ material retained, however with appropriate caveats to describe the shortcomings, as laid out below.

RC: "The chosen approach of steady-state analysis combined with optimization is a way, which certainly has benefits, but also shortcomings. I believe the benefits lie in the possibility to test a great number of parameter values, and this is certainly analysed with great effort and detail and worth publishing (but see my recommendation on shortenings of certain parts below). However, there is little learned on the potential short- comings and pitfalls, which in my view need to be discussed more deeply. I believe where this approach is falling to short is the following: By analysing only steady-state the authors miss out the opportunities to judge the results based on the timing (when do processes change leading to what results). I provide one example where the article nicely fails, producing a potenially right answer for very likely the wrong reason: One of the dominant features of atmospheric δ13C during the last glacial cycle is a drop by about 0.5‰ during MIS4. The steady-state approach now leads to the evalution of a mean value of atmospheric δ13C which does not really cover this decrease at all, it shows about a decline by about 0.2‰ from MIS5a to MIS4 (Fig 4). So, any explanation for this drop would be falling too short in the observed amplitude by 0.3‰."

RC: This is an interesting debate, and we thank the reviewer for the opportunity to explore this.

The aim of our study is to help diagnose the causes of the major changes in atmospheric $CO_2$ during the last glacial cycle. As we identify in our manuscript, there are three particularly large, sustained falls in atmospheric $CO_2$ between the penultimate interglacial (~125 ka) and the LGM (18-24 ka). These three major changes in atmospheric $CO_2$ are summarised well in the literature, for example, in Kohfeld and Chase (2017). Our aim, with our model-data study, is to understand if plausible changes in ocean circulation (GOC and AMOC) and marine biological productivity can explain the major falls in atmospheric $CO_2$. Other proxy data (e.g. δ13C) provide useful data constraints for a model-data study to help solve this problem. Our approach is to apply a model-data optimisation with a simple carbon cycle box model, to solve for major ocean carbon cycle parameter values during the last glacial-interglacial cycle, and to explain the major, non-transient, falls in atmospheric $CO_2$. To our knowledge, this is the first time that someone has attempted a multiple proxy model-data optimisation, that is optimised against atmospheric $CO_2$, $\delta^{13}C$, $\Delta^{14}C$, ocean $\delta^{13}C$, ocean $\Delta^{14}C$, and carbonate ion proxy data, and hard-constrained by many observational data (SST, salinity, ocean volume, sea-ice cover proxy, coral reef carbonates, atmospheric [14]C production rate) for the last glacial-interglacial cycle of 130 kyr. This study is quite different and unique in this regard.

This is done in an average sense across each MIS (nine of them over the last 130 kyr), using average proxy data values for each MIS and solving for the average parameter values at each MIS over the last 130 kyr. The MIS timeframes were chosen as an accessible reference point to the scientific community and because they are also simple reference points for the major atmospheric $CO_2$ declines in the last glacial cycle. In this way, we may not solve for maximum or minimum values in the parameters, "overshoots" and "undershoots", within each MIS, but the changes in the average values across the last glacial-interglacial cycle. We think the article has been successful in achieving what it set out to do.

The aim of our study is not to disentangle the transient or shorter-term changes in the carbon cycle within MIS stages. Other studies (e.g. Eggleston et al., 2016) have done that excellently for their area of focus (e.g. MIS 3-4 atmospheric $\delta^{13}C$), and other modelling studies have attacked this using transient simulations (e.g. Ganopolski, 2010; Menviel et al., 2012). Our study does successfully diagnose the timing of changes in major oceanic processes that drive major changes in atmospheric $CO_2$ during the last glacial-interglacial cycle, that are hard-constrained/brutally optimised by a host of data and observations.

With regard to your comments about MIS 3-5 and our manuscript and modelling, below, we address that in more detail to clarify what it is (and isn't) that our model-data results are telling us, and what we should have said about transient changes in our original manuscript. We clarify things quite substantially, but we are still very happy to explore the shortcomings of the approach and we have amended the manuscript with additional caveats, as explained below.

**RC: Note that this δ13C feature is not rapid, it is an anomaly that has been detected from raw data by spline smoothing and is alltogether nearly 20 kyr long, however the decreasing flank falls in MIS4, the increasing flank in MIS3, thus the signal is largely smoothed out in the chosen MIS-centric analysis. The analysis of the results now comes to the conclusion that very likely changes in terrestrial carbon storage was responsible for a change in atmospheric δ13Caˇ of -0.2‰ (as said explaining a too little amplitude), it is furthermore said that the drop is accompanied by a 30 ppm fall in CO2 (page 12, lines 1-5), citing Hoogakker et al., 2016. I believe this is entirely wrong: The drop in CO2 happens clearly a few kyr before the drop in atmospheric δ13C, as seen in Fig. 4. Furthermore, since both CO2 and δ13C are meassured at the same samples and are both derived from gases in ice cores, this temporal offset between CO2 and atmospheric δ13C can not be explained by chronological issues. The anomalies in biosphere as documented by Hoogakker et al., 2016 all fall in line with the CO2 changes, but not with the δ13C changes, also note that Hoogakker et al., 2016 was published before the atmospheric δ13C data set of Eggleston et al. (2016). In that respect citations from Hoogakker on page 19 are also missing the correct timing: In Hoogakker NPP drops between around 70 ka (parallel to the drop in CO2), while the δ13C drop occurs 5 ka later. Also note, that in Eggleston et al. (2016) the authors of this atmospheric δ13C record tried to make sense of it by focusing on the part in which δ13C falls, but CO2 rises again (Fig 2 in that paper) focusing on an opposite behaviour than described here.""**

AC: There is a major confusion here, that we will spend some time below to help with.

There are two minor comments in our original manuscript about the terrestrial biosphere and atmospheric $\delta^{13}$C at MIS4. They are not conclusions of our work, nor are they a result of our model-data experiments, and thus don't reflect any obvious or glaring deficiency in our modelling or model-data analysis that we are aware of. The two comments are just peripheral statements we made about the terrestrial biosphere in MIS 4, with a very quick look at the atmospheric $\delta^{13}$C data, without looking in any detail, as this excursion in the $\delta^{13}$C pattern is not the focus of our work. In the two short sentences in our manuscript, we made casual reference of the transient, reversing change in atmospheric $\delta^{13}$C across MIS 4 and MIS 3 (termed an "excursion" by Eggleston et al., 2016). We stated that the transient drop in $\delta^{13}$C probably reflects a weaker terrestrial biosphere, based on reconstruction of the terrestrial biosphere for the same period, by Hoogakker et al. (2016). The first of these comment is in the "Data Analysis" section, and was just the result of a quick eyeballing of the atmospheric $\delta^{13}$C data and another study on the terrestrial biosphere covering approximately the same time period (Hoogakker et al., 2016). You correctly point out our oversimplification of the true complexity of MIS 3-5, in our short statements.

This is simply an oversight on our part, in drafting the text and not joining the dots between the various data sources we have gathered, and our model-data results. If we look at this in a little more detail, with reference to the figures and tables in our manuscript (drawing the reviewer's attention to them here), we can provide the following (below).

If we look at Figure 2 in the manuscript, we can see that there are dramatic changes in SST (top panel), and less dramatic changes in salinity, sea-ice proxy, sea level/ocean volume, and reef C carbon between MIS 3-5. These data are well incorporated in our model-data experiments as forcings – or constraints, or, another way of saying it - values fed into the model - they hard hard-baked into our model-data results and are influencing the results. With those forcings included, our model-data experiments solve for changes in GOC, AMOC and SO Bio across MIS 3-5, and we find important changes in these parameters across MIS 3-5 (see Figure 8 where GOC, AMOC drop and Southern Ocean biological productivity increases). Therefore, our model-data experiments, and what we should say in our manuscript, is that there are large changes in SST and other observations in the ocean during MIS 4 and MIS 3 (Figure 2 top panel), as well as the changes we estimate for GOC, AMOC and Southern Ocean biological export productivity. It is likely that the combination of these features, led to the $\delta^{13}$C pattern during MIS 3 and 4. We also note that Eggleston et al. (2016) posited changes in SST, ocean biological productivity, AMOC and Southern Ocean upwelling to explain the $\delta^{13}$C "excursion" at MIS 3-4. There are also changes in the terrestrial biosphere, but as per the reviewer comments about timing with regards of changes in the terrestrial biosphere, atmospheric $CO_2$ and $\delta^{13}$C, and on closer inspection we can see that this is perhaps not a dominant driver but some background factor or simply a part of the MIS 3-4 $\delta^{13}$C pattern.

However, it is clear, as you point out, that our MIS-averaging approach does not capture the full extent, the overshoots and undershoots, of the changes in atmospheric $\delta^{13}$C across this period of MIS 3-5.

To address the reviewer comments, we simply reword the sentence in "Data analysis" you refer to (i.e. before the model results section), to better reflect the data we have used and

how that data is described (e.g. Eggleston et al., 2016) and also the literature that has focussed in detail on atmospheric $\delta^{13}C$ at MIS 3-4.

Original text: "*The large drop in $\delta^{13}C$ in MIS 4 accompanies a ~30 ppm fall in $CO_2$. The drop in $\delta^{13}C$ is likely caused by a reduction in the terrestrial biosphere, itself driven by the fall in $CO_2$ (Hoogakker et al., 2016).*"

We reword this and include the caveat about how our MIS-averaging does not include the full amplitude of changes within MIS 4 and 3, at P15 L23:

"The large drop in $\delta^{13}C$ in MIS4, reverses in MIS 3 (Fig. 4(B)). This excursion in the $\delta^{13}C$ pattern likely resulted from sequential changes in SST (cooling), AMOC, Southern Ocean upwelling and marine biological productivity (Eggleston et al., 2016). Eggleston et al. (2016) parsed the atmospheric $\delta^{13}C$ signal into its component drivers across MIS 3-5, using a stack of proxy indicators, and highlighted the sequence of events between the end of MIS 5 and beginning of MIS 3, and their cumulative effects to deliver the full change in atmospheric $\delta^{13}C$. Our MIS-averaging approach fails to capture the full amplitude of the changes in atmospheric $\delta^{13}C$ during MIS 3-5, and only captures the changes in the mean-MIS value, serving to understate the full extent of transient changes in responsible processes. In addition, the MIS-averaging approach misses the sequential timing of changes in processes within each MIS. These are limitations of our steady-state, MIS-averaging approach. "

Then on P22 L24, in reference to the terrestrial biosphere:

Original text: "*Notably, there is a distinct drop in NPP at MIS 4, a period where atmospheric $CO_2$ falls by ~30 ppm (Fig. 4(A)). Falling NPP and persistent respiration of the terrestrial biosphere carbon stock during MIS 4, which releases δ13C-negative carbon to the atmosphere, can explain the steep drop in atmospheric δ13C during the same period (Fig. 4(B)).*"

We simply delete the reference to the terrestrial biosphere and atmospheric δ13C (in red above).

Plus, we have added a caveat to the discussion on limitations of the work that our MIS-averaging approach misses the full amplitude of transient changes (P34, L21).

"Our MIS time-slicing obscures details in the proxy records within MIS. For example, Yu et al. (2013) observed a transient drop in carbonate ion concentrations in the deep Pacific Ocean during MIS 4, and there are large transient changes in atmospheric $\delta^{13}C$ during MIS 3-4. Ganopolski et al. (2010) and Menviel et al. (2012) modelled transient collapses and rebounds in AMOC during MIS 4 (and other short-term changes in atmospheric dust supply and depth of biological nutrient remineralisation), which could have contributed to the full observed magnitude of changes in atmospheric $\delta^{13}C$ across this period (e.g. Eggleston et al., 2016) - not captured with our MIS-averaging approach."

However, what we are getting at in our response here, is that although our MIS-averaging strategy misses the full amplitude of transient changes or "excursions" in the proxy record, this doesn't mean that we don't meaningfully capture the data signals across the glacial-interglacial cycle, in our model-data experiments, as data constraints on our model-data experiments. This is a more nuanced but very important point, that we explore in more detail in the following.

A closer look at MIS 3-5 atmospheric $CO_2$ and $\delta^{13}C$ and our model-data results

Our model-data experiments at MIS 3,4 and 5, contain forcings of the model with observationally-derived SST, salinity, sea-ice proxy, sea level/ocean volume, and reef carbonate carbon fluxes. In addition to the model forcings, our MIS-averaged model-data results show a fall in GOC and AMOC, and an increase in Southern Ocean biological export productivity from MIS 5 into MIS 4. This outcome is supported by many proxy observations from the ocean for this time period including ocean carbonate ion proxy, ocean δ13C and dust records for the Southern Ocean and intense cooling in the North Atlantic Ocean (e.g. Oliver et al., 2010; Yu et al., 2016; Kohfeld and Chase, 2017). In addition, our results agree with transient modelling of the last glacial-interglacial cycle, across this period MIS 3-5. Ganopolski et al. (2010), Brovkin et al. (2012) and Menviel et al. (2012) all show a slowdown in AMOC at this time. Ganopolski and Brovkin (2017), in Figure 9(c) in their paper, model a contribution to atmospheric $CO_2$ drawdown from dust iron-fertilisation of Southern Ocean marine biological productivity in MIS 4. Therefore, our model-data conclusions for MIS 4 are consistent with the proxy data and also transient modelling exercises for this period.

Further to the review comments, we undertook a simple reconnaissance modelling experiment to test our MIS-average model-data results, at more detailed time intervals, against the non-MIS averaged data for atmospheric $CO_2$ and $\delta_{13}C$ through the MIS 3-5 period, to see if they hold up.

Eggleston et al. (2016) attempted to disentangle transient changes in the atmospheric $\delta^{13}C$ pattern during MIS 4 and MIS 3 (Heading 4.2 in their paper "Transient Changes at the Onset and End of MIS 4"). The first process they identify is iron fertilisation from dust over the Southern Ocean and a possible increase in SO biological export productivity (as above, we modelled increased SO biological export in this period too) to lower atmospheric $CO_2$ (but this would increase atmospheric $\delta^{13}C$). Then, Eggleston et al. (2016) mention cooling SST (a key part of our model constraints), where they plot a global average (we model latitude bands), which would also lead to lower atmospheric $CO_2$ as well as lower $\delta^{13}C$ (colder ocean fractionates more $\delta^{13}C$). Then, they mention slowing AMOC as a minor cause of lower atmospheric $CO_2$ and higher $\delta^{13}C$. Then, Eggleston et al. (2016) mention the effects of carbonate compensation and ocean alkalinity in lowering atmospheric $CO_2$, and with minor effects on $\delta^{13}C$ (captured in our model in MIS average).

Then, Eggleston et al. (2016) mention that Nd isotope and Pa/Th ratios in proxy data support a more pronounced slowdown in AMOC, which lasted until the end of MIS 4 (also in our model-data result for MIS 4, as discussed above and in the manuscript). Eggleston et al. (2016) discuss a weakening and shoaling of AMOC and expansion of AABW at this time, and

quote the hypothesised changes to AABW and AMOC of Ferrari et al. (2014). This would have lowered atmospheric $CO_2$ but increased atmospheric $\delta^{13}C$.

Then, Eggleston et al. (2016) mention that iron dust fertilisation may have reduced at the end of MIS 4 (showing the dust proxy data as evidence), leading to a drop in SO biological productivity, which would increase $CO_2$ and lower atmospheric $\delta^{13}C$, reversing the hypothesised changes in early part of MIS 4 (note our results show SO biological export productivity drops off from MIS 4 levels at MIS 3 – Figure 8 in the manuscript). At this time, SST warmed a small amount in the SO, and cooled in the North Atlantic, with presumably offsetting effects. According to Eggleston et al. (2016), quoting opal flux data, a short-term increase in SO upwelling likely led to the final drop in atmospheric $\delta^{13}C$ to reach the trough of the $\delta^{13}C$ pattern near the boundary if MIS 3-4 (not captured in our MIS-averaged modelling).

We forced our model with the data in Figure 2 in the manuscript, without averaging for the MIS stages, over 1kyr intervals for the period 47-75 ka. We then took our model-data results for the average parameter values across MIS 3-5 as shown in the manuscript, and profiled them to vary within each MIS according to the hypothesised changes from Eggleston et al. (2016), described above, also at 1 kyr intervals. In this way, we allow the parameters to vary within the MIS stages, but constrained to meet the MIS-average values, in their average, from our model-data experiments as shown in the manuscript.

Figure 1 below (top panel) shows model-data results compared with the proxy data for atmospheric $CO_2$ and Figure 1 (bottom panel) shows the same for atmospheric δ13C. This shows that taking the forcings for SST, salinity, sea-ice cover proxy, sea level/volume and coral reef carbonates (as per Table 2 of the manuscript), and time-profiled average MIS values for MIS 3-4-5 from the model-data experiments (taking the averages from Figure 8), accounts for the full amplitude of changes in atmospheric $CO_2$ and atmospheric $\delta^{13}C$ across this period, the overshoots and undershoots. The model-data results also account for the MIS-averaged proxy data across MIS 3, MIS 4 and MIS 5, as per our manuscript. The model results oscillate relative to the δ13C data, due to the 1 kyr intervals we have applied (we understand the $\delta^{13}C$ data has been smoothed), but it is easy to make this a 1 year/1 second interval exercise and it will produce a smoother set of results, for future analysis (in another body of work).

Completing this analysis to match the transient atmospheric $CO_2$ and $\delta^{13}C$ data across MIS 3-5, does not change our findings as presented in the manuscript, but actually reinforces them.

**Figure 1**: 1kyr-interval model results for MIS 3-5 compared to proxy data for atmospheric $CO_2$ (top panel) and atmospheric $\delta^{13}C$ (bottom panel). These model runs take as inputs the carbon cycle forcings from Table 1 in the manuscript, and our average values for GOC, AMOC and Southern Ocean biological export profiled with the pattern described by Eggleston et al. (2016). Atmospheric $CO_2$ data from Bereiter et al. (2015), and $\delta^{13}C$ data from Eggleston et al. (2016)

[Figure]

Therefore, while the model-data results we present in the manuscript do not describe fully the transient or short-term changes in the carbon cycle within each MIS, they are not inconsistent with the transient data observations – as evidenced by a 1 kyr-interval extension of our model-data results for MIS 3-5 (Figure 1), and comparison with proxy data and other modelling studies. Our model-data results show that, on average, GOC and AMOC weakened in MIS 4, and SO biology on average, was stronger, although these values fluctuated around their mean values within the MIS. We emphasise that these findings above are still peripheral to the main objective of our manuscript (major, sustained changes in atmospheric $CO_2$ through the last glacial-interglacial cycle), although this helps shed some more light on our model-data results in the context of the reviewer comments.

**RC: "The second most dramatic change in atmospheric δ13C is a sharp drop by 0.2‰ during Termination I, a time window which has been chosen to be not be included in this steady-state analysis, again missing the opportunity to use 13C to pin down responsible processes.."**

AC: We disagree. We argue that the responsible processes for the major and sustained changes in atmospheric $CO_2$ over the last glacial-interglacial cycle (e.g. SST, ocean circulation, biological export productivity, sea level, coral reefs, salinity, terrestrial biosphere) actually show themselves much more clearly, in sequence over the last 130 kyr, than the very short last glacial termination – whereby many processes were interacting in a relatively, very short period of time, and not easily untangled.

Our approach to attempting to solve for large changes in atmospheric $CO_2$, is to study the 100 kyr lead-up to the LGM, where the large changes separate out much more clearly into unique events over 100 kyr. Many studies have attempted to answer the problem of glacial-interglacial $CO_2$ by focussing on the LGM and Holocene periods alone (e.g. Peterson et al.,

2014; Menviel et al. 2016; Muglia et al., 2018). Others may try to get at this by looking at 10-18 ka period with transient modelling, where all the changes in the carbon cycle rapidly unwound (e.g. Menviel et al., 2012; Joos et al., 2004, Ganopolski and Brovkin (2017)), but that's not our paper. That's almost an entirely different approach to the explicit approach of our paper which was NOT to focus on the transient termination of the last glacial maximum, which has been studied in great detail elsewhere. Our paper is focussed on the major, non-transient, drops in atmospheric $CO_2$ in the lead-up to the last glacial maximum over 100 kyr – a much longer period that nicely shows up the sequential changes in the carbon cycle.

We have added references to the studies mentioned, to point readers in that direction if that is their area of focus, at P8 L7.

"We are interested in the LGM and Holocene as discrete periods, so our experiment time slice for MIS 2 is truncated at 18 ka, and our MIS 1 simply covers the Holocene, removing overlaps with the glacial termination. Therefore, our modelling excludes the last glacial termination (~11-18 ka). The glacial termination period was highly transient, with atmospheric $CO_2$ varying by ~85 ppm in <10 kyr, and large changes in carbon isotopes. Thus, it is anticipated that in a model-data reconstruction, model parameters would vary substantially for this period. Our strategy of integrating the model forward to an equilibrium state for each MIS as intervals of discrete climate and $CO_2$, would be unsuitable when applied to the last glacial termination. Joos et al. (2004), Ganopolski et al. (2010), Menviel et al. (2012), Menviel and Joos (2012), Brovkin et al. (2012) and Ganopolski and Brovkin (2017) provide coverage of the termination period with transient simulations of the last glacial-interglacial cycle, using intermediate complexity models (more complex than our model)."

Further, in our discussion of limitations of the study (P34 L27):

"We omitted the transient last glacial termination from our analysis, a period in which atmospheric $CO_2$ rose ~85 ppm in 8 kyr. Future model-data optimisation work could probe this period at 1 kyr intervals, or with transient, data-optimised simulations, to profile the unwinding of processes that led to the last glacial cycle $CO_2$ drawdown."

**RC: Only the long-term trend in δ13C of +0.2‰ from the penultimate interglacial to the Holocene seemed to be meaningful covered by the approach**

AC: The change in $\delta^{13}C$ across the last glacial-interglacial cycle is a bit larger than +0.2 per mil, as stated by the reviewer comment. The change in atmospheric $\delta^{13}C$ is quoted as +0.4 per mil in the literature (e.g. Schneider et al., 2013; Eggleston et al., 2016) and is a very important feature of the last glacial-interglacial cycle of atmospheric $\delta^{13}C$. Noted our MIS-averaging also understates this full variation, but it is a very important long-term and sustained feature of the last glacial-interglacial cycle, so it is important that our analysis meaningfully captures this feature. As per Eggleston et al. (2016):

"*Due to the lack of a complete δ13C(atm) record connecting the various data sets, unanswered questions remained. **Most importantly,** the penultimate glacial maximum (PGM) was found to be 0.4‰ isotopically lighter in δ13C(atm) than the Last Glacial Maximum (LGM), and the penultimate warm period (marine isotope stage (MIS) 5e) was*

*also more negative in $\delta13C(atm)$ by a similar amount. This is a surprisingly large difference, on the order of the changes in $\delta13C(atm)$ observed during glacial terminations*."

While we don't focus on the MIS 3-5 transient $\delta^{13}$C excursion, with better explanation provided above we can demonstrate our approach produces results that are consistent with more detailed interpretations of the transient proxy record, such as Eggleston et al. (2016).

**Reviewer comments: the δ13C cycle**

AC: We address the reviewer comments individually, and then provide annotated snapshots of the model code as supporting evidence in the Attachment A to our author comments.

**RC: "As already seen above the steady-state approach might not be the best way to tackle atmospheric δ13C. Furthermore, for an evaluation of δ13C in general in such steady- state experiments as performed here the fluxes (e.g. as mol C/yr) and δ13C-signatures in/out of the simulated atmosphere/ocean carbon cycle are essential: atmosphere- land carbon fluxes, volcanic CO2 outgassing, weathering, and burial of organic and inorganic carbon in the sediments. Little to non of those fluxes (and δ13C-signatures) are given in the text itself. If I dig into the python source code of the model (or the description of version 1 in O'Neill et al. (2019)) I find a few information, but the source code is difficult to interpret as a non-user and some information seemed to be either misleading or wrong. An examples: Continental weathering consists of two different processes depending on the rock type that is weathered. In carbonate weathering 1 mol of CaCO3 together with 1 mol of CO2 from the atmosphere leads to the entry of 2 mol of HCO−3 into the surface ocean. In silicate weathering 2 mol of atmospheric CO2a˘ are necessary to weather 1 mol of CaSiO3 leading again to the entry of 2 mol of HCO−3 into the surface ocean. For details see, for example Lord et al. (2016). From the description of weathering in O'Neill et al. (2019) I have the impression that the carbonate weathering is not depicted correctly (no consumption of atmospheric CO2). "**

AC: Re carbon fluxes/$\delta^{13}$C. We have added the below table to the Supplementary Information to describe the various prescribed fluxes of C and $\delta^{13}$C signatures in our (revised) modelling exercise. This includes some changes from the original model/and model-data runs, from this set of review comments, and also the other reviewer comments. These changes in the model from the revised model-data experiments, will be uploaded with the final manuscript to a new Zenodo link.

Further below, we have clarified our treatment of carbonate and silicate weathering, carbonate weathering $\delta^{13}$C signature, and we have modified our volcanic $\delta^{13}$C and silicate weathering $\delta^{13}$C signatures in the model (incorporated in the revised manuscript model-data results provided in the revised manuscript), in response to the reviewer comments.

**Table 1**: parameterisation of various fluxes of C and δ13C in the modelling experiments (**CP_RC1_Tab1.png**)

| Parameter (units) | Value |
|---|---|
| Terrestrial biosphere $\delta^{13}C$ (‰) | -23 |
| Marine biological productivity $\delta^{13}C$ (‰) | -19 |
| Carbonate weathering DIC flux $\delta^{13}C$ (‰) | 0 |
| Silicate weathering $CO_2$ flux $\delta^{13}C$ (‰) | Atmosphere $\delta^{13}C$ |
| Volcanic $CO_2$ $\delta^{13}C$ (‰) | -4.5 |
| Marine carbonate $\delta^{13}C$ (‰) | 0 |
| Air-sea $\delta^{13}C$ fractionation factors | 0.9989-0.999 |
| Air-sea D14C fractionation factors | 0.98-0.998 |
| Volcanic $CO_2$ emissions (mol (GtC) yr$^{-1}$) | 6 x 10$^{12}$ (0.1) |
| Carbonate weathering flux of C (mol m$^{-3}$ yr$^{-1}$) | 1.5 |
| Silicate base weathering flux of C (mol m$^{-3}$ yr-1) | 7.5 x 10$^{-3}$ |
| Silicate weathering slope with respect of atmospheric $CO_2$ (m$^{-3}$ yr$^{-1}$) | 0.7 |
| Calculated carbonate weathering flux of C at 275 ppm atmospheric $CO_2$ (Tmol yr$^{-1}$) | 10 |
| Calculated carbonate weathering flux of C at 190 ppm atmospheric $CO_2$ (Tmol yr$^{-1}$) | 7 |
| Calculated silicate weathering flux of C at 275 ppm atmospheric $CO_2$ (Tmol yr$^{-1}$) | 6 |
| Calculated silicate weathering flux of C at 190 ppm atmospheric $CO_2$ (Tmol yr$^{-1}$) | 5 |
| Terrestrial NPP interglacial base rate (PgC yr$^{-1}$) | 66 |

**Re carbonate and silicate weathering fluxes**

We consulted Lord et al. (2016), as recommended by the reviewer in the RC above, and we note that the approach to carbonate weathering of Lord et al. (2016) is identical to ours, in that the activity of carbonate rock weathering simply transfers fluxes of DIC and Alk (in ratio 1:2) to the ocean via rivers, which causes a sink of $CO_2$ to the ocean, and their treatment of silicate weathering is very similar to ours (see below, where we looked into more detail in the rock weathering model of Lord et al. (2016), which is described in detail in Colbourn et al. (2013)).

For example, in Lord et al. (2016):

"In all schemes, the terrestrial rock-weathering module calculates global fluxes of ALK and

DIC from carbonate and silicate rock weathering and routes them to the coastal ocean".

And importantly, as described in in Colbourn et al. (2013), the carbonate weathering model used in Lord et al. (2016):

"Note that there is only one mole of DIC for each mole of $Ca^{2+}$; this is a short-circuiting of the atmosphere based on the assumption that the atmosphere and surface ocean are well mixed on the timescales considered here. Instead of removing one mole of CO2 from the atmosphere – and by implication the ocean – and adding two **moles of bicarbonate to the ocean nothing is taken from the atmosphere and one mole of bicarbonate is added to the ocean**."

In addition to Lord et al. (2016), we also found our approach for carbonate and silicate weathering to be identical to a range of other studies – they are shown and referenced below. We also found our approach to δ13C in carbonate weathering, as shown by the model code as shown in the Attachment to these comments (with line-by-line annotation) was identical to that used in Sano and Williams (1996) and Mook (1986), the references suggested by the reviewer for us to consult (see RC below).

As pointed out by the reviewer, some confusion for the reader about carbonate and silicate weathering, is perhaps contributed from our simple, high level model description paper which glosses over some details (O'Neill et al., 2019) and perhaps non-user friendliness of the model code. We will add better descriptive text in our model code for the final model upload to this manuscript upon finalisation. We've provided line-by-line references to our model code in the Attachment to these responses, to help understanding.

**Further on the treatment of carbonate and silicate weathering in SCP-M**

The treatment of carbonate and silicate weathering in SCP-M is described in O'Neill et al. (2019) and mainly takes into account Walker and Kasting (1992), Toggweiler (2008) and Zeebe (2012) for its basis. Walker and Kasting (1992) provides the theoretical basis for treatment of carbonate and silicate rock weathering/river fluxes in many carbon cycle models (e.g. Zeebe, 2012; Colbourn et al., 2013; Lord et al., 2016).  For example, Zeebe (2012) applies to the LOSCAR carbon cycle model a simple, parameterised weathering scheme based on Walker and Kasting (1992) and the same scheme was applied in simple carbon cycle feedback modelling applied by Toggweiler (2008) and Hogg (2008). An almost identical approach, was also applied by Lenton and Britton (2006), and Colbourn et al. (2013) and Lord et al. (2016). The only difference with Lenton and Britton (2006) and Colbourn et al. (2013) from our simple model, is that they applied additional temperature and terrestrial biosphere dependencies for rock weathering.

In summary, continental silicate and carbonate rock weathering are both represented in the SCP-M model. Both supply alkalinity and carbon to the surface ocean in ratio 2:1 (e.g. more alkalinity than DIC).

The weathering equation used in the model, are as per the model documentation (O'Neill et al., 2019), and the original model code at (https://doi.org/10.5281/zenodo.1310161), and is reproduced here:

$$dC/dt_{weath} = (W_{SC} + (W_{SV} + W_{CV})AtCO_2)$$

where $W_{SC}$ is a constant silicate weathering term set at $0.75 \times 10^{-4}$ mol m$^{-3}$ year$^{-1}$, $W_{SV}$ is a variable rate of silicate weathering per unit of atmosphere $CO_2$ (ppm), set to 0.5 mol m$^{-3}$ atm$^{-1}$ $CO_2$ year$^{-1}$ and $W_{CV}$ is the variable rate of carbonate weathering with respect to atmosphere $CO_2$, set at 1.5-2.0 mol m$^{-3}$ atm$^{-1}$ $CO_2$ year$^{-1}$ (Toggweiler, 2008).

There is a slight difference between carbonate weathering versus silicate weathering, in our model, in terms of the direct consumption of $CO_2$ from the atmosphere when weathering takes place. This direct consumption of $CO_2$ is assumed to be fully reversed in the case of carbonate weathering, but is only partially reversed in the case of silicate weathering. The main $CO_2$ sink activity of the carbonate weathering, is therefore is in the alkalinity fluxes to the ocean and its effects on relative $pCO_2$ in the ocean versus the atmosphere (e.g. Colbourn et al., 2013).

Carbonate weathering

Weathering of carbonate rocks initially takes up $CO_2$ from the atmosphere (one mol), and supplies calcium and bicarbonate ions to the ocean (an additional mol of carbon), as per the following equation:

$$CaCO_3 + H_2O + CO_2 = Ca^{2+} + 2HCO_3^-$$

Therefore, two moles of carbon and one mole of calcium enter the ocean for each mole of $CaCO_3$ weathered. This raises ocean carbon and alkalinity by two units each. In steady state, subsequent precipitation of $CaCO_3$ releases the same amount of $CO_2$ back to the atmosphere that was consumed by weathering (Zeebe, 2012) – a short-term circular loop that leads to a net zero direct consumption of $CO_2$ from the atmosphere from carbonate weathering (e.g. Colbourn et al., 2013; Lord et al., 2016). This is described in detail in Zeebe (2012).

**Figure 2**: Extract of the Zeebe (2012) schematic description of carbonate weathering (**CP_RC1_Fig2.png**)

[Figure]

This return of $CO_2$ to the atmosphere (one mol of carbon) leaves a net addition to the ocean of carbon and alkalinity from carbonate weathering in 1:2 ratio (Zeebe, 2012). The ocean carbon and alkalinity balance is later restored due to subsequent burial and $CaCO_3$ and carbonate compensation (Zeebe, 2012).

According to Zeebe (2012):

*"As a result, although the addition of Ca2+ and 2 HCO 3 increases ocean TCO2 : TA in a 2:2 ratio, on a net basis CaCO3 weathering increases ocean TCO2 : TA in a 1:2 ratio because one mole of $CO_2$ returns to the atmosphere. If influx equals burial, carbonate weathering thus represents a zero net balance for atmospheric $CO_2$."*

For our steady state modelling, we assume the $CO_2$ consumed directly by the carbonate weathering process is returned to the atmosphere – a net zero of **direct** consumption of $CO_2$ from the atmosphere. This is a short-circuiting of the process, but not incorrect (refer Colbourn et al. (2013) quote reproduced above, about "short-circuiting" direct atmospheric $CO_2$ effect of carbonate weathering). Therefore, the fluxes associated with carbonate weathering are those of DIC and alkalinity into the surface ocean boxes of the model. This is the same approach applied by Toggweiler (2008), and Lenton and Britton (2006), and identical to the approach of Lord et al. (2016), and Colbourn et al. (2013). For these studies, the sink of atmospheric $CO_2$ from carbonate weathering comes indirectly through the effects of alkalinity supplied to the surface ocean which lowers $pCO_2$ and draw $CO_2$ into the ocean.  Some interesting quotes from those references below, with bolded parts for emphasis.

The approach for carbonate weathering in Lord et al. (2016) (the reference suggested by the reviewer), is referenced in that study to Colbourn et al. (2013), and is described in Colbourn et al. (2013) as:

*"Note that there is only one mole of DIC for each mole of Ca2+; **this is a short-circuiting of the atmosphere based on the assumption that the atmosphere and surface ocean are well mixed on the timescales considered here. Instead of removing one mol of CO2 from the atmosphere – and by implication the ocean – and adding two moles of bicarbonate to the ocean (as in Eq. 1), nothing is taken from the atmosphere and one mole of bicarbonate is added to the ocean.**"*

And "**The fluxes are then used to calculate fluxes of DIC (FDIC) and Alkalinity (FAlk)**".

We note further from Colbourn et al. (2013):

*"**In the case of carbonate weathering there is an overall null cycle for CO2, whereas silicate weathering transfers CO2 to the Earth's crust**."*

In summary, our simple box modelling representation of carbonate weathering is consistent with the theory of carbonate chemistry, and the literature on modelling of carbonate weathering. Our calculated estimate of 10 Tmol C $yr^{-1}$ from carbonate weathering supplied to the ocean at 275 ppm atmospheric $CO_2$, is comparable to that of 12 Tmol mol C $yr^{-1}$ in Morse and Mackenzie (1990) and Zeebe (2012), Archer et al. (1998), but higher than that assumed by Colbourn et al. (2013) and Lord et al. (2016) of 5 Tmol C $yr^{-1}$. In those latter two studies, they simply assume an even, equal split of the fluxes of silicate and carbonate weathering in their model spin-up Fsil=Fcarb=5 Tmol C $yr^{-1}$. However, Colbourn et al. (2013) quote post-spin-up, pre-industrial total flux of weathering of 12-20 mol C C $yr^{-1}$, split equally between carbonate and silicate weathering (6-10 mol C $yr^{-1}$ each).

We have supplied an annotated snapshot of our model code, in the Attachment A to these comments (below).

Silicate weathering

Silicate rock weathering can be described by the following chemical equation:

$$CaSiO_3 + H_2O + 2\ CO_2 = Ca^{2+} + 2HCO^-_3 + SiO_2$$

Silicate rock weathering removes 2 mols of $CO_2$ from the atmosphere for each mole of $CaSiO_3$ weathered. The subsequent precipitation of $CaCO_3$ in the ocean releases one mole of $CO_2$ back to the atmosphere, with the other mole of $CO_2$ consumed by the atmosphere, taken up in $CaCO_3$, which may end up buried in the marine sediments (Zeebe, 2012). In steady state, over long timeframes, the silicate weathering direct consumption of atmospheric $CO_2$ balances out volcanic emissions of $CO_2$ (*Berner et al., 1983; Zeebe and Caldeira, 2008, Zeebe, 2012).* Because the steady state in the silicate weathering is achieved over a much longer timeframe (1e5-1e6 years), it is appropriate to model a direct sink of $CO_2$ from the atmosphere associated with silicate weathering. The steady state atmosphere-ocean response to carbonate weathering only requires a relatively short timeframe, hence we can model the steady state assumption of carbonate weathering returning its direct consumption of $CO_2$ to the atmosphere (Walker and Kasting, 1992; Lenton and Britton,

2006; Toggweiler, 2008, Zeebe, 2012).

**Therefore, relative to carbonate weathering, there is an additional step applied with silicate weathering**. To account for the unit of $CO_2$ consumed directly from the atmosphere in silicate weathering that is not returned (one more unit than carbonate weathering, as per Zeebe, 2012), and using the approach of Toggweiler (2008), we also subtract an amount equal to a unit of silicate weathering directly from the atmosphere. This is the same approach of Zeebe (2012) who applies a doubling of the molar flux of silicate weathering (to replicate two mols of $CO_2$ initially drawn from the atmosphere), and that of Toggweiler who subtracts a flux of $CO_2$ directly from the atmosphere (but no direct consumption of $CO_2$ in the case of carbonate weathering) to account for the additional unit of $CO_2$ consumed by silicate weathering (when compared with carbonate weathering). This flux is subtracted directly from Atmospheric $CO_2$ in SCP-M as referenced in the model equation above (and described in the code in Attachment A to these comments). This flux, subtracted from the atmosphere, negates the effects on atmospheric $CO_2$ of the units of C added to the ocean by the silicate weathering flux of C. Volcanic $CO_2$ emissions are set equal to the amount of $CO_2$ taken directly from the atmosphere by silicate weathering, to reflect the long-term offset of volcanic emissions by silicate weathering (Walker and Kasting, 1992; Archer et al., 1998, Toggweiler, 2008; Zeebe, 2012, Colbourn et al., 2013; Brault et al., 2017).

As described in Walker and Kasting (1992), Toggweiler (2008), Zeebe (2012) Brault et al. (2017), Colbourn et al. (2013, 2015) and Lord et al. (2016), in steady state the silicate weathering flux feedback for $CO_2$ matches the volcanic $CO_2$ emissions, which we have set in SCP-M. Note, for anthropogenic scenarios we separate volcanic emissions from weathering flux, because the silicate weathering feedback under the forcing of atmospheric $CO_2$, is expected to increase at a greater rate than volcanic emissions (volcanic emissions do not respond to anthropogenic emissions of $CO_2$).

Our calculation for silicate weathering yields a flux of carbon to the oceans 6.3 T mol C yr$^{-1}$. At 275 ppm atmospheric $CO_2$. Our volcanic emissions rate is set to this figure, which is in good agreement with Lord et al. (2016) who set their volcanic C flux at 5.6 Tmol yr$^{-1}$ to balance the silicate weathering component.

**AC: Furthermore, from the python code I learned that weathering (probably meaning carbonate weathering, since in silicate weathering all CO2 comes from the atmosphere with its δ13C-signature) has a δ13C-signature of –6.9‰, similarly as volcanic CO2. While the volcanic δ13C seems to be in the expected range (although on the lower side) I believe the weathering δ13C-signature is wrong, since carbonate rocks have a typical δ13C-signature of about +1-2‰, see for example Sano and Williams (1996); Mook (1986).**

AC: Re carbonate weathering. The δ$^{13}$C of carbonate weathering in our model is not -6.90 per mil, as stated in the reviewer comment above, but it is 0 per mil, via our application of the reference standard value for δ$^{13}$C (the Pee Dee Belemnite) = 0. This feature is shown clearly in the annotated excerpt of the model code in the Attachment A. 0 per mil is the identical value for carbonate weathering used in the first reference provided by the reviewer (Sano and Williams, 1986), and precisely in the middle of the range (+/- 1 per mil)

used in the second reference provided by reviewer (Mook, 1986). We have added text to our model code to make this more obvious.

With regards to silicate weathering $\delta^{13}$C. In the SCP-M model the $\delta^{13}$C of silicate weathering $CO_2$ drawdown was originally set at -6.90 per mil, which is the same as the volcanic $\delta^{13}$C we had assumed. This approach was consistent with offsetting volcanic $CO_2$ emissions with silicate weathering (Zeebe, 2012; Toggweiler, 2008, Lord et al., 2016; Colbourn et al., 2013, 2015; Walker and Kasting, 1991). This is a simplification with regards to the $\delta^{13}$C, and therefore we have changed this, and now applied the atmospheric $\delta^{13}$C signature output from the model to the silicate weathering flux (this is now updated in our model results/re-runs). As per the reviewer comments we have now set the $\delta^{13}$C of the direct consumption of $CO_2$ by silicate weathering, to take atmospheric $\delta^{13}$C value. This is a modest change, however, as atmospheric $\delta^{13}$C is in the range -6.3-7 per mil in the last glacial-interglacial cycle, and we had initially assumed a fixed value of -6.90 per mil.

We also note the reviewer comment that our assumption of -6.90 per mil for volcanic $CO_2$ emissions is at the low end of literature estimates. We have modified this to -4.5 (compared with -4.0 in Zeebe, 2012).

In summary, the changes we have incorporated in the final set of model runs for this manuscript, guided by the reviewer comments:
  - We have changed the $\delta^{13}$C of silicate weathering direct consumption of $CO_2$ from the atmosphere, to the atmospheric $\delta^{13}$C signature outputted from model at each time step, as suggested in the reviewer comments (previously it was set as -6.90 per mil).
  - We have adjusted our $\delta^{13}$C of volcanic emissions from -6.90 to -4.50 per mil, which is more of a "middle of the range" value.
  - We have tidied up the model code description of carbonate weathering and its $\delta^{13}$C (for upload to the Zenodo repository upon finalisation of the manuscript).

**RC: I also do not understand how their approach with not explicitly considering terrestrial carbon change (terrestrial carbon to my understanding is covered as externally to the atmosphere/ocean system, fluxes in/out of it prescribed by optimization) covers changes in C3 vs C4 photosynthesis (which have a significantly different isotopic frac- tionation) on glacial/interglacial timescales (Collatz et al., 1998; Köhler and Fischer, 2004) which leads to differences in the mean terrestrial δ13C and therefore also the changes in the δ13C-cycle as a whole (Kaplan et al., 2002).**

AC: Thanks for the comment. Our response is broken in two parts 1) the terrestrial biosphere and 2) C3 vs C4 photosynthesis.

In summary, the terrestrial biosphere is explicitly considered in our modelling. It is two boxes within the carbon cycle box model we have used. It is not prescribed by optimisation.

We have decided not to assess C3 versus C4 photosynthesis and its effects on δ13C fractionation.

We discuss both of these points in more detail below.

**Terrestrial biosphere in SCP-M**

The terrestrial biosphere is treated in SCP-M as two boxes that exchange carbon with the atmosphere based on fluxes of net primary productivity (NPP) (carbon in) and respiration (carbon out). It is part of the carbon cycle that includes the terrestrial biosphere-atmosphere-ocean-sediments-volcanoes etc. Our box model applies a simple representation of the terrestrial biosphere, whereby biological productivity responds to carbon fertilisation. Therefore, $CO_2$ is the driver of terrestrial biosphere productivity in this model. We apply the two-box terrestrial box model scheme of Harman et al. (2011). The inputs are starting estimates of net primary productivity (NPP), the terrestrial biosphere carbon stock, plant respiration rate and atmospheric $CO_2$. The approach of Harman et al. (2011) is to split the terrestrial biosphere into a fast-response (grasslands and grassy components of savannah systems) and a slow-response (woody trees) component. In this model, the productivity is mostly focussed on the plants/grasses component.

The formula is shown in the model documentation paper (O'Neill et al., 2019) and Harman et al. (2011), and is reproduced here:

$$dAtCO2/dt = -N_{pre}RP[1+\beta LN(AtCO2)] + Cstock/k + D_{forest}$$

Where Npre is NPP at a reference pre-industrial level of atmospheric CO2, RP is a parameter to split NPP between short-term terrestrial biosphere carbon stock and the longer term stock (Cstock1 and Cstock2). B is a parameter with a value typically in the range 0.4-0.8 (Harman et al., 2011). Cstock is the carbon stock in each terrestrial biosphere box, k is the respiration timeframe for each box. Dforest is the prescribed rate of deforestation emissions for present day simulations and projections. A terrestrial biosphere fractionation factor is applied for the carbon isotopes.

This flux out of the atmosphere feeds into the two terrestrial biosphere stocks of carbon (Cstock1 and Cstock2), and the boxes lose carbon to the atmosphere by respiration, as per the equation above. This differential equation for NPP, respiration, and the net flux into and out of the terrestrial biosphere (increase or decrease in the terrestrial biosphere carbon stock), solves at each time step of the model, taking the model's output of $CO_2$ and then the NPP, respiration, Cstock1 and Cstock2 carry forward into the next simulation. The time step of the model is one year, with 10,000 years for each model-data simulation, so this is appropriate to allow the terrestrial biosphere to adjust within each simulation.

Harman et al. (2011) model the terrestrial biosphere primarily as a function of atmospheric $CO_2$. They also incorporate an optional temperature dependency. This is the same approach used in the simplest 4Box terrestrial biosphere module of the Bern Simple Carbon Model (Strassman and Joos, 2018; Seigenthaler and Joos, 1992; Kicklighter et al., 1999; Meyer et

al., 1999), and described by Enting (1994) – although we understand that there are various terrestrial biosphere modules applied with the Bern models, and most are more complex. As far as we can discern, the simple carbon fertilisation approach is also used in Jelstch-Thommes et al. (2019), which also applies the simplest 4Box terrestrial biosphere of the simple Bern model.

There are other possible drivers of the NPP – temperature, precipitation, soil nutrient levels. In the context of our simple carbon cycle model, we are mainly interested in $CO_2$. We don't model atmospheric temperature, and if we were to try to incorporate atmospheric temperature as a driver of terrestrial biosphere, we would also need to incorporate it for terrestrial weathering. There is a limit to how much detail we want to include in the model given we are conducting many simulations (~80,000) in our model-data optimisations across the MIS of the last glacial-interglacial cycle.

We do note that there are studies devoted to determining whether the $CO_2$ fertilisation effect or climate is the dominant control on terrestrial biosphere NPP and the size of the terrestrial biosphere carbon stock. According to Hoogakker et al. (2016), $CO2$ fertilization, rather than climate, is the primary driver of lower glacial net primary productivity by the terrestrial biosphere, accounting for around 85% of the reduction in global NPP at the LGM. Kaplan et al. (2002) also concluded that over glacial-interglacial timescales, global terrestrial carbon storage is controlled primarily by atmospheric $CO_2$, while the climate has more influence on the isotopic composition. Otto et al. (2002) also found that the $CO2$ fertilization effect is mostly responsible for the total increase in vegetation and soil carbon stocks since the last glacial maximum. Kohler et al. (2010) prioritised $CO_2$ fertilisation as the driver of terrestrial biosphere in their "control" main simulation scenario for glacial-interglacial cycles over the last 740 kyr, but also ran scenarios with a climatic driver for the terrestrial biosphere to estimate the effects of "fast" climate changes on atmospheric $\delta^{13}C$. Other studies arguing that atmospheric $CO_2$ is an important, or is the main driver of terrestrial biosphere productivity include Kicklighter et al. (1999), Joos et al. (2004), Schimel et al. (2015), Sitch et al. (2008), Arneth et al. (2017). This view has been contested by van der Sleen et al. (2015).

Given we don't model the atmospheric temperature or precipitation, we saw limited additional benefit to introduce them into our model of the terrestrial biosphere, although it would not be difficult to do this. Finally, given that $CO_2$ and atmospheric temperature co-vary closely, across glacial cycles, it seems of limited benefit to split these effects out in our simple carbon cycle modelling exercise. For example, Meyer et al. (1999) found similar results for modelling carbon uptake in the terrestrial biosphere whether only $CO_2$ fertilisation, or $CO_2$ fertilisation + climate, were included as drivers of NPP – but noting this was not tested for the LGM.

Our aim is not to contribute new findings on the terrestrial biosphere, but we present the behaviour of the terrestrial biosphere in our manuscript to confirm that our exhaustively multi-proxy constrained model-data output is consistent with the range of literature estimates of variations in the terrestrial biosphere in the last glacial-interglacial cycle and LGM-Holocene period, and we show this. For example, our experiment shows a change in the terrestrial biosphere carbon stock of +630 PgC between the MIS 2 (LGM) and MIS 1

(Holocene) period. This compares with other estimates of +540 PgC (Brovkin et al., 2007), +~820-850 PgC (Joos et al., 2004) – with the majority by $CO_2$ fertilisation, ~+500 PgC (Kohler et al., 2010), +~500 PgC (Brovkin et al., 2012), +850 PgC (Jeltsch-Thommes et al., 2019), +511 +/- 289 PgC (Peterson et al., 2014), +378 +/- 88 PgC (Menviel et al., 2016). Another estimate of the LGM-Holocene terrestrial biosphere change is 550-694 Pg C (Prentice et al., 2011), which our result of 630 Pg C sits comfortably within.

Our estimate is actually towards the upper end of the literature ranges, suggesting if anything we could exaggerate the effects of the terrestrial biosphere from the LGM to the Holocene period, with perhaps little to gain by splitting out temperature and precipitation effects. If did, we would probably also need to consider other important features such as soil nutrients and local humidity.

While we have a simple, but explicit two-box representation of the terrestrial biosphere, we don't believe that this detracts from our model-data results, as shown in Figures 9-11 and Figure 12 specifically for the terrestrial biosphere.

**C3 and C4 photosynthesis.**

In summary, our model exercise doesn't take account of C3 versus C4 photosynthesis in the terrestrial biosphere, or consider its effects on the $\delta^{13}$C signature of the terrestrial biosphere. In response to the reviewer comments, we looked into this in more detail to see if we can improve our modelling – noting that it is very easy to update the model code for something like this. For example, we re-ran the model-data experiments as part of one of the other reviewer comments, so could easily incorporate more detail for the terrestrial biosphere, such as C3%/C4% variation in $\delta^{13}$C.

Our approach was to understand and quantify the references provided by the reviewer, review approaches by other modelling exercises for the glacial-interglacial cycle of last 130 kyr, and decide whether we should re-run the modelling with an alternative treatment of the terrestrial biosphere to cater for C3%/C4% and $\delta^{13}$C. As part of investigation, we also constructed the C3/C4 model of Collatz et al. (1998)/Kohler and Fischer (2004) in a python module that easily fits into the carbon cycle box model, to evaluate whether it would improve our modelling (described below and attached to these comments).

Kohler and Fischer (2004), suggested reading by the reviewer, in their excellent paper do make a very good point about C3/C4 photosynthesis in the context of glacial-interglacial $\delta^{13}$C, that is worth reproducing here as a summary:

"*Oceanic inorganic carbon is becoming 0.4 heavier during the G/IG transition, which is in good agreement with both modelling studies and data constraints (Curry et al., 1988; Duplessy et al., 1988; Michel et al., 1995).* **It should be noted that 85% of this calculated oceanic change in $\delta^{13}$C can be explained by the increase in the terrestrial carbon stock and only the missing fraction of 15% by changes in the abundance of the two photosynthetic pathways.** Thus, uncertainties in the current knowledge on C$_3$/C$_4$ plant distribution during the LGM are of minor importance for the overall simulation results. "

Collatz et al. (1998)/Kohler and Fischer (2004) modelling approach

The Collatz et al. (1998) approach to modelling C3 vs C4 %, is based on the estimation of a "cross-over temperature" for dominance of C4 or C3 plants. Above the cross-over temperature, C4 plants are favoured. Below the cross-over temperature, C3 plants are favoured. Collatz et al. (1998) derived a simple equation for the cross-over temperature of C3 vs C4. The cross-over temperature exhibits a positive relationship with atmospheric $CO_2$. Therefore, as $CO_2$ goes up, the cross-over "hurdle" temperature for C4 dominance also increases, so C4% has a negative relationship with $CO_2$. While increasing temperatures may favour C4 plants, if $CO_2$ was also increasing, this would tip the advantage back towards C3 plants. The cross-over temperature calculation of Collatz et al. (1998) is shown as (**CP_RC1_T50.png**):

$$T_{50}(^\circ C) = \frac{10}{\ln Q_{10}} \ln \left( \frac{pO_2(1 + 0.5\frac{\alpha_{C3}}{\alpha_{C4}})}{0.8 \cdot pCO_2 \cdot s_{25}(\frac{\alpha_{C3}}{\alpha_{C4}} - 1)} \right) + 25.$$

Where $T_{50}$ is the crossover temperature for C4 and C3 dominance, where $a_{C3}$ is the "intrinsic quantum yield for C3 photosynthesis" and $p_i$ is the leaf internal $pCO_2$, assumed to be equal to 0.8 x atmospheric $pCO_2$. $s_{25}$ is the value of s at 25°C and $Q_{10}$ is the relative change in s for a change in temperature.

s is defined as (**CP_RC1_s.png**):

$$s = 2,600 \; Q_{10}^{\frac{T_X - 25}{10}}$$

To analyse C4%, Kohler and Fisher (2004) extended the Collatz et al. (1998) equation and provide a simple set of equations to estimate C4% and C3% between the glacial and interglacial periods, using the change in temperature relative to changes in the cross-over temperature between the two periods (**CP_RC1_C3C4.png**):

$$\widetilde{C^*_{C4}} = C^*_{C4} \cdot (1 - a_{C3/C4} \cdot (\Delta T - \Delta T_{50})),$$
$$\widetilde{C^*_{C3}} = C^*_{C3} \cdot (1 + a_{C3/C4} \cdot (\Delta T - \Delta T_{50})).$$

We reconstructed this model of Collatz et al. (1998)/Kohler and Fischer (2004) in the attached python script (python cannot be uploaded, so we have attached a pdf of the model code, the data dependencies of atmospheric $CO_2$ and temperature for the last glacial-interglacial cycle (.txt data files), can also be provided). We use the cross-over temperature

calculation of Collatz et al. (1998), the C4% model of Kohler and Fischer (2004), and estimate an average terrestrial biosphere $\delta^{13}$C using the C4% and C3% output from this model and estimates of $\delta^{13}$C for C4 and C3 plants.

https://zenodo.org/record/3889704#.XuH3Ji1L0_U

To test our simple model works, we satisfy the estimate of $T_{50}$ of 22 deg C at atmospheric $pCO_2$ of 350 ppm from Collatz et al. (1998), and, as per Kohler and Fischer (2004) Figure 4, ~18 deg C at atmospheric $pCO_2$ of ~280 ppm, and ~11 deg C for atmospheric $pCO_2$ of ~190 ppm.

We forced our version of the C4% model of Kohler and Fischer (2004)/Collatz et al. (1998) with atmospheric temperature and $CO_2$ through the last glacial-interglacial cycle (Figure 3 below). The atmospheric temperature data of Jouzel et al. (2007) is derived from Antarctic ice cores, so it likely overstates the amplitude global average temperature cooling during the glacial period. Jouzel et al. (2007) show peak cooling of ~11 degrees C, which is greater than global estimates in the range 3-6 deg C (Schneider von Deimling et al., 2006a; Holden et al., 2009; Schmittner et al., 2011; Annan and Hargreaves, 2013). We take an intermediate average global LGM cooling of 4.5 degrees, and scale the profile of Jouzel et al. (2007) to the average global amplitude of cooling of 4.5 deg C for the LGM, which is the middle of the range of global estimates. This is a simplification, but appropriate for our reconnaissance exercise. We also apply the last glacial-interglacial cycle atmospheric $CO_2$ data of Bereiter et al. (2015) – Figure 3 below.

In terms of what starting values to use for $\delta^{13}$C for the C3 and C4 plants, we note a huge variation in the possible values to use for $\delta^{13}$C of C3 plants, and also note a large variation in the estimates for average $\delta^{13}$C of the terrestrial biosphere applied in terrestrial biosphere and carbon cycle modelling exercises for the last glacial-interglacial cycle. We discuss in more detail below, but flag that natural variation in the average values assumed for $\delta^{13}$C fractionation of the terrestrial biosphere, and variation in $\delta^{13}$C values assumed between modelling studies, greatly outweigh the posited variation in $\delta^{13}$C fractionation from C4% vs C3%.

Carbon cycle modelling exercises show a large range (e.g. Brovkin et al., 2002 (-16 per mil), Menviel et al. (2016) (-23.3 per mil), Jelstch-Thommes et al. (-24 per mil), and the study of Kohler and Fisher applied an average of -16 per mil (C3 -19 per mil, C4 -5 per mil). For this simple exercise, we take the starting average δ13C for terrestrial biosphere taken from Jeltsch-Thommes et al. (2019) (this text was a suggested reference by the reviewer) of -24 per mil, and back out the average starting C3 and C4 $\delta^{13}$C assuming the PI value of C4% of 20% applied in Kohler and Fischer (2004) (the reference suggested by the reviewer). This yields a starting $\delta^{13}$C for C3 plants of -27 per mil, and -14 per mil for C4 plants. For comparison, Kohn et al. (2010) provided a range of $\delta^{13}$C estimates for C3 plants of -20 to -37 per mil, with a global average of -27 per mil. O'Leary et al. (1988) provided a synthesis of global data of -27.1 per mil for C3 plants and -13 per mil for C4 plants.

We model C4% to vary from the preindustrial starting estimate of 20% (Kohler and Fischer, 2004), up to an average of 25% during the LGM (Figure 3). We model average $\delta^{13}$C for the terrestrial biosphere to vary between the range -24.2-23.6 per mil during the last glacial-interglacial cycle, a variation of 0.6 per mil (Figure 3 below).

**Figure 3**: Modelling of the share of C4 photosynthetic plants (C4%) (bottom panel) and average terrestrial biosphere $\delta^{13}$C fractionation factor (middle panel) as a function of atmospheric $CO_2$ and temperature for the last glacial-interglacial cycle (**CP_RC1_Fig3.png**).

[Figure]

Our estimated C4% from using the Collatz et al. (1998) equation (25%) is a little higher than Kohler and Fischer (2004) (24%) and this likely reflects differing atmospheric $CO_2$ and temperature assumptions. For example, Kohler and Fischer (2004) take average northern hemisphere average temp change of -5 degrees, and southern of -8 degrees. We have inputted a global average change of -4.5 degrees C as per the literature range of 3-6 degrees C cooling. However, there must be something else being applied by Kohler and Fischer (2004) to achieve their LGM "target" C4% of 30-33%.

The approach of Kohler and Fischer (2004) was to establish a target variation of C4% between the LGM and the PI and then to see what parameterisations of their model runs could reach that target. Our estimate of LGM C4% is of 25% is far below the "targeted" C4% of 30-33% from Kohler and Fischer (2004). Their study found that varying the C4% amplitude

in the Collatz et al. (1998) C3/C4% share model could increase C4% from 20% to 24%, but increasing the grassland succession amplitude increased the C4% up to 42%, a much bigger change than the C3/C4% share model alone. Furthermore, according to Huang et al. (2001), local moisture conditions might be even more important than any temperature or $CO_2$ effects on C3/C4%.

The grassland succession factor is an equation contributed by Kohler and Fischer (2004) to estimate the effects of changes in the tree-line (the divide between where trees and grasses grow) as a function of changes in temperature, between the LGM and PI. According to Kohler and Fisher (2004), this is the main driver for the C4% change and change in the terrestrial biosphere $\delta^{13}$C fractionation, perhaps not the temperature and CO2-dependant equation of Collatz et al. (1998).

Kaplan et al. (2002) posit something different again, that the major driver of changed terrestrial biosphere $\delta^{13}$C discrimination since the LGM is retreating ice sheets, with an additional or ancillary role for C3/C4 plant substitution.

Our estimated change in terrestrial biosphere $\delta^{13}$C fractionation of ~+0.6 per mil, is below the estimate from Kohler and Fischer of 1.3 per mil, and that reflects that they include the grassland succession factors in their LGM-PI analysis. The offset in assumed $\delta^{13}$C fractionation between C3 and C4 of -13 per mil (-27 per mil less -14 per mil) is very similar to their chosen -14 per mil (-19 per mil less -5 per mil), suggesting that the differences reflect the use of another factor outside C3/C4%, the grassland succession factors, to drive their results.

Beyond the simple exploratory attempt above, modelling highly uncertain grassland succession factors, or ice sheet retreat/advance, or localised moisture and temperature changes, to try and explain uncertain changes in C3% vs C4%, for which the starting values themselves could fall within huge ranges of uncertainty, looks beyond the scope of our study.

We note that, with regard to the estimates of C4% used by Kohler and Fisher to create "targets" for pre-industrial and LGM periods, Kohler and Fisher (2004) say the following:

"NPP and $f_{C4}$ for the LGM are based on modelling studies only and, thus, represent only weak indicators which were only used for uncertainty estimates." And furthermore, on P16:

"However, because the constraints on NPP and the **fraction of $C_4$ plants were based on only a few mostly modelling studies, we merely interpret those as a model evaluation**."

These findings underscore the uncertainty of estimates for quantifying C4/C3 and therefore $\delta^{13}$C of the terrestrial biosphere. This uncertainty is amplified in the actual estimates of $\delta^{13}$C for C3 and C4 plants, as we discuss below.

The Figure below is reproduced from Kohn (2010), and shows the range in $\delta^{13}$C fractionation for C3 plants alone, which spans -20 to -37 per mil, and is impacted by many factors including temperature, precipitation, and effects of canopies and new growth (**CP_RC1_Kohn1_extract.png**).

[Figure]

**Fig. 1.** (*A*) Histogram of MAP values for isotopically charact.erized C3 plants, showing emphasis on relatively arid ecosystems (MAP ≤ 500 mm/yr) and tropical rainforests (spikes at MAP ~ 2,000, 3,000 mm/yr). (*B*) Histogram of $\delta^{13}C$ values of modern C3 plants. Data compiled in this study average −27.0‰, excluding analyses from the understory of closed-canopy forests. Estimated global average composition, based on global trends in precipitation and vegetation, is approximately −28.5‰, significantly lower than typically assumed. An accurate average $\delta^{13}C$ value for C3 plants is needed for accurate models of carbon fluxes, atmospheric $CO_2$ compositions, and soil organic matter. (*C*) $\delta^{13}C$ values vs. MAP showing increasing $\delta^{13}C$ with aridity. Data sources are listed in *SI Text*. White dots are average compositions of data from a large collection made in a single month during a wet year (35).

Furthermore, more recently, Kohn (2016) attempted to estimate the change in $\delta^{13}C$ for C3 plants from the LGM to modern day, based on atmospheric CO2, and also to quantify the effects of precipitation on C3 plant $\delta^{13}C$. This Figure shows the variation in C3 $\delta^{13}C$ discrimination itself, is even bigger than the posited effect of C3/C4% (see Figure below from Kohn (2016) - **CP_RC1_Kohn2_extract.png**).

To model C3 and C4 $\delta^{13}C$ properly, there are other important effects in C3 plants (on their own), that would need to be taken into account. For example, Francois et al. (1999) point out that changes in the $\delta^{13}C$ fractionation from a changing C4% were partially offset by changes in the opposite sign in the fractionation of C3 plants due to the modification of the intercellular $CO_2$ pressure within their leaves.

[Figure]

**Figure 1** Proposed models for factors that influence δ¹³C of C3 plants. **(a)** $p_{CO2}$. Differences are illustrated between geological conditions vs. AD 2000 ($p_{CO2}$ = 370 ppmv, average δ¹³C = -28.5 for C3 biomass). LGM = Last Glacial Maximum. Note inverse relationship between δ¹³C and Δ¹³C. Experiments are for above-ground biomass (Schubert and Jahren, 2012), shifted to fit preferred curve. **(b)** Mean annual precipitation (data and data averages from Kohn, 2010).

**Peer group/modelling approaches**

In exploring this issue of C3 vs C4% and δ¹³C further, and to benchmark our work against the peers who are modelling and analysing the last glacial-interglacial cycle (0-130 ka), we investigated the literature. C3 versus C4 fraction in photosynthesising plants is not discussed much in the literature of modelling of the last 130 kyr glacial-interglacial cycle of carbon. We couldn't find any reference to C3/C4 photosynthesis and δ¹³C in Eggleston et al. (2016), who contributed the atmospheric δ¹³C data we used in our model-data analysis. We don't find any mention of C3/C4 photosynthesis and its effects on δ¹³C in any of Ganopolski et al. 2010, Brovkin et al. 2012, Eggleston et al., 2016; Ganopolski and Brovkin, 2017; Kohfeld and Chase, 2017.

Brovkin et al. (2002) simply state, with reference to their CLIMBER-2 model of the last glacial-interglacial cycle:

*"Most of the carbon (ca. 85%) is allocated to the C3 photosynthesis pathway and the remaining carbon (15%) to the C4 pathway. The globally averaged δ¹³C fractionation factor for terrestrial biosphere is 0.984."* (-16 per mil).

We find no reference to any changes for glacial interglacial C3 and C4 and terrestrial biosphere δ¹³C modelled in Brovkin et al. (2007, 2012), or Ganopolski (2010, 2017).

The transient modelling of the last glacial-interglacial cycle, undertaken by Menviel et al. (2012b), does not mention C3 and C4 photosynthesis, or its effects on δ13C fractionation.

We note that Kohler et al. (2010), mention the parameterisation of C4% in the terrestrial biosphere in their 740 kyr transient simulations with the BICYCLE model. In their control simulation (CTRL) they had a representation of the terrestrial biosphere that emphasised $CO_2$ fertilisation as the dominant control on terrestrial biosphere NPP, and limited or no change (hard to tell from reading) in C4% on the glacial-interglacial δ¹³C of the terrestrial

biosphere. There is an extended scenario TB+ which emphasises climate as the driver of the terrestrial biosphere, faster response of NPP/terrestrial biosphere and parameterises higher C4% in the LGM (and associated change in the $\delta^{13}$C of the terrestrial biosphere), leading to a combined small effect on deep Pacific $\delta^{13}$C of 0.1 per mil.

However, in discussing the all-important drivers of the changes in atmospheric pCO2, $\delta^{13}$C and deep Indo-Pacific $\delta^{13}$C, and mean ocean δ13C, for termination I, as listed in Kohler et al. (2010) Table 3, C4% and terrestrial $\delta^{13}$C changes are not mentioned. The features listed by Kohler et al. (2010) as the drivers are: lower ocean temperatures, smaller terrestrial carbon storage, lower sea level, weaker NADW formation, enhanced marine export production, larger sea ice cover (gas exchange), higher Southern Ocean stratification.

There is a little more discussion of the C4% and terrestrial biosphere in terms of the LGM and Holocene, which unfortunately is only a small fraction of our 130 kyr period of interest.

For example, Joos et al. (2004) modelled a change in terrestrial biosphere $\delta^{13}$C between the LGM and Holocene of 0.5 per mil. However, they observed the following:

"Changes in the mean terrestrial isotopic signature have a minor impact on the modeled changes in $\delta^{13}$C of DIC……….The estimated oceanic $\delta^{13}$C shift is 0.05% smaller than in the standard case, if the land biosphere-atmosphere $\delta^{13}$C difference is kept at the Holocene value of -17 per mil."

Menviel et al. (2012a) provided an interesting quote and the following caveat with their modelling of the last glacial termination and Holocene:

"A caveat is that a constant atmosphere-land isotopic fractionation factor is applied in the inverse approach by Elsig et al. [2009] and in this study, therefore not taking into account any relative changes in the occurrence of C3/C4 plants and other influences on fractionation. However, using the LPJ-DGVM vegetation model, Joos et al. [2004] found that changes in fractionation and C3/C4 plant abundance due to climate and CO2 changes lead to a decrease in $\delta^{13}$C signature of the terrestrial biosphere of about 0.5 per mil from the early Holocene (10 ka B.P.) to pre-industrial times. **A 0.5 permil decrease in biosphere $\delta^{13}$C translates into an atmospheric δ13C decrease of about 0.02 permil. This suggests that changes in the atmosphere-land isotopic fractionation have a small influence on the results presented above**. "

We note another paper relevant to our manuscript, by Menviel et al. (2016) and focussed on the LGM (18-24 ka), made brief mention of C3/C4, and described that they undertook a sensitivity of -0.7 per mil and +0.5 mil around their average estimate of -23.3 per mil $\delta^{13}$C for the terrestrial biosphere, but the modelling results of that sensitivity are not discussed further in the paper. That type of sensitivity is pretty easy to undertake for analysing only the LGM and the Holocene, as any studies on C3 vs C4 (Kohler and Fisher, 2004; Kaplan et al. 2002, Francois et al., 1999; Joos et al., 2004) have looked at this time period – even though they produce uncertain estimates for the % C3 vs C4 and therefore $\delta^{13}$C fractionation factor. It is a much more difficult proposition to come up with values for a sensitivity for the last

glacial-interglacial cycle in its entirety (130,000 years), but that may be an interesting piece of work on its own – future work.

Studies focussed on the terrestrial biosphere

We note the references that focussed specifically on the terrestrial biosphere in detail as the major focus of their work, in the early 2000's, or example those provided by the reviewer (e.g. Collatz et al., 1998; Kaplan et al., 2002, Kohler and Fisher, 2004), and another (e.g. Francois et al., 1999), focused only for the Last Glacial Maximum and PI/modern periods None of them examined the last glacial-interglacial cycle which was ~130 kyrs in duration. All of these studies above, to our understanding, produced uncertain results.

A recent study devoted to analysing the terrestrial biosphere in detail/major focus (Jeltsche-Thommes et al. (2019) - suggested by the reviewer), does not mention this feature C3 vs C4%. Jeltsche-Thommes (2019), in their study focussed on the terrestrial biosphere from the last glacial maximum to the Holocene, simply state:

"The $\delta^{13}$C signature of terrestrial carbon is set to −24 ‰." (at the top of page 856).

We wondered whether we can contribute something important here with regard to C3/%/C4% and the terrestrial biosphere $\delta^{13}$C that has not been considered by any of our peer group of model-data analysis of the last glacial-interglacial cycle.

In summary, there are studies that focussed specifically on the terrestrial biosphere, using dedicated vegetation models. We see that these studies had great detail for the terrestrial biosphere, but were very light on detail for other features of the carbon cycle (ocean circulation and biology, volcanism, weathering, the effects of calcium carbonate compensation). In reviewing these papers, and consistent with our prior understanding, there is not great confidence on quantifying the change in C3 and C4 proportions during the LGM and Holocene, and this is particularly worse during the time period we have analysed up to 130 ka. The papers of Collatz et al. (1998), Kaplan et al. (2002), Kohler and Fisher (2002), all focus on the period LGM-present. There is no coverage of the last glacial cycle 130-20 ka, which is the focus of our study. Furthermore, studies that do focus on the last glacial-interglacial cycle of atmospheric $CO_2$, eg Brovkin, Ganopoloski, do not mention C3 versus C4 fractionation in their papers – making difficult any comparison. We even note that a paper we have referenced in our manuscript, Hoogakker et al. (2016), a paper devoted entirely to the terrestrial biosphere in the last glacial-interglacial cycle, does not address C3 versus C4 plant composition.

As shown above, it is actually an easy process to add the C3 and C4 equations of Collatz et al. (1999) and Kohler and Fischer (2004), and also a temperature dependency for NPP, as we have shown above and with the attached code (Attachment B). We could do this and then re-do the simulations as an appendix or addendum (or a sensitivity).

We could even just apply a sensitivity on the $\delta^{13}$C of terrestrial biosphere of +1/-1 per mil change between LGM and Holocene. However, that's a straightforward exercise for the

LGM and Holocene comparison, but it would involve us trying to fit the uncertain LGM-Holocene changes back for the entire last glacial cycle, which is another highly uncertain exercise. We note all of the studies referenced in the reviewer comments and described here, considered C3 vs C4 only for the LGM to Holocene-modern period, but we've explicitly looked at the lead-up to the LGM over the period from 130 ka. We would not like to try to extrapolate changes in C3 v C4 for the LGM over the entire last glacial-interglacial cycle, and implementing the Collatz et al. (1998)/Kohler and Fischer (2004) module would not help us much in that regard as it only explains less than half of the change $\delta^{13}$C of terrestrial biosphere from the LGM (the rest explained by changes in grassland vs forest succession).

There is huge uncertainty around average $\delta^{13}$C factors for plants, and that extends even further to C3 and C4 $\delta^{13}$C, and their possible respective shares and variations. The indicated changes of 0.3-1.8 per mil terrestrial biosphere $\delta^{13}$C between the Holocene and LGM, from the literature described above, are very minor compared to the absolute uncertainties and range in $\delta^{13}$C of the terrestrial biosphere itself.

Summary on terrestrial biosphere and C3 vs C4 photosynthesis

We investigated these topics enthusiastically, based on the reviewer's comments. We're very confident, based on our assessment of the papers above, that our model results will not change by much at all, and the paper conclusions by nothing at all, by varying our approach to the terrestrial biosphere (equally for rock weathering as discussed above). If the CP Journal Editors and the reviewer feel greatly compelled that we need to modify our modelling approach, we certainly can (these would not be major model revisions, only minor adjustments). Our preferred approach, is to simply add a caveat that our model-data experiments don't consider the effects of C3/C4 photosynthesis on $\delta^{13}$C fractionation of the terrestrial biosphere.

**Amendments to the manuscript**

We have added the following text to the model description (P5 L24):

[revised manuscript text omitted]

We have also updated the discussion of our model results for the terrestrial biosphere, to provide a bit more detail and some additional references (Section 5.3), plus an additional

caveat in the "advantages and limitations section" (P34, L18).

"Furthermore, we apply a simple representation of the terrestrial biosphere in our model-data experiments, relying primarily on atmospheric $CO_2$ as the driver for NPP. This approach provided reasonable results for the terrestrial biosphere carbon stock and NPP, on the whole, but may miss some detail in the terrestrial biosphere during the last glacial-interglacial cycle."

Future work could enhance this set of modelling results with more detail in the terrestrial biosphere. For example, the modelling values for ocean circulation and biology derived here, could be used to solve for the optimal data-matching values for C3 and C4 plant productivity, with separate $\delta^{13}$C-fractionation factors, to help inform that area of study.

**Attachment A**

**Carbonate rock weathering in SCP-M**

The reviewer mentioned the model code. In terms of the model equations and model code, the flux of carbon to the ocean from carbonate weathering is set in our model by the following equation (please see O'Neill et al. (2019) and the annotated model code snapshot below):

RVCARB=WCARB*AtCO2  (1)

Where WCARB is a weathering parameter with respect of atmospheric $CO_2$ and is set at 1.5-2.0 mol C/m3/atmosphere. At 275 ppm atmospheric CO2, this is a flux of 10 x1012 mol C annum. (for comparison, this flux is 12 x 1012 mol C annum in Morse and Mackenzie (1990), Zeebe (2012) and Archer et al. (1998), and 14.9 Tmol C annum in Toggweiler (2008). This flux of carbon is added to the low latitude surface box of the model (as per Toggweiler (2008), Zeebe (2012), Hogg (2008)), and alkalinity is added in the ratio ALK:DIC 2:1 (as per Toggweiler (2008), Zeebe (2012), Colbourn et al.,2013) by multiplying RVCARB by 2.0 to create the river flux of alkalinity to the ocean surface boxes. This 2:1 flux of alkalinity:carbon reflects that the initial one mol of CO2 consumed by the carbonate weathering equation, has been returned to the atmosphere (the DIC proportion of 1 is 2 mols less one mol returned to the atmosphere) as per Zeebe (2012) and Lenton and Britton (2006).

The fluxes of DIC and Alk from carbonate weathering are added to the ocean via the river fluxes of C and Alk (see below). This lowers pCO2 in the ocean surface box and therefore draws CO2 from the atmosphere into the ocean, a net sink of CO2 from carbonate rock weathering. We do not subtract a mol of CO2 directly from the atmosphere in our equation for atmospheric CO2, as for the time scale modelled ~10 kyr, we are taking the short-cut of assuming the CO2 taken up directly from the atmosphere from carbonate weathering, is released back to the atmosphere upon precipitation of CaCO3 into the ocean (Zeebe (2012), Toggweiler (2008)). Carbonate weathering is therefore a flux of carbon and alkalinity to the surface ocean via a river flux, leading to lowering of pCO2 in the surface ocean box and subsequent drawdown of CO2 from the atmosphere. An almost identical approach to ours, was applied by Lenton and Britton (2006), a paper devoted to the study of rock weathering as a sink of atmospheric CO2. The only difference is that Lenton and Britton (2006) applied an additional temperature and terrestrial biosphere dependency on weathering.

We consulted Lord et al. (2016) as suggested in the reviewer comments. Lord et al. (2016) use the cGenie model to estimate weathering feedbacks from atmospheric CO2 emissions. Lord et al. (2016) is a paper that is devoted to the feedback of rock weathering on atmospheric CO2. The treatment of carbonate weathering, in terms of setting this simply as fluxes of DIC and Alk in the ratio of 1:2 to the surface ocean box, is identical to ours. Where they differ, is because they are looking in much more detail at the effects of terrestrial rock weathering, they also explore other dependencies for rock weathering, such as temperature, terrestrial biosphere productivity and run-off rates. Ours has an atmospheric CO2 dependency, as per Zeebe (2012), Toggweiler (2008), Walker and Kasting (1992).

**Silicate rock weathering in SCP-M**

The treatment of silicate weathering in the SCP-M model is:

RVSIL=(BSIL+WSIL*AtCO2) (2)

Where BSIL is a constant weathering rate of 0.75 e-4 mol/m3/yr (Toggweiler, 2008), and WSIL is a rate varying with atmospheric CO2, set at 0.5 mol/m3/atmosphere as per Toggweiler (2008). For atmospheric CO2 of 275 ppm, this is a weathering flux of 5.7 x 1012 mol C annum (5 x 1012 mol in Zeebe (2012) and 5.63 x 1012 mol annum in Toggweiler (2008)).

The silicate and carbonate weathering fluxes of carbon, are added to the surface ocean boxes of the box model. Alkalinity is also added, in a ratio of 2:1 to the carbon fluxes (Sarmiento and Gruber (2006), Toggweiler (2008), Zeebe (2012)).

**However, there is an additional step applied with silicate weathering**. To account for the unit of CO2 consumed directly from the atmosphere in silicate weathering (one more unit than carbonate weathering, as per Zeebe, 2012), and using the approach of Toggweiler (2008), we also subtract an amount equal to a unit of silicate weathering directly from the atmosphere. This is the same approach of Zeebe (2012) who applies a doubling of the flux of silicate weathering, and that of Toggweiler who subtracts a flux of CO2 directly from the atmosphere to account for the additional unit of CO2 consumed by silicate weathering (when compared with carbonate weathering). This flux is subtracted directly from Atmospheric CO2 in SCP-M. This flux subtracted from the atmosphere negates the effects on atmospheric CO2 of the units of C added to the ocean by the silicate weathering flux of C. The effect of the more alkaline ocean (alk:C is 2:1 in the silicate weathering flux) is to draw down the volcanic emissions of CO2. Volcanic CO2 emissions are set equal to the amount of CO2 taken directly from the atmosphere by silicate weathering, to reflect the long-term offset of volcanic emissions by silicate weathering (Walker and Kasting, 1992; Archer et al., 1998, Toggweiler, 2008; Zeebe, 2012, Brault et al., 2017). In Walker and Kasting, 1992; Toggweiler, 2008; Zeebe, 2012; Brault et al., 2007, volcanic emissions are also set to the silicate weathering drawdown of CO2.

As described in Walker and Kasting, 1992; Toggweiler, 2008; Zeebe, 2012; Brault et al., 2007, Colbourn et al. (2013, 2015); Lord et al. (2016), in steady state the silicate weathering flux feedback for CO2 matches the volcanic CO2 emissions, which we have set in SCP-M. Note, for anthropogenic scenarios we separate weathering flux from volcanic emissions, as it is clearly a non-steady state simulation, and the silicate weathering feedback, under the forcing of atmospheric CO2, is expected to increase at a greater rate than volcanic emissions.

We note that Zeebe (2012) implements the scheme slightly differently to ours, by subtracting fluxes of carbonate and silicate weathering from the atmosphere, but by doubling the silicate flux to account for the net removal of CO2 from the atmosphere (balanced by volcanic emissions). In Zeebe (2012) when CO2 is returned to the atmosphere from precipitation of CaCO3 in the ocean surface boxes, there is a net zero direct flux of CO2

from the atmosphere from carbonate weathering and a direct flux of CO2 from the atmosphere of 1 mol from silicate weathering.

**The SCP-M model code for carbonate and silicate weathering**

Below is the description and extract of the original model code presented in O'Neill et al. (2019) as referenced by the reviewer. A revised model code, incorporating the changes described in this response, will be uploaded with the final manuscript.

Line 418 shows the equation (1) above, where the carbonate rock weathering (RVCARB) is calculated from atmospheric CO2 with the WCARB parameter.

Line 419 shows the equation (X) above where the silicate rock weathering (RVSIL) is calculated from atmospheric CO2 and a constant.

Line 420 the silicate weathering amount to be directly subtracted from the atmosphere, as described above, "weaths", is identified.

Line 423 Volcanic emissions is set to equal "weaths", the direct (net) amount of CO2 taken from the atmosphere by silicate weathering, as described above.

In line 425-428 there is the option to apply an input value for volcanic emissions instead of setting it to equal silicate weathering. This is for the model runs with analysis of anthropogenic emissions/short time frames and is switched off for our experiments.

In Line 431 the net effect of volcanic emissions and silicate weathering on atmospheric CO2 is calculated

In Line 432 the above terrestrial fluxes of carbon can be disabled by a switch (for sensitivities and model testing) via "TerrestrialGeo" (1 is on, 0 = off).

In line 435 the δ13C of silicate weathering drawdown of CO2 from the atmosphere is set to the hardwired value of -6.90. In the revised model code it is now set to atmospheric δ13C within each model time step.

In line 436 the δ13C for silicate weathering direct atmospheric CO2 flux and volcanic emissions of CO2 are applied to their fluxes of carbon and converted to molar concentrations in the atmosphere. (we have now amended the volcanic emissions δ13C to a value of -4.5 per mil).

In line 437 the terrestrial δ13C fluxes can be switched on or off (for model testing or sensitivity) via "TerrestrialGeo" (1 is on, 0 = off).

In line 438 the radiocarbon content (zero, dead) of volcanic emissions and weathering fluxes is applied.

In line 442 both RVCARB and RVSIL fluxes of carbon are added to the surface ocean box via river flux.

In line 443 alkalinity flux is added to the surface box in ration 2:1, leading to a lowering in pCO2 in the surface box and a drop in atmospheric CO2.

In line 475 ocean δ13C is calculated. **The river flux of C (derived from weathering) is introduced to the surface ocean box with a δ13C of the standard value "Sstand" (δ13C=0) as discussed above. The dissolution of marine carbonates also introduces carbon with the standard value for δ13C (δ13C = 0) to the ocean boxes**.

In line 481 the net fluxes of volcanic emissions and silicate weathering drawdown of CO2 are added to the equation for atmospheric CO2.

In line 484 the δ13C of net fluxes of volcanic emissions and silicate weathering drawdown of CO2 are added to the equation for atmospheric δ13C.

The confusion with the reviewer likely comes from our comment in the model code in line 416 "# As per Toggweiler (2008) only silicate weathering is a sink of CO2 from the atmosphere". We will delete this statement as it is a poor descriptor.

In addition, we should modify the following comment
 "# Weathering of carbonate rocks is a source of carbon to the low latitude surface ocean via rivers" with "…source of carbon **and alkalinity**"
Therefore, it is indeed the case in SCP-M that both carbonate and silicate weathering ultimately work as sinks of atmospheric CO2 by altering the surface boxes' alkalinity.

Figure 1: Original model documentation paper (O'Neill et al., 2019) model code extract from https://zenodo.org/record/1310161#.Xm7Mby17E_U (**CP_RC1_code1.png**)

```
414  ## Weathering, river fluxes and volcanic emissions-------------------------
415
416      # As per Toggweiler (2008) only silicate weathering is a sink of CO2 from the atmosphere
417      # Weathering of carbonate rocks is a source of carbon to the low latitude surface ocean via riv
418      RVCARB=WCARBs*AtCO2
419      RVSIL=(BSILs+WSILs*AtCO2)
420      weaths=RVSIL*Varr[0,0] #silicate rock weathering sink of CO2, carbonate weathering is a source
421
422      # Volcanic carbon emissions
423      volcs=weaths # volcanic emissions in step with silicate weathering as per Toggweiler (2008)
424      # unless anthropocene scenario, hardwired estimate
425      if AnthEmits==1:
426          volcs=volcs1
427      else:
428          volcs=weaths
429
430      # net source/sink of terrestrial carbon
431      TerrC=(volcs-weaths)/Varrat
432      TCflux=TerrC*TerrestrialGeo
433
434      # weathering and volcanism 13C and 14C fluxes
435      weathd13C=weathd13C
436      TerrSC=(-weaths*weathd13C+volcs*volcd13C)/Varrat
437      TSCflux=TerrSC*TerrestrialGeo
438      TRCflux=TerrC*TerrRC*TerrestrialGeo
439
440      # River fluxes
441      RiverCflux=np.zeros([7,1])
442      RiverCflux[0,0]=(RVCARB+RVSIL) # mol/m3 Incorporates source of carbonate weathering
443      RiverAlkflux=RiverCflux*2.0 #Alk:C ratio 2:1 as per Toggweiler (2008)
444      RiverPflux=np.zeros([7,1])
445      RiverPflux[0,0]=RiverP_mols/Varr[0,0]
446      PSedflux=np.zeros([7,1])
447      PSedflux[5,0]=RiverP_mols/Varr[5,0]
448
```

(**CP_RC1_code2.png**)

```
464  ## Step forward model calculations-----------------------------------
465
466      # Model equations
467
468      # Ocean boxes
469      Parr = Parr +dt*secsyr*(np.dot(PhysMat, Parr)+BioP+RiverPflux*Rivers-PSedflux*Rivers)
470      Carr = Carr + dt*secsyr*(np.dot(PhysMat, Carr)+BioC+cflux+NetCflux+RiverCflux*Rivers)
471      Alkarr = Alkarr + dt*secsyr*(np.dot(PhysMat, Alkarr)+NetAlkflux+RiverAlkflux*Rivers)
472      Fearr = Fearr + dt*secsyr*(np.dot(PhysMat, Fearr)+BioFe)
473      Siarr = Siarr + dt*secsyr*(np.dot(PhysMat, Siarr)+BioSi)
474      Oarr = Oarr + dt*secsyr*(np.dot(PhysMat, Oarr)+BioO)
475      SCarr = SCarr+ dt*secsyr*(np.dot(PhysMat, SCarr)+BioSC+Scflux+NetCflux*Sstand+
476                       RiverCflux*Sstand*Rivers)
477      SCratio=SCarr/Carr
478      RCarr = RCarr + dt*secsyr*(np.dot(PhysMat, RCarr)+BioRC+Rcflux+NetCflux*(RCarr/Carr)-(RCD1*R
479
480      # Atmosphere
481      AtCO2 = AtCO2 + dt*secsyr*(Atcflux+TCflux-((CFert-Respire)/Varrat)+
482                       ((AnthEmit+DeforestC)/Varrat)*AnthEmits)
483      pCO2a=np.append(pCO2,AtCO2) # create an array of all pCO2
484      SCAt = SCAt+dt*secsyr*(AtSCflux+TSCflux-((CFert-Respire)/Varrat)*TerrBioSC*
485                       TerrestrialBios+(AnthSC1+DeforestSC)/Varrat
486                       *AnthEmits)
487      RCAt = RCAt+dt*secsyr*(AtRCflux+RCS1At/Varrat-RCD1At*RCAt*Varrat/Varrat-TRCflux-((CFert-Resp
488                       *TerrestrialBios+((AnthRC1+DeforestRC)/Varrat)*AnthEmits+(Bomb14C/Var
```

The code extract below shows the values chosen for weathering input parameters, as described in the text above. At line 321, weathδ13C is the value that was applied to silicate weathering, NOT carbonate weathering as assumed by the reviewer.
(**CP_RC1_code3.png**)

```
310 # Continental weathering atmospheric CO2 sink and flux into oceans
311 # Generally following Toggweiler (2008)
312 WCARB=2.0 # mol/m3/atm/yr as per Toggweiler (2008)
313 WCARBs=WCARB/secsyr
314 BSIL=0.75e-4 # mol/m3/yr as per Toggweiler (2008)
315 BSILs=BSIL/secsyr
316 WSIL=0.5 #mol/m3/atm/yr as per Toggweiler (2008)
317 WSILs=WSIL/secsyr
318
319 # Carbon isotops for volanic emissions and weathering
320 volcd13C=-0.90 #per mil
321 weathd13C=-6.90 #per mil
322 volcd13C=(volcd13C/1000+1)*Sstand
323 weathd13C=(weathd13C/1000+1)*Sstand
324 TerrRC=0.0 # 14C dead
325
```

**RC: Furthermore, it is worth pointing out somewhere in the discussion that this modelling exercise only examines the potential role of sea ice as a barrier to CO2 exchange, and not its synergistic (and likely more important) roles in influencing nutrient distributions, marine productivity, and a trigger for deep ocean circulation changes. The authors state this somewhat in their "Advantages and limitations" section, but I think that this point could be made more explicitly.**

AC: To address this comment, we have added the following (==P27, L23==):

"This finding may reflect our approach to treat polar sea-ice cover simply as a regulator of the rate of air-sea gas exchange in the polar oceans. This approach may neglect other effects of sea-ice cover including as a trigger for changes in Southern Ocean upwelling, NADW formation rates, deep ocean stratification, nutrient distributions and biological productivity (Morrison et al., 2011; Brovkin et al., 2012; Ferrari et al., 2014; Kohfeld and Chase, 2017; Jansen, 2017; Marzocchi and Jansen, 2017). For example, Brovkin et al. (2012) found that in the CLIMBER-2 model, atmospheric CO2 was more sensitive to sea ice cover when it was linked to weakened vertical diffusivity in the Southern Ocean of tracers such as DIC, thereby reducing outgassing of $CO_2$."

**RC: Another larger issue that the sea ice proxy highlights is the spatial heterogeneity of the Southern Ocean and how the model results are linked with reality: the sea ice proxy likely represents changes very close to the continent and early glacial changes in sea ice are not well reproduced in the few long sea ice records that are found near the APF. This not only suggests that a barrier effect of sea ice would be limited to only part of the Southern Ocean, it points to larger issues with treating the Southern Ocean as one box, with an unclear delineation of how much of the S. Oc. this box is presumed to cover. If the**

**box is supposed to ONLY cover those areas close to the continent where AABW and Circumpolar Deepwater processes that influence GOC are most important, then the authors' main conclusion of increases in S. Oc. export production aren't well supported by paleoceanographic data which show reductions in export South of the APF for the majority of the glacial cycle between MIS5d and MIS2. Some discussion of what the Southern Ocean box actually represents - and this potential disconnect with paleoceanographic data - is warranted.**

AC: SCP-M has two Southern Ocean boxes in each basin: a polar and sub polar Southern Ocean box. These are: polar Southern Ocean box for both basins (box 12 in Figure 1) which covers 60-80 deg S, sub polar Atlantic box (box 7 in Figure 1, 40-60 deg S) and sub polar Pacific-Indian box (box 11, 40-60 deg S). The sea ice forcing/air-sea gas exchange is undertaken for the polar Southern Ocean box. The biological export productivity experiment is undertaken for the sub polar Southern Ocean boxes in each basin, as per the regions highlighted for increased glacial period biological activity by Martinez-Garcia (2014) and Lambert et al. (2015), Shoenfelt et al. (2018). Put another way, our Southern Ocean biological flux experiments are not concerned with the APF, but with the open Southern Ocean box.

We have added the following text in the model description in Section 2.1 (P5, L6):

"The Southern Ocean is split into two boxes, including a polar box which covers latitude range 60-80 degrees South (box 12 in Fig. 1) and sub polar boxes in the Atlantic (box 7) and Pacific-Indian (box 12) basins, which cover latitude range 40-60 degrees South. See O'Neill et al. (2019) for a discussion of the choice of box depth and latitude dimensions."

We have also added the following text in the first paragraph of Section 2.2.1 Model parameters and forcing (P8, L26):

"Note the polar Southern Ocean box which is forced with reduced air-sea exchange, is separate from the sub polar Southern Box in which the biological export productivity parameter is varied in the model-data experiment."

**RC: Throughout the paper the authors refer to "abyssal" and "deep" water masses for all basins, but I was never able to find the depth cut-offs that were used to distinguish these depths in the different basins. Please put them in the figure captions and text (not just supplemental information, if it is there.)**

AC: We have added the following text in the model description in Section 2.1 (P5, L4):

"SCP-M splits out depth regions of the ocean between surface boxes (100-250m average depth), intermediate (1,000m average depth), deep (2,500m average depth) and abyssal depth boxes (3,700-4,000m average depth). The Southern Ocean is split into two boxes, including a polar box which covers latitude range 60-80 degrees South (box 12 in Fig. 1) and sub polar boxes in the Atlantic (box 7) and Pacific-Indian (box 12) basins, which cover latitude range 40-60 degrees South. See O'Neill et al. (2019) for a discussion of the choice of box depth and latitude dimensions."

We have also added depth references to the Caption on Figure 1, Figures 5-7, Figures 9-11

**RC: The authors discuss briefly that previous studies have only used the C. wuellerstorfi data to reconstruct deep ocean δ13C (Peterson et al. study; Kohfeld and Chase study). Which data did these authors select from Oliver et al. (2010)? They mention only using "deep" and "abyssal" sites (again, depths undefined) on page 11, but they do not indicate whether they have filtered the data to only include C. wuellerstorfi (or even Cibicidoides spp), which they SHOULD be doing if they haven't. Otherwise, the changes in δ13C described on page 12 are invalid as descriptions of deep ocean circulation changes in δ13C.**

AC: The work of Oliver et al. (2010) was to aggregate ocean δ13C data, estimate and correct for species-related problems or errors, and thereby provide a dataset to be used for assessing ocean circulation changes. The Oliver et al. (2010) dataset is split into Planktonic and Benthic species data. We had used the benthic datasets. We had given Oliver et al. (2010) the benefit of the doubt, in our first manuscript, as they had gone to substantial effort to produce a δ13C dataset for paleooceanographic purposes.

However, on the suggestion of the reviewer, we have revisited the data and filtered the Cibicides species for the δ13C dataset, which also includes Cibicides data contributed by Govin et al. (2009) and Piotrowski et al. (2009).

We have re-constructed our ocean δ13C database using only the Cibicides species δ13C data, re-calibrated the model for a new set of (penultimate) interglacial starting data, and re-run all of our model-data experiments. The revised manuscript (attached) incorporates these changes in the text, charts and tables.

The data section is updated as follows (==P13, L4==):

"Oliver et al. (2010) compiled a global dataset of 240 cores of marine $\delta^{13}$C data encompassing benthic and planktonic species over the last ~150 kyrs. Oliver et al. (2010) observed considerable uncertainties associated with the broad range of species included, particularly for the planktonic foraminifera. By comparison, Peterson et al. (2014) aggregated marine $\delta^{13}$C for the LGM and late Holocene periods, as time period averages, exclusively sampling benthic *C. wuellerstorfi* data, which is a more reliable indicator of marine $\delta^{13}$C (Oliver et al., 2010; Peterson et al., 2014). To narrow the range of uncertainty, we constrain our use of marine $\delta^{13}$C data to the deep and abyssal benthic *Cibicides* species foraminifera samples in the Oliver et al. (2010) dataset, supplemented with *Cibicides* species $\delta^{13}$C proxy data from Govin et al. (2009) and Piotrowski et al. (2009) (Table 2). Figure 3 shows the $\delta^{13}$C data locations from Oliver et al. (2010), which are concentrated in the Atlantic Ocean. We mapped and averaged the carbon isotope data into SCP-M's boxes on depth and latitude coordinates (Fig. 1), and averaged for each MIS time slice."

**RC: On Page 12, the authors qualitatively describe the differences between "deep" and "abyssal" changes in δ13C. Why leave this discussion qualitative, when the data are available and quantification would be hugely useful. These data in the Pacific that are described are the data that pin the authors' entire argument surrounding early changes in GOC. I think that this warrants a bit more quantification of these data (once species other than Cibicidoides are filtered out of the dataset). I would be interested to know if the differences between deep and abyssal δ13C in the Indo-pacific are statistically significant, and I think plots of the probability distribution functions of these data would be very useful.**

AC: Thanks. Following from this comment we've investigated a number of ways to analyse the data. We have focussed on the $\delta^{13}C$ data (*Cibicides*, as above) only, for this analysis, as there is continuous coverage for deep and abyssal boxes for the Atlantic and Pacific-Indian oceans across all of the MIS stages we are interested in.

We applied some tests for statistical significance of the various boxes throughout the MIS stages. We used a Welch's paired unequal variance t-test for statistically different mean $\delta^{13}C$ between deep and abyssal boxes, and also for differences in the offsets in mean $\delta^{13}C$ between deep and abyssal boxes, between MIS stages. We have added this to the supplementary information file and referenced its location from the main document (P17 L13).

As per the reviewer comment, we first plot the distribution of mean $\delta^{13}C$ values for each of the deep and abyssal boxes across the MIS stages.

**Figure 1**: Distribution histograms of $\delta^{13}C$ data for the Pacific-Indian (left column) and Atlantic Ocean (right column) deep (100/1,000-2,500m) and abyssal (>2,500m) boxes. Plots also show the mean $\delta^{13}C$ for each box (vertical dashed lines), and the calculated offset between the deep and abyssal mean $\delta^{13}C$ values (**CP_RC2_Fig1.png**).

[Figure]

We applied a Welch's paired t-test to test for statistical independence of the means of $\delta^{13}$C in the deep and abyssal ocean boxes for Atlantic and Pacific-Indian, within each MIS. This returns p-values very close to zero for every box pair and every MIS. A p-value <0.05 means that we reject the null hypothesis that the abyssal and deep ocean boxes are statistically the same. That is, our deep and abyssal boxes in the model are statistically independent of each other, in terms of mean $\delta^{13}$C. This simply confirms that our abyssal and deep ocean boxes are not the same in terms of mean $\delta^{13}$C in each MIS.

**Table 1**: Tests for statistical independence of the mean $\delta^{13}$C between deep and abyssal boxes (**CP_RC2_Tab1.png**)

| MIS | Abyssal-deep Pacific-Indian | | Abyssal-deep Atlantic | |
|---|---|---|---|---|
| | t-statistic | p-value | t-statistic | p-value |
| MIS5e | -7.0 | 0 | 5.1 | 0 |
| MIS5d | 16.1 | 0 | 9.8 | 0 |
| MIS5c | 13.0 | 0 | 7.8 | 0 |
| MIS5b | 9.5 | 0 | 6.3 | 0 |
| MIS5a | 13.2 | 0 | 6.9 | 0 |
| MIS4 | 24.0 | 0 | 17.6 | 0 |
| MIS3 | 23.3 | 0 | 21.6 | 0 |
| MIS2 | 18.8 | 0 | 31.6 | 0 |
| MIS1 | 14.2 | 0 | 11.9 | 0 |

Given we discuss in the manuscript (qualitatively) the changes in the offset between deep and abyssal ocean δ13C through the MIS, we can test to see if the changes in deep-abyssal offset from the penultimate interglacial (MIS 5e) to the glacial periods are statistically significant. The chart below shows the deep-abyssal offsets in δ13C for the Pacific-Indian and Atlantic Ocean boxes through each MIS of the last glacial-interglacial cycle. We show the absolute deep-abyssal δ13C offsets for Pacific-Indian and Atlantic Ocean boxes, for each MIS (columns). We also show the deep-abyssal δ13C offsets relative to the penultimate interglacial in MIS 5e (lines).

The Pacific-Indian δ13C offset shows a widening in MIS 5d, relative to MIS 5e, which is maintained until MIS 5a, and then begins a slow decline. The offset declines to a similar value to MIS 5e, by MIS 1 (the Holocene). The Atlantic deep-abyssal δ13C offset does not increase meaningfully until MIS 4, and then peaks at MIS 2 (the LGM), before contracting at MIS 1 to a value almost the same as MIS 5e.

**Figure 2**: Offsets between mean deep and abyssal box δ13C for each MIS in the last glacial-interglacial cycle for the Pacific-Indian (blue columns) and Atlantic Ocean (grey columns). Changes in the offsets from the penultimate interglacial (MIS 5e) are shown by the blue (Pacific-Indian) and grey (Atlantic) lines (**CP_RC2_Fig2.png**).

[Figure]

We further undertook Welch's paired T-tests for the independence of deep-abyssal offsets in mean δ13C with respect of the penultimate interglacial period (MIS 5e), for the periods MIS 1-5e. The null hypothesis is that the deep-abyssal offset in mean δ13C in each MIS is not statistically independent of MIS 5e (i.e. statistically the same and not supportive of a change in deep-abyssal δ13C distribution that may be delivered by a changed ocean process). p-values>0.05 lead to the null hypothesis being accepted, whereas p-values <0.05 lead to the null hypothesis being rejected and confirm statistical independence of the deep-abyssal offsets relative to MIS 5e (perhaps supportive of a changed ocean distributive process in the glacial period). Deep-abyssal offsets for the Pacific-Indian during MIS 2-MIS5d are statistically independent of MIS 5e, supportive of a changed oceanic distribution of δ13C throughout the glacial period. The MIS 1 Pacific-Indian deep-abyssal δ13C offset is not statistically independent of MIS 5e, indicating a similar deep-abyssal δ13C distribution between the last and penultimate interglacial periods. For the Atlantic Ocean, deep-abyssal mean δ13C offsets are not statistically independent with respect to MIS 5e (p-value >0.05, accept null hypothesis), until the period MIS 2-4. Atlantic deep-abyssal mean δ13C offset in MIS 1 is not statistically different from MIS 5e.

**Table 2**: Statistical tests for significance of difference in deep-abyssal δ13C offsets versus penultimate interglacial (MIS 5e). 'Accept'/red is to accept the null hypothesis - no statistically significant difference, 'Reject'/green is to reject the null hypothesis – statistically significant difference with respect of MIS 5e (**CP_RC2_Tab2.png**).

| MIS | Pacific-Indian (vs MIS 5e) | | | Atlantic (vs MIS 5e) | | |
|---|---|---|---|---|---|---|
| | t-statistic | p-value | Accept/ reject null | t-statistic | p-value | Accept/ reject null |
| MIS 5e | 0.0 | 0.500 | Accept | 0.0 | 0.500 | Accept |
| MIS5d | 3.8 | 0.000 | Reject | 1.4 | 0.079 | Accept |
| MIS5c | 3.0 | 0.002 | Reject | 0.7 | 0.253 | Accept |
| MIS5b | 2.9 | 0.002 | Reject | 1.5 | 0.074 | Accept |
| MIS5a | 4.5 | 0.000 | Reject | 1.4 | 0.082 | Accept |
| MIS4 | 3.7 | 0.000 | Reject | 4.5 | 0.000 | Reject |
| MIS3 | 2.1 | 0.017 | Reject | 3.6 | 0.000 | Reject |
| MIS2 | 1.7 | 0.044 | Reject | 5.9 | 0.000 | Reject |
| MIS1 | 0.7 | 0.246 | Accept | 0.1 | 0.478 | Accept |

The statistical analysis above is helpful and provides support for our model-data experiment results – that GOC slowed in MIS 5d and AMOC slowed in MIS 4. However, we do want to make the point that our model-data results don't hang on one particular data point to deliver these findings, in any MIS. They are constrained and optimised with many observations. The model-data results in the first instance are telling us that, the many observational forcings we have imposed in each MIS (SST, salinity, sea-ice cover proxy, coral reef carbonates) are not enough to deliver the change in atmospheric CO2, atmospheric and ocean $\delta 13C$, D14C and CO23 proxy data. Changes from within the set of ocean circulation, mixing and/or biology parameters are needed. Note the result for GOC that is hinted at by the $\delta 13C$ data, that we model in our experiments, is sustained throughout the last glacial cycle, not just at MIS 5d.

The main point of our work, and what has taken substantial effort, is to undertake an exhaustive model-data optimisation using a carbon cycle box model and multiple atmospheric and ocean proxy data. The model-data results don't just rely on one data point, the results need to be the best fit for all the data used, in each MIS. This is where this model-data experiment differentiates itself from many others.

We have included the distribution plot and T-test table above, in the manuscript's Supporting Information. We make reference to this material in the manuscript when discussing the data charts in the "Data Analysis" section. This chart/table provide supplemental support to the model-data analysis, the latter being the focus of our manuscript. We feel that the manuscript is becoming very voluminous and we also think that this analysis would require its own section in the manuscript (However, it is presented in this response to the discussion (preserved online) and in the SI).

**RC: Some type of quantification would also be very useful for the authors' description of the "transient drop in abyssal Atlantic ocean CO3= at MIS5b" on page 14. I was not convinced that this transient drop exists from the figure presented.**

AC: Yes, the axes on these charts are a little difficult to decipher small changes in the data. We wish to show the range of shallow-deep $CO_3^{2-}$ data (not just deep-abyssal), as the pattern is quite interesting at the LGM-Holocene. Our suggestion is to add the changes in units for the pattern that we wish to describe (P17, L35):

"There is a modest drop in abyssal Atlantic Ocean $CO_3^{2-}$ at MIS 5b (-13 µmol kg$^{-1}$ relative to MIS 5c), which coincides with a minor drop in abyssal Atlantic Ocean $\delta^{13}C$ (-0.19‰) and atmospheric $CO_2$ (-14 ppm), suggesting a possible common link."

**RC: Please note on the bottom of page 13 and top of page 14 that the authors mean to refer to Figure 7 (not 6) to describe carbonate ion concentration data.**

AC: Thank you, we have corrected these references in the manuscript.

**RC: Last sentence before Results section: Please cite the figures you are using to make these observations about changes in δ13C and DD14C**

AC: We have added the figure references

**RC: Similar quantification would be useful in the comparison between the carbonate ion concentration model output and data in Figure 9 and in the discussion on page 16-17.**

AC: Figure references for Figure 9 added to the text here

**CP reviewer comments #3 and author responses**

**AC**: We thank the reviewer for their comments, suggestions and input into this manuscript. These comments make a strong contribution to improving the quality of our work. Please see below our responses to the individual comments.

We have made reference to changes to the manuscript, which are included as a supplement to the author comments, in track changes. Page and line references below refer to locations in the revised document with track changes.

Please note that we have changed our treatment of ocean $\delta^{13}$C proxy data, stemming from one of the other reviewer comments, to only include $\delta^{13}$C from *Cibicides* species of benthic foraminifera. We have also made some small changes to the parameterisation of the volcanic and weathering isotopic signatures in the model, from reviewer comments. These changes required the re-calibration of our model and re-running of our model-data experiments. The model-data results changed modestly. We have updated the figures and text (tracked in the attachment) in the manuscript, accordingly.

**Major comments:**

**RC 1) The "data analysis" section 3 presents the changes in atm. CO2, d13CO2, oceanic d13C, D14C and CO3(2-) as inferred from proxy records from the LIG to the LGM. This is obviously a huge task, but which I am afraid can give rise to approximations and simplifications. I would consider seriously amending this section. How can the "increase in d13C across the glacial cycle be attributed to the growth of tundra at high latitudes"? (p12, L. 2-3).**

**AC**: Thanks for the comment. In this instance (P12, L2-3 in the original manuscript) and throughout our manuscript, we have been a bit loose with our references to tundra, permafrost and peat, as you point out in this comment and a few below.

What we mean to refer to here is the storage of carbon by the accumulation and freezing, or burial, of peat and other soil organic matter under soil overburden, and growth of cold-climate vegetation, throughout the glacial cycle (e.g. Tarnocai et al., 2009; Ciais et al., 2012; Schneider et al., 2013; Eggleston et al., 2016; Treat et al., 2019).

We have corrected the statement on P12 L2-3, and expanded a bit, including a few more references and other possible causes of the atmospheric $\delta^{13}$C pattern, now at P15 L10:

"Atmospheric $\delta^{13}$C (Fig. 4(B)) increased by ~0.4‰ between the penultimate interglacial (MIS 5e) and the Holocene (MIS 1), with temporary falls at MIS 5d, MIS 4 and in the last glacial termination (between MIS 1 and 2). The cause of the observed increase in atmospheric $\delta^{13}$C across the last glacial-interglacial cycle may be the effect of accumulation and freezing, or burial in glacial sediments, of peat and other soil organic matter at the high latitudes (e.g. Tarnocai et al., 2009; Ciais et al., 2012; Schneider et al., 2013; Eggleston et al., 2016; Ganopolski and Brovkin, 2017; Treat et al., 2019). According to Treat et al. (2019),

peatlands and other vegetation accumulated carbon in the relatively warm periods, and these carbon stocks were then frozen and/or buried in glacial and other sediments during the cooler periods, throughout the last glacial cycle. This buried or frozen stock of carbon persists to the present day (Tarnocai et al., 2009), although according to Ciais et al. (2012) it may be smaller now than in the LGM. Schneider et al. (2013) evaluated several possible candidates for the rising atmospheric $\delta^{13}$C pattern across the last glacial-interglacial cycle and could not discount any of (1) changes in the carbon isotope fluxes of carbonate weathering and sedimentation on the seafloor, (2) variations in volcanic outgassing or (3) peat and permafrost build-up throughout the last glacial-interglacial cycle.

The large drop in $\delta^{13}$C in MIS4, reverses in MIS 3 (Fig. 4(B)). This excursion in the $\delta^{13}$C pattern likely resulted from sequential changes in SST (cooling), AMOC, Southern Ocean upwelling and marine biological productivity (Eggleston et al., 2016). Eggleston et al. (2016) parsed the atmospheric $\delta^{13}$C signal into its component drivers across MIS 3-5, using a stack of proxy indicators, and highlighted the sequence of events between the end of MIS 5 and beginning of MIS 3, and their cumulative effects to deliver the full change in atmospheric $\delta^{13}$C. Our MIS-averaging approach fails to capture the full amplitude of the changes in atmospheric $\delta^{13}$C during MIS 3-5, and only captures the changes in the mean-MIS value, serving to understate the full amount of transient changes in responsible processes. In addition, the MIS-averaging approach misses the sequential timing of changes in processes within each MIS. These are limitations of our steady-state, MIS-averaging approach. The reduction in atmospheric $\delta^{13}$C at the last glacial termination, between MIS 1 and MIS 2, coincident with a large atmospheric $CO_2$ increase, is attributed to the release of deep-ocean carbon to the atmosphere resulting from increased ocean circulation and Southern Ocean upwelling (Schmitt et al., 2012). The subsequent rebound of $\delta^{13}$C in the termination period and the Holocene is believed to result from terrestrial biosphere regrowth, in response to increased $CO_2$ and carbon fertilisation (Schmitt et al., 2012; Hoogakker et al., 2016). "

Other amendments to this section are shown in track changes.

**RC: p12, L. 11-14: How were the values for MIS3 DD14C in the Atlantic derived? From Fig. 6a, it looks like there is no data across MIS3.**

**AC**: Thanks, this was a charting error and now the chart has been corrected to show the data for MIS 3.

**RC: This is quite a shortcut to explain the deglacial D14C decrease, and maybe you want to check the references and include " increase in Southern Ocean ventilation" above anything else.**

**AC**: P16, L4 modified to "….an acceleration in atmospheric $\Delta^{14}$C decline at the last glacial termination is attributed to the release of old, $^{14}$C -depleted waters from the deep ocean, **due mainly to increased Southern Ocean upwelling** (e.g. Sikes et al., 2000; Marchitto et al., 2007; Skinner et al., 2010; Burke and Robinson, 2012; Siani et al., 2013; Skinner et al., 2017)."

**AC**: This sentence re-worded as:
(P17, L35) "There is a modest drop in abyssal Atlantic Ocean $CO_3^{2-}$ at MIS 5b (-13 µmol $kg^{-1}$ relative to MIS 5c), which coincides with a minor drop in abyssal Atlantic Ocean $\delta^{13}C$ (-0.19‰) and atmospheric $CO_2$ (-14 ppm), indicating a common link. Menviel et al. (2012) modelled a transient slowdown in North Atlantic overturning circulation for this period, which could explain these features. "

**RC: 2) Fit with the data: 50 umol/L as an "arbitrary standard deviation' for [CO3] is huge and represents more than the [CO3] changes (0-30 umol/L) recorded across the G-IG cycles. How much was taken for the standard deviation for d13C and D14C? It looks quite large. Figures 9-11 would gain in having a more appropriate range in the y axis. At the moment the ranges and std are large, so that it almost looks like there are no changes from MIS5 to MIS 2.**

**AC**: Re $CO_3^{2-}$. In response to this reviewer's comments, and a change to our data approach from the other reviewer comments (using only *Cibicides* species for $\delta^{13}C$), we have been able to reduce our default standard deviation for ocean $CO_3^{2-}$ from 50 umol $kg^{-1}$ to 15 umol $kg^{-1}$, a substantial improvement. The rationale for setting the $CO_3^{2-}$ SD at an artificial level for the weighting in our model-data optimisation is dealt with in Section 2.3.2. This is an unfortunate feature of using a box model with large boxes and applying sparse proxy data. The relatively small number of $CO_3^{2-}$ data points in clustered locations leaves relatively small standard deviations, giving $CO_3^{2-}$ a disproportionate weighting in the model-data optimisation versus the other proxies. Therefore, we overcome the issue by scaling up the $CO_3^{2-}$ standard deviations and applying as default across all boxes and MIS time slices.

Re $\delta^{13}C$ and $\Delta^{14}C$. The standard deviations are calculated from box-averaged published proxy data and shown in the supporting information. The standard deviations look large for these box-averaged and MIS-averaged values, because the boxes in the box model are large. The ocean box $\delta^{13}C$ standard deviation is now lower in the revised manuscript due to filtering out only *Cibicides* species, from the other reviewer comments.

The issue of box size and standard deviation is addressed again in the discussion of limitations of the study (P34 L7):

"However, given the large spatial coverage of the SCP-M boxes, data for large areas of the ocean are averaged, and some detail is lost. For example, in the case of the carbonate ion proxy, we apply a default estimate of standard deviation to account for the large volume of ocean covered by SCP-M's boxes relative to the proxy data locations, and to enable the normalisation of the carbonate ion proxy data in a procedure that uses the data standard deviation as a weighting. Despite this caveat, we argue that the model-data experiment results provide a good match to the data across the various atmospheric and ocean proxies as shown in Figs 9-11."

Re Figs 9-11. The standard deviation ranges for $CO_3^{2-}$ and $\delta^{13}C$ are now narrower following the improvements we have made, which improves the resolution of Figs 9-11. In addition,

we have expanded y-axes where we can to help with reading the figures.

**RC: 3) References: In general I find that only a few references are used over and over and sometimes not appropriately. A few additional references are included in this review. Please note the typo throughout the document in "Ridgwell".**

**AC:** Thanks, we've now added the references suggested by the reviewer, throughout the manuscript, and we corrected the typo for Ridgwell throughout.

References added following this reviewers' comments:

Watson et al., 2000
Joos et al., 2004
Tarnocai et al., 2009
Ganopolski et al., 2010
Menviel et al., 2012
Menviel and Joos, 2012
Brovkin et al., 2012
Jaccard et al., 2013
Schneider et al., 2013
Menviel et al., 2015
Yu et al., 2016
Ganopolski and Brovkin, 2017
Lindgren et al., 2018
Mauritz et al., 2018
Yamamoto et al., 2019
Treat et al., 2019

**Specific comments:**

**RC: 1) Abstract: The first line does not make sense. Please reformulate. L. 3 Please add "SO" in front of "biological productivity"**

**AC**: Re-formulated as: "We conduct a model-data analysis of the marine carbon cycle to understand and quantify the drivers of atmospheric $CO_2$ during the last glacial cycle".

Southern Ocean added to the sentence P1 L3.

**RC: 2) Introduction: - L.15-19: please be more specific. Instead of "Ocean biology" you might want to refer to "iron fertilisation and its impact on nutrient utilisation", or changes in remineralisation depth (e.g. Kwon et al. 2009).**

**AC**: text modified to (P2 L18):

"Hypotheses for an ocean biological role include the effects of iron fertilisation on biological export productivity (e.g. Martin, 1990; Watson et al., 2000; Martinez-Garcia et al., 2014),

the depth of remineralisation of particulate organic carbon (POC) (e.g. Matsumoto, 2007; Kwon et al., 2009; Menviel et al., 2012), changes in the organic carbon:carbonate ("the rain ratio") or carbon:silicate constitution of marine organisms (e.g. Archer and Maier-Reimer, 1994; Harrison, 2000), and increased biological utilisation of exposed shelf-derived nutrients such as phosphorus (e.g. Menviel et al., 2012)."

**RC: What do you mean by composite mechanisms?**

**AC**: we have amended this to "the aggregate effects of several mechanisms" throughout the document

**RC: It would be good to also introduce the numerous modelling studies that have been done on the topic of G-IG changes in pCO2, and notably transient simulations of the G-IG trying to understand the changes in pCO2 (e.g. Ganopolski & Brovkin 2017, Menviel et al., 2012).**

**AC**: Thanks, we have added to our introduction (P2 L23):

"Several studies have attempted to solve the problem of glacial-interglacial $CO_2$ by modelling either the last glacial-interglacial cycle in its entirety, or multiple glacial-interglacial cycles (e.g. Ganopolski et al., 2010; Menviel et al., 2012; Brovkin et al., 2012; Ganopolski and Brovkin, 2017). These studies highlight the roles of orbitally-forced Northern Hemisphere ice sheets in the onset of the glacial periods, and important feedbacks from ocean circulation, carbonate chemistry and marine biological productivity throughout the glacial cycle (Ganopolski et al., 2010; Brovkin et al., 2012; Ganopolski and Brovkin, 2017). Menviel et al. (2012) modelled a range of physical and biogechemical mechanisms to deliver the full amplitude of atmospheric $CO_2$ variation in the last glacial-interglacial cycle, using transient simulations with the Bern3D model. According to Brovkin et al. (2012), a ~50 ppm drop in atmospheric $CO_2$ early in the last glacial cycle was caused by cooling sea surface temperatures (SST), increased Northern hemisphere ice sheet cover, and expansion of southern-sourced abyssal waters in place of North Atlantic Deep Water (NADW) formation. Ganopolski and Brovkin (2017) modelled the last four glacial cycles with orbital forcing as the singular driver of carbon cycle feedbacks. They described the "carbon stew", a feedback of combined physical and biogeochemical changes in the carbon cycle, to drive the last four glacial-interglacial cycles of atmospheric $CO_2$."

And also, a few lines down to explain how our approach differs (P3 L23):

"Our modelling approach differs from other model studies of the last glacial-interglacial cycle (e.g. Ganopolski et al., 2010; Menviel et al., 2012; Brovkin et al., 2012; Ganopolski and Brovkin, 2017), in that we constrain several physical processes from observations (SST, sea level, sea-ice cover, salinity, coral reef fluxes of carbon), then solve for the values of model parameters for ocean circulation and biology based on an optimisation against atmospheric and ocean proxy data. "

And at P8 L14:

"Joos et al. (2004), Ganopolski et al. (2010), Menviel et al. (2012), Menviel and Joos (2012), Brovkin et al. (2012) and Ganopolski and Brovkin (2017) provide coverage of the termination period with transient simulations of the last glacial-interglacial cycle, using intermediate complexity models (more complex than our model). "

**RC: 3) Methods: - Variables included in the model: surely the model includes Dissolved Inorganic Carbon.**

**AC**: yes, the model includes DIC and we have added DIC to the sentence.

**RC: By "CO2", do you mean atmospheric CO2?**

**AC**: yes, we have added "atmospheric" to the sentence at P3 L33.

**RC: Does the model really includes "carbonate ions" as a prognostic tracer?**

**AC**: Yes. SCP-M calculates $CO_3^{2-}$ concentration in umol $kg^{-1}$, by calculating the three species of DIC. First, $pCO_2$ is calculated using the method of Follows et al. (2006) which takes as inputs DIC, alkalinity, pH, SST, salinity and phosphorus in each box in the model. Then $H_2CO_3$, $HCO_3^-$ and $CO_3^{2-}$ are calculated using coefficients for the solubility of $CO_2$ ($K_0$) and coefficients for carbonic acid of $K_1$ and $K_2$ using Lueker et al. (2000). In the model documentation paper (O'Neill et al., 2019) the SCP-M model estimates for $CO_3^{2-}$ in a modern ocean setting are demonstrated to align with modern data from the ocean, using data from Key at al (2004).

We have added a summary sentence to describe this, in section 2.1 "Model description" on P4.

**RC: p4, L. 2: please refer to section 2.2.1 and Figure 2.**

**AC**: Added

**RC: p7: I am very confused by the treatment of the terrestrial biosphere in the model and the paragraph L. 19-27. It reads like there is an interactive terrestrial module. But how can NPP be calculated with significance if there is no atm. Temperature or precipitation in the model?**

Our box model applies a simple representation of the terrestrial biosphere, whereby biological productivity responds to carbon fertilisation. Therefore, $CO_2$ is the driver of terrestrial biosphere productivity in this model. We use a two-box terrestrial box model scheme, presented in Harman et al. (2011). The inputs are starting estimates of net primary productivity (NPP), the terrestrial biosphere carbon stock, plant respiration rate and atmospheric $CO_2$. The approach of Harman et al. (2011) is to split the terrestrial biosphere into two boxes, a fast-response (grasslands and grassy components of savannah systems)

and a slow-response (woody trees) component. In this model, the productivity is mostly focussed on the plants/grasses component.

The formula is shown in the model documentation paper (O'Neill et al., 2019) and Harman et al. (2011), and extract is reproduced here:

$$dAtCO2/dt = -N_{pre}RP[1+\beta LN(AtCO2)] + Cstock/k + D_{forest}$$

Where Npre is NPP at a reference pre-industrial level of atmospheric CO2, RP is a parameter to split NPP between short-term terrestrial biosphere carbon stock and the longer term stock (Cstock1 and Cstock2). B is a parameter with a value typically in the range 0.4-0.8 (Harman et al., 2011). Cstock is the carbon stock in each terrestrial biosphere box, k is the respiration timeframe for each box. Dforest is the prescribed rate of deforestation emissions for present day simulations and projections. A terrestrial biosphere fractionation factor is applied for the carbon isotopes.

Harman et al. (2011) model the terrestrial biosphere primarily as a function of atmospheric $CO_2$. They also incorporate an optional temperature dependency. This is the same approach used in the simplest 4Box terrestrial biosphere module of the Bern Simple Carbon Model (Strassman and Joos, 2018; Seigenthaler and Joos, 1992; Kicklighter et al., 1999; Meyer et al., 1999), and described by Enting (1994) – although we understand that there are various terrestrial biosphere modules applied with the Bern models, and most are more complex. As far as we can discern, the simple carbon fertilisation approach is also used in Jelstch-Thommes et al. (2019), which also applies the simplest 4Box terrestrial biosphere of the simple Bern model.

There are other possible drivers of the NPP – temperature, precipitation, soil nutrient levels. In the context of our simple carbon cycle model, we are mainly interested in $CO_2$. We don't model atmospheric temperature, and if we were to try to incorporate atmospheric temperature as a driver of terrestrial biosphere, we would also need to incorporate it for terrestrial weathering. There is a limit to how much detail we want to include in the model given we are conducting many simulations (~80,000) in our model-data optimisations across the MIS of the last glacial-interglacial cycle.

We do note that there are studies devoted to determining whether the $CO_2$ fertilisation effect or climate is the dominant control on terrestrial biosphere NPP and the size of the terrestrial biosphere carbon stock. According to Hoogakker et al. (2016), CO2 fertilization, rather than climate, is the primary driver of lower glacial net primary productivity by the terrestrial biosphere, accounting for around 85% of the reduction in global NPP at the LGM. Kaplan et al. (2002) also concluded that over glacial-interglacial timescales, global terrestrial carbon storage is controlled primarily by atmospheric $CO_2$, while the climate has more influence on the isotopic composition. Otto et al. (2002) also found that the CO2 fertilization effect is mostly responsible for the total increase in vegetation and soil carbon stocks since the last glacial maximum. Kohler et al. (2010) prioritised $CO_2$ fertilisation as the driver of terrestrial biosphere in their "control" main simulation scenario for glacial-interglacial cycles over the last 740 kyr, but also ran scenarios with a climatic driver for the terrestrial biosphere to estimate the effects of "fast" climate changes on atmospheric $\delta^{13}C$. Other

studies arguing that atmospheric $CO_2$ is an important, or is the main driver of terrestrial biosphere productivity include Kicklighter et al. (1999), Joos et al. (2001), Schimel et al. (2015), Sitch et al. (2008), Arneth et al. (2017)). This view has been contested by Francois et al. (1999) and van der Sleen et al. (2015).

Given we don't model the atmospheric temperature or precipitation, we saw limited additional benefit to introduce them into our model of the terrestrial biosphere, although it would not be difficult to do this. Finally, given that $CO_2$ and atmospheric temperature co-vary closely, across glacial cycles, it seems of limited benefit to split these effects out in our simple carbon cycle modelling exercise. For example, Meyer et al. (1999) found similar results for modelling carbon uptake in the terrestrial biosphere whether only $CO_2$ fertilisation, or $CO_2$ fertilisation + climate, were included as drivers of NPP – but noting this was not tested for the LGM.

In summary, our aim is not to contribute new findings on the terrestrial biosphere, but we present the behaviour of the terrestrial biosphere in our manuscript to confirm that our exhaustively multi-proxy constrained model-data output is consistent with the range of literature estimates of variations in the terrestrial biosphere in the last glacial-interglacial cycle and LGM-Holocene period, and we show this. For example, our experiment shows a change in the terrestrial biosphere carbon stock of +630 PgC between the MIS 2 (LGM) and MIS 1 (Holocene) period. This compares with other estimates of +540 PgC (Brovkin et al., 2007), +~820-850 PgC (Joos et al., 2004) – with the majority by $CO_2$ fertilisation, ~+500 PgC (Kohler et al., 2010), +~500 PgC (Brovkin et al., 2012), +850 PgC (Jeltsch-Thommes et al., 2019), +511 +/- 289 PgC (Peterson et al., 2014), +378 +/- 88 PgC (Menviel et al., 2016). Another estimate of the LGM-Holocene terrestrial biosphere change is 550-694 Pg C, which our result of 630 Pg C sits comfortably within (Prentice et al., 2011)

Our estimate is actually towards the upper end of the literature ranges, suggesting if anything we could exaggerate the effects of the terrestrial biosphere from the LGM to the Holocene period, with perhaps little to gain by splitting out temperature and precipitation effects. If did, we would probably also need to consider other important features such as soil nutrients and local humidity.

While we have a simple, but explicit two-box representation of the terrestrial biosphere, we don't believe that this detracts from our model-data results, as shown in Figures 9-11 and Figure 12 specifically for the terrestrial biosphere.

If there is some reason to examine the terrestrial biosphere in more detail, we suggest for our study this would be done simply by a sensitivity, as applied in Menviel et al. (2016) with regard to C3/C4 plants and the relative proportional influence of C3 and C4 plants on terrestrial biosphere $\delta^{13}$C fractionation.

We have added some text to explain that we have a simplified representation of the terrestrial biosphere employing $CO_2$ fertilisation, and that we don't take account of temperature and precipitation, in the methods section, P5 L24. This also includes discussion of the isotopic fractionation factor in response to one of the other reviewers:

[revised manuscript text omitted]

We have also updated the discussion of our model results for the terrestrial biosphere, to provide a bit more detail and some additional references (Section 5.3), plus an additional caveat in the "advantages and limitations section" (P34, L18).

"Furthermore, we apply a simple representation of the terrestrial biosphere in our model-data experiments, relying primarily on atmospheric $CO_2$ as the driver for NPP. This approach provided reasonable results for the terrestrial biosphere carbon stock and NPP, on the whole, but may miss some detail in the terrestrial biosphere during the last glacial-interglacial cycle."

**RC: Why is "tundra" discussed with such emphasis in this paragraph?.**

**AC**: Thanks for picking up on this. We have substantially revised this paragraph as follows (P10 L25):

"The terrestrial biosphere module in SCP-M does not explicitly represent the carbon stored in buried peat, permafrost and also cold-climate vegetation that may have expanded its footprint in the glaciation, such as tundra biomes (e.g. Tarnocai et al., 2009; Ciais et al., 2012; Schneider et al., 2013; Eggleston et al., 2016; Ganopolski and Brovkin, 2017; Treat et al., 2019). The freezing and burial of organic matter across the glacial cycle may significantly imprint the terrestrial biosphere $CO_2$ size and $\delta^{13}$C signature (Tarnocai et al., 2009; Ciais et al., 2012; Schneider et al., 2013; Eggleston et al., 2016; Ganopolski and Brovkin, 2017; Mauritz et al., 2018; Treat et al., 2019). Schneider et al. (2013) and Eggleston et al. (2016) both observed a permanent increase in atmospheric $\delta^{13}$C during the last glacial cycle, of ~0.4‰, and attributed its cause likely due to soil storage of carbon in peatlands which were buried or frozen as permafrost as the glacial cycle progressed. Ganopolski and Brovkin (2017) incorporated permafrost, peat, and buried carbon into their transient simulations of the last four glacial- interglacial cycles, observing that these features dampened the amplitude of glacial-interglacial variations in terrestrial biosphere carbon stock, in the CLIMBER-2 model. As a crude measure to account for this counter-$CO_2$ cycle storage of carbon in the terrestrial biosphere and frozen soils, we force the terrestrial biosphere productivity parameter in SCP-M in the range ~+5-10 PgC yr$^{-1}$, increasing into the LGM (MIS

2), and maintained in the Holocene (MIS 1). We maintain the forcing of the terrestrial biosphere in the Holocene, as the posited effects of buried peat and permafrost storage of carbon on atmospheric $CO_2$ and $\delta^{13}C$ during the lead-up and into the LGM, were likely not fully reversed after the glacial termination (Tarnocai et al., 2009; Eggleston et al., 2016; Mauritz et al., 2018; Treat et al., 2019), and were partially or wholly replaced by other soil stocks of carbon (e.g. Lindgren et al., 2018). SCP-M calculates net primary productivity (NPP) using this productivity input parameter, as a function of carbon fertilisation (Harman et al., 2011)."

**RC: Tundra is not an "inert" carbon pool**

**AC**: we've modified the sentence as per above excerpt to refer to carbon stored in frozen peat, permafrost soils.

**RC: and I don't think "permafrost" is a vegetation type**

**AC**: We've modified this sentence as per above excerpt, to remove the reference to permafrost as a vegetation type.

**RC: What is "pre-carbon fertilisation"?**

**AC**: This is just the $N_{pre}$ in the equation for NPP from the model documentation, reproduced above. We can refer to this as "undisturbed" (by $CO_2$) NPP. The equations for NPP takes an input value $N_{pre}$, which is subsequently varied due to any change in atmospheric $CO_2$. This is our model representation of $CO_2$ fertilisation of the terrestrial biosphere.

**RC: p8: what is the point of Table 1 if all the values of GOC, AMOC, biology are the same? It would be interesting to mention the PI control values though.**

**AC**: Thanks, we've consolidated Table 1 to show the MIS model-data experiment ranges and the PI control values.

**RC: - p10-11: The 'depth issue' should also be discussed in 2.3.1 and 2.3.2.**

**AC**: Re 2.3.1 – there is a much greater coverage of $\delta^{13}C$ and $\Delta^{14}C$ data for the ocean boxes so we have not applied a default weighting for those data in our model-data optimisation. For $CO_3^{2-}$, a problem presents because there are only 1 or 2 data points in some boxes, and they are clustered near the box boundary, so we end up with unrepresentative data for some boxes for $CO_3^{2-}$. So, we applied a larger weighting for $CO_3^{2-}$ data, as discussed in 2.3.2.

4) Discussion:

**RC: p20, L. 3-6: It is not what the simulations tell you, but the proxy data!**

**AC**: We've removed this reference to the modelling and replaced with reference to the proxy data shown in Figure 4 (P23, L7).

**RC: p21, L. 1-2: This is wrong → you are forcing your model with SST, Sea-ice. . . . so all these factors contribute to the pCO2 decrease. The experiments show that changes in oceanic circulation and SO biological productivity also contribute to that pCO2 de- crease.**

**AC**: We have reworded this sentence to list the full set of changes modelled (P24 L7)

**RC: Please take into consideration that G-IG pCO2 changes have been previously successfully simulated with models of intermediate complexity (e.g. e.g. Ganopolski & Brovkin 2017, Menviel et al., 2012) and box models.**

**AC**: We have added a sentence at the start of the discussion to reference these studies (P23, L5) and they are referenced throughput the Discussion.

**RC: p21, L. 3-4: I don't understand the meaning**

**AC**: This sentence has been reworded (P24 L6).

**RC: p21, L. 7: Might want to check Piotrowski et al., 2008, Yu et al., 2016. (Nat. Geo).**

**AC**: We have picked up the citation of Yu et al. (2016) in reference to AMOC in the MIS 4, a little further down in the manuscript (P29 L28). We have added a reference to Piotrowski et al. (2009) in the same place (P29 L29).

We have also added the Piotrowski et al. (2009) $\delta^{13}C$ data to our dataset and cited it in the manuscript (Table 2).

**RC: p21, L. 10 –p22, L. 5: This section really has to be discussed in light of all the work that has been done on the impact of iron fertilisation in the Southern Ocean. Some work on the topic: Watson et al., 2000, Nature; Jaccard et al., 2013, Science; Yamamoto et al., 2019, Climate of the Past;**

**AC**: Text added (P31 L2):

"Our finding of increased biological productivity, while mostly constrained to MIS 2 and MIS 4, and a modest contributor to the overall glacial $CO_2$ drawdown, corroborates proxy data (e.g. Martinez-Garcia et al., 2014; Lambert et al., 2015; Kohfeld and Chase, 2017) and recent model-data exercises (e.g. Menviel et al., 2016; Muglia et al., 2018; Khatiwala, 2019). Martin (1990) pioneered the "iron hypothesis", which invoked the increased supply of continent-borne dusts to the Southern Ocean in glacial periods. Increased dust supply stimulated more plankton productivity where plankton were bio- limited in nutrients supplied in the dust, such as iron (Martin, 1990). Since then, the iron hypothesis has retained an important place in the debate over glacial-interglacial cycles of $CO_2$. Watson et al. (2000) took experimental data on the effects of iron supply on plankton productivity in the Southern Ocean (Boyd, 2000) and applied this to a carbon cycle model across glacial- interglacial cycles. Their modelling, informed by the ocean experiment data, suggested that variations in the

Southern Ocean iron supply and plankton productivity could account for large (~40 ppm) swings in atmospheric $CO_2$, with peak activity in the last glacial cycle at MIS 2 and MIS 4. Debate has continued over the magnitude of the contribution of Southern Ocean biological productivity to the glacial $CO_2$ drawdown. According to Kohfeld et al. (2005), based on sediment data, the Southern Ocean biological productivity mechanism could account for no more than half of the glacial $CO_2$ drawdown. Others emphasise that Southern Ocean biological export productivity fluxes may have been weaker in the LGM, in absolute terms, but that with weaker Southern Ocean upwelling, the iron-enhanced productivity contributed to a stronger biological pump of carbon and was a major contributor to the LGM $CO_2$ drawdown (Jaccard et al., 2013; Martinez-Garcia et al., 2014; Yamamoto et al., 2019). "

**RC: p22, L. 18: "sea-ice cover"**

**AC**: Thanks, corrected

**RC: p23, L. 1-12: Figure 13 is interesting but care has to be taken here given the large size of the "boxes". This should at least be discussed in light of previous modelling studies on the subject (e.g. Menviel et al., 2015, GBC).**

**AC**: This figure has changed from the original manuscript due to a change in our data method for $\delta^{13}C$, stemming from the other reviewer comments. We are now only using *Cibicides* species $\delta^{13}C$ data, and we re-ran our model-data experiments. There are only slight variations to our model-data results. However, a narrower spread of standard deviations of the $\delta^{13}C$ data necessitates us to change this Figure. We do think it's an important figure that provides some insights into our model, the results in this manuscript and how they might differ from other studies that simply rely on qualitative and simple statistical analysis of proxy data (without models).

Text added P29 L3:

[revised manuscript text omitted]

Mauritz, M., Celis, G., Ebert, C., Hutchings, J., Ledman, J., Natali, S., Pegoraro, E., Salmon, V.,

Schädel, C., Taylor, M., , and Schuur, E.: Using stable carbon isotopes of seasonal ecosystem respiration to determine permafrost carbon loss, Journal of Geophysical Research: Biogeosciences, 124, 46–60, 2018.

Menviel, L. and Joos, F.: Toward explaining the Holocene carbon dioxide and carbon isotope records: Results from transient ocean carbon cycle-climate simulations, Paleoceanography, 27, PA1207, doi:10.1029/2011PA002 224, 2012.

Menviel, L., Joos, J., and Ritz, S.: Simulating atmospheric CO2, 13C and the marine carbon cycle during the Last Glacial-Interglacial cycle: possible role for a deepening of the mean remineralization depth and an increase in the oceanic nutrient inventory, Quaternary Science Reviews, 56, 46–68, 2012.

Menviel, L., Mouchet, A., Meissner, K. J., Joos, F., and England, M. H.: Impact of oceanic circulation changes on atmospheric d13CO2, Global Biogeochemical Cycles, 29, 1944–1961, 2015.

Menviel, L., Yu, J., Joos, F., Mouchet, A., Meissner, K. J., and England, M. H.: Poorly ventilated deep ocean at the Last Glacial Maximum inferred from carbon isotopes: A data-model comparison study, Paleoceanography, 31, 2–17, 2016.

Meyer, R., F. Joos, G. Esser, M. Heimann, G. Hooss, G. Kohlmaier, W. Sauf, R. Voss, and U. Wittenberg. The substitution of high-resolu- tion terrestrial biosphere models and carbon sequestration in response to changing CO2 and climate, Global Biogeochem. Cycles, 13, 785-802, 1999.

O'Neill, C., A. Mc. Hogg, M.J. Ellwood, S. E., and Opdyke, B.: The [simple carbon project] model v1.0, Geosci. Model Dev., 12, 1541–1572, https://doi.org/10.5194/gmd–12–1541–2019, 2019.

Otto, D., Rasse, D., Kaplan, J., Warnant, P., and Francois, L.: Biospheric carbon stocks reconstructed at the Last Glacial Maximum: com- parison between general circulation models using prescribed and computed sea surface temperatures, Global and Planetary Change, 33, 117–138, 2002.

Piotrowski, A., Banakar, V., Scrivner, A., Elderfield, H., Galy, A., and Dennis, A.: Indian Ocean circulation and productivity during the last glacial cycle, Earth and Planetary Science Letters, 285, 179–189, 2009.

Prentice, I. C., Harrison, S. P. & Bartlein, P. J. Global vegetation and terrestrial carbon cycle changes after the last ice age. New Phytologist (2011) 189: 988–998 doi: 10.1111/j.1469-8137.2010.03620.x

Schmitt, J., Schneider, R., Elsig, J., Leuenberger, D., Lourantou, A., Chappellaz, J., Köhler, P., Joos, F., Stocker, T., Leuenberger, M., and Fischer, H.: Carbon Isotope Constraints on the Deglacial CO2 Rise from Ice Cores, Science, 336, 711–714, 2012.

Schneider, R., Schmitt, J., Kohler, P., Joos, F., and Fischer, H.: A reconstruction of atmospheric carbon dioxide and its stable carbon isotopic composition from the

penultimate glacial maximum to the last glacial inception, Climate of the Past, 9, 2507–2523, 2013.

Schubert, B. and Jahren, A.: The effect of atmospheric CO2 concentration on carbon isotope fractionation in C3 land plants, Geochimica et Cosmochimica Acta, 96, 29–43, 2012.

Siegenthaler, U. & F. Joos (1992) Use of a simple model for studying oceanic tracer distributions and the global carbon cycle, Tellus B: Chemical and Physical Meteorology, 44:3, 186-207, DOI: 10.3402/tellusb.v44i3.15441

Strassmann, K. M. and Joos, F.: The Bern Simple Climate Model (BernSCM) v1.0: an extensible and fully documented open-source re-implementation of the Bern reduced-form model for global carbon cycle–climate simulations, Geosci. Model Dev., 11, 1887–1908, https://doi.org/10.5194/gmd-11-1887-2018, 2018.

Tarnocai, C., Canadell, J., Schuur, E., Kuhry, P., Mazhitova, G., and Zimov, S.: Soil organic carbon pools in the northern circumpolar permafrost region, Global Biogeochemical Cycles, 23, GB2023, doi:10.1029/2008GB003 327, 2009.

Treat, C., Kleinen, T., Broothaerts, N., Dalton, A., Dommain, R., Douglas, T., Drexler, J., Finkelstein, S., Grosse, G., Hope, G., Hutchings, J., Jones, M., Kuhry, P., Lacourse, T., Lähteenoja, O., Loisel, J., Notebaert, B., Payne, R., Peteet, D., Sannel, A., Stelling, J., Strauss, J., Swindles, G., Talbot, J., Tarnocai, C., Verstraeten, G., C.J. Williams, Z. X., Yu, Z., Väliranta, M., Hättestrand, M., Alexanderson, H., and Brovkin, V.: Widespread global peatland establishment and persistence over the last 130,000 y, PNAS, 116, 4822–4827, 2019.

van der Sleen, P., P. Groenendijk, M. Vlam, N. P. R. Anten, A. Boom, F. Bongers, T. L. Pons, G. Terburg, and P. A. Zuidema (2015), 
[revised manuscript text omitted]

---

## Referee Report (RR1)

**Re-review on**

*Sequential changes in ocean circulation and biological export productivity during the last glacial cycle: a model-data study*

**by CM O'Neill et al**

submitted to *Climate of the Past*, article reference: cp-2019-146

**Date: August 20, 2020**

This is a re-review on a major revision of the paper. I have already been involved as reviewer in evaluating the original version.

In principle, I can agree with the rebuttal and the effort undertaken to revise the paper. However, I have to say that the rebuttal was focusing too much on lengthly arguments why things have (or have not) been done ending in 77 pages of responses. In this sense less is more. One can not expect to read that much arguments in detail. Over the length of arguments, however, the changes in the paper have been forgotten here and there. For example. Table S13 in the SI gives numbers for various parameters and processes, but the Table is to my knowledge not cited in the main text, and also never refered to its content (I did not check on all Tables and figure in the SI).

Before acceptance I ask the authors to correct my issues below, mainly minor, but 1–2 potentialla major (#2, #6).

1. Model description: The paragraph on C3, C4 and terrestrial biosphere (mainly on page 6) gives mainly arguments why something is NOT done. This is a discussion and should be put in section 5.3.

2. Model description: Nothing is said on weathering, volcanic outgassing and the corresponding 13C, the only details to that are found in the unreferred Table S13. Please include a paragraph here (and not only in the rebuttal), and check if all material in the SI is addressed at least once. Here also, some explanation is necessary for the weathering fluxes, since they come in units "mol/m$^3$/yr. Does this imply the weathering is put in the entire water mass, or only surface boxes (which?)? Also

extend the Table S12 on the references, on which the chosen parameter values are based on, or extend the table with footnotes, in which you explain your choice, if one reference is not possible (I believe those details have been in the rebuttal already). Air-sea fractionation factors: Are they fixed? Do you fractionate both fluxes air2sea and sea2air, as typically done (e.g. Mook, 1986)? The line "silicate weathering $CO_2$ flux $\delta^{13}C$" is not necessary, since this is obvious. Maybe add another column for the units and make the table wide enough, that no line-breaks in individual entries are necessary.

3. Page 10: Schneider et al (2013) gives 3 potential processes for the 0.4‰ change from PGM to LGM, not only the "likely cause by land C" mentioned here. These three causes are given later-on in section 3, page 15, but I believe they are better suited here.

4. page 11, line 6. You need to start a sentence with a word, not with "$\sim$".

5. page 15, lines 9ff: The 0.4‰ rise in $\delta^{13}CO_2$ is between PGM and LGM, not between last interglacial and Holocene.

6. Terrestrial biosphere: There is still something wrong here. In Fig 12 you show a decline in land C from 2200 Pgc (MIS 5e) to 1700 PgC (MIS 2). This release of 500 PgC leads to a RISE in $CO_2$ on a hundred-thousands years time-scale of about 25 ppm (airborne fraction for 100 kyr should be about 10%), see also Köhler et al. (2010) cited here. However, in Figure 14 it is suggestes that the contribution of terrestrial carbon is always more or less the same. I believe this is obtained by switching land C on/off for equilibrium runs, but never doing transient runs. I therfore believe, this is wrong, land C should be extracted from Fig 14. (also, what does "(RHS)" (added to the label of terrestrial biosphere in Fig 14) mean? Right-hand-side? of what? y-axes are the same left and right???). So, I am not sure what the correct answer from that model to the contribution of land C on $CO_2$ is, but Figure 12 should guide the solution. Maybe the runs were too long, leading more or less to similar oceanic C uptake of the C released from land? Anyhow, if a decent

answer can be found here it should be included in the list of processes changing $CO_2$ given on page 30, which discuss fig 14, even if land C is NOT contributing to the deglacial $CO_2$ rise, but make the $CO_2$ changes, that need to be explained, larger. If no decent answer comes up for the land C contribution (e.g. due to the setup with equilibrium runs), this should also be stated here.

7. section 5.2. You might add Köhler et al. (2010), which is already cited, to the list of references that claim that a number of processes are necessary to explain the LGM-Holocene $CO_2$ change (line 20), and also to those which claim a contribution of wind-borne iron-induced Southern Ocean productivity (line 30). If interested in more detail on both, they are found in previous papers (Köhler et al., 2005; Köhler and Fischer, 2006).

8. Data and code (section 7). What is found at https://doi.org/10.5281/zenodo.3559339 is a V2 from December 2019, suggesting that the final changes necessary for this revision here, have not yet been uploaded.

9. Throughout the draft: "Kohler", should be "Köhler"

10. Throughout the draft: "Francois et al, 1999", should be "François et al, 1999"

11. page 39, line 5: Second author of Kohfeld et al. 2005, is "Le Quéré, C.", not "Quéré, C.L."

12. page 36: Authors missing in "Arneth et al 2017".

13. Check reference list for details, e.g "$CO_2$", not CO2; not DOI and http, etc.

14. I believe it should be "$C_3$" and "$C_4$", not "C3" and "C4".

**References**

Köhler, P. and Fischer, H.: Simulating low frequency changes in atmospheric $CO_2$ during the last 740 000 years, Climate of the Past, 2, 57–78, doi:10.5194/cp-2-57-2006, 2006.

Köhler, P., Fischer, H., Munhoven, G., and Zeebe, R. E.: Quantitative interpretation of atmospheric carbon records over the last glacial termination, Global Biogeochemical Cycles, 19, GB4020, doi:10.1029/2004GB002345, 2005.

Mook, W. G.: $^{13}$C in atmospheric $CO_2$, Netherlands Journal of Sea Research, 20, 211–223, 1986.

---

## Referee Report (RR2)

**Sequential changes in ocean circulation and biological export productivity during the last glacial cycle: a model-data study.**

Pearse J. Buchanan, University of Liverpool.

First, I appreciate the enormous amount of work that the lead author has done to address the comments of the three prior reviewers, evident by the additions to the methods section and the response to reviewers. In undertaking this review, I have therefore chosen not to focus on minor issues, nor comment on the length, writing or quality of figures, but will concentrate on whether the interpretations/lessons from the study are valid.

The authors use a simple box model of the ocean with attached atmospheric and terrestrial components to understand what happened to ocean circulation and Southern Ocean biological carbon export over the full glacial cycle, from MIS-5e to the Holocene. As far as I can tell, the model takes SST, global mean salinity, sea volume, sea ice cover, carbonate reef production, global overturning circulation rate, Atlantic meridional overturning rate and biological carbon export as inputs. Values of SST, global mean salinity, sea volume, sea ice cover, carbonate reef production and biological carbon export outside of the Southern Ocean are fixed for each experiment at each MIS. The authors chose to vary the remaining three inputs, global overturning, Atlantic overturning, and biological carbon export in the Southern Ocean in thousands of combinations at each MIS. They then compared simulated atmospheric $p$CO$_2$, $\delta^{13}$C$_{CO2}$ and $\Delta^{14}$C$_{CO2}$, as well as oceanic $\delta^{13}$C, $\Delta^{14}$C and carbonate ion concentrations in deep and abyssal waters with the equivalent paleoproxy values, and fit a linear least-squares optimisation with using these 9,000 simulations to find the "optimal" values of global overturning, Atlantic overturning, and biological carbon export in the Southern Ocean at each MIS.

The authors then discuss the trends in atmospheric and oceanic proxy data and review the physical mechanisms that drove these trends, referencing prior work. This analysis lays the foundation for their quantitative work with the model.

Following a comprehensive introduction and description of their tools and palaeoproxy data, the authors make a brief description of their results, finding declines in both overturning rates (global and atlantic) at slightly different times during the glacial, and an increase in southern ocean carbon export at MISs 4 and 2, as the major changes needed to explain the proxy records. A large contribution of SST decline to CO$_2$ drawdown was also found. Other processes (salinity, ocean volume, reef calcification/dissolution, terrestrial carbon store) were of minor importance.

The approach is very interesting and insightful. Although it is not a truly transient simulation, it is a welcome addition to the field and deserves publication. It also is not hugely controversial, as the results echo other studies calling for a decline in overturning rate and increased southern ocean productivity during the glacial, something that the authors recognise and discuss.

However, one finding that is particularly interesting is how the slowdown of the GOC at MIS 5e-5d explains the first drop in CO$_2$ while showing little change in $\delta^{13}$C. As far as I am aware, this is an important finding that should be shared with the community so that more complex models can be used to further test this, as the authors allude to in their discussion. I have long wondered at the absence of change in $\delta^{13}$C at the transition between MIS 5e-5d and thought that this must be explained because surely such a drop in CO$_2$ must involve changes in ocean circulation.

**Overall, I strongly advocate for publication with minor revisions/clarifications. I disagree with reviewer 1's request to remove the $\delta^{13}C$ from the paper. Sure, the paper like any other has its shortcomings and limitations, but the results are in my opinion worth publishing. It would be a shame to bury them. Moreover, the substantial work done in the response to reviewer 1, particularly in regard to their concerns about MIS-averaging and the treatment of carbonate chemistry, clearly shows the legitimacy of the model, their approach and the findings.**

I ask for the following revisions/clarifications:

1. It should be more clearly stated how the authors calculate their standard deviations in their optimisation approach. I would also appreciate a figure/table/paragraph that ranks the most important variables in the optimisation procedure per box or per MIS, whichever makes the most sense and doesn't add too much writing. Clearly, atmospheric $CO_2$ and SST, with small standard deviations, will likely be strong contributors to the optimisation, while $CO_3$ with a standard deviation of 15 uM and changes of < 30 uM must be small. But some more clarity on how the different variables contributed would be good.

2. I needed to read O'Neill et al (2019) GMD to find out that the SCP-M does not hold biological production (Z) constant in non-Southern Ocean boxes. What happens to these boxes in the simulations? Even after reviewer 2 asked for this, I still think that this should be made much clearer. Is it, for instance, dependent on phosphate?
Logically, extra-SO carbon export should decline as the GOC declines because less nutrients would be supplied to surface waters in the lower latitudes. Surely, if you simulate phosphate concentrations explicitly then the carbon pump should respond? Is this so?
I need more information on what is happening. The reason I need more information is because I would expect the slowdown in GOC to slow down carbon export. If this does not happen, then this could be the reason why purely physical mechanisms are responsible for a drawdown of 70 ppm, and biological effects are less important. If the biological pump were allowed to respond to the global slowdown in overturning, then the physical contribution would be weaker than presented because more carbon would escape via the tropical upwelling as your so rightly state with the reference to the Takahashi paper. However, if a more effective carbon pump was to develop in the low latitudes, for instance by increases in C:P ratios (Matsumoto et al., 2020) and/or via $N_2$ fixation (Buchanan et al., 2019), this would ensure that the influx of carbon into the ocean via physical mechanisms would be prevented from outgassing by a tighter biological lid. The two must ultimately work together.

3. As far as I can tell, the authors simulate an increase in Southern Ocean production during MIS 2 and MIS 4 without applying the increase in dust deposition that is seen during these periods. It is therefore noteworthy, and should be emphasised further, that the increase in Southern Ocean production at MIS2 and MIS4 emerged independently without needing to provide the dust record to the model, and yet aligns with it. I think that this is a striking finding that needs more emphasis. Although, it is important to say that the model does not (or does it?) alter carbon export outside of the Southern Ocean, which may also have been important for "tightening the lid" on the ocean carbon store (as I've discussed above).

4. I don't follow the logic of paragraph 4 in the Discussion. First talks about the model results compared with the analysis of Kohfeld & Chase (2017), then diverts to Stephen & Keeling (2000) Antarctic sea ice changes. I suggest making the narrative of your discussion clearer here.

5.  It is important to mention that the contribution of SST is very likely overestimated, given your use of box model rather than the general circulation model. Box model atmospheric $CO_2$ is known to be more sensitive to the SST changes in the higher latitudes compared with general circulation models (Archer et al., 2000). This should be stated clearly in the discussion section as a caveat. I want to know how the results might change if the exercise were repeated with a GCM, as this may inspire others.
6.  Page 5, line 8 – Don't you mean 40S-60N?

References

Archer, D. E., Eshel, G., Winguth, A., Broecker, W., Pierrehumbert, R., Tobis, M., et al. (2000). Atmospheric pCO 2sensitivity to the biological pump in the ocean. *Global Biogeochem. Cycles* 14, 1219.

Buchanan, P. J., Chase, Z., Matear, R. J., Phipps, S. J., and Bindoff, N. L. (2019). Marine nitrogen fixers mediate a low latitude pathway for atmospheric CO2 drawdown. *Nat. Commun.* 10, 1–10. doi:10.1038/s41467-019-12549-z.

Matsumoto, K., Tanioka, T., and Rickaby, R. (2020). Linkages Between Dynamic Phytoplankton C:N:P and the Ocean Carbon Cycle Under Climate Change. *Oceanography* 33. doi:10.5670/oceanog.2020.203.

---

## Author Response (AR2)

**Thanks very much for the Editor and Reviewer comments below. We have worked hard to address the points and have thereby further improved the work.**

**Editor's comments**

1) In general, I think the text could be tightened as there are a lot of repetitions throughout the different sections (e.g. P33, L9-10 is similar to P5, L. 28-31).
I would also reiterate Reviewer 1 comment to tighten Section 2 by more directly describing what you have done.

**The earlier paragraph has been removed and the sentence is left in the discussion. Discussion material has been moved from the Methods section to the Discussion section.**

2) Data analysis section: I am not sure what the goal of this section is, and I think it needs to be amended. I think it provides an incomplete review of all the changes in the carbon cycle occurring during the last glacial-interglacial cycle. This data is used to constrain the model, so has to be introduced in section 2, however the processes leading to changes in these variables across the last G-IG cycle should be compared to your results in the discussion.

**The aim of this section is to describe the patterns in the data that tell the stories captured in the model-data results. This is done to frame the data in terms of the changes between the MIS as we have averaged the data into MIS. This section also establishes the proxy data patterns, in terms of the MIS averages, that the model-data experiments seek to explain. We simply refer to the major changes in atmospheric $CO_2$ in our MIS-averaged data, then look for changes in the other proxies for the same events.**

**We have added a paragraph at the start of Section 2 to set out the objectives more clearly. We have also added more references for the reader to consult if they seek more detail on the G-IG proxy data.**

3) Please make sure to define the acronyms (GOC, AMOC, NADW, AABW…) in their first appearance and then use them throughout. Please be consistent in the way you refer to figures, and remove the parentheses around the panels of figures in the text: i.e. change Fig.X(A) into Fig.XA throughout.

**Thanks, NADW references and Figure references amended throughout**

4) Supplementary material: Currently only Table S8 is expressly called in the manuscript. Please make sure the supplementary material includes only appropriate material and that each table and figure are called in the main text when necessary.

**Thanks, we have added references to the SI tables and figures throughout the text.**

P 5, L. 10: remove the comma after "study" **Done**
P 5, L. 13: remove the commas before and after "at 100m depth" **Done**

P5, L. 28: Please remove the reference to Menviel et al. 2012 here, as the land-atmosphere $CO_2$ fluxes were imposed in that study. **Done**

P7, L. 15: Please move this information to the code availability section. **Done**

P8, L. 13-14: I don't think this sentence is necessary. **Deleted**

P8, L. 31: Please remove "serve to" **Deleted**

P12, L. 20: Please add concentration to "carbonate ion" and define as [$CO_3^{2-}$] **Done**

Table 2: Change last row to: [$CO_3^{2-}$] as deduced from B/Ca **Done**

P15, L. 7: Please replace "littered" by a more appropriate verb. **Changed to "occurred"**

P18, L. 2: It seems you are showing ocean – atmospheric $D^{14}C$, no? **Yes, reworded as such**

P19, L. 9: "its" **Corrected**

P20, L. 3: It might be good to add the depth that you are referring to here, and also add a reminder about the fact that there is no distinction between north and south in the abyssal Atlantic (i.e. 1 box). **Added**

P20, L. 4: "its" **Corrected**

P21, L. 1: "shows" **Corrected**

P23, L. 3: How can you be so sure this is just the result of $CO_2$ fertilization? As $CO_2$ decreases so do temperature and precipitation. **This sentence is amended to say it is the case in our model that carbon fertilisation drives NPP and terrestrial carbon stock but that temperature and precipitation also impact NPP. NB we are confident our model captures sufficient changes in NPP and terrestrial carbon stock during the G-IG cycle because of the calibration of the "Beta" parameter for carbon fertilisation when we built and documented the model, and compared its output with others. Therefore, our changes in the NPP and terrestrial carbon stock fall in the range of literature estimates from other models.**

P25, L. 4,5: "Last Interglacial" **Corrected**

P26, L. 10-12: I don't see the link between this sentence and the previous ones. **Sentence removed**

P26, L. 35: remove "and" before "rich" **Done**

P26, L. 35: Add "enhanced" in front of "biological" **Done**

P27, L. 2-5: I am not sure this sentence is really correct. I would suggest to revise and correct the reference issue. **This sentence amended and the reference issue fixed**

P27, L.8: Please be more specific here, and add processes and direction of changes instead of the repeated "other changes". **This sentence changed to show we are talking about the model forcings and the reference to the Figure they are shown in.**

P28, L. 16-17: Please remove "observed during the last glacial termination" as this was not studied in Menviel et al., (2015), which was a broad sensitivity study (whereas the last glacial termination was studied in particular in Menviel et al. (2018)). You can reformulate the sentence as follows "in the context of past changes in atmospheric $CO_2$ and $d^{13}CO_2$." **Done**

P28, L.20-21: What do you mean here by "but also quite a variation across …"? **Happy to remove that part of the sentence as it only a minor point in Menviel et al (2015) that effects of NADW on deep ocean $d^{13}C$ might vary between models, whereas we are looking at GOC.**

P30, L. 26: "Last Interglacial" **Done**

P 30, L. 30: Replace "cooler SST" by "lower SST" **Done**

P30, L. 33-34: What do you mean here by polar sea-ice? Only changes in Antarctic sea-ice cover are included in Figure 14, and from Figure 14, an Antarctic sea-ice increase always

seems to lead to a small (2ppm?) pCO2 decrease. **The sentence corrected to show sea-ice decreases CO2 and the text corrected for Antarctic**

P 34, L. 13: Please replace "d13C negative CO2" by "13C depleted CO2" **Done**

P 34, L. 16-17: The grammatical structure of that sentence is odd. Please rephrase. **Modified to "Our results agree with hypotheses for glacial-interglacial cycle CO$_2$ that include varying ocean circulation, marine biological export productivity and other physical and biogeochemical changes in the marine and terrestrial carbon cycle".**

Figure 14: Please remove "proxy" after "Antarctic sea-ice" **Done**

**Anonymous reviewer 1**

1. Model description: The paragraph on C3, C4 and terrestrial biosphere (mainly on page 6) gives mainly arguments why something is NOT done. This is a discussion and should be put in section 5.3.

**This is moved to the Discussion section**

2. Model description: Nothing is said on weathering, volcanic outgassing and the corresponding 13C, the only details to that are found in the unreferred Table S13. Please include a paragraph here (and not only in the rebuttal), and check if all material in the SI is addressed at least once.

**We have added the following to the description of the model:**

**SCP-M contains a simple parameterisation of the terrestrial carbon cycle. For continental rock weathering, we apply the simple scheme of Walker and Kasting (1992) as implemented in Toggweiler (2008). The scheme supplies DIC and alkalinity from carbonate and silicate rock weathering to the low latitude surface ocean boxes (boxes 1 and 8 in Figure 1) in units of mol m$^{-3}$ yr$^{-1}$. The parameter values used are shown in Table S13. For the model's weathering equations please see O'Neill et al (2019). $\delta^{13}$C fluxes for carbonate and silicate weathering are shown in Table S13. A volcanic flux of carbon (and $\delta^{13}$C) is also assumed, following the method to set the rate of volcanic CO2 outgassing roughly to the rate of silicate rock weathering (Walker and Kasting, 1992; Toggweiler, 2008; Zeebe, 2012). Parameters are shown in Table S13.**

**We have added references to the SI tables and figures throughout the text.**

Here also, some explanation is necessary for the weathering fluxes, since they come in units "mol/m$^3$/yr. Does this imply the weathering is put in the entire water mass, or only surface boxes (which?)?

**Clarified in the new text as above.**

Also extend the Table S13 on the references, on which the chosen parameter values are based on, or extend the table with footnotes, in which you explain your choice, if one reference is not possible (I believe those details have been in the rebuttal already).

**References added to Table S13**

Air-sea fractionation factors: Are they fixed? Do you fractionate both fluxes air2sea and sea2air, as typically done (e.g. Mook, 1986)?

**Clarified in the new text in SI Table S13. Yes, the air-sea and sea-air fluxes fractionate $\delta^{13}$C in our model.**

The line "silicate weathering $CO_2$ flux $\delta^{13}$C" is not necessary, since this is obvious.

**Removed**

Maybe add another column for the units and make the table wide enough, that no line-breaks in individual entries are necessary.

**Table amended as such**

3. Page 10: Schneider et al (2013) gives 3 potential processes for the 0.4 change from PGM to LGM, not only the "likely cause by land C" mentioned here. These three causes are given later-on in section 3, page 15, but I believe they are better suited here.

**We prefer to amend this sentence to "possible cause" for the terrestrial biosphere and leave the reference in Section 3 to describe the three possible causes from Schneider et al (2013).**

4. page 11, line 6. You need to start a sentence with a word, not with "~".

**Changed to "More than".**

5. page 15, lines 9ff: The 0.4 rise in $\delta^{13}CO_2$ is between PGM and LGM, not between last interglacial and Holocene.

**If we consult the source of the atmospheric $\delta^{13}$C data, Eggleston et al (2016) we find the following (quote):**

**"As mentioned above, the last interglacial (around 120 kyr B.P., also known as MIS 5e) was characterized by about 0.4‰ lower $\delta^{13}$C(atm) values than the Holocene, an offset also seen when comparing the PGM and LGM"**

**and:**

**"Most importantly, the penultimate glacial maximum (PGM) was found to be 0.4‰ isotopically lighter in $\delta^{13}$C(atm) than the Last Glacial Maximum (LGM), and the penultimate warm period (marine isotope stage (MIS) 5e) was also more negative in $\delta^{13}$C(atm) by a similar amount"**

**Therefore, we have amended the text to say that it was both the last interglacial/Holocene and PGM/LGM:**

**"Atmospheric $\delta^{13}$C (Fig. 4B) was ~0.4‰ higher in the Holocene (MIS 1) and LGM (MIS 2) periods than in the last 10 interglacial (MIS 5e) and penultimate glacial periods (MIS 6, not shown in Fig. 4B)"**

6. Terrestrial biosphere: There is still something wrong here. In Fig 12 you show a decline in land C from 2200 Pgc (MIS 5e) to 1700 PgC (MIS 2). This release of 500 PgC leads to a RISE in $CO_2$ on a hundred-thousands years time-scale of about 25 ppm (airborne fraction for 100 kyr should be about 10%), see also Kohler et al. (2010) cited here. However, in Figure 14 it is suggestes that the contribution of terrestrial carbon is always more or less the same. I believe this is obtained by switching land C on/off for equilibrium runs, but never doing transient runs. I therfore believe, this is wrong, land C should be extracted from Fig 14. (also, what does "(RHS)" (added to the label of terrestrial biosphere in Fig 14) mean? Right- hand-side? of what? y-axes are the same left and right???). So, I am not sure what the correct answer from that model to the contribution of land C on $CO_2$ is, but Figure 12 should guide the solution.

Maybe the runs were too long, leading more or less to similar oceanic C uptake of the C released from land? Anyhow, if a decent answer can be found here it should be included in the list of processes changing $CO_2$ given on page 30, which discuss fig 14, even if land C is NOT contributing to the deglacial $CO_2$ rise, but make the $CO_2$ changes, that need to be explained, larger. If no decent answer comes up for the land C contribution (e.g. due to the setup with equilibrium runs), this should also be stated here.

**We have re-worked the Figure 14 and show the effects of the terrestrial biosphere and the other mechanisms with the net effect on atmospheric CO₂ also shown. This analysis shows a peak effect of the reduced terrestrial biosphere of +13 ppm in MIS 2. We are comfortable with the pattern of atmospheric CO₂ and the magnitude of the impact when we compare this with the references below.**

**Amended Fig. 14.**

[Figure]

**If we consult Kohler et al (2010) we find the following:**

**"The release of about 500 PgC of $^{13}$C-depleted terrestrial carbon (with d$^{13}$C between −20‰ and −25‰) results in a drop in d$^{13}$CO$_2$ by 0.44‰ in the LGM and a rise in pCO$_2$ by only 12 uatm."**

**Table 3.** Summary of the Contribution of Different Processes to the Simulated Changes in Atmospheric $p$CO$_2$ and $\delta^{13}$CO$_2$ and Deep Pacific $\delta^{13}$C$_{DIC}$ and Mean Ocean $\delta^{13}$C$_{DIC}$ During Termination I[a]

|  | Atmosphere $\Delta(\delta^{13}$CO$_2)$ (‰) | Atmosphere $\Delta(p$CO$_2)$ ($\mu$atm) | Deep Pacific $\Delta(\delta^{13}$C$_{DIC})$ (‰) | Mean Ocean $\Delta(\delta^{13}$C$_{DIC})$ (‰) |
|---|---|---|---|---|
| Process |  |  |  |  |
| Lower ocean temperatures | −0.49 | −30 | −0.08 | −0.07 |
| Smaller terrestrial carbon storage | −0.44 | +12 | −0.44 | −0.44 |
| Lower sea level | +0.06 | +14 | −0.01 | +0.02 |
| Weaker NADW formation | +0.10 | −24 | −0.08 | −0.08 |
| Enhanced marine export production | +0.19 | −28 | −0.16 | −0.10 |
| Larger sea ice cover (gas exchange) | +0.18 | +12 | +0.03 | +0.03 |
| Higher Southern Ocean stratification | +0.31 | −46 | −0.26 | −0.16 |
| Summed-up changes | −0.09 | −90 | −1.00 | −0.80 |
| Simulated changes in CTRL | +0.00 | −85 | −0.47 | −0.43 |
| Nonlinearities[b] | −0.09 | +5 | +0.53 | +0.40 |

[a]The contributions are calculated by subtracting results with the process in question switch off (all but one) from the control run.
[b]Nonlinearity values are sum minus CTRL.

**Furthermore, in Eggleston et al (2016):**

[Figure]

**Figure 4.** Effects of various processes on [CO$_2$] and $\delta^{13}$C(atm); the changes of the forcing parameters with respect to modeled LGM values are indicated. For example, SST (+4K) represents an increase in average global sea surface temperature of 4 K from the LGM to preindustrial era with a response of +0.5‰ and +30 ppm in $\delta^{13}$C(atm) and [CO$_2$], respectively, as indicated by the respective black dots [*Köhler et al.*, 2010]. The sea ice coverage was reduced by 35% and 25% in the Northern and Southern Hemispheres, respectively. NADW and SO upwelling strength increased by 60% and 200%, respectively. Note that we are approximating the response of $\delta^{13}$C(atm) and [CO$_2$] to each of these processes as linear.

**Menviel et al (2012) showed a contribution of +11 ppm in the period 125-20 ka and -17 ppm in the period 20-0 ka as per the extract below.**

**Table 2**

Attribution of $pCO_2$ (ppmv), $\delta^{13}CO_2$ (permil) and $\delta^{13}C_{DIC}$ (permil) changes to processes for the period 125−20 ka B.P. and 20−0 ka B.P. $\delta^{13}C_{DIC}$ reflects the whole ocean change in $\delta^{13}C$ of DIC. As the experiments are performed with a sediment module, the processes described below include the $pCO_2$ response to sediment interactions. Physical processes denote all the processes included in experiment OC (T, S, ocean circulation and export production).

| Processes | 125−20 ka B.P. | | | 20−0 ka B.P. | | |
|---|---|---|---|---|---|---|
| | $\Delta CO_2$ | $\Delta(\delta^{13}CO_2)$ | $\Delta(\delta^{13}C_{DIC})$ | $\Delta CO_2$ | $\Delta(\delta^{13}CO_2)$ | $\Delta(\delta^{13}C_{DIC})$ |
| S1$_{(Rem,P)}$ | −112 | +0.078 | −0.12 | +44 | −0.04 | +0.12 |
| S2$_{(Rem,Br)}$ | −72 | −0.39 | −0.42 | +50 | +0.12 | +0.16 |
| S3$_{(Br,P)}$ | −102 | +0.033 | −0.24 | +26 | +0.02 | +0.18 |
| Remineralization rate (REM−FE) | −31 | −0.04 | −0.02 | +21 | −0.054 | −0.1 |
| Brine param. (BR−FE) | −12 | −0.1 | −0.1 | +2 | 0 | −0.02 |
| P inventory (PO−FE) | −50 | +0.4 | +0.2 | +5 | −0.2 | −0.08 |
| Terrestrial carbon (VG−FE) | +11 | −0.1 | −0.08 | −17 | +0.2 | +0.18 |
| Fe fertilization (FE−OC) | −10 | +0.12 | −0.002 | +10 | −0.14 | −0.014 |
| Physical processes (OC) | −31 | −0.22 | −0.24 | +20 | +0.18 | +0.16 |
| Shallow-water CaCO$_3$ deposition | − | − | − | +12 | +0.026 | +0.013 |

**Furthermore, Ganpoloski and Brovkin (2017) performed a factorial experiment by running their model through the last G-IG cycle with and without the terrestrial biosphere. Ganpoloski and Brovkin (2017) found that the terrestrial biosphere impacted the G-IG atmospheric CO2 by 10-15 ppm, as shown in the extract below. Their time profiling of the terrestrial biosphere through the last G-IG cycle (shown below) looks comparable to ours.**

[Figure]

**Figure 9.** Results of factor separation analysis. **(a)** Simulated $CO_2$ (ppm) in one-way coupled ONE_1.1 experiment (purple line) and reconstructed $CO_2$ concentrations (black dashed line, Lüthi et al., 2008). **(b)–(d)** Contributions to simulated atmospheric $CO_2$ (ppm) of terrestrial carbon cycle **(b)**, ONE_S4–ONE_S3; iron fertilisation **(c)**, ONE_S3–ONE_S2; variable volcanic outgassing **(d)**, ONE_S2–ONE_S1; temperature-dependent remineralisation depth **(e)**, ONE_S1–ONE_1.1.

7. section 5.2. You might add Kohler et al. (2010), which is already cited, to the list of references that claim that a number of processes are necessary to explain the LGM-Holocene $CO_2$ change (line 20), and also to those which claim a contribution of windborne iron-induced Southern Ocean productivity (line 30). If interested in more detail on both, they are found in previous papers (Köhler et al., 2005; Köhler and Fischer, 2006).

**Thanks, reference added**

8. Data and code (section 7). What is found at https://doi.org/10.5281/zenodo.3559339 is a V2 from December 2019, suggesting that the final changes necessary for this revision here, have not yet been uploaded.

**The modified model files, model and parameters, are appended at the following repository uploaded on 13/10/20:**

https://doi.org/10.5281/zenodo.4084586

**Once there are no further iterations, we will create a dedicated new version 3.0 model DOI and upload all the model files, data and model-data results for the paper.**

9. Throughout the draft: "Kohler", should be "Kohler"

**Thanks, amended**

10. Throughout the draft: "Francois et al, 1999", should be "Francois et al, 1999"

**Thanks, amended**

11. page 39, line 5: Second author of Kohfeld et al. 2005, is "Le Quere, C.", not "Quere, C.L."

**Amended**

12. page 36: Authors missing in "Arneth et al 2017".

**Amended**

13. Check reference list for details, e.g "$CO_2$", not CO2; not DOI and http, etc.

**Thanks, we will address this in the final edit and production process.**

14. I believe it should be "$C_3$" and "$C_4$", not "C3" and "C4".

**Thanks, amended throughout text**

**Reviewer #4**

1. It should be more clearly stated how the authors calculate their standard deviations in their optimisation approach. I would also appreciate a figure/table/paragraph that ranks the most important variables in the optimisation procedure per box or per MIS, whichever makes the most sense and doesn't add too much writing. Clearly, atmospheric $CO_2$ and SST, with small standard deviations, will likely be strong contributors to the optimisation, while $CO_3$ with a standard deviation of 15 uM and changes of < 30 uM must be small. But some more clarity on how the different variables contributed would be good.

We use the simple standard deviation of data points average for each box and MIS. The proxy data for each ocean box is binned into model boxes on depth, latitude and longitude assigns the data to either Atlantic or Pacific-Indian basins. These are then binned into MIS age groups and the sample population is then averaged and the standard deviation is calculated. The SD is then used as a weighting in the optimisation procedure which 1) reduces the weighting assigned to data points with higher SD (more variability within each box or within each MIS period) and 2) normalises data points with different units (e.g. ppm, umol/kg and per mil) into units of mean/SD. Some text with the above is added at the start of the data methodology Section 2.

Re contribution

The contribution that each variable makes to the optimisation is mainly determined by its standard deviation (SD) in the model box and MIS. High SD de-weights the variable, a lower SD increases the weighting. It is important to note that generally the SDs for the atmospheric $CO_2$, $\delta^{13}C$ and D14C are smaller relative to the mean than the ocean boxes. This is due to the relatively lower number of sampling locations for the atmosphere data. The ocean boxes, and particularly in a box model, average data for large parts of the ocean and therefore there is more variability in the ocean box sample.

The optimisation solves for the mean values of each variable in each box per MIS, not the changes from a base value. It does this by minimising the difference between the model result for a box and the mean value of the data observation for each box ("the residual"), weighted by the standard deviation. The size of the SD relative to the mean therefore determines the contribution of each data observation to the optimisation result.

Therefore, it is the relative standard deviation that approximates the contribution of each variable to the optimised outcome. Relative standard deviation is calculated by dividing the standard deviations by the means, using the data shown in Tables S3-S6. The ranking of the RSD's is from smallest to largest. The smaller the RSD, the smaller the standard

deviation is relative to the mean, and the larger the proportional weighting of that variable in the optimisation.

Please note that while undertaking this analysis we adjusted the default SD for ocean $CO_3^{2-}$ data in the optimisation from 15 umo/kg to 20 umol/kg to de-weight this variable a bit relative to the ocean $\delta^{13}C$ data. This is updated in the manuscript, and the model results text, tables and figures are also updated throughout.

The table ranking the RSDs is shown in the document in the discussion section 5.1, with the following text.

Table 3 shows contribution analysis for the data observations in each MIS model-data optimisation. The ranking is based on the relative standard deviation for each data observation (or set of data observations) in each MIS. The contribution analysis shows that atmospheric $\delta^{13}C$ and $CO_2$ exert the greatest influence on the optimisation results throughout the MIS experiments. This reflects that each of the atmospheric time series is derived from a single source of data and does not require locational averaging as in the ocean boxes. For the atmosphere data, only MIS-averaging takes place. For the ocean boxes, averaging on depth and latitude takes place as well as MIS-averaging, to derive a box/MIS mean data value. Using a box model with large boxes such as SCP-M, means that large parts of the ocean are averaged into the box mean value and therefore there is an increased spread of data values around the mean. The model-data results show a precise fit to the atmospheric $CO_2$ and $\delta^{13}C$ data as shown in Figs 9-11. However, the results for oceanic variables also fall within the standard deviations of the data observations for each box and MIS (Figs 9-11). Some other studies have attempted model-data studies focusing only on the ocean data without matching atmospheric data. While these studies could elucidate more detail on oceanic processes, they are also potentially fraught due to the high spread of data values for the oceanic data and could return results that are not consistent with the well constrained glacial-interglacial atmosphere data. For our study, the express purpose is to identify causes of changes in atmospheric $CO_2$, so it is appropriate that atmospheric data observations contribute most to the model results. However, as shown in Figs 9-11 this is not at the expense of providing plausible results for the ocean variables.

| MIS / variable | AtCO2 | Atd13C | Atd14C | Oc d13C | Oc CO23 | Oc D14C |
|---|---|---|---|---|---|---|
| MIS5e | 1 | 2 | nan | 4 | 3 | nan |
| MIS5d | 2 | 1 | nan | 4 | 3 | nan |
| MIS5c | 2 | 1 | nan | 4 | 3 | nan |
| MIS5b | 2 | 1 | nan | 4 | 3 | nan |
| MIS5a | 2 | 1 | nan | 4 | 3 | nan |
| MIS4 | 2 | 1 | nan | 4 | 3 | nan |
| MIS3 | 2 | 1 | 3 | 5 | 4 | 6 |
| MIS2 | 2 | 1 | 3 | 6 | 4 | 5 |
| MIS1 | 2 | 1 | 4 | 5 | 3 | 6 |

2. I needed to read O'Neill et al (2019) GMD to find out that the SCP-M does not hold biological production (Z) constant in non-Southern Ocean boxes. What happens to these boxes in the simulations? Even after reviewer 2 asked for this, I still think that this should be made much clearer. Is it, for instance, dependent on phosphate?
   Logically, extra-SO carbon export should decline as the GOC declines because less nutrients would be supplied to surface waters in the lower latitudes. Surely, if you simulate phosphate concentrations explicitly then the carbon pump should respond? Is this so? I need more information on what is happening. The reason I need more information is because I would expect the slowdown in GOC to slow down carbon export. If this does not happen, then this could be the reason why purely physical mechanisms are responsible for a drawdown of 70 ppm, and biological effects are less important. If the biological pump were allowed to respond to the global slowdown in overturning, then the physical contribution would be weaker than presented because more carbon would escape via the tropical upwelling as your so rightly state with the reference to the Takahashi paper. However, if a more effective carbon pump was to develop in the low latitudes, for instance by increases in C:P ratios (Matsumoto et al., 2020) and/or via N$_2$ fixation (Buchanan et al., 2019), this would ensure that the influx of carbon into the ocean via physical mechanisms would be prevented from outgassing by a tighter biological lid. The two must ultimately work together.

**Thanks for raising an interesting and important discussion.**

**First, with regards to O'Neill et al (2019) in GMD. In that paper, we applied the first version of our simple carbon cycle box model (seven ocean boxes plus atmosphere) in a model-data study of the LGM and Holocene. In that study, we optimised the values for GOC, AMOC, deep-abyssal mixing and global ocean biological export productivity.**

**In that model, the biological export productivity (BEP) at 100m depth in each box is set with reference to a global base value. Each surface box was calibrated as a fixed % of the global base productivity value in model spinup. In the model-data experiment, the global**

base value of biological export productivity at 100m depth was solved for in the model-data optimisation. The study found that GOC and AMOC were weaker and that global ocean biological export productivity was unchanged at the LGM. With regards to biology, that could be interpreted to show that at a global level, positive and negative changes in BEP offset each other (e.g. stronger in SO, weaker in low latitudes). However, we did not explore that in the paper.

We have had various incarnations of the BEP function in the model during its development, including setting BEP as a time function of the ocean concentration of phosphate and then calculating BEP for other nutrients by applying Redfield ratios. Given we want to solve for the value of BEP in the model data optimisation, our approach was to apply the Martin function for BEP where there is a depth decay function for export, from a productivity assumption at 100m. This productivity assumption at 100m is our parameter Z to be solved in the model-data experiments.

The second paper differs from the first in that we focussed on Southern Ocean BEP in our G-IG model-data experiments. We solved for values for SO BEP in each MIS, but we left the values in the other surface boxes unchanged.

For us, it is quite uncertain whether biological export productivity was weaker or stronger outside the Southern Ocean in the LGM and prior glacial periods. As you point out, there are arguments and evidence both for weaker or stronger productivity.

As you point out, a logical argument is that if GOC and AMOC slow, then the BEP would weaken due to weakened supply of nutrients into the intermediate and surface ocean. You make the point that if we didn't factor in a weaker biological pump, that our model would provide an exaggerated "lid" on carbon outgassing in the low latitudes.

However, as you point out, there are arguments that global BEP including at the lower latitudes might have increased at the LGM.

Broecker (1981, 1982) first proposed that variations in the carbon-to-phosphorous ratio, or the amount of phosphorous supplied to the ocean from shelf sediments (supplying increased biological productivity), could increase low-latitude biological export productivity and drive glacial $CO_2$ lower, alongside lower sea-level, exposed sea shelf dissolution and associated alkalinity fluxes into the ocean.

Filippelli et al. (2007) found evidence for increased glacial ocean phosphate inventory, from the combination of terrestrial weathering, and reduced shelf-deposition of phosphate due to lower sea level. Filippelli et al. (2007) also found proxy evidence for increased biological productivity near the end of glaciations, carrying over into the early part of interglacial periods. Tamburini and Föllmi (2009) analysed ocean sedimentary

proxies and calculated a 17-40% increase in the ocean phosphate inventory in glacial periods. Tamburini and Föllmi (2009) posited the transfer of phosphate from shelf deposits to the deep ocean, caused by lower sea level, as the driver. A number of modelling studies quantified a 30-60 ppm biological drawdown of atmospheric CO2 (Heinze et al., 1991; Sigman et al., 1998; Tschumi et al., 2011). Menviel et al. (2012) modelled the last G-IG cycle with a medium complexity model and found that a 10% increase in the ocean phosphate inventory could have led to a 5 ppm reduction in atmospheric $CO_2$ during the last glaciation. However, due to the relatively long residence time of phosphate in the ocean, the positive impact on atmospheric $CO_2$ of increased sea level, and reduction of phosphate in the ocean, is only 5 ppm over the shorter deglacial and interglacial period (0-20 ka) (Menviel et al., 2012). This suggests that shelf phosphate supply could account for the last glacial CO2 drawdown, but not the deglacial CO2 increase. Menviel et al. (2012) found increased biological export productivity from ocean phosphate inventory as possible glacial CO2 hypothesis, and in good agreement with atmospheric $\delta^{13}C$ data.

According to Kohfeld and Ridgwell (2009), the biologically-driven G-IG hypotheses have suffered due to a lack of direct proxy evidence preserved in the geological record. A traditional source of paleo proxy data are cores of marine sediments, which preserve biological and geological matter which landed on the seafloor, such as deceased organisms, shells, coral, muds, sands, and rocks. It would be expected that increased biological productivity during the LGM may have shown up in marine sediments and cores. However, according to some proxy data, marine biological sedimentation rates were lower during glacial periods (Bradtmiller et al., 2007; Anderson et al., 2009). This either reflects lower productivity/export or lower preservation/higher dissolution rates. Sigman and Boyle (2000) concluded that export of biogenic carbon was lower during glacial periods, citing sedimentary core data.

Matsumoto (2007) found that a cooling global ocean in glacial periods could lead to deepening of organic remineralisation and a drop in $CO_2$ of 30 ppm. The bacteria which break down dead, sinking organic matter and return their elements to the ocean in dissolved form, are believed to be more sluggish in colder water, allowing deeper penetration of sinking particulate organic carbon, and more ocean sequestration of carbon. Menviel et al. (2012) modelled relatively large $CO_2$ effects (-27 ppm) from increasing the depth of remineralisation of particulate organic matter, which stores more carbon in the ocean interior, in the last glacial lead up, and that this feature was quickly reversible, but to a lesser extent (+21 ppm) at the last deglaciation. While a plausible theory, according to Kohfeld and Ridgwell (2009), there is limited geological evidence for such changes over G-IG cycles due to the difficulty of reconstructing the past biological remineralisation depth with geochemical proxies.

Finally, variations in dissolved silica supply to the surface ocean can influence the carbon composition of marine organisms (Harrison, 2000). Increased glacial silica supply from continental dust could steer the biological species mix towards diatoms, away from coccoliths, which make opal shells (diatoms) rather than calcium carbonate (coccoliths), a decreased "rain ratio" (Archer and Maier-Reimer, 1994). The silicate hypothesis posits that increased continental silicate dust supply during glacial periods led to greater $CO_2$ drawdown (Harrison, 2000). Based on the slow cycling time of silica in the ocean of ~23 kyr, and it's predominant source to the surface ocean by marine upwelling and not continental dust, the silicate hypothesis has been assigned only a small possible role in G-IG CO2 (Ridgwell et al., 2002; Kohfeld and Ridgwell, 2009).

Other studies suggest that equatorial Pacific BEP was weaker in the glacial periods (Calvo et al, 2011; Hayes et al, 2011; Winckler:2016aa), citing increases in upwelling driven BEP increase at the last glacial termination, not during the LGM.

Given the lack of consensus around the global ocean productivity outside of the Southern Ocean, we are comfortable with our assumption to leave the extra-SO BEP parameters unchanged in our MIS model-data experiments. However, we have added the following text to flag an additional caveat around this:

"It is important to note our model-data experiments assume unchanged biological export productivity in surface boxes outside of the Atlantic and Pacific-Indian subpolar Southern Ocean boxes across the last glacial-interglacial period. Some authors posit that low latitude biological export productivity may have been stronger at the LGM due to increased shelf-sourced phosporus (Broecker, 1981, 1982; Filippelli et al., 2007; Tamburini and Föllmi, 2009; Menviel et al., 2012) or increased biological matter remineralisation depth (Matsumoto, 2007; Menviel et al., 2012). Others argue that low latitude biological export productivity was weaker at the LGM due to lessened upwelling of thermocline waters and lower nutrient levels (Calvo et al., 2011; Hayes et al., 2011; Winckler et al., 2016). Weaker (stronger) glacial biological export productivity in the low latitude surface boxes would reduce (increase) the sensitivity of atmospheric CO2 to ocean circulation in our model-data experiments."

3. As far as I can tell, the authors simulate an increase in Southern Ocean production during MIS 2 and MIS 4 without applying the increase in dust deposition that is seen during these periods. It is therefore noteworthy, and should be emphasised further, that the increase in Southern Ocean production at MIS2 and MIS4 emerged independently without needing to provide the dust record to the model, and yet aligns with it. I think that this is a striking finding that needs more emphasis. Although, it is important to say that the model does not (or does it?) alter carbon export outside of the Southern Ocean, which may also have been important for

"tightening the lid" on the ocean carbon store (as I've discussed above).

**Text added**

**"Importantly, our finding for increased biological export productivity at MIS 2 and 4 is delivered without any model-simulated iron dust fertilisation of the Southern Ocean and entirely on account of model results best-fit to the atmospheric and ocean proxy data used. Therefore. the finding is a robust independently-derived support for increased biological export productivity at MIS 2 and 4."**

4. I don't follow the logic of paragraph 4 in the Discussion. First talks about the model results compared with the analysis of Kohfeld & Chase (2017), then diverts to Stephen & Keeling (2000) Antarctic sea ice changes. I suggest making the narrative of your discussion clearer here.

**Reworded to emphasise the debate we wish to address.**

5. It is important to mention that the contribution of SST is very likely overestimated, given your use of box model rather than the general circulation model. Box model atmospheric $CO_2$ is known to be more sensitive to the SST changes in the higher latitudes compared with general circulation models (Archer et al., 2000). This should be stated clearly in the discussion section as a caveat. I want to know how the results might change if the exercise were repeated with a GCM, as this may inspire others.

**This is an interesting and important debate. Thanks for the opportunity to explore.**

**A number of earlier studies including one mentioned by the reviewer (e.g. Broecker et al, 1999; Archer et al, 2000; Kohfeld and Ridgwell, 2009) had highlighted the tendency for simple box models to overestimate the effects of changes in high latitude SST on atmospheric CO2. However, these studies were focussed on the pioneering three or four box models from the 1980's and 1990's (e.g. Knox and McElroy, 1984; Seigenthaler and Wenk, 1984; Sarmiento and Toggweiler, 1984).**

**Furthermore, when we compare the results of our analysis with a range of models from the literature (table below), including GCM's we find our estimate of the G-IG contribution of SST to atmospheric $CO_2$ of 28 ppm as shown in Figure 14 fits very comfortably within the range (see table below). For example, Kohfeld and Ridgwell (2009) sampled studies using GCMs exclusively to yield a range of estimates of 21-30 ppm and average value of 26 ppm. Our estimate of 28 ppm within the range of GCMs. Finally, a recent study by Khatiwala et al (2019) using a GCM model found for a much greater contribution of lower SST in the LGM to atmospheric $CO_2$ of 44 ppm.**

| Estimate of G-IG impact of SST on atmospheric $CO_2$ (ppm) | Study | Model |
|---|---|---|
| 21-30 (mean 26) | Kohfeld and Ridgwell (2009) | Selection of general circulation models from the literature |
| 30 | Sigman and Boyle, 2000 | CYCLOPS box model |
| 18 | Brovkin et al, 2007 | CLIMBER-2 Earth system model of intermediate complexity |
| 30 | Kohler et al, 2010 | BICYCLE box model |
| 27.5 | Menviel et al, 2012 | Bern 3D model: 36 x 36 grid boxes and 32 unevenly spaced layer, three-dimensional frictional geostrophic balance ocean model |
| 21 | O'Neill et al, 2019a | SCP-M 7 box plus atmosphere box model |
| 44 | Khatiwala et al, 2019 | UVic ESCM ocean general circulation model |
| **28** | **O'Neill et al, 2020 (this study)** | **SCP-M 12 box plus atmosphere box model** |

To address the reviewer's points we have added a brief summary of the above to our discussion of the model results for SST in reference to Figure 14.

"Some studies observed that early versions of box models tended to overstate the effects of SST and other processes at high latitudes on atmospheric CO2, relative to general circulation models (GCMs) (Broecker et al., 1999; Archer et al., 2000; Ridgwell, 2001; Kohfeld and Ridgwell, 2009). However, our modelled estimate of 28 ppm for the contribution of SST to the glacial-interglacial atmospheric CO2 (Fig. 14) falls within the range of GCM-derived estimates of 21-30 ppm (mean value 26 ppm) compiled by Kohfeld and Ridgwell (2009), is similar to that of Menviel et al. (2016) (27.5 ppm) and substantially less than another recent GCM-derived estimate of 44 ppm (Khatiwala et al., 2019)."

6. Page 5, line 8 – Don't you mean 40S-60N?

This is 40-60 South is it is referring to the subpolar Southern Ocean boxes in each basin. We have checked this is clear.

**References**

Anderson, R., Ali, S., Bradtmiller, L., Nielsen, S., Fleisher, M., Anderson, B., and Burckle, L.: Wind-driven upwelling in the Southern Ocean and the deglacial rise in atmospheric CO2, Science, 323, 1443–1448, 2009.

[revised manuscript text omitted]

---

## Author Response (AR3)

**Thanks for the suggested edits and changes. Please see responses and marked up manuscript below.**

As a general comment, I would suggest to read the manuscript carefully, try and tighten the text by avoiding repetition or information that is not directly relevant. The parts of the text describing the terrestrial biosphere and changes in peats and permafrosts are usually hard to follow. Also, I would suggest to be more precise when describing the timing of changes that occur in proxy records. I think you should use "during" to mention changes occurring during a certain period and not "at": e.g. atmospheric CO2 decreased during MIS 4.

Please use the appropriate tense throughout the manuscript, and make sure you do not switch tense without a reason. For example, the tense of the Methods consistently varies between present and past tense (Methods should be in present tense).

**Changes made:**

**-Methods section changed to present tense, including table captions**
**-We have changed "at" to "during" for MIS references throughout**
**-We have followed a consistent approach when describing interglacial-glacial changes in time series proxy data, modelling experiments or results, to write all the MIS timings with the earlier MIS first – e.g. MIS "5e-5d", or "MIS 4-2", or "between MIS 4 and MIS 2". This is to reflect the time direction of the changes in the proxy data or model results. However, when we are describing the MIS in Section 2.2 or data time ranges we leave the descriptions as directed back in time.**
**-We have added a subheading for "Contribution and attribution analysis" in the Discussion**
**-We have trimmed parts of the terrestrial biosphere and peat discussions**
**-We have added more specific time references to proxy data throughout**

**Examples of removed duplication include:**

**-removed the second reference to Follows et al (2006) in the model description**
**-we removed the mention of atmospheric d13C increase during the G-IG cycle from our experiment forcings discussion, as this is covered in the data analysis section**
**-Trimmed the mention of sea-ice cover in the experiment description as this is covered in the discussion**
**-Removed duplicate description of AMOC shoaling in the attribution analysis, as it was already discussed earlier in the discussion.**
**-Removed duplicate description of the "iron hypothesis" at the end of the Section 5.3 "LGM and Holocene", due to its previous coverage in the discussion section.**

Ideally, the order of the material in the supplementary information should follow their mention in the main manuscript (which is not the case for Table S13 for example). Please consider changing the order of the material shown in SI.

**Table S13 is now Table S1**

P1, L.1: please amend as follows: "of changes in atmospheric CO2 concentration" **Changed**

P1, L. 13: Please move the part on MIS5b after MIS5d and thus before MIS4. **Changed**

P2, L. 2: "in the concentration of atmospheric CO2" (and elsewhere as you feel appropriate as it the concentration of atm. CO2 that changes in time). **Added**

P2, L. 31: "biogeochemical" **Fixed**

P3, L.10-11: "A second phase of 40ppm CO2 drawdown between 72 and 65 ka.." **Amended**

P3, L. 17: "and other proxy records presented in Kohfeld & Chase, 2017, covering the last glacial cycle." **Added**

P3, L. 32: "We use" **Changed**

P6, L. 17: delta sign. **Fixed**

P6, L. 20: "with respect to" **Fixed**

P6, L. 28: "we undertake" and "biological parameters" **Fixed**

P6, L. 29: "We target" (and please use present throughout the methods as sometimes you switch to the past). **Done**

P9, L. 33: "the terrestrial biosphere carbon reservoir and its d13C signature" **Changed**

Table 2: For atmospheric CO2 and d13C, please make sure the correct references are included: Bereiter et al., 2015 paper focuses on 600 to 800ka whereas your simulation focuses on the last 130ka. **Individual data sources added and the Bereiter compilation of all sources is also included, as also for Eggleston et al (2016).**

Figure 3 legend: Throughout please simply state "Figure SX" and remove "Supp. Info" in front. **Done**

Please remove L. 20-28, p14. **Removed**

From p15, L. 13 to p16, L. 1-2: Please add appropriate references. **Schneider et al (2013) and Eggleston et al (2016) references are added. Schneider et al (2013) reference a 0.4 per mil offset between the LGM and the penultimate glacial period, whereas Eggleston et al (2016) refer to a 0.4 per mil offset between Holocene and last interglacial, and also LGM and penultimate glacial.**

P16, L.2-3: Please be more precise as to when the d13Catm decreases occur. **More precise timing added**

P16, L. 21 and 22: "between the LGM and the Holocene" or MIS 2 and MIS 1 (but not the reverse). **Amended**

P18, L.6: Please replace "North Atlantic overturning circulation" by AMOC. **Amended**

P19, Legend of Figure 6: Please amend as follows:

"DD14C is ocean minus atmospheric D14C" (with the correct Delta sign).

"rather DD14C represents the difference between each ocean data point and the contemporary atm. 14C value. **Changed**

P20 L. 7, please change to "MIS3, then increases from 16 Sv to 29 Sv between MIS 2 (the LGM) and the Holocene." **Changed**

Figure 10: Please move the labels so that they do not cover the data. **Done, moved to centre of chart**

P24, Legend of Figure 11: "DD14C is ocean minus atmospheric D14C" **Corrected**

P24, L.6-7 – Figure 12 and legend: How did/can you correct NPP to take into account permafrosts and peats?
I can't see how this can be justified and would thus suggest to remove that part. I also note that both your NPP and the estimate from Hoogakker et al. do not take into account permafrosts and peats. It might be worth emphasing in the manuscript that there is only one box for the terrestrial biosphere and that the changes in NPP in that box are a simple function of pCO2.

**Best to simply remove the green lines and associated text for Figure 12.**

P26, L.2-7: This is an almost word for word repeat of P. 14, L.30 through p15 L. 1. Please amend. **The text is reworded to simplify and remove duplication**
P28, L. 14: "atmospheric" **Corrected**
P28, L. 18: Please remove "in the experiment" **Removed**
P30, L.5-6: "enhanced Southern ocean biological productivity could account for" **Amended**
P32, L. 13: The correct ref is Menviel et al. 2012. **Corrected**
P34, L. 4-6: This sentence is not really correct as in Menviel et al 2016 the impact of these different processes on pCO2 was not studied. Menviel et al., 2016 showed that the oceanic d13C, and D14C records were most consistent with a weak GOC and AMOC. Menviel et al., 2016 further showed that this weak oceanic circulation would significantly increase the deep ocean carbon content (and thus significantly contribute to the pCO2 decrease).
**Amended accordingly**

P34, L. 8: You should add a sentence of caution here stating that your analysis looks at the mean MIS state. The oceanic circulation was highly variable during MIS3. Your result could also be an artefact of averaging across MIS 3 D-O variability, and HS6. **Sentence added including reference to MIS 3 modelling of D-O events of Menviel et al (2014).**

P34, L. 34 and P35, L.1: I don't think that is the way to put it. Temperature and precipitation exert a dominant control on the vegetation type and will thus impact NPP. This is simply not taken into account in your model. **OK, we have reworded this sentence.**

P36, L. 21: please remove "transient" **Removed**

[revised manuscript text omitted]